# OLinear: A Linear Model for Time Series Forecasting in Orthogonally Transformed Domain

**Wenzhen Yue**[1], **Yong Liu**[2], **Hao Wang**[3], **Haoxuan Li**[4,*], **Xianghua Ying**[1,*],
**Ruohao Guo**[1], **Bowei Xing**[1], **Ji Shi**[1]

[1]State Key Laboratory of General Artificial Intelligence, School of Intelligence Science
and Technology, Peking University, [2]School of Software, BNRist, Tsinghua University,
[3]College of Control Science and Engineering, Zhejiang University,
[4]Center for Data Science, Peking University
`{yuewenzhen,hxli}@stu.pku.edu.cn, xhying@pku.edu.cn`

## Abstract

This paper presents **OLinear**, a **linear**-based multivariate time series forecasting model that operates in an **o**rthogonally transformed domain. Recent forecasting models typically adopt the temporal forecast (TF) paradigm, which directly encode and decode time series in the time domain. However, the entangled step-wise dependencies in series data can hinder the performance of TF. To address this, some forecasters conduct encoding and decoding in the transformed domain using fixed, dataset-independent bases (e.g., sine and cosine signals in the Fourier transform). In contrast, we utilize **OrthoTrans**, a data-adaptive transformation based on an orthogonal matrix that diagonalizes the series' temporal Pearson correlation matrix. This approach enables more effective encoding and decoding in the decorrelated feature domain and can serve as a plug-in module to enhance existing forecasters. To enhance the representation learning for multivariate time series, we introduce a customized linear layer, **NormLin**, which employs a normalized weight matrix to capture multivariate dependencies. Empirically, the NormLin module shows a surprising performance advantage over multi-head self-attention, while requiring nearly half the FLOPs. Extensive experiments on 24 benchmarks and 140 forecasting tasks demonstrate that OLinear consistently achieves state-of-the-art performance with high efficiency. Notably, as a plug-in replacement for self-attention, the NormLin module consistently enhances Transformer-based forecasters. The code and datasets are available at `https://github.com/jackyue1994/OLinear`.

## 1 Introduction

Multivariate time series forecasting is critical in fields such as weather [Wu et al., 2023a], transportation [Ma et al., 2021], energy [Zhou et al., 2021], and finance [Chen et al., 2023]. Time series forecasters typically adopt the temporal forecast (TF) paradigm [Wang et al., 2025a, Liu et al., 2024a, Nie et al., 2023, Wang et al., 2024a], which encodes time series into latent representations and decodes them back, all within the time domain. However, this paradigm struggles to fully exploit the forecasting potential in the presence of entangled intra-series dependencies [Yi et al., 2023, Yue et al., 2025]. To mitigate this issue, recent studies apply Fourier [Yi et al., 2024a, 2023, Yue et al., 2025] or wavelet [Masserano et al., 2024] transforms to obtain the decorrelated feature sequence and perform encoding and decoding in the transformed domain. Nevertheless, these methods rely on dataset-independent bases, which fail to exploit the dataset-specific temporal correlation information.

---

[*]Haoxuan Li and Xianghua Ying are the corresponding authors.

39th Conference on Neural Information Processing Systems (NeurIPS 2025).

In this paper, we introduce **OrthoTrans**, a dataset-adaptive transformation scheme that constructs an orthogonal basis via eigenvalue decomposition of the temporal Pearson correlation matrix [Gray and Davisson, 2004]. Projecting the series onto this basis obtains the decorrelated feature domain, providing a disentangled input for linear encoding and empirically improving forecasting performance. Notably, OrthoTrans is modular and can be integrated into existing forecasters to enhance their performance. In-depth ablation studies reveal that OrthoTrans promotes representation diversity and increases the rank of attention matrices in Transformer-based models.

As OrthoTrans transforms complex temporal variations into decorrelated features, the representation learning process can be effectively handled by linear layers [Yi et al., 2023]. Specifically, we employ a linear-based Cross-Series Learner (CSL) and Intra-Series Learner (ISL) to model multivariate correlations and sequential dynamics, respectively. To motivate our design, we note that in the classic self-attention mechanism, the attention entries are all positive with the row-wise L1 norm fixed as 1. Inspired by this, we design a new **lin**ear layer in CSL, called **NormLin**, where the weight matrix entries are made positive via the `Softplus` function and then row-wise **norm**alized. Surprisingly, the NormLin module consistently outperforms the self-attention mechanism while improving computational efficiency in the field of time series forecasting. The overall model is referred to as **OLinear**, and our contributions can be summarized as follows:

- In contrast to the commonly used TF paradigm, we introduce **OrthoTrans**, a dataset-adaptive transformation scheme which leverages the orthogonal matrix derived from the eigenvalue decomposition of the temporal Pearson correlation matrix. As a plug-in, it consistently improves the performance of existing forecasters.

- For better representation learning, we present the **NormLin** layer, which employs a row-normalized weight matrix to capture multivariate correlations. Notably, as a plug-in, the NormLin module improves both the accuracy and efficiency of Transformer-based forecasters. It also adapts well to decoder architectures and large-scale time series models.

- Extensive experiments on 24 benchmarks and 140 forecasting tasks (covering various datasets and prediction settings) demonstrate that **OLinear** consistently achieves state-of-the-art performance with competitive computational efficiency.

## 2 Related work

### 2.1 Transformed domain in time series forecasting

Deep learning–based forecasters typically adopt the TF paradigm [Wang et al., 2025a, 2024a, Yu et al., 2024, Wang et al., 2024b, Liu et al., 2024a, Nie et al., 2023, Wu et al., 2023b, Zeng et al., 2023], with the entire process performed in the time domain. However, this paradigm may underperform in the presence of strong intra-series correlations [Yi et al., 2023, Wang et al., 2025b]. To mitigate this, recent models propose to forecast in the frequency domain [Yi et al., 2024a, Xu et al., 2024, Yi et al., 2023] or the wavelet domain [Masserano et al., 2024] where step-wise correlations are reduced and performance improvements are observed. FreDF [Wang et al., 2025b] incorporates a frequency-domain regularization term into the loss function to address the issue of biased forecast. However, classical Fourier and wavelet transforms employ fixed, dataset-agnostic bases that do not explicitly use the specific statistical characteristics of the dataset. In this work, we leverage orthogonal matrices derived from the temporal Pearson correlation matrix to decorrelate the series data. This approach provides a more suitable input for linear encoding and improves forecasting performance.

### 2.2 Time series forecasters

Deep learning based time series forecasters can generally be categorized into Transformer-based [Wang et al., 2025a, Nie et al., 2023, Liu et al., 2024a, Wang et al., 2024b, Yu et al., 2024], linear-based [Wang et al., 2024a, Zeng et al., 2023, Yi et al., 2023, Xu et al., 2024, Yi et al., 2024a], TCN-based [Wu et al., 2023b, Luo and Wang, 2024], RNN-based [Lai et al., 2018, Rangapuram et al., 2018], and GNN-based [Huang et al., 2023a, Yi et al., 2024b] models. Recently, research interests increasingly focus on Transformer-based and linear-based methods, each offering distinct advantages. Transformer-based forecasters typically exhibit strong expressiveness, whereas linear-based models offer better computational efficiency. In this paper, we aim to achieve state-of-the-art performance using an efficient linear-based model. Specifically, we employ the linear-based NormLin module to

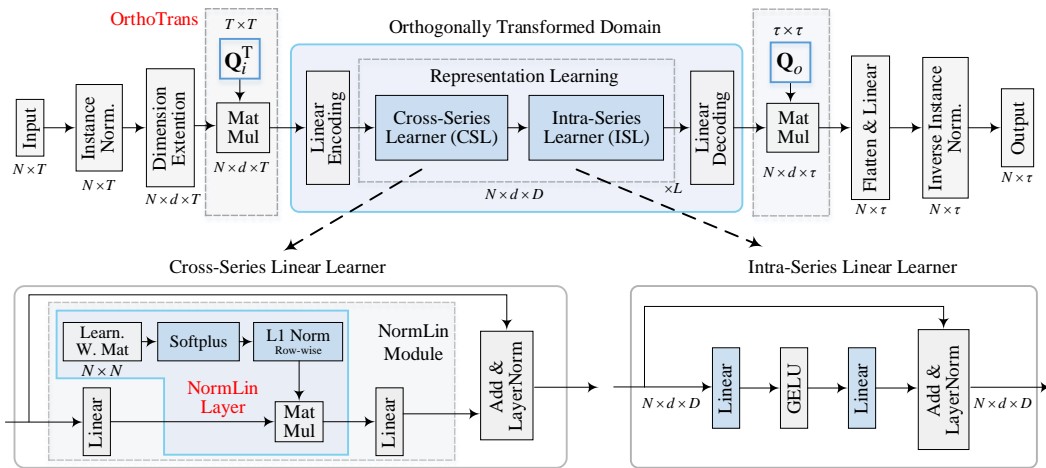

Figure 1: Overall structure of OLinear. The orthogonal matrix derived from the series' temporal Pearson correlation matrix transforms the series into the new feature domain with removed sequential correlations. The cross-series learners and intra-series learners are then employed for robust representation learning. Specifically, we introduce the NormLin layer, whose weight matrix is processed by the `Softplus` function and row-wise L1 normalization, to capture multivariate correlations.

robustly model inter-variate correlations. Remarkably, the NormLin module consistently outperforms the self-attention mechanism in Transformer-based forecasters, highlighting the strong capability of linear layers for time series forecasting.

## 3 Preliminaries

Real-world time series often exhibit temporal dependencies, where each time step is influenced by its predecessors. In this section, we derive the expected value of a future step given its past, assuming a multivariate Gaussian distribution. Without loss of generality, we consider a series $\mathbf{x} \in \mathbb{R}^t$, and the following time step $y \in \mathbb{R}$. We now present the following theorem.

**Theorem 1** (Expected value of $y$). *We assume that* $\mathbf{x} \sim \mathcal{N}\left(\mu_{\mathbf{x}}, \Sigma_{\mathbf{x}}\right)$, $y \sim \mathcal{N}\left(\mu_y, \sigma_y^2\right)$, *and the joint distribution* $[\mathbf{x}, y] \sim \mathcal{N}\left([\mu_{\mathbf{x}}, \mu_y], \begin{bmatrix} \Sigma_{\mathbf{x}} & \Sigma_{\mathbf{x}y} \\ \Sigma_{\mathbf{x}y}^{\mathsf{T}} & \sigma_y^2 \end{bmatrix}\right)$. *Here,* $\mathcal{N}\left(\cdot, \cdot\right)$ *denotes a Gaussian distribution with the specified mean vector and covariance matrix.* $\Sigma_{\mathbf{x}} \in \mathbb{R}^{t \times t}$ *is symmetric and positive definite.* $\Sigma_{\mathbf{x}y} \in \mathbb{R}^{t \times 1}$ *denotes the covariance between* $\mathbf{x}$ *and* $y$. *Then, the expected value of* $y$ *given* $\mathbf{x}$ *is*

$$\mu_{y|\mathbf{x}} = \mu_y + \Sigma_{\mathbf{x}y}^{\mathsf{T}} \Sigma_{\mathbf{x}}^{-1} \left(\mathbf{x} - \mu_{\mathbf{x}}\right) \tag{1}$$

The proof of Theorem 1 is provided in Appendix A. Equation 1 shows that for a temporally correlated series $[\mathbf{x}, y]$, the expected value of $y$ depends not only on its own mean $\mu_y$, but also on its past observations $\mathbf{x}$. The second term in Equation 1 introduces additional difficulty to forecasting under the TF paradigm. In contrast, decorrelating the series simplifies the forecasting task. In this work, we introduce OrthoTrans to transform the original series into a decorrelated transformed domain, converting the temporal forecasting problem into the *feature prediction* task. Experiments in Section 5.2 validate the effectiveness and generality of this transformation strategy.

## 4 Method

**Problem formulation** For multivariate time series forecasting, given a historical sequence $\mathbf{X} = \{\mathbf{x}_1, \cdots, \mathbf{x}_T\} \in \mathbb{R}^{N \times T}$ with $T$ time steps and $N$ variates, the task is to predict the future $\tau$ time steps $\mathbf{Y} = \{\mathbf{x}_{T+1}, \cdots, \mathbf{x}_{T+\tau}\} \in \mathbb{R}^{N \times \tau}$. Our goal is to approximate the ground truth $\mathbf{Y}$ as closely as possible with predictions $\hat{\mathbf{Y}}$.

## 4.1 Overall architecture

As shown in Figure 1, OLinear adopts a simple yet effective architecture. Given the input series $\mathbf{X}$, a RevIN [Kim et al., 2021] layer first performs instance normalization to mitigate non-stationarity. A dimension extension module then enhances expressiveness [Yi et al., 2023] by computing the outer product with a learnable vector $\phi_d \in \mathbb{R}^d$, where $d$ is the embedding size. Next, the time domain is decorrelated by multiplying with a transposed orthogonal matrix $\mathbf{Q}_i^{\mathsf{T}} \in \mathbb{R}^{T \times T}$, which will be detailed in Section 4.2. This process can be formulated as $\mathbf{Z} = (\text{RevIN}_{\text{Norm}}(\mathbf{X}) \otimes \phi_d) \mathbf{Q}_i^{\mathsf{T}} \in \mathbb{R}^{N \times d \times T}$, where $\otimes$ denotes the outer product.

We then perform encoding and forecasting in the transformed domain. Specifically, a linear layer first encodes the decorrelated features $\mathbf{Z}$ to the model dimension $D$. The cross-series learner (CSL) and intra-series learner (ISL) subsequently capture multivariate correlations and model intra-series dynamics, respectively. After passing through $L$ stacked blocks, the representation is decoded to the desired prediction length $\tau$, and then mapped back to the time domain via multiplication with the orthogonal matrix $\mathbf{Q}_o$. The overall process is summarized as:

$$
\begin{aligned}
\tilde{\mathbf{H}}^0 &= \text{LinearEncode}(\mathbf{Z}) \in \mathbb{R}^{N \times d \times D}, \\
\tilde{\mathbf{H}}^{l+1} &= \text{ISL}\left(\text{CSL}\left(\tilde{\mathbf{H}}^l\right)\right), \quad l = 0, \cdots, L-1, \\
\tilde{\mathbf{Y}} &= \text{LinearDecode}\left(\tilde{\mathbf{H}}^L\right) \cdot \mathbf{Q}_o \in \mathbb{R}^{N \times d \times \tau}.
\end{aligned}
\tag{2}
$$

Finally, the flattened output is mapped to shape $N \times \tau$ via a linear layer, then de-normalized to yield the final prediction: $\hat{\mathbf{Y}} = \text{RevIN}_{\text{DeNorm}}\left(\text{FlattenLinear}\left(\tilde{\mathbf{Y}}\right)\right)$.

## 4.2 Orthogonal transformation (OrthoTrans)

One effective approach for decorrelating the series is based on the Pearson correlation matrix. Let $\mathbf{X}^{train} \in \mathbb{R}^{N \times M}$ denote the training set, where $M$ is the length of the training series. For each variate $j$, we generate $T$ lagged series with temporal offsets from $0$ to $T-1$: $\mathbf{s}_i^j = \mathbf{X}[j, i:M-T+i]$, $i = 0, \cdots T-1$, where $\mathbf{s}_i^j$ denotes the $i$-th temporally lagged series of variate $j$. We then compute the Pearson correlation matrix $\text{CorrMat}_t^j$ of $\{\mathbf{s}_i^j\}$, whose $(p,q)$-th entry is $\frac{\text{Cov}(\mathbf{s}_p, \mathbf{s}_q)}{\sqrt{\text{Var}(\mathbf{s}_p) \cdot \text{Var}(\mathbf{s}_q)}}$, $p, q \in \{0, \cdots, T-1\}$.

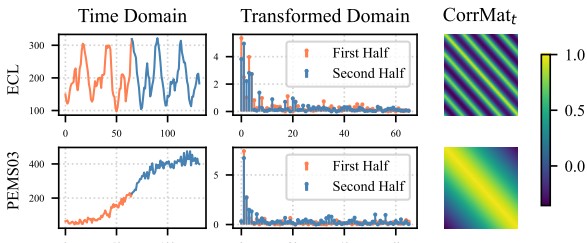

Here, $\text{Cov}(\cdot)$ denotes the covariance of two series, and $\text{Var}(\cdot)$ denotes the variance. The final Pearson correlation matrix is then obtained by averaging over all variates: $\text{CorrMat}_t = \frac{1}{N} \sum_{j=0}^{N-1} \text{CorrMat}_t^j$.

Figure 2: Comparison between the time and transformed domains. (1) The series exhibits strong consistency in the transformed domain, which is desirable for forecasting. (2) Temporal correlations are effectively reduced in the transformed domain. (3) Periodicity is clearly reflected in the correlation matrix, which facilitates modeling. (4) The new domain typically exhibits energy compaction, with only a few dominant components. More cases are shown in Figure 7.

Following the above procedure, the resulting $\text{CorrMat}_t$ is symmetric with all diagonal entries equal to 1. Based on the properties of symmetric matrices, we perform eigenvalue decomposition as $\text{CorrMat}_t = \mathbf{Q}_i \Lambda \mathbf{Q}_i^{\mathsf{T}}$, where $\mathbf{Q}_i$ is an orthogonal matrix [Horn and Johnson, 2012] whose columns are the eigenvectors of $\text{CorrMat}_t$ [Bishop and Nasrabadi, 2006], and $\Lambda$ is a diagonal matrix containing the corresponding eigenvalues.

For the input series $\mathbf{x}$ with temporal correlation matrix $\text{CorrMat}_t$ and unit variance, the covariance matrix of the transformed vector $\mathbf{Q}_i^{\mathsf{T}} \mathbf{x}$ is $\text{CovMat}\left(\mathbf{Q}_i^{\mathsf{T}} \mathbf{x}\right) = \mathbf{Q}_i^{\mathsf{T}} \text{CorrMat}_t \mathbf{Q}_i = \mathbf{Q}_i^{\mathsf{T}} \mathbf{Q}_i \Lambda \mathbf{Q}_i^{\mathsf{T}} \mathbf{Q}_i = \Lambda$, which is a diagonal matrix. Therefore, the entries of $\mathbf{Q}_i^{\mathsf{T}} \mathbf{x}$ are linearly uncorrelated, removing sequential dependencies. Similarly, we can compute $\mathbf{Q}_o \in \mathbb{R}^{\tau \times \tau}$, and recover the temporal correlations

by multiplying with $\mathbf{Q}_o$. Note that $\mathbf{Q}_i$ and $\mathbf{Q}_o$ are pre-computed and used throughout training and inference. Ablation studies on these two matrices are discussed in Appendix J.3. Moreover, as shown in Appendix I.10, OLinear remains robust even when the orthogonal matrices $\mathbf{Q}_i$ and $\mathbf{Q}_o$ are computed with limited training data.

**Discussion** (1) Mathematically, $\mathbf{Q}_i^\top \mathbf{x}$ projects $\mathbf{x}$ onto the eigenvectors of the temporal correlation matrix $\mathrm{CorrMat}_t$. The transformed domain exhibits energy compaction, and **noise is suppressed** in the primary components [Bishop and Nasrabadi, 2006]. (2) Compared to DFT and wavelet transforms, or to the TF paradigm without any transformation, OrthoTrans produces **higher-rank attention matrices** for Transformer-based forecasters (see Figures 10– 14), indicating greater representation diversity and enhanced model expressiveness. This partly explains why OrthoTrans enhances other forecasters when used as a plug-in module.

## 4.3 Representation learning

**CSL and NormLin** Inspired by the properties of attention matrices in self-attention [Vaswani et al., 2017], we impose two constraints on the weight matrix of the linear layer for robust multivariate representation learning: (1) all entries must be positive, and (2) each row must sum to 1. To enforce these constraints, we apply the `Softplus` function followed by row-wise L1 normalization to the learnable weight matrix $\mathbf{W}$. The resulting layer is referred to as NormLin, and is defined as follows:

$$\mathrm{NormLin}\left(\mathbf{x}\right) = \mathrm{RowNorm}_{\mathrm{L1}}\left(\mathrm{Softplus}\left(\mathbf{W}\right)\right)\mathbf{x}, \mathbf{W} \in \mathbb{R}^{N \times N}, \mathbf{x} \in \mathbb{R}^N. \tag{3}$$

The variants of NormLin are discussed in Appendix J.1. Incorporating the other components in Figure 1(b), the CSL process is formulated as:

$$\mathrm{CSL}\left(\tilde{\mathbf{H}}\right) = \mathrm{LayerNorm}\left(\tilde{\mathbf{H}} + \mathrm{Linear}\left(\mathrm{NormLin}\left(\mathrm{Linear}\left(\tilde{\mathbf{H}}\right)\right)\right)\right), \tag{4}$$

where $\mathrm{Linear}(\cdot)$ and $\mathrm{NormLin}(\cdot)$ operate on the sequence and variate dimensions, respectively. We define **the NormLin module** as $\mathrm{NormLinModule}\left(\cdot\right) \triangleq \mathrm{Linear}\left(\mathrm{NormLin}\left(\mathrm{Linear}\left(\cdot\right)\right)\right)$. Ablation studies on the two linear layers in $\mathrm{NormLinModule}\left(\cdot\right)$ are presented in Appendix J.2.

**ISL** We adopt two linear layers separated by the `GELU` activation function as a powerful predictive representation learner [Li et al., 2023]. It has been well established that MLPs are highly effective for encoding sequential dynamics and decoding future series [Zeng et al., 2023]. Similar to CSL, residual connections and LayerNorm are also applied. The complete ISL process is defined as:

$$\mathrm{ISL}\left(\tilde{\mathbf{H}}\right) = \mathrm{LayerNorm}\left(\tilde{\mathbf{H}} + \mathrm{Linear}\left(\mathrm{GELU}\left(\mathrm{Linear}\left(\tilde{\mathbf{H}}\right)\right)\right)\right). \tag{5}$$

**Discussion** Compared to the self-attention mechanism, the NormLin module offers the following advantages: (1) As shown in Table 1, it reduces **computational complexity** by half and decreases memory footprint by a factor of $1/h$, where $h$ is the number of attention heads; (2) The learned weight matrix in NormLin naturally exhibits **high rank**, in contrast to the low-rank nature of self-attention (see Appendix E.2). The low-rank issue in self-attention could arise from the sharp value concentration of the Softmax function. A higher-rank weight

Table 1: Comparison of FLOPs and memory usage between the NormLin module and multi-head self-attention (MHSA). Here, $h$ denotes the number of attention heads, typically set to 8.

| Module | NormLin Module | MHSA |
|---|---|---|
| FLOPs | $\mathcal{O}\left(N^2 D + 2ND^2\right)$ | $\mathcal{O}\left(2N^2 D + 4ND^2\right)$ |
| Memory | $\mathcal{O}\left(N^2 + ND\right)$ | $\mathcal{O}\left(hN^2 + ND\right)$ |

(or attention) matrix often better preserves the rank of the representation space and thus improves the model's expressiveness [Han et al., 2023]. (3) From the perspective of **gradient flow**, the NormLin layer provides a more direct backpropagation path for optimizing weight entries. Appendix B shows that the Jacobian matrices of $\mathrm{Softmax}(\cdot)$ in self-attention and $\mathrm{Norm}_{\mathrm{L1}}(\mathrm{Softplus}(\cdot))$ in the NormLin layer share a similar structure, but the latter offers greater flexibility. (4) The query-key mechanism of self-attention induces correlations in attention matrix entries, restricting the **optimization space**. In contrast, NormLin decouples these dependencies, allowing for more flexible parameter updates.

Table 2: Long-term forecasting results with prediction lengths $\tau \in \{12, 24, 48, 96\}$ for PEMS and $\tau \in \{96, 192, 336, 720\}$ for others. Lookback horizon $T = 96$. Results are averaged over four prediction lengths. *Avg* denotes further averaging over subsets. Full results are shown in Table 17.

| Model | OLinear (Ours) | | TimeMix. 2024a | | FilterNet 2024a | | DLinear 2023 | | TimeMix.++ 2025a | | Leddam 2024 | | CARD 2024b | | Fredformer 2024 | | iTrans. 2024a | | PatchTST 2023 | | TimesNet 2023b | |
|---|---|---|---|---|---|---|---|---|---|---|---|---|---|---|---|---|---|---|---|---|---|---|
| Metric | MSE | MAE | MSE | MAE | MSE | MAE | MSE | MAE | MSE | MAE | MSE | MAE | MSE | MAE | MSE | MAE | MSE | MAE | MSE | MAE | MSE | MAE |
| ETT(Avg) | 0.359 | **0.376** | 0.367 | 0.388 | 0.375 | 0.394 | 0.442 | 0.444 | **0.349** | 0.377 | 0.367 | 0.387 | 0.366 | 0.380 | 0.366 | 0.385 | 0.383 | 0.399 | 0.380 | 0.396 | 0.391 | 0.404 |
| ECL | **0.159** | **0.248** | 0.182 | 0.273 | 0.173 | 0.268 | 0.212 | 0.300 | 0.165 | 0.253 | 0.169 | 0.263 | 0.168 | 0.258 | 0.176 | 0.269 | 0.178 | 0.270 | 0.208 | 0.295 | 0.192 | 0.295 |
| Exchange | 0.355 | 0.399 | 0.387 | 0.416 | 0.388 | 0.419 | 0.354 | 0.414 | 0.357 | 0.409 | 0.354 | 0.402 | 0.362 | 0.402 | **0.333** | **0.391** | 0.360 | 0.403 | 0.367 | 0.404 | 0.416 | 0.443 |
| Traffic | 0.451 | **0.247** | 0.485 | 0.298 | 0.463 | 0.310 | 0.625 | 0.383 | **0.416** | 0.264 | 0.467 | 0.294 | 0.453 | 0.282 | 0.433 | 0.291 | 0.428 | 0.282 | 0.531 | 0.343 | 0.620 | 0.336 |
| Weather | 0.237 | **0.260** | 0.240 | 0.272 | 0.245 | 0.272 | 0.265 | 0.317 | **0.226** | 0.262 | 0.242 | 0.272 | 0.239 | 0.265 | 0.246 | 0.272 | 0.258 | 0.279 | 0.259 | 0.281 | 0.259 | 0.287 |
| Solar | 0.215 | 0.217 | 0.216 | 0.280 | 0.235 | 0.266 | 0.330 | 0.401 | **0.203** | 0.258 | 0.230 | 0.264 | 0.237 | 0.237 | 0.226 | 0.262 | 0.233 | 0.262 | 0.270 | 0.307 | 0.301 | 0.319 |
| PEMS03 | **0.095** | **0.199** | 0.167 | 0.267 | 0.145 | 0.251 | 0.278 | 0.375 | 0.165 | 0.263 | 0.107 | 0.210 | 0.174 | 0.275 | 0.135 | 0.243 | 0.113 | 0.221 | 0.180 | 0.291 | 0.147 | 0.248 |
| PEMS04 | **0.091** | **0.190** | 0.185 | 0.287 | 0.146 | 0.258 | 0.295 | 0.388 | 0.136 | 0.251 | 0.103 | 0.210 | 0.206 | 0.299 | 0.162 | 0.261 | 0.111 | 0.221 | 0.195 | 0.307 | 0.129 | 0.241 |
| PEMS07 | **0.077** | **0.164** | 0.181 | 0.271 | 0.123 | 0.229 | 0.329 | 0.395 | 0.152 | 0.258 | 0.084 | 0.180 | 0.149 | 0.247 | 0.121 | 0.222 | 0.101 | 0.204 | 0.211 | 0.303 | 0.124 | 0.225 |
| PEMS08 | **0.113** | **0.194** | 0.226 | 0.299 | 0.172 | 0.260 | 0.379 | 0.416 | 0.200 | 0.279 | 0.122 | 0.211 | 0.201 | 0.280 | 0.161 | 0.250 | 0.150 | 0.226 | 0.280 | 0.321 | 0.193 | 0.271 |
| 1st Count | 5 | 9 | 0 | 0 | 0 | 0 | 0 | 0 | 4 | 0 | 0 | 0 | 0 | 0 | 1 | 1 | 0 | 0 | 0 | 0 | 0 | 0 |

Table 3: Short-term forecasting results. Two settings are applied: S1 (Input-12, Predict-$\{3, 6, 9, 12\}$) and S2 (Input-36, Predict-$\{24, 36, 48, 60\}$). Average results across eight prediction lengths are reported. Full results are provided in Tables 18 and 19.

| Model | OLinear (Ours) | | TimeMix. 2024a | | FilterNet 2024a | | DLinear 2023 | | TimeMix.++ 2025a | | Leddam 2024 | | CARD 2024b | | Fredformer 2024 | | iTrans. 2024a | | PatchTST 2023 | | TimesNet 2023b | |
|---|---|---|---|---|---|---|---|---|---|---|---|---|---|---|---|---|---|---|---|---|---|---|
| Metric | MSE | MAE | MSE | MAE | MSE | MAE | MSE | MAE | MSE | MAE | MSE | MAE | MSE | MAE | MSE | MAE | MSE | MAE | MSE | MAE | MSE | MAE |
| ILI | **1.429** | **0.690** | 1.864 | 0.806 | 1.793 | 0.791 | 2.742 | 1.126 | 1.805 | 0.793 | 1.725 | 0.777 | 1.959 | 0.822 | 1.732 | 0.797 | 1.715 | 0.773 | 1.905 | 0.804 | 1.809 | 0.807 |
| COVID-19 | **5.187** | **1.211** | 5.919 | 1.350 | 5.607 | 1.322 | 8.279 | 1.601 | 5.974 | 1.369 | 5.251 | 1.285 | 5.536 | 1.314 | 5.279 | 1.287 | 5.301 | 1.293 | 5.836 | 1.362 | 6.106 | 1.369 |
| METR-LA | 0.587 | **0.311** | 0.608 | 0.372 | 0.603 | 0.366 | 0.580 | 0.422 | **0.567** | 0.363 | 0.603 | 0.367 | 0.639 | 0.350 | 0.617 | 0.369 | 0.627 | 0.373 | 0.614 | 0.372 | 0.617 | 0.370 |
| NASDAQ | 0.121 | 0.201 | 0.120 | 0.204 | 0.127 | 0.211 | 0.150 | 0.251 | 0.125 | 0.210 | 0.128 | 0.211 | 0.125 | 0.207 | 0.127 | 0.210 | 0.133 | 0.217 | 0.128 | 0.209 | 0.161 | 0.247 |
| Wiki | 6.395 | **0.415** | 6.443 | 0.439 | 6.457 | 0.439 | 6.420 | 0.510 | 6.430 | 0.443 | 6.417 | 0.433 | 6.419 | 0.427 | **6.318** | 0.429 | 6.422 | 0.432 | 6.368 | 0.424 | 7.633 | 0.572 |
| SP500 | **0.146** | **0.250** | 0.153 | 0.265 | 0.164 | 0.279 | 0.178 | 0.298 | 0.157 | 0.270 | 0.163 | 0.282 | 0.147 | 0.252 | 0.167 | 0.286 | 0.161 | 0.279 | 0.159 | 0.277 | 0.150 | 0.262 |
| DowJones | 7.686 | **0.619** | 8.499 | 0.633 | 8.283 | 0.633 | 7.893 | 0.626 | 8.895 | 0.643 | 8.257 | 0.633 | 7.699 | **0.619** | 8.041 | 0.625 | 8.177 | 0.630 | 7.991 | 0.626 | 10.960 | 0.737 |
| CarSales | 0.330 | **0.305** | 0.333 | 0.322 | 0.328 | 0.319 | 0.387 | 0.376 | 0.337 | 0.321 | 0.335 | 0.322 | 0.347 | 0.324 | 0.335 | 0.325 | **0.311** | 0.307 | 0.327 | 0.318 | 0.334 | 0.328 |
| Power | 1.248 | **0.835** | 1.234 | 0.840 | 1.309 | 0.870 | 1.278 | 0.870 | **1.234** | 0.841 | 1.295 | 0.868 | 1.288 | 0.847 | 1.302 | 0.870 | 1.324 | 0.874 | 1.311 | 0.873 | 1.317 | 0.871 |
| Website | 0.225 | 0.311 | 0.279 | 0.358 | 0.297 | 0.367 | 0.302 | 0.389 | 0.260 | 0.344 | 0.264 | 0.348 | 0.303 | 0.366 | 0.266 | 0.351 | **0.179** | **0.297** | 0.284 | 0.362 | 0.251 | 0.341 |
| Unemp | 0.729 | **0.461** | 1.581 | 0.708 | 1.286 | 0.627 | **0.565** | 0.509 | 1.506 | 0.678 | 1.502 | 0.689 | 1.163 | 0.596 | 2.048 | 0.789 | 1.408 | 0.666 | 1.237 | 0.624 | 2.328 | 0.852 |
| 1st Count | 4 | 10 | 2 | 0 | 0 | 0 | 1 | 0 | 2 | 0 | 0 | 0 | 0 | 1 | 1 | 0 | 2 | 1 | 0 | 0 | 0 | 0 |

## 5 Experiments

**Datasets and implementation details** We extensively evaluate OLinear using 24 diverse real-world datasets: **ETT** (four subsets), **Weather**, **ECL**, **Traffic**, **Exchange**, **Solar-Energy**, **PEMS** (four subsets), **ILI**, **COVID-19**, **METR-LA**, **NASDAQ**, **Wiki**, **SP500**, **DowJones**, **CarSales**, **Power**, **Website**, **Unemp**. The weighted L1 loss function from CARD [Wang et al., 2024b] is adopted. The embedding size $d$ is set as 16. Dataset description and more implementation details are presented in Appendices C and D, respectively.

### 5.1 Forecasting performance

**Baselines** We carefully choose 11 well-acknowledged state-of-the-art forecasting models as our baselines, including (1) Linear-based models: TimeMixer [Wang et al., 2024a], FilterNet [Yi et al., 2024a], FITS [Xu et al., 2024], DLinear [Zeng et al., 2023]; (2) Transformer-based models: TimeMixer++ [Wang et al., 2025a], Leddam [Yu et al., 2024], CARD [Wang et al., 2024b], Fredformer [Piao et al., 2024], iTransformer [Liu et al., 2024a], PatchTST [Nie et al., 2023]; (3) TCN-based model: TimesNet [Wu et al., 2023b].

**Main results** Comprehensive long-term and short-term forecasting results are presented in Tables 2 and 3, respectively, with the best results highlighted in **bold** and the second-best underlined. Lower MSE/MAE values indicate more accurate predictions. Across a wide range of benchmarks, OLinear

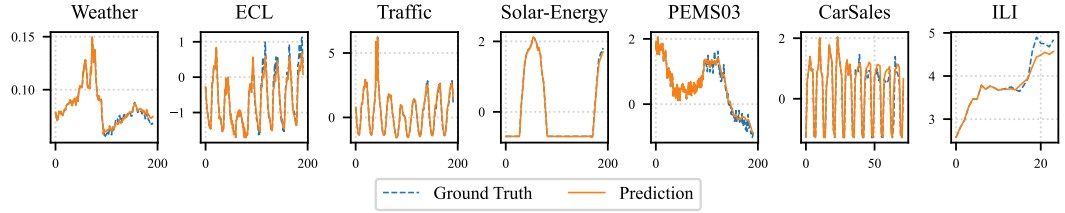

Figure 4: Visualization of the forecasting results of OLinear, demonstrating great accuracy.

Table 4: Ablation study on transformation bases. *S1* and *S2* represent 'Input-12, Predict-$\{3, 6, 9, 12\}$' and 'Input-36, Predict-$\{24, 36, 48, 60\}$', respectively. *Wavelet1* and *Wavelet2* use the Haar and discrete Meyer wavelets. *Identity* denotes no transformation. Full results are in Table 20.

| Dataset | ECL | | Solar-Energy | | PEMS03 | | Power (S2) | | ILI (S1) | | COVID (S2) | | METR-LA (S2) | |
|---|---|---|---|---|---|---|---|---|---|---|---|---|---|---|
| Metric | MSE | MAE | MSE | MAE | MSE | MAE | MSE | MAE | MSE | MAE | MSE | MAE | MSE | MAE |
| Ours | **0.159** | **0.248** | **0.215** | **0.217** | **0.095** | **0.199** | **1.487** | **0.922** | **1.094** | **0.578** | **8.467** | **1.754** | **0.838** | **0.402** |
| Fourier | 0.161 | 0.250 | 0.219 | 0.219 | 0.101 | 0.204 | 1.614 | 0.967 | 1.268 | 0.584 | 9.165 | 1.839 | 0.843 | 0.403 |
| Wavelet1 | 0.160 | 0.249 | 0.221 | 0.221 | 0.107 | 0.210 | 1.663 | 0.987 | 0.116 | 0.580 | 8.666 | 1.799 | 0.840 | 0.404 |
| Wavelet2 | 0.162 | 0.251 | 0.226 | 0.224 | 0.108 | 0.210 | 1.664 | 0.987 | 1.177 | 0.594 | 8.949 | 1.840 | 0.843 | 0.406 |
| Chebyshev | 0.218 | 0.295 | 0.226 | 0.226 | 0.105 | 0.207 | 1.570 | 0.965 | 1.217 | 0.597 | 9.330 | 1.875 | 0.854 | 0.407 |
| Laguerre | 0.167 | 0.255 | 0.233 | 0.230 | 0.111 | 0.214 | 1.659 | 0.984 | 1.353 | 0.651 | 9.302 | 1.890 | 0.868 | 0.420 |
| Legendre | 0.161 | 0.250 | 0.243 | 0.235 | 0.109 | 0.213 | 1.685 | 0.995 | 1.177 | 0.603 | 8.550 | 1.798 | 0.841 | 0.404 |
| Identity | 0.163 | 0.252 | 0.227 | 0.225 | 0.106 | 0.209 | 1.542 | 0.945 | 1.153 | 0.587 | 8.856 | 1.819 | 0.848 | 0.408 |

consistently outperforms state-of-the-art Transformer-based and linear-based forecasters. Notably, these gains are achieved with high computational efficiency. (Figure 3 and Table 41).

We attribute this superior performance to the adopted OrthoTrans and NormLin modules. The effectiveness of the simpler NormLin module challenges the necessity of the widely adopted multi-head self-attention mechanism, which has been a dominant design in prior works. To further validate robustness, we evaluate OLinear under varying lookback lengths (Table 29), where it consistently outperforms existing state-of-the-art methods.

Figure 4 shows the prediction visualizations of OLinear. Moreover, Table 14 demonstrates that OLinear exhibits greater robustness to random seeds compared to state-of-the-art Transformer-based forecasters such as TimeMixer++ and iTransformer.

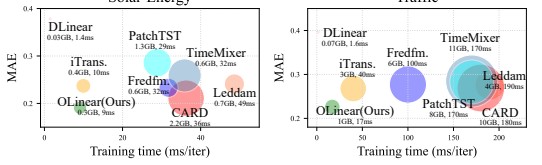

Figure 3: Model efficiency comparison. Bubble areas represent GPU memory usage during training, scaled independently in the two subfigures. Resource footprint data are from Table 41.

## 5.2 Model analysis

**Transformation bases** We replace Ortho-Trans with several commonly used bases, including the Fourier basis, two wavelet bases (Haar and discrete Meyer wavelets), and three polynomial bases. As shown in Table 4, our method consistently outperforms all alternatives. Specifically, it achieves a 5.5% reduction in MSE over the Fourier basis, and a 10.0% improvement over the no-transformation baseline (i.e., the TF paradigm) on PEMS03.

**OrthoTrans as a plug-in** We further integrate OrthoTrans into three classic forecasters: iTrans-

Table 5: Applying OrthoTrans (O.Trans) to iTransformer, PatchTST and RLinear. Average MSEs are reported. Full results are in Table 21.

| Model | iTrans. 2024a | | PatchTST 2023 | | RLinear 2023 | |
|---|---|---|---|---|---|---|
| | Van. | +O.Trans | Van. | +O.Trans | Van. | +O.Trans |
| ETTm1 | 0.407 | **0.404** | 0.387 | **0.384** | 0.414 | **0.408** |
| ECL | 0.178 | **0.171** | 0.208 | **0.181** | 0.219 | **0.214** |
| PEMS03 | 0.113 | **0.103** | 0.180 | **0.163** | 0.495 | **0.477** |
| PEMS07 | 0.101 | **0.085** | 0.211 | **0.147** | 0.504 | **0.485** |
| Solar | 0.233 | **0.228** | 0.270 | **0.239** | 0.369 | **0.354** |
| Weather | 0.258 | **0.252** | 0.259 | **0.246** | 0.272 | **0.269** |
| METR-LA | 0.338 | **0.329** | 0.335 | **0.333** | 0.342 | **0.341** |

former, PatchTST, and RLinear [Li et al., 2023]. As shown in Table 5, OrthoTrans yields average MSE improvements of 5.1% and 10.1% for iTransformer and PatchTST, respectively, highlighting the benefit of incorporating dataset-specific statistical information into the model design. This improvement can be attributed to the increased attention matrix rank introduced by OrthoTrans (see Appendix E.3), which indicates enlarged representation space and enhanced model capacity.

**Representation learning**   To validate the rationality of our CSL and ISL designs, we conduct ablation studies by replacing or removing their core components—NormLin and (standard) linear layers. As shown in Table 6, our design—applying NormLin along the variate dimension and standard linear layers along the temporal dimension—consistently achieves the best performance. Notably, applying NormLin along the *variate* dimension consistently outperforms its **self-attention** counterpart (last row), with reduced computational cost. Furthermore, removing the ISL module (third-last row) results in a 6.2% performance decline, highlighting the importance of updating temporal representations. Interestingly, on small-scale datasets with fewer variates (e.g., NASDAQ and ILI), the model with only temporal linear layers (third row) exhibits competitive performance, implying that NormLin is more beneficial when handling a larger number of variates.

Table 6: Ablations on the CSL and ISL design. 'NormLin (Temporal)' replaces the linear layer in Eq. 5 with the NormLin layer, and 'Linear (Variate)' replaces the NormLin layer in Eq. 4 with a standard linear layer. 'Attn.' denotes the self-attention mechanism. Results are averaged over four prediction lengths. Full results are shown in Table 22.

| Variate | Temp. | ECL | | Traffic | | Solar | | PEMS03 | | Weather | | ETTm1 | | NASDAQ (S1) | | ILI (S2) | |
|---|---|---|---|---|---|---|---|---|---|---|---|---|---|---|---|---|---|
| | | MSE | MAE | MSE | MAE | MSE | MAE | MSE | MAE | MSE | MAE | MSE | MAE | MSE | MAE | MSE | MAE |
| NormLin | Linear | **0.159** | **0.248** | **0.451** | **0.247** | **0.215** | **0.217** | **0.095** | **0.199** | **0.237** | **0.260** | **0.374** | **0.377** | 0.055 | 0.125 | **1.764** | **0.802** |
| Linear | Linear | 0.178 | 0.272 | 0.606 | 0.320 | 0.246 | 0.238 | 0.121 | 0.226 | 0.238 | 0.261 | 0.377 | 0.380 | 0.057 | 0.132 | 1.938 | 0.837 |
| w/o | Linear | 0.178 | 0.259 | 0.482 | 0.257 | 0.241 | 0.232 | 0.147 | 0.234 | 0.247 | 0.266 | 0.378 | 0.379 | **0.054** | **0.124** | 1.864 | 0.823 |
| NormLin | NormLin | 0.169 | 0.257 | 0.460 | 0.275 | 0.252 | 0.240 | 0.112 | 0.214 | 0.239 | 0.261 | 0.381 | 0.382 | 0.055 | 0.126 | 1.947 | 0.836 |
| Linear | NormLin | 0.183 | 0.276 | 0.578 | 0.339 | 0.262 | 0.254 | 0.143 | 0.246 | 0.240 | 0.261 | 0.383 | 0.384 | 0.057 | 0.133 | 2.037 | 0.867 |
| w/o | NormLin | 0.185 | 0.266 | 0.493 | 0.290 | 0.283 | 0.262 | 0.182 | 0.269 | 0.246 | 0.265 | 0.384 | 0.385 | 0.055 | 0.125 | 2.093 | 0.874 |
| NormLin | w/o | 0.169 | 0.257 | 0.460 | 0.275 | 0.253 | 0.241 | 0.114 | 0.215 | 0.239 | 0.261 | 0.380 | 0.382 | 0.055 | 0.126 | 1.940 | 0.837 |
| Linear | w/o | 0.183 | 0.276 | 0.591 | 0.341 | 0.262 | 0.254 | 0.142 | 0.246 | 0.240 | 0.262 | 0.384 | 0.384 | 0.057 | 0.132 | 2.073 | 0.874 |
| Attn. | Linear | 0.166 | 0.255 | 0.457 | 0.251 | 0.220 | 0.221 | 0.097 | 0.202 | 0.244 | 0.265 | 0.391 | 0.389 | 0.056 | 0.126 | 2.022 | 0.847 |

**OLinear-C**   As shown in Figure 5, the learned weight matrix resemble $\mathrm{Softmax}\left(\mathrm{CorrMat}_v\right)$, where $\mathrm{CorrMat}_v$ is the Pearson correlation matrix across variates. Motivated by this, we replace the learnable weights in NormLin with the pre-computed $\mathrm{Softmax}\left(\mathrm{CorrMat}_v\right)$, resulting in a simplified variant: $\mathrm{NormLin}_c\left(\mathbf{x}\right) = \mathrm{Softmax}\left(\mathrm{CorrMat}_v\right)\mathbf{x}$. **This indicates that the weight matrix in NormLin is optimized to align with the correlation matrix among channels** $\mathrm{CorrMat}_v$. We refer to the model with this NormLin variant as **OLinear-C**. Table 7 shows that OLinear-C performs comparably to OLinear with reduced learnable parameters. Comprehensive evaluation of OLinear-C and the variants of $\mathrm{NormLin}_c$ are discussed in Appendix H.



Figure 5: The learned NormLin weights exhibit similarity to the multivariate correlation matrix (after `Softmax`).

## 5.3   Generality and scalability of the NormLin module

In our CSL module, we employ the NormLin module—comprising the NormLin layer and its associated pre- and post-linear layers (Equation 4)—to capture multivariate correlations. Despite its simple architecture, the NormLin module demonstrates strong capability, generality, and scalability. Specifically, it consistently outperforms the classic multi-head self-attention mechanism and its variants, offering a compelling alternative for token dependency modeling in time series forecasting.

**Comparison with attention mechanisms**   To assess its effectiveness, we compare NormLin with classic attention variants, such as Reformer [Kitaev et al., 2020], Flowformer [Wu et al., 2022], FLatten [Dao et al., 2022], and Mamba [Gu and Dao, 2023]. As shown in Table 8, NormLin

Table 7: Performance comparison of OLinear and OLinear-C. Full results are shown in Table 25.

| Dataset | ECL | | Traffic | | ETT | | Solar | | PEMS | | CarSales | | ILI | | COVID-19 | | Unemp | |
|---|---|---|---|---|---|---|---|---|---|---|---|---|---|---|---|---|---|---|
| Metric | MSE | MAE | MSE | MAE | MSE | MAE | MSE | MAE | MSE | MAE | MSE | MAE | MSE | MAE | MSE | MAE | MSE | MAE |
| OLinear | **0.159** | **0.248** | **0.451** | **0.247** | **0.359** | **0.376** | **0.215** | **0.217** | **0.094** | **0.187** | **0.330** | **0.305** | **1.429** | **0.690** | **5.187** | **1.211** | **0.729** | **0.461** |
| OLinear-C | 0.161 | 0.249 | **0.451** | **0.247** | **0.359** | **0.376** | **0.215** | **0.217** | **0.094** | **0.187** | **0.330** | **0.305** | 1.463 | 0.698 | 5.346 | 1.247 | 0.766 | 0.474 |

Table 8: Comparison of the NormLin module with state-of-the-art attention variants. Average MSEs are reported. Full results are in Table 23.

| Dataset | NormLin (Ours) | Trans. 2017 | Refm. 2020 | Flowfm. 2022 | Flatten 2023 | Mamba 2023 | E.Attn. 2025 |
|---|---|---|---|---|---|---|---|
| ECL | **0.159** | 0.166 | 0.167 | 0.165 | 0.164 | 0.176 | **0.159** |
| Traffic | 0.451 | 0.457 | 0.459 | 0.460 | 0.464 | 0.456 | **0.439** |
| PEMS03 | **0.095** | 0.097 | 0.096 | 0.099 | 0.101 | 0.104 | 0.097 |
| Weather | **0.237** | 0.244 | 0.241 | 0.242 | 0.246 | 0.242 | 0.241 |
| Solar | **0.215** | 0.223 | 0.216 | 0.222 | 0.231 | 0.228 | 0.217 |
| ILI | **1.764** | 2.022 | 1.821 | 1.881 | 2.134 | 1.950 | 1.878 |
| NASDAQ | **0.055** | 0.056 | **0.055** | **0.055** | 0.057 | **0.055** | **0.055** |

Table 9: Applying NormLin (N.Lin) to state-of-the-art Transformer-based forecasters. Average MSEs are reported. Full results are in Table 24.

| Dataset | iTrans. | | PatchTST | | Leddam | | Fredformer | |
|---|---|---|---|---|---|---|---|---|
| | Van. | N.Lin | Van. | N.Lin | Van. | N.Lin | Van. | N.Lin |
| ETTm1 | 0.407 | **0.388** | 0.387 | **0.379** | 0.386 | **0.381** | 0.384 | **0.381** |
| ECL | 0.178 | **0.166** | 0.208 | **0.181** | 0.169 | **0.165** | 0.176 | **0.169** |
| PEMS03 | 0.113 | **0.102** | 0.180 | **0.146** | 0.107 | **0.103** | 0.134 | **0.108** |
| PEMS07 | 0.101 | **0.086** | 0.211 | **0.168** | 0.084 | **0.082** | 0.121 | **0.096** |
| Solar | 0.233 | **0.226** | 0.270 | **0.237** | 0.230 | **0.222** | **0.226** | **0.226** |
| Weather | 0.258 | **0.245** | 0.259 | **0.245** | 0.242 | **0.242** | 0.246 | **0.240** |
| METR-LA | 0.338 | **0.328** | 0.335 | **0.333** | 0.327 | **0.320** | 0.336 | **0.329** |

consistently outperforms these methods, indicating that simply normalizing the weight matrices can be more effective than self-attention for time series forecasting (see Paragraph 4.3).

**NormLin as a plug-in** To demonstrate the generality of the NormLin module, we replace self-attention with NormLin in Transformer-based forecasters. As shown in Table 9, this substitution leads to notable MSE improvements—6.7% for iTransformer and 10.3% for PatchTST—validating its plug-and-play effectiveness. These results also validate NormLin's capability to model dependencies across multiple token types (e.g., variate, temporal patch, and frequency-domain patch), highlighting its potential as a universal token dependency learner for time series forecasting. Furthermore, the NormLin module consistently improves both training and inference efficiency across these models (see Table 42); for example, it boosts iTransformer's inference efficiency by an average of 53%.

**Scalability of the NormLin module** Decoder-only Transformers are currently the *de facto* architecture of large time series models [Liu et al., 2024b, Ansari et al., 2024, Das et al., 2024]. We take Timer [Liu et al., 2024b] as a representative model and replace its self-attention with our NormLin module. To align with Timer's decoder-only structure, we apply a causal mask by zeroing out the upper triangular part of NormLin's weight matrix. The modified model is pre-trained on the UTSD dataset [Liu et al., 2024b], which spans seven domains and contains up to 1 billion time points. As shown in Table 10, NormLin improves performance in both zero-shot and fine-tuning scenarios. For example, the zero-shot MSE on ETTh2 and ETTm1 is reduced by 12.7% and 8.4%, respectively. These results demonstrate that the NormLin module adapts well to decoder-only architectures and scales effectively to large-scale pre-training scenarios.

Table 10: Applying NormLin to the large time series model Timer. For fine-tuning, 5% samples of trainset are used. MSEs are reported.

| Dataset | Fine-tuning | | Zero-shot | |
|---|---|---|---|---|
| | Van. | NormLin | Van. | NormLin |
| ETTh1 | 0.362 | **0.360** | 0.438 | **0.404** |
| ETTh2 | 0.280 | **0.269** | 0.314 | **0.274** |
| ETTm1 | 0.321 | **0.309** | 0.690 | **0.632** |
| ETTm2 | 0.176 | **0.172** | 0.213 | **0.212** |
| ECL | 0.132 | **0.130** | 0.192 | **0.183** |
| Traffic | 0.361 | **0.353** | **0.458** | 0.462 |
| Weather | 0.151 | **0.149** | 0.181 | **0.174** |

## 6 Conclusion

In this work, we present OLinear, a simple yet effective linear-based forecaster that achieves state-of-the-art performance, built on two core components: (1) OrthoTrans, an orthogonal transformation that decorrelates temporal dependencies to facilitate better encoding and forecasting, and (2) the NormLin module, a powerful and general-purpose token dependency learner. Notably, both modules consistently improve existing forecasters when used as plug-ins. We hope OLinear will serve as a strong baseline and inspire future studies in time series forecasting.

## Acknowledgments

This work was supported by Beijing Natural Science Foundation under Grant No. L247029, and the National Natural Science Foundation of China (NSFC) under Grant No. 62371009 and Grant No. 623B2002.

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

# A Proof of Theorem 1

*Proof.* For clarity, we denote $\mathbf{z} \triangleq \begin{bmatrix} \mathbf{x} \\ y \end{bmatrix} \in \mathbb{R}^{t+1}$, $\mu_z \triangleq \begin{bmatrix} \mu_\mathbf{x} \\ \mu_y \end{bmatrix} \in \mathbb{R}^{t+1}$, and $\Sigma \triangleq \begin{bmatrix} \Sigma_\mathbf{x} & \Sigma_{\mathbf{x}y} \\ \Sigma_{\mathbf{x}y}^\mathsf{T} & \sigma_y^2 \end{bmatrix}$.
According to the definition, the probability density function of z, i.e, the joint density of $\mathbf{x}$ and $y$, is

$$p(\mathbf{z}) = p(\mathbf{x}, y) = \frac{1}{(2\pi)^{(t+1)/2}|\Sigma|^{1/2}} \exp\left( -\frac{1}{2}(\mathbf{z} - \mu_z)^\mathsf{T}\Sigma^{-1}(\mathbf{z} - \mu_z) \right), \tag{6}$$

where $|\Sigma|$ denotes the determinant of $\Sigma$. In the following, we ignore the constant coefficient and focus on the exponential term.

Using the standard block matrix inverse formula [Horn and Johnson, 2012], we have

$$\Sigma^{-1} = \begin{bmatrix} \Sigma_\mathbf{x}^{-1} + d^{-1}\Sigma_\mathbf{x}^{-1}\Sigma_{\mathbf{x}y}\Sigma_{\mathbf{x}y}^\mathsf{T}\Sigma_\mathbf{x}^{-1} & -d^{-1}\Sigma_\mathbf{x}^{-1}\Sigma_{\mathbf{x}y} \\ -d^{-1}\Sigma_{\mathbf{x}y}^\mathsf{T}\Sigma_\mathbf{x}^{-1} & d^{-1} \end{bmatrix} \triangleq \begin{bmatrix} \mathbf{A} & -\mathbf{b} \\ -\mathbf{b}^\mathsf{T} & d^{-1} \end{bmatrix}, \tag{7}$$

where the scalar $d = \sigma_y^2 - \Sigma_{\mathbf{x}y}^\mathsf{T}\Sigma_\mathbf{x}^{-1}\Sigma_{\mathbf{x}y}$ is the Schur complement [Horn and Johnson, 2012]. Letting $\mathbf{b} \in \mathbb{R}^{t \times 1}$ denote $d^{-1}\Sigma_\mathbf{x}^{-1}\Sigma_{\mathbf{x}y}$, the matrix $\mathbf{A}$ becomes $\Sigma_\mathbf{x}^{-1} + d \cdot \mathbf{b}\mathbf{b}^\mathsf{T}$.

For notational simplicity, we define $\alpha \triangleq \mathbf{x} - \mu_\mathbf{x} \in \mathbb{R}^{t \times 1}$ and $\beta \triangleq y - \mu_y \in \mathbb{R}$. Then, the exponent in Equation 6 (ignoring the constant factor $-1/2$) becomes

$$\begin{aligned}
(\mathbf{z} - \mu_z)^\mathsf{T}\Sigma^{-1}(\mathbf{z} - \mu_z) &= \begin{bmatrix} \alpha^\mathsf{T} & \beta \end{bmatrix} \begin{bmatrix} \Sigma_\mathbf{x}^{-1} + d \cdot \mathbf{b}\mathbf{b}^\mathsf{T} & -\mathbf{b} \\ -\mathbf{b}^\mathsf{T} & d^{-1} \end{bmatrix} \begin{bmatrix} \alpha \\ \beta \end{bmatrix} \\
&= \alpha^\mathsf{T}\Sigma_\mathbf{x}^{-1}\alpha + d \cdot \alpha^\mathsf{T}\mathbf{b}\mathbf{b}^\mathsf{T} - 2\beta\alpha^\mathsf{T}\mathbf{b} + d^{-1}\beta^2 \\
&= \alpha^\mathsf{T}\Sigma_\mathbf{x}^{-1}\alpha + d^{-1}\left( \beta^2 - 2d \cdot \alpha^\mathsf{T}\mathbf{b} \cdot \beta + d^2 \cdot \alpha^\mathsf{T}\mathbf{b}\mathbf{b}^\mathsf{T}\alpha \right) \\
&= \alpha^\mathsf{T}\Sigma_\mathbf{x}^{-1}\alpha + d^{-1}\left( \beta - d\alpha^\mathsf{T}\mathbf{b} \right)^2.
\end{aligned} \tag{8}$$

Since $p(\mathbf{x}) \propto \exp\left( -\frac{1}{2}\alpha^\mathsf{T}\Sigma_\mathbf{x}^{-1}\alpha \right)$, it follows that

$$\begin{aligned}
p(y \mid \mathbf{x}) &= \frac{p(\mathbf{x}, y)}{p(\mathbf{x})} \\
&\propto \exp\left( -\frac{1}{2}\left( \alpha^\mathsf{T}\Sigma_\mathbf{x}^{-1}\alpha + d^{-1}\left( \beta - d\alpha^\mathsf{T}\mathbf{b} \right)^2 - \alpha^\mathsf{T}\Sigma_x^{-1}\alpha \right) \right) \\
&= \exp\left( -\frac{\left( \beta - d\alpha^\mathsf{T}\mathbf{b} \right)^2}{2d} \right) \\
&= \exp\left( -\frac{\left( y - \mu_y - d\alpha^\mathsf{T}\mathbf{b} \right)^2}{2d} \right).
\end{aligned} \tag{9}$$

Therefore, the conditional density of $y$ give $\mathbf{x}$ is also a Gaussian distribution, whose mean is

$$\begin{aligned}
\mu_y + d\alpha^\mathsf{T}\mathbf{b} &= \mu_y + d\mathbf{b}^\mathsf{T}\alpha \\
&= \mu_y + d \cdot d^{-1}\Sigma_{\mathbf{x}y}^\mathsf{T}\Sigma_\mathbf{x}^{-1}(\mathbf{x} - \mu_x) \\
&= \mu_y + \Sigma_{\mathbf{x}y}^\mathsf{T}\Sigma_\mathbf{x}^{-1}(\mathbf{x} - \mu_\mathbf{x}),
\end{aligned} \tag{10}$$

and the variance is $d = \sigma_y^2 - \Sigma_{\mathbf{x}y}^\mathsf{T}\Sigma_\mathbf{x}^{-1}\Sigma_{\mathbf{x}y}$. This completes the proof.

$\square$

# B Jacobian matrix comparison of self-attention and NormLin

In this section, we analyze the gradients of the non-linear transformations in the self-attention mechanism and the NormLin module, focusing on the attention/weight rows, denoted as $\mathbf{a} \in \mathbb{R}^N$.

## B.1 Jacobian matrix of Softmax in self-attention

Let $\mathbf{c} \triangleq \mathrm{Softmax}(\mathbf{a}) \in \mathbb{R}^N$. We first compute the partial derivatives element-wise, and then rewrite the results in matrix form. Based on the definition of $\mathrm{Softmax}$, the $i$-th element of $\mathbf{c}$ is $\mathbf{c}_i = \frac{e^{\mathbf{a}_i}}{\sum_{k=1}^N e^{\mathbf{a}_k}}$. The partial derivative of $\mathbf{c}_i$ with respect to $\mathbf{a}_i$ is:

$$\frac{\partial \mathbf{c}_i}{\partial \mathbf{a}_i} = \frac{e^{\mathbf{a}_i}}{\sum_{k=1}^N e^{\mathbf{a}_k}} - \frac{e^{\mathbf{a}_i} \cdot e^{\mathbf{a}_i}}{(\sum_{k=1}^N e^{\mathbf{a}_k})^2} = \mathbf{c}_i - \mathbf{c}_i^2. \tag{11}$$

For $j \neq i$, the partial derivative of $\mathbf{c}_i$ with respect to $\mathbf{a}_j$ is:

$$\frac{\partial \mathbf{c}_i}{\partial \mathbf{a}_j} = -\frac{e^{\mathbf{a}_i} \cdot e^{\mathbf{a}_j}}{(\sum_{k=1}^N e^{\mathbf{a}_k})^2} = -\mathbf{c}_i \mathbf{c}_j \tag{12}$$

Combining Equations (11) and (12), we obtain:

$$\frac{\partial \mathbf{c}_i}{\partial \mathbf{a}} = \mathbf{c}_i[\delta_{1i}, \delta_{2i}, \ldots, \delta_{Ni}] - \mathbf{c}_i \mathbf{c}^\top, \tag{13}$$

where $\delta_{ji} = 1$ if $j = i$, otherwise 0. Therefore, the Jacobian matrix $\frac{\partial \mathbf{c}}{\partial \mathbf{a}}$ can be written as

$$\frac{\partial \mathbf{c}}{\partial \mathbf{a}} = \mathrm{Diag}(\mathbf{c}) - \mathbf{c}\mathbf{c}^\top, \tag{14}$$

where $\mathrm{Diag}(\mathbf{c})$ is the diagonal matrix with $\mathbf{c}$ as its diagonal. We can observe that the Jacobian matrix is a function of $\mathbf{c}$. Since the Softmax function could cause sharp value concentration, the Jacobian matrix may exhibit sparsity, with most entries being close to zero [Surya Duvvuri and Dhillon, 2024].

## B.2 Jacobian matrix of the NormLin layer

Let $\mathbf{c} \triangleq \mathrm{Norm}_{\mathrm{L1}}(\mathrm{Softplus}(\mathbf{a})) \in \mathbb{R}^N$, where $\mathrm{Norm}_{\mathrm{L1}}(\cdot)$ denotes the L1 normalization. For notational simplicity, let $\mathbf{b} \triangleq \mathrm{Softplus}(\mathbf{a})$. We first analyze $\frac{\partial \mathbf{c}}{\partial \mathbf{b}}$. Since all entries of $\mathbf{b}$ are positive, we have:

$$\mathbf{c}_i = \frac{\mathbf{b}_i}{\sum_{k=1}^N \mathbf{b}_k}. \tag{15}$$

Therefore, it follows that

$$\frac{\partial \mathbf{c}_i}{\partial \mathbf{b}_i} = \frac{1}{\sum_{k=1}^N \mathbf{b}_k} - \frac{\mathbf{b}_i}{(\sum_{k=1}^N \mathbf{b}_k)^2} = \frac{1}{\|\mathbf{b}\|_1^2} \left(\|\mathbf{b}\|_1 - \mathbf{b}_i\right). \tag{16}$$

For $j \neq i$, the partial derivative $\frac{\partial \mathbf{c}_i}{\partial \mathbf{b}_j}$ is:

$$\frac{\partial \mathbf{c}_i}{\partial \mathbf{b}_j} = -\frac{\mathbf{b}_i}{(\sum_{k=1}^N \mathbf{b}_k)^2} = -\frac{1}{\|\mathbf{b}\|_1^2} \mathbf{b}_i. \tag{17}$$

Combining Equations (16) and (17), we obtain the Jacobian matrix:

$$\frac{\partial \mathbf{c}}{\partial \mathbf{b}} = \frac{1}{\|\mathbf{b}\|_1^2} \left(\|\mathbf{b}\|_1 \mathbf{I} - \mathbf{b}\mathbf{1}^\top\right), \tag{18}$$

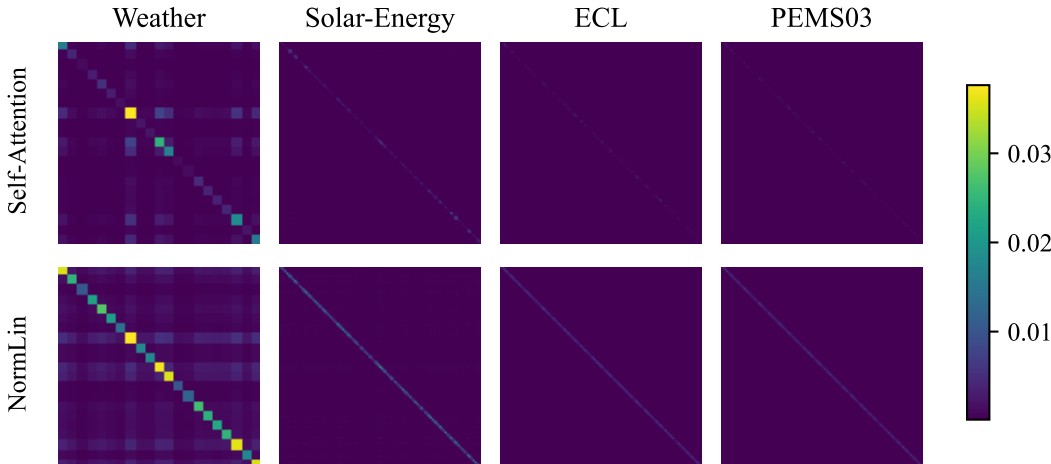

Figure 6: Comparison of Jacobian matrices of $\text{Softmax}(\cdot)$ in self-attention and $\text{Norm}_{\text{L1}}(\text{Softplus}(\cdot))$ in the NormLin layer. The absolute values of the matrix entries are visualized. NormLin and self-attention exhibit similar patterns; however, NormLin generally yields diagonal entries of higher magnitude, suggesting more effective gradient propagation. The same input vector is used for both to ensure a fair comparison.

where $\mathbf{1} \in \mathbb{R}^N$ is the all-ones vector.

For the derivative of $\mathbf{b}_i$ with respect to $\mathbf{a}_j$ with $1 \leq i, j \leq N$, we have

$$
\begin{aligned}
\frac{\partial \mathbf{b}_i}{\partial \mathbf{a}_j} &= \delta_{ij} \cdot \frac{\partial \mathbf{b}_i}{\partial \mathbf{a}_i} = \delta_{ij} \cdot \frac{\partial}{\partial \mathbf{a}_i} \ln(1 + e^{\mathbf{a}_i}) \\
&= \delta_{ij} \cdot \frac{e^{\mathbf{a}_i}}{1 + e^{\mathbf{a}_i}} = \delta_{ij} \cdot \text{Sigmoid}(\mathbf{a}_i).
\end{aligned}
\tag{19}
$$

Therefore, $\frac{\partial \mathbf{b}}{\partial \mathbf{a}}$ can be written as

$$
\frac{\partial \mathbf{b}}{\partial \mathbf{a}} = \text{Diag}\left(\text{Sigmoid}(\mathbf{a})\right) \triangleq \text{Diag}\left(\tilde{\mathbf{b}}\right),
\tag{20}
$$

where $\text{Sigmoid}(\cdot)$ operates element-wise, and $\tilde{\mathbf{b}} \triangleq \text{Sigmoid}(\mathbf{a})$ is defined for clarity. Using the chain rule, the Jacobian matrix of $\mathbf{c}$ with respect to $\mathbf{a}$ can be derived as follows:

$$
\begin{aligned}
\frac{\partial \mathbf{c}}{\partial \mathbf{a}} &= \frac{\partial \mathbf{c}}{\partial \mathbf{b}} \cdot \frac{\partial \mathbf{b}}{\partial \mathbf{a}} = \frac{1}{\|\mathbf{b}\|_1^2} \left(\|\mathbf{b}\|_1 \mathbf{I} - \mathbf{b}\mathbf{1}^\mathsf{T}\right) \text{Diag}\left(\tilde{\mathbf{b}}\right) \\
&= \frac{1}{\|\mathbf{b}\|_1^2} \left(\|\mathbf{b}\|_1 \text{Diag}\left(\tilde{\mathbf{b}}\right) - \mathbf{b}\tilde{\mathbf{b}}^\mathsf{T}\right) \\
&= \frac{1}{\|\mathbf{b}\|_1} \left(\text{Diag}\left(\tilde{\mathbf{b}}\right) - \bar{\mathbf{b}}\tilde{\mathbf{b}}^\mathsf{T}\right),
\end{aligned}
\tag{21}
$$

where we use the fact that $\mathbf{1}^\mathsf{T}\text{Diag}\left(\tilde{\mathbf{b}}\right) = \tilde{\mathbf{b}}^\mathsf{T}$, and $\bar{\mathbf{b}} = \frac{\mathbf{b}}{\|\mathbf{b}\|_1}$ is the normalized $\mathbf{b}$.

Equations (21) and (14) share similar structures, with the former introducing an additional learnable scaling factor $\frac{1}{\|\mathbf{b}\|_1}$, which offers greater flexibility. The entries of $\tilde{\mathbf{b}} = \text{Sigmoid}(\mathbf{a})$ in Equation (21) are generally larger than those of $\mathbf{c} = \text{Softmax}(\mathbf{a})$ in Equation (14), particularly when $\mathbf{a}$ contains small values near zero. This property may contribute to stronger gradients in NormLin. Figure 6 illustrates the Jacobian matrices of the self-attention mechanism and the NormLin layer under the same input $\mathbf{a}$, highlighting that NormLin tends to produce stronger gradient values.

## C Dataset description

In this work, the following real-world datasets are used for performance evaluation. The dataset details are presented in Table 11.

- **ETT** datasets [Zhou et al., 2021] record seven channels related to electricity transformers from July 2016 to July 2018. It contains four datasets: ETTh1 and ETTh2, with hourly recordings, and ETTm1 and ETTm2, with 15-minute recordings.
- **Weather** [Wu et al., 2021] contains 21 meteorological variables (e.g., air temperature, humidity) recorded every 10 minutes in 2020.
- **ECL** [Wu et al., 2021] records hourly electricity consumption of 321 consumers from July 2016 to July 2019.
- **Traffic** [Wu et al., 2021] includes hourly road occupancy rates from 862 sensors in the Bay Area from January 2015 to December 2016.
- **Exchange** [Wu et al., 2021] collects daily exchange rates for eight countries from January 1990 to October 2010.
- **Solar-Energy** [Lai et al., 2018] records the solar power output every 10 minutes from 137 photovoltaic plants in 2006.
- **PEMS** [Liu et al., 2022] provides public traffic sensor data from California, collected every 5 minutes. We use its four subsets (PEMS03, PEMS04, PEMS07, PEMS08) in this study.
- **ILI** [2] contains weekly records of influenza-like illness patient counts provided by the U.S. CDC from 2002 to 2021.
- **COVID-19** [Chen et al., 2022] includes daily records of COVID-19 hospitalizations in California in 2020, provided by Johns Hopkins University.
- **METR-LA** [3] collects traffic network data in Los Angeles every 5 minutes from March to June 2012. A total of 207 channels are included.
- **NASDAQ** [4] includes daily NASDAQ index and key economic indicators (e.g., interest rate and gold price) from 2010 to 2024.
- **Wiki** [5] records daily page view counts for Wikipedia articles over two years (2018–2019). The first 99 channels are used in this study.
- **SP500** records daily SP500 index data (e.g., opening price, closing price, and trading volume) from January 1993 to February 2025.
- **DowJones** collects daily stock prices of 27 Dow Jones Industrial Average (DJIA) component companies from January 1999 to March 2025.
- **CarSales** collects daily sales of 10 vehicle brands (e.g., Toyota, Honda) in the U.S. from January 2005 to June 2023. The data are compiled from the *Vehicles Sales* dataset [6] on Kaggle.
- **Power** contains daily wind and solar energy production (in MW) records for the French grid from April 2020 to June 2023. The data are compiled from the *Wind & Solar Daily Power Production* dataset [7] on Kaggle.
- **Website** [8] contains six years of daily visit data (e.g., first-time and returning visits) to an academic website, spanning from September 2014 to August 2020.
- **Unemp** contains monthly unemployment figures for 50 U.S. states and three other territories from January 1976 to March 2020, sourced from the official website of the U.S. Bureau of Labor Statistics[9].

---

[2] https://gis.cdc.gov/grasp/fluview/fluportaldashboard.html
[3] https://github.com/liyaguang/DCRNN
[4] https://www.kaggle.com/datasets/sai14karthik/nasdq-dataset
[5] https://www.kaggle.com/datasets/sandeshbhat/wikipedia-web-traffic-201819
[6] https://www.kaggle.com/datasets/crisbam/vehicles-sales/data
[7] https://www.kaggle.com/datasets/henriupton/wind-solar-electricity-production
[8] https://www.kaggle.com/datasets/bobnau/daily-website-visitors
[9] https://www.bls.gov/web/laus.supp.toc.htm

Table 11: Detailed dataset descriptions and statistics. *Dim* denotes the number of variates for each dataset. *Frequency* refers to the time interval between consecutive steps. *Split* indicates the (Train, Validation, Test) ratio. *Prediction len.* represents the prediction lengths. For long-term forecasting, the input length is fixed at 96. For short-term forecasting, we adopt two settings: S1 (Input-12, Predict-$\{3, 6, 9, 12\}$) and S2 (Input-36, Predict-$\{24, 36, 48, 60\}$). In total, 140 prediction tasks across various datasets and prediction length settings are evaluated in this work.

| Dataset | Dim | Frequency | Total len. | Split | Prediction len. | Information |
|---|---|---|---|---|---|---|
| ETTh1, ETTh2 | 7 | Hourly | 17420 | 6:2:2 | {96,192,336,720} | Electricity |
| ETTm1, ETTm2 | 7 | 15 mins | 69680 | 6:2:2 | {96,192,336,720} | Electricity |
| Weather | 21 | 10 mins | 52696 | 7:1:2 | {96,192,336,720} | Weather |
| ECL | 321 | Hourly | 26304 | 7:1:2 | {96,192,336,720} | Electricity |
| Traffic | 862 | Hourly | 17544 | 7:1:2 | {96,192,336,720} | Transportation |
| Exchange | 8 | Daily | 7588 | 7:1:2 | {96,192,336,720} | Economy |
| Solar-Energy | 137 | 10 mins | 52560 | 7:1:2 | {96,192,336,720} | Energy |
| PEMS03 | 358 | 5 mins | 26209 | 6:2:2 | {12,24,48,96} | Transportation |
| PEMS04 | 307 | 5 mins | 16992 | 6:2:2 | {12,24,48,96} | Transportation |
| PEMS07 | 883 | 5 mins | 28224 | 6:2:2 | {12,24,48,96} | Transportation |
| PEMS08 | 170 | 5 mins | 17856 | 6:2:2 | {12,24,48,96} | Transportation |
| ILI | 7 | Weekly | 966 | 7:1:2 | {3,6,9,12}
{24,36,48,60} | Health |
| COVID-19 | 55 | Daily | 335 | 7:1:2
6:2:2 | {3,6,9,12}
{24,36,48,60} | Health |
| METR-LA | 207 | 5 mins | 34272 | 7:1:2 | {3,6,9,12}
{24,36,48,60} | Transportation |
| NASDAQ | 12 | Daily | 3914 | 7:1:2 | {3,6,9,12}
{24,36,48,60} | Finance |
| Wiki | 99 | Daily | 730 | 7:1:2 | {3,6,9,12}
{24,36,48,60} | Web |
| SP500 | 5 | Daily | 8077 | 7:1:2 | {3,6,9,12}
{24,36,48,60} | Finance |
| DowJones | 27 | Daily | 6577 | 7:1:2 | {3,6,9,12}
{24,36,48,60} | Finance |
| CarSales | 10 | Daily | 6728 | 7:1:2 | {3,6,9,12}
{24,36,48,60} | Market |
| Power | 2 | Daily | 1186 | 7:1:2 | {3,6,9,12}
{24,36,48,60} | Energy |
| Website | 4 | Daily | 2167 | 7:1:2 | {3,6,9,12}
{24,36,48,60} | Web |
| Unemp | 53 | Monthly | 531 | 6:2:2 | {3,6,9,12}
{24,36,48,60} | Society |

# D Implementation details

OLinear is optimized using the ADAM optimizer [Kingma and Ba, 2015], with the initial learning rate selected from $\{10^{-4}, 2 \times 10^{-4}, 5 \times 10^{-4}\}$. The model dimension $D$ is chosen from $\{128, 256, 512\}$, while the embedding size $d$ is set to 16. The batch size is selected from $\{4, 8, 16, 32\}$ depending on the dataset scale. The block number $L$ is chosen from $\{1, 2, 3\}$. Training is performed for up to 50 epochs with early stopping, which halts training if the validation performance does not improve for 10 consecutive epochs. We adopt the weighted L1 loss function following CARD [Wang et al., 2024b]. The experiments are implemented in PyTorch [Paszke et al., 2019] and conducted on an NVIDIA GPU with 24 GB of memory. Hyperparameter sensitivity is discussed in Appendix I.8. For baseline models, we use the reported values from the original papers when available; otherwise, we produce the results using the official code. For the model FilterNet [Yi et al., 2024a], TexFilter is adopted in this work. The code and datasets are available at the following anonymous repository: `https://github.com/jackyue1994/OLinear`.

# E Showcases

## E.1 Time domain and transformed domain

Figure 7 illustrates the temporal domain and its corresponding transformed domain. In the transformed domain, the correlations along the sequence are effectively suppressed. Adjacent time series exhibit strong consistency in this new domain, which is desirable for forecasting tasks. The orthogonal transformation corresponds to projecting the time series onto the eigenvectors of the temporal Pearson correlation matrix $\mathrm{CorrMat}_t$. The eigenvalues of this symmetric positive semi-definite matrix $\mathrm{CorrMat}_t$ typically decay rapidly [Jolliffe, 2002], with only a few being dominant. Consequently, the transformed series exhibits sparsity, with most of the energy concentrated in just a few dimensions. Moreover, since noise tends to be evenly distributed across the transformed dimensions, the signal-to-noise ratio (SNR) in the leading components is improved [Bishop and Nasrabadi, 2006].

## E.2 Low-rank attention matrix and high-rank NormLin weight matrix

We replace the NormLin module in OLinear with a standard self-attention mechanism and observe the typical low-rank property in the resulting attention matrices, as shown in Figure 9. This phenomenon can be attributed to the sparsity induced by the transformed domain and the sharp focus introduced by the Softmax operation. The low-rank attention matrix could limit the expressive capacity of the model. In contrast, the weight matrices in NormLin exhibit higher rank, better preserving representation diversity.

Furthermore, the learned NormLin weights closely resemble the across-variate Pearson correlation matrix $\mathrm{CorrMat}_v$, suggesting that the NormLin layer effectively captures multivariate correlations. By directly replacing the learnable weight matrix with $\mathrm{Softmax}(\mathrm{CorrMat}_v)$, we obtain **OLinear-C**, which also achieves competitive performance, as demonstrated in Appendix H.

**Reason analysis** A potential reason behind this phenomenon is that OrthoTrans can mitigate the intrinsic low-rank property of time series data. The empirical results are presented below. Following TimeBase [Huang et al., 2025c], we compute the median singular values of the correlation matrix among non-overlapping time series patches (patch length = 24, number of patches = 30). As shown in Table 12, OrthoTrans yields larger singular values than other transformation bases (with "Identity" denoting no transformation). In other words, OrthoTrans alleviates low-rank tendencies of time series data and enhances data diversity.

## E.3 OrthoTrans enhances attention matrix rank

Figures 10–12 visualize the attention matrices of OLinear (with the NormLin module replaced by self-attention) under different transformation bases. Similar results for iTransformer and PatchTST are presented in Figures 13 and 14, respectively. As shown, OrthoTrans typically yields higher-rank attention matrices compared to DFT, wavelet transforms, or no transformation. As stated earlier,

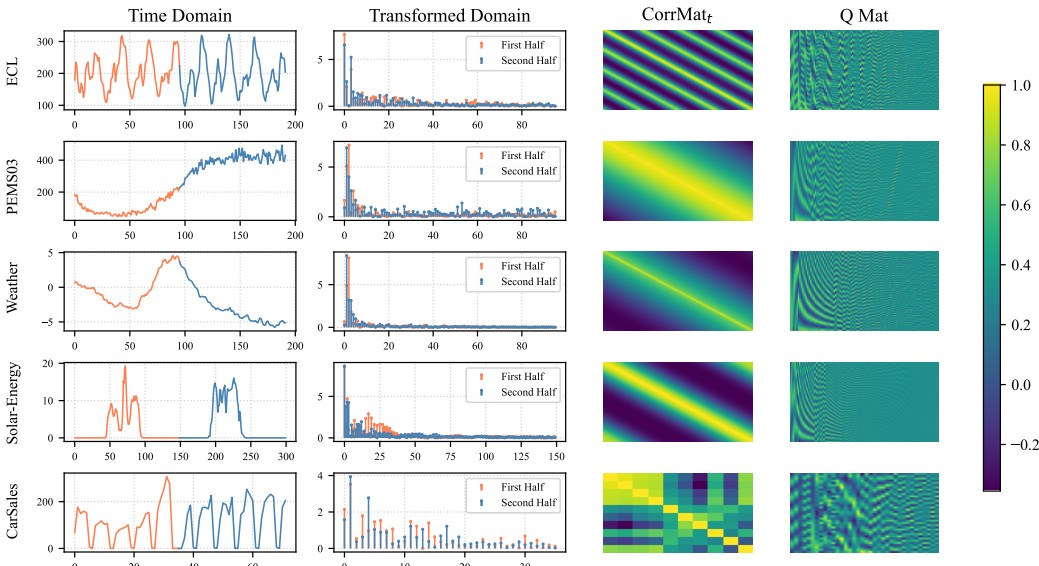

Figure 7: Visualization of temporal series, transformed series, the temporal Pearson correlation matrix, and its corresponding orthogonal matrix. The series are normalized prior to transformation, and the absolute values in the transformed domain are shown for clarity. The X-axis in the first and second columns represents the temporal and feature dimensions, respectively. Adjacent series exhibit consistency in the transformed domain, with sparsity clearly observable. Temporal Pearson correlation matrices on more datasets are presented in Figure 8.

higher-rank attention matrices better preserve the representation space and can potentially enhance the model's expressive capacity [Han et al., 2023]. This may explain why integrating OrthoTrans as a plug-in consistently improves the performance of Transformer-based forecasters (as shown in Table 5).

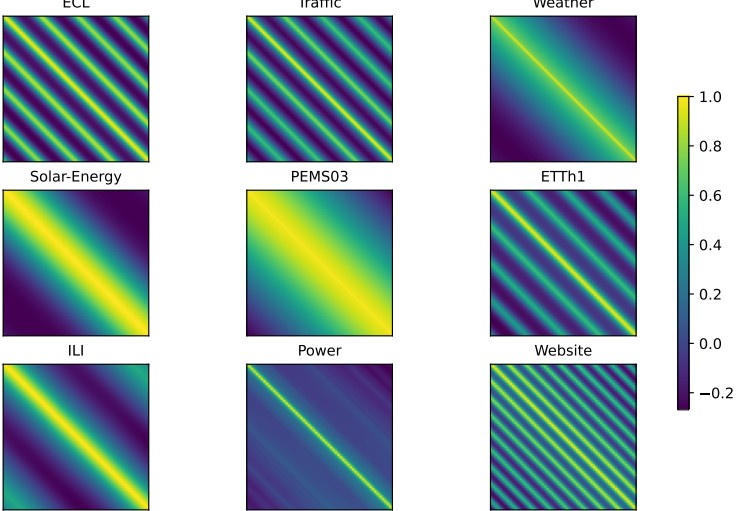

Figure 8: Temporal Pearson correlation matrices on various datasets. Periodic patterns are clearly reflected in the correlation matrices. We incorporate such *a priori* statistical information into the model design, leading to improved forecasting performance.

Table 12: OrthoTrans can effectively mitigate the low-rank characteristics of time series data. The median singular values of the correlation matrices among temporal patches are reported. The orthogonally transformed domain exhibits better data diversity.

| Dataset | OrthoTrans | DFT | Wavelet | Identity |
|---------|-----------|-----|---------|----------|
| ECL | **0.446** | 0.332 | 0.381 | 0.162 |
| Traffic | **0.568** | 0.177 | 0.049 | 0.002 |
| Weather | **0.572** | 0.352 | 0.292 | 0.012 |
| Solar | **0.481** | 0.032 | 0.023 | 0.007 |
| PEMS03 | **0.450** | 0.356 | 0.395 | 0.241 |

Table 13: Robustness of OLinear performance. Standard deviations are calculated over seven random seeds. *S1* and *S2* correspond to $\text{Input} - 12, \text{Predict} - \{3, 6, 9, 12\}$ and $\text{Input} - 36, \text{Predict} - \{24, 36, 48, 60\}$, respectively.

| Dataset | ECL | | Traffic | | ETTm1 | | Solar-Energy | |
|---------|-----|-----|---------|-----|-------|-----|--------------|-----|
| Metric | MSE | MAE | MSE | MAE | MSE | MAE | MSE | MAE |
| 96 | 0.131±4e-4 | 0.221±4e-4 | 0.398±3e-3 | 0.226±2e-4 | 0.302±7e-4 | 0.334±3e-4 | 0.179±6e-4 | 0.191±5e-4 |
| 192 | 0.150±1e-3 | 0.238±1e-3 | 0.439±3e-3 | 0.241±4e-4 | 0.357±9e-4 | 0.363±5e-4 | 0.209±8e-4 | 0.213±2e-4 |
| 336 | 0.165±1e-3 | 0.254±1e-3 | 0.464±4e-3 | 0.250±3e-4 | 0.387±2e-3 | 0.385±5e-4 | 0.231±8e-4 | 0.229±7e-5 |
| 720 | 0.191±2e-3 | 0.279±2e-3 | 0.502±4e-3 | 0.270±4e-4 | 0.452±1e-3 | 0.426±6e-4 | 0.241±1e-3 | 0.236±4e-4 |

| Dataset | Weather | | PEMS03 | | NASDAQ (S2) | | Wiki (S1) | |
|---------|---------|-----|--------|-----|-------------|-----|-----------|-----|
| Metric | MSE | MAE | MSE | MAE | MSE | MAE | MSE | MAE |
| H1 | 0.153±1e-3 | 0.190±1e-3 | 0.060±3e-4 | 0.159±3e-4 | 0.121±1e-3 | 0.216±9e-4 | 6.161±1e-2 | 0.368±7e-4 |
| H2 | 0.200±2e-3 | 0.235±2e-3 | 0.078±6e-4 | 0.179±5e-4 | 0.163±7e-4 | 0.261±9e-4 | 6.453±9e-3 | 0.385±1e-3 |
| H3 | 0.258±3e-3 | 0.280±2e-3 | 0.104±6e-4 | 0.210±6e-4 | 0.205±2e-3 | 0.296±2e-3 | 6.666±6e-3 | 0.398±1e-3 |
| H4 | 0.337±4e-3 | 0.333±2e-3 | 0.140±2e-3 | 0.247±1e-3 | 0.259±2e-3 | 0.336±2e-3 | 6.834±4e-3 | 0.406±4e-4 |

| Dataset | DowJones (S2) | | SP500 (S2) | | CarSales (S1) | | Power (S1) | |
|---------|---------------|-----|------------|-----|---------------|-----|------------|-----|
| Metric | MSE | MAE | MSE | MAE | MSE | MAE | MSE | MAE |
| H1 | 7.432±3e-2 | 0.664±9e-4 | 0.155±2e-3 | 0.271±2e-3 | 0.303±2e-3 | 0.277±1e-3 | 0.864±7e-3 | 0.688±4e-3 |
| H2 | 10.848±7e-2 | 0.799±1e-3 | 0.209±2e-3 | 0.317±2e-3 | 0.315±2e-3 | 0.285±2e-3 | 0.991±7e-3 | 0.742±3e-3 |
| H3 | 14.045±1e-1 | 0.914±1e-3 | 0.258±2e-3 | 0.358±1e-3 | 0.327±8e-4 | 0.293±8e-4 | 1.062±1e-2 | 0.770±4e-3 |
| H4 | 16.959±8e-2 | 1.017±3e-3 | 0.305±3e-3 | 0.387±2e-3 | 0.336±5e-4 | 0.301±4e-4 | 1.119±2e-2 | 0.789±6e-3 |

## F   Robustness under various random seeds

Table 13 presents the standard deviations for different datasets and prediction lengths using seven random seeds. OLinear demonstrates strong robustness across independent runs. Furthermore, as shown in Table 14, our model exhibits better robustness than state-of-the-art Transformer-based models, TimeMixer++ and iTransformer, as measured by the 99% confidence intervals.

Note that averaging over $\mathcal{M}$ prediction lengths reduces the standard deviation by a factor of $\frac{1}{\sqrt{\mathcal{M}}}$, making the results more robust to the choice of random seeds. Therefore, we prefer to report the average results in this work to mitigate the influence of randomness.

We perform a significance test with 7 random seeds using Student's t-test to compare OLinear and iTransformer across multiple datasets and prediction horizons. As shown in Table 15, OLinear consistently outperforms iTransformer, with the improvements being statistically significant at the 0.05 level ($p < 0.05$).

## G   Full results

Table 16 lists the simplified tables from the main text and their full versions in the appendix.

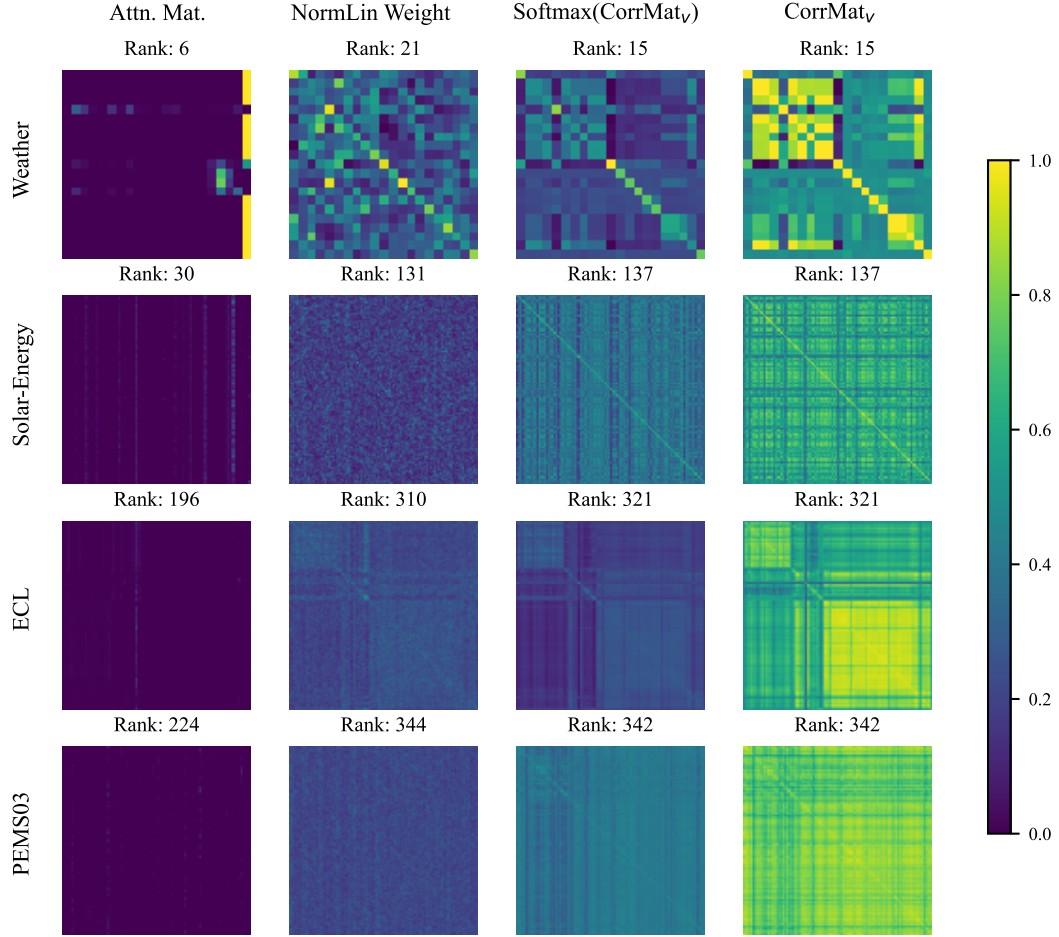

Figure 9: Illustration of attention/weight matrices from self-attention, NormLin, and the across-variate Pearson correlation matrices (with and without Softmax). When applied in OLinear, the self-attention mechanism results in low-rank attention matrices, whereas NormLin produces high-rank weight matrices that better preserve representation diversity. Moreover, the learned NormLin weights resemble the across-variate correlation matrices, suggesting effective modeling of multivariate relationships.

# H    OLinear-C

## H.1    Forecasting performance

Table 25 compares OLinear and OLinear-C on both short- and long-term forecasting tasks. OLinear-C achieves performance comparable to OLinear while using fewer learnable parameters and offering improved efficiency (see Table 41). However, due to its fixed weight matrix, $\mathrm{NormLin}_c$ lacks the flexibility required to serve as a plug-and-play module for other forecasters. Therefore, we focus on OLinear in this work.

## H.2    Robustness

Table 26 reports the standard deviation of OLinear-C across seven random seeds, demonstrating its robustness to independent runs. For a broader robustness comparison, Table 14 presents 99% confidence intervals for OLinear, OLinear-C, TimeMixer++, and iTransformer.

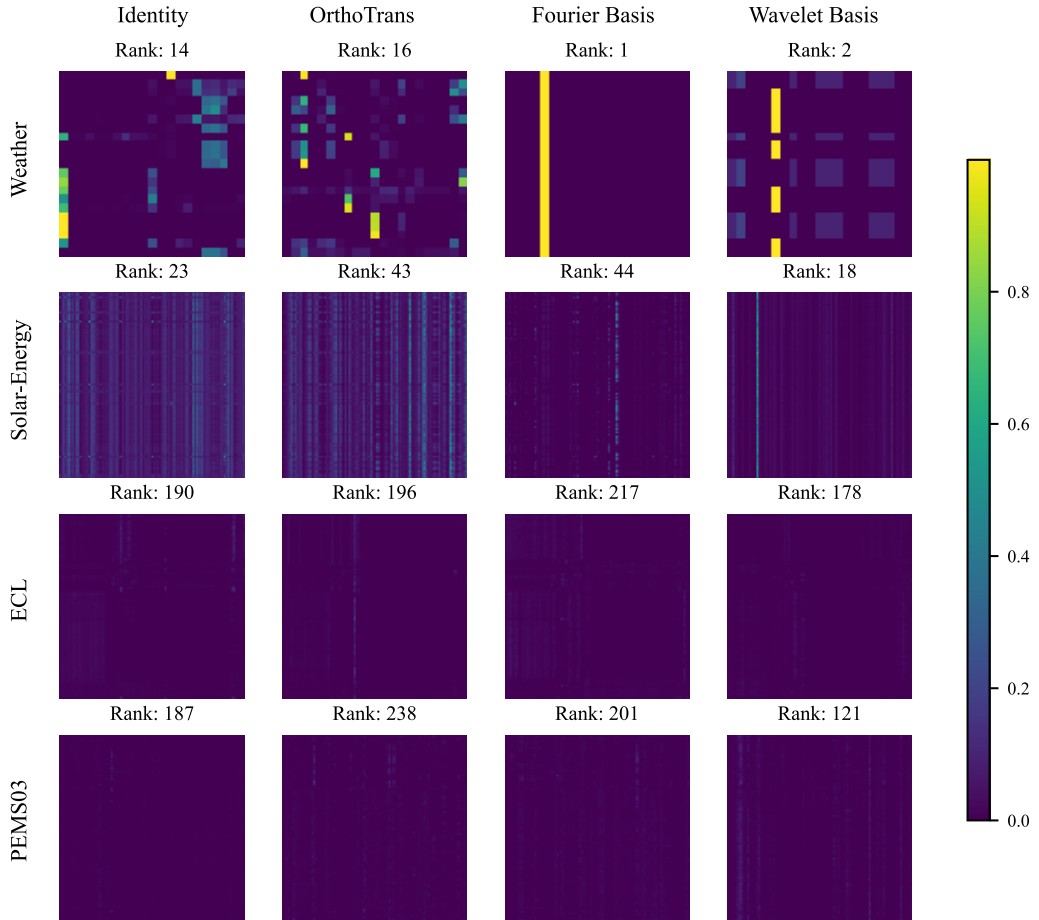

Figure 10: Comparison of attention matrices under various transformation bases for OLinear with the NormLin module replaced by self-attention (**Example 1**). For a fair comparison, the **same** input series is used for different bases on each dataset. 'Identity' denotes no transformation, and the Haar wavelet is used as the wavelet basis. In general, OrthoTrans yields higher-rank attention matrices and thus better preserves the representation space.

### H.3 Ablation studies

We conduct ablation studies on different transformations of the Pearson correlation matrix $\mathrm{CorrMat}_v$, including $\mathrm{Softplus}$, $\mathrm{Sigmoid}$, $\mathrm{ReLU}$, and the identity function (i.e., no transformation). All variants are followed by row-wise L1 normalization. As shown in Table 27, $\mathrm{Softmax}(\mathrm{CorrMat}_v)$ consistently outperforms the other options and is adopted in the main experiments.

## I More experiments

### I.1 Quantitation of temporal decorrelation of OrthoTrans

We evaluate the decorrelation effectiveness of OrthoTrans, DFT, and Haar wavelet transforms using a window size of 96. The metric is the off-diagonal Frobenius norm of the temporal correlation matrix, normalized by the number of off-diagonal elements. As shown in Table 28, OrthoTrans reduces the correlation in the original series by 96%, and achieves relative improvements of 84% and 93% over DFT and Haar wavelet transforms, respectively.

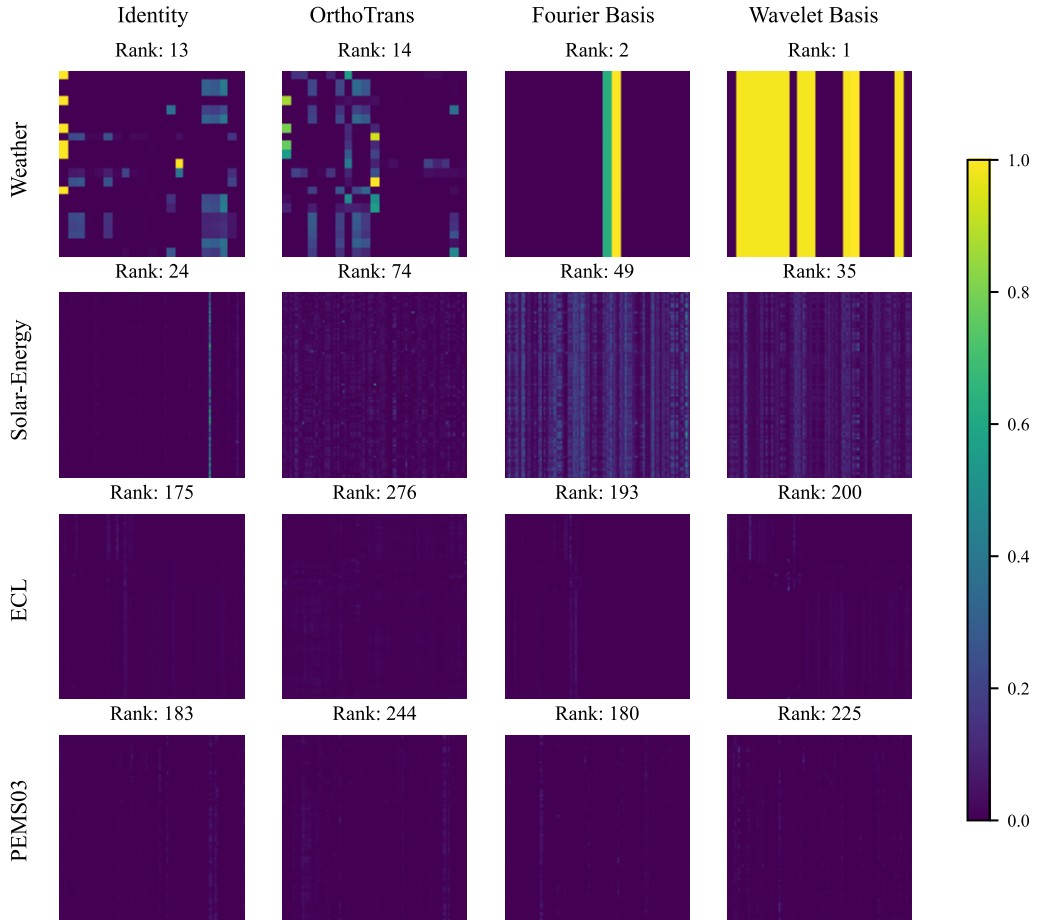

Figure 11: Comparison of attention matrices under various transformation bases for OLinear with the NormLin module replaced by self-attention (**Example 2**). For a fair comparison, the **same** input series is used across bases on each dataset. 'Identity' denotes no transformation, and the Haar wavelet is used as the wavelet basis. In general, OrthoTrans yields higher-rank attention matrices and thus better preserves the representation space.

## I.2 Performance under varying lookback horizons

We evaluate the forecasting performance of OLinear and its variant, OLinear-C, under varying lookback horizons. Instead of using a fixed lookback length, we search for the optimal horizon within the range of 96 to 720. As shown in Table 29, both models consistently achieve state-of-the-art results across different forecasting lengths and datasets, demonstrating their robustness and strong performance.

## I.3 More baselines

We further compare OLinear with several lightweight baselines, including **SimpleTM** [Chen et al., 2025], **TQNet** [Lin et al., 2025], **TimePro** [Ma et al., 2025g], and **TimeBase** [Huang et al., 2025c]. The lookback length is uniformly set to 96 for all methods. As shown in Table 30, OLinear achieves the lowest MAE across all datasets, outperforming SimpleTM, TQNet, TimePro, and TimeBase by 5%, 5%, 7%, and 16% on average, respectively. These results demonstrate that OLinear serves as a strong and efficient new baseline for lightweight time series forecasting.

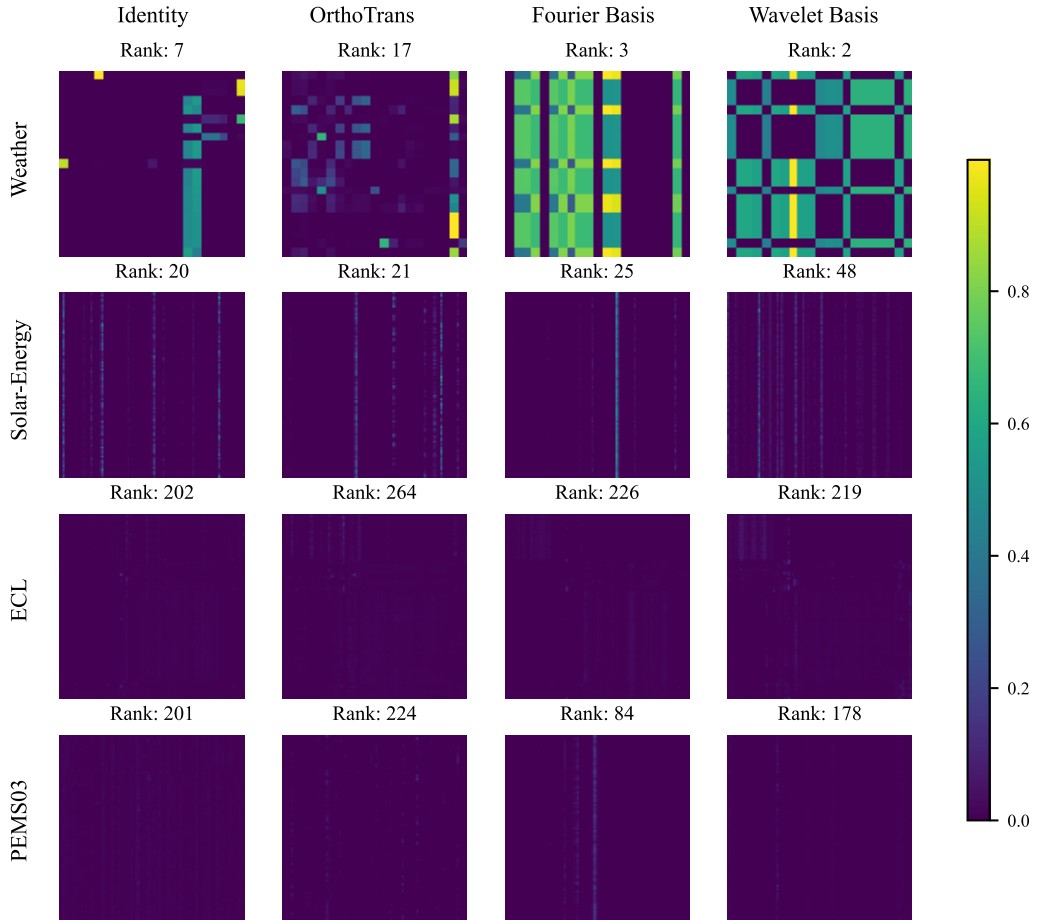

Figure 12: Comparison of attention matrices under various transformation bases for OLinear with the NormLin module replaced by self-attention (**Example 3**). For a fair comparison, the **same** input series is used across bases on each dataset. 'Identity' denotes no transformation, and the Haar wavelet is used as the wavelet basis. In general, OrthoTrans yields higher-rank attention matrices and thus better preserves the representation space.

## I.4   Few-shot and zero-shot forecasting

To evaluate OLinear's adaptability to sparse data and its transferability to unseen datasets, we conduct few-shot and zero-shot learning experiments, with the results shown in Tables 31 and 32, respectively. As a linear-based model, OLinear exhibits surprisingly strong generalization capabilities, a characteristic more commonly associated with Transformer-based models [Brown et al., 2020]. Notably, OLinear achieves this with high computational efficiency (see Table 41). Furthermore, OLinear delivers performance on par with or superior to TimeMixer++ [Wang et al., 2025a], despite the latter's architectural complexity, which includes multi-scale design, series imaging, and dual-axis attention mechanisms.

This performance advantage may be attributed to the high-rank property of the NormLin weight matrix (see Appendix E.2) and its enhanced gradient flow (see Appendix B). In contrast, attention matrices derived from the classic query-key mechanism typically exhibit low-rank characteristics (see Figures 10- 14), leading to a compressed representation space [Han et al., 2023].

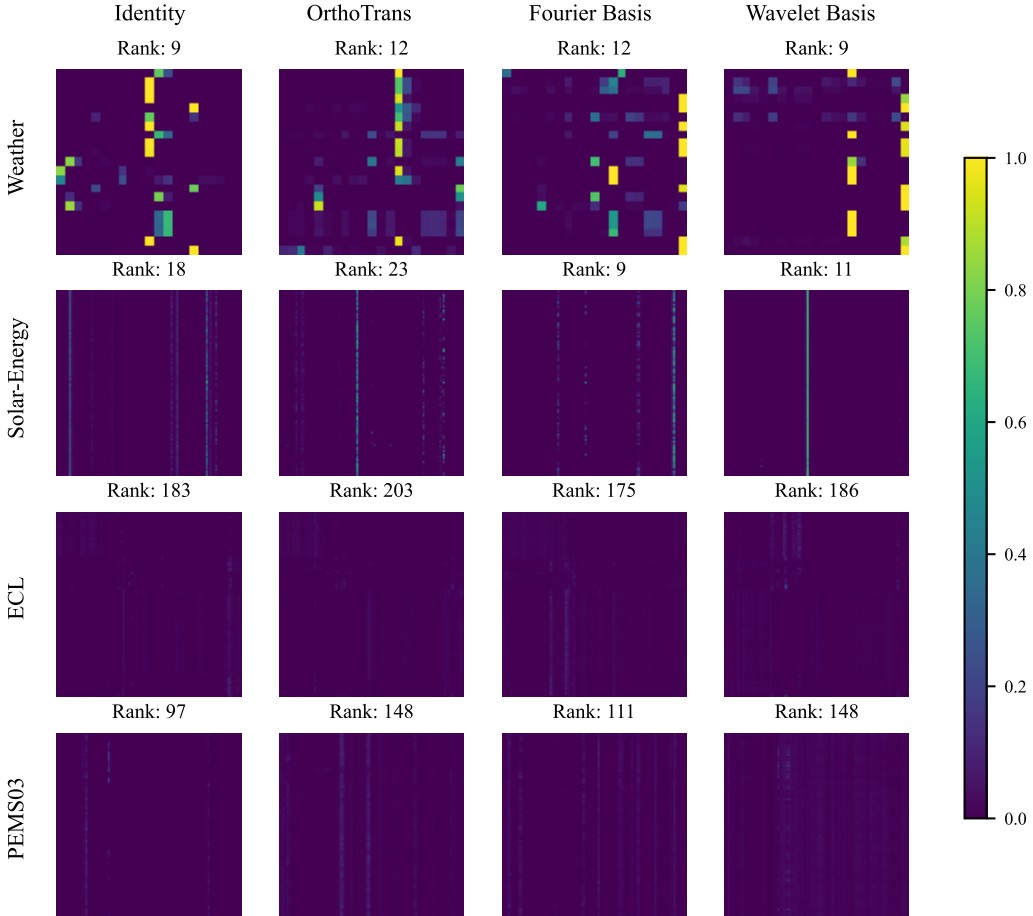

Figure 13: Comparison of attention matrices under various transformation bases for **iTransformer**. For a fair comparison, the **same** input series is used across bases on each dataset. 'Identity' denotes no transformation, and the Haar wavelet is used as the wavelet basis. In general, OrthoTrans yields higher-rank attention matrices and thus better preserves the representation space.

## I.5 Increasing lookback lengths

Performance under increasing lookback lengths reflects a model's ability to effectively utilize historical information [Liu et al., 2024a]. Linear-based forecasters have strong potential to benefit from longer lookback windows [Zeng et al., 2023]. As shown in Figure 15, OLinear demonstrates consistent improvements as the lookback horizon increases from 48 to 720, and consistently outperforms state-of-the-art forecasters.

## I.6 Training with less data

Training with less data reflects a model's adaptability and learning efficacy. As shown in Figure 16, OLinear adapts well to decreasing training set sizes, from 100% down to 5%. Our model consistently outperforms others across all training ratios, highlighting its robustness to data-sparse conditions.

## I.7 OLinear versus large time series models

We further compare OLinear with the large pre-trained time series models in the full-shot, few-shot and zero-shot scenarios. As shown in Table 33, OLinear surprisingly achieves comparable

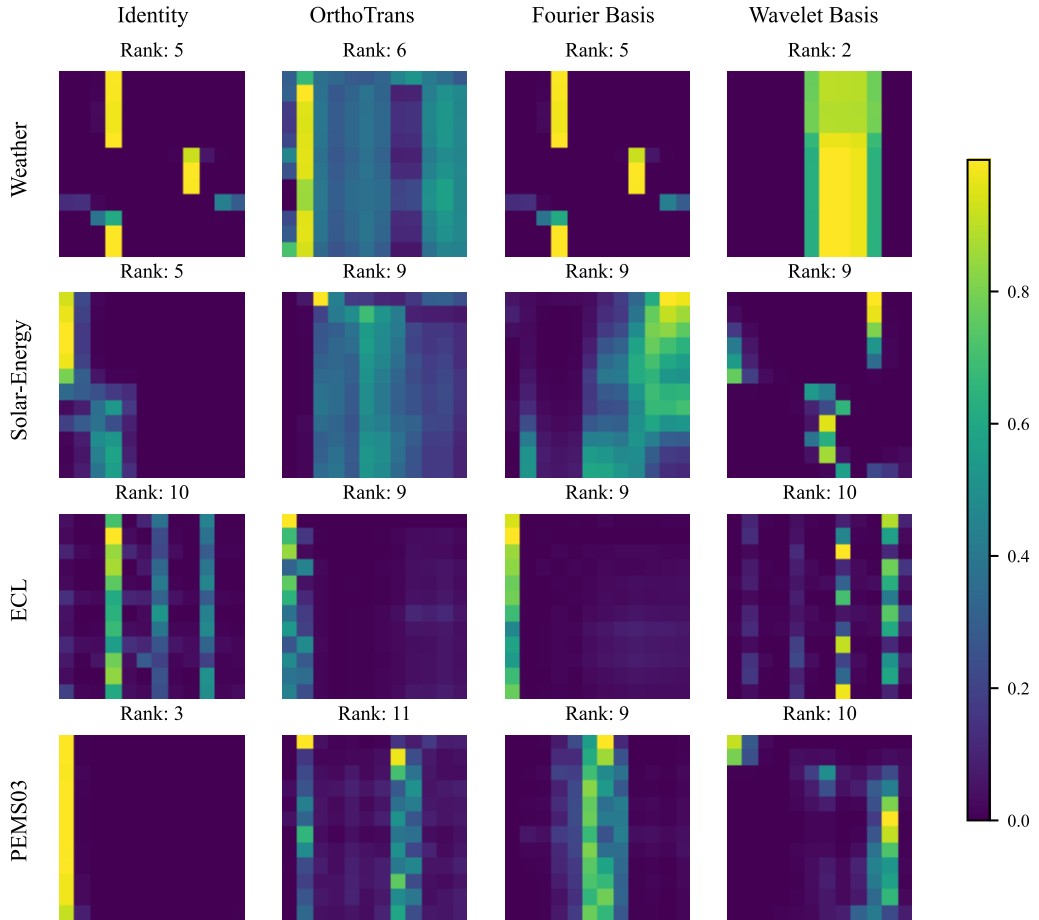

Figure 14: Comparison of attention matrices under various transformation bases for **PatchTST**. For a fair comparison, the **same** input series is used across all bases for each dataset. 'Identity' denotes no transformation, and the Haar wavelet is used as the wavelet basis. In general, OrthoTrans yields higher-rank attention matrices and thus better preserves the representation space.

performance to Timer [Liu et al., 2024b], despite not being pre-trained on large-scale datasets. When the full training set is used, OLinear outperforms the fine-tuned Timer on the ECL, Traffic, Weather, and PEMS datasets. Even when only 5% of training data are available, OLinear still performs better on the ETTm1 and ETTm2 datasets. In zero-shot forecasting, our model—trained on ETTh1 and ETTm1 respectively—outperforms state-of-the-art large time series models when evaluated on ETTh2 and ETTm2. These results not only demonstrate OLinear's strong generalization capability, but also imply that large time series models are still in the early stages of development.

## I.8 Hyperparameter sensitivity

We evaluate the hyperparameter sensitivity of OLinear with respect to four key factors: the learning rate ($lr$), the number of blocks ($L$), the model dimension ($D$), and the embedding size ($d$). As shown in Figure 17, OLinear generally maintains stable performance across different choices of $lr$, $L$, and $D$, although each dataset exhibits its own preferences. Regarding the embedding size $d$, Table 34 indicates that $d = 16$ represents a *sweet spot*, which we adopt as the default setting in our experiments.

Table 14: 99% confidence intervals of the average performance across four prediction lengths, computed using three times the standard deviation from seven random seeds. The smallest standard deviations are highlighted in **bold**. *OLinear-C* denotes the variant of OLinear where the weight matrix in NormLin is set to $\mathrm{Softmax}\left(\mathrm{CorrMat}_v\right)$, with $\mathrm{CorrMat}_v$ being the multivariate correlation matrix.

| Model | OLinear (Ours) | | OLinear-C (Ours) | | TimeMixer++ 2025a | | iTransformer 2024a | |
|---|---|---|---|---|---|---|---|---|
| Metric | MSE | MAE | MSE | MAE | MSE | MAE | MSE | MAE |
| Weather | 0.237±0.006 | 0.260±**0.003** | 0.238±**0.005** | 0.259±0.004 | 0.226±0.008 | 0.262±0.007 | 0.258±0.009 | 0.278±0.006 |
| Solar | 0.215±**0.001** | 0.217±0.001 | 0.215±**0.001** | 0.217±**3e-4** | 0.203±**0.001** | 0.238±0.010 | 0.233±0.009 | 0.262±0.007 |
| ECL | 0.159±**0.001** | 0.248±0.002 | 0.161±0.006 | 0.249±0.005 | 0.165±0.011 | 0.253±**0.001** | 0.178±0.002 | 0.270±0.005 |
| Traffic | 0.451±**0.003** | 0.247±**0.001** | 0.451±0.006 | 0.247±**0.001** | 0.416±0.015 | 0.264±0.013 | 0.428±0.008 | 0.282±0.002 |
| ETTh1 | 0.424±0.003 | 0.424±0.002 | 0.424±**0.002** | 0.424±**0.001** | 0.419±0.011 | 0.432±0.015 | 0.454±0.004 | 0.447±0.007 |
| ETTh2 | 0.367±**0.002** | 0.388±0.002 | 0.368±**0.002** | 0.389±**0.001** | 0.339±0.009 | 0.380±0.002 | 0.383±0.004 | 0.407±0.007 |
| ETTm1 | 0.374±**0.001** | 0.377±**0.001** | 0.375±**0.001** | 0.378±**0.001** | 0.369±0.005 | 0.378±0.007 | 0.407±0.004 | 0.410±0.009 |
| ETTm2 | 0.270±**3e-4** | 0.313±**2e-4** | 0.270±4e-4 | 0.313±**2e-4** | 0.269±0.002 | 0.320±0.012 | 0.288±0.010 | 0.332±0.003 |

Table 15: Significance test with Student's t-test and performance comparison between OLinear and iTransformer across various datasets and prediction horizons.

| Dataset & Horizon | OLinear (MAE) | iTrans.(MAE) | p-value between OLinear and iTrans. |
|---|---|---|---|
| ECL: 96 | **0.221±4e-4** | 0.240±4e-4 | 5.12E-11 |
| ECL: 192 | **0.238±1e-3** | 0.253±2e-3 | 3.06E-06 |
| ECL: 336 | **0.254±1e-3** | 0.269±1e-3 | 9.96E-07 |
| ECL: 720 | **0.279±2e-3** | 0.317±7e-3 | 5.59E-05 |
| Traffic: 96 | **0.226±2e-4** | 0.268±1e-3 | 1.61E-12 |
| Traffic: 192 | **0.241±4e-4** | 0.276±1e-3 | 9.48E-10 |
| Traffic: 336 | **0.250±3e-4** | 0.283±4e-4 | 1.87E-13 |
| Traffic: 720 | **0.270±4e-4** | 0.302±4e-4 | 6.15E-13 |
| Weather: 96 | **0.190±1e-3** | 0.214±3e-4 | 1.29E-08 |
| Weather: 192 | **0.235±2e-3** | 0.254±1e-3 | 1.04E-06 |
| Weather: 336 | **0.280±2e-3** | 0.296±1e-3 | 1.93E-06 |
| Weather: 720 | **0.333±2e-3** | 0.349±4e-4 | 5.68E-06 |

## I.9 More metrics

We further conduct a comprehensive evaluation of forecasting performance using three scale-free metrics: the Coefficient of Determination ($R^2$), the Pearson Correlation Coefficient ($r$), and the Mean Absolute Scaled Error (MASE), which are defined as follows:

$$
\begin{aligned}
R^2 &= \frac{1}{N}\sum_{n=1}^{N}\left(1 - \frac{\left|\hat{\mathbf{Y}}_{n:} - \mathbf{Y}_{n:}\right|_2^2}{\left|\mathbf{Y}_{n:} - \overline{\mathbf{Y}_{n:}}\right|_2^2}\right), \\
r &= \frac{1}{N}\sum_{n=1}^{N}\frac{\sum_{t=1}^{\tau}\left(\hat{\mathbf{Y}}_{n,t} - \overline{\hat{\mathbf{Y}}_{n:}}\right)\left(\mathbf{Y}_{n,t} - \overline{\mathbf{Y}_{n:}}\right)}{\sqrt{\sum_{t=1}^{\tau}\left(\hat{\mathbf{Y}}_{n,t} - \overline{\hat{\mathbf{Y}}_{n:}}\right)^2}\sqrt{\sum_{t=1}^{\tau}\left(\mathbf{Y}_{n,t} - \overline{\mathbf{Y}_{n:}}\right)^2}}, \\
\mathrm{MASE} &= \frac{1}{N}\sum_{n=1}^{N}\frac{\frac{1}{\tau}\left\|\hat{\mathbf{Y}}_{n:} - \mathbf{Y}_{n:}\right\|_1}{\frac{1}{\tau-1}\sum_{t=1}^{\tau-1}\left|\mathbf{Y}_{n,t} - \mathbf{Y}_{t-1,n}\right|}.
\end{aligned}
\tag{22}
$$

Table 16: Overview of tables in the main paper and their full versions in the appendix.

| Tables in paper | Tables in Appendix | Content |
|---|---|---|
| Table 2 | Table 17 | Long-term forecasting |
| Table 3 | Tables 18 and 19 | Short-term forecasting |
| Table 4 | Table 20 | Ablation studies on various bases |
| Table 5 | Table 21 | OrthoTrans as a plug-in |
| Table 7 | Table 25 | Performance of OLinear-C |
| Table 6 | Table 22 | Ablation studies of OLinear |
| Table 8 | Table 23 | NormLin vs self-attention and its variants |
| Table 9 | Table 24 | NormLin as a plug-in |

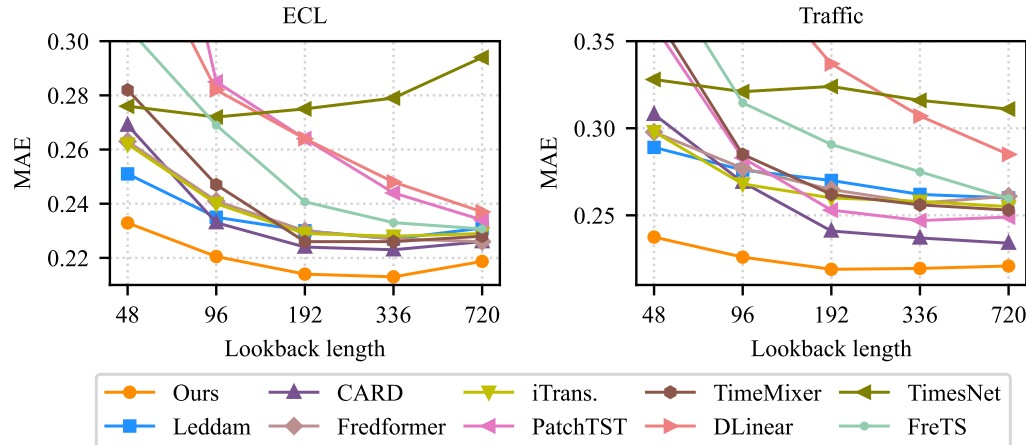

Figure 15: Performance change with increasing lookback lengths. The prediction length is $\tau = 96$.

Here $\mathbf{Y}_{n:}$ and $\hat{\mathbf{Y}}_{n:}$ denote the $n$-th variate of the ground truth $\mathbf{Y}$ and the prediction $\hat{\mathbf{Y}}$, respectively. $\mathbf{Y}_{n,t}$ refers to the $t$-th timestep of the $n$-th variate, and $\overline{(\cdot)}$ represents the mean.

As shown in Table 35, OLinear consistently achieves state-of-the-art performance on these metrics. For example, it an average Pearson correlation coefficient of $r = 0.903$ on the ECL dataset, indicating strong alignment between predictions and ground truth. On the Traffic dataset, OLinear achieves a MASE of 0.764, outperforming the second-best model, CARD [Wang et al., 2024b], by a notable margin of 11.4%.

### I.10  Less training data to compute Q matrices

To evaluate the amount of training data required for computing the orthogonal transformation matrices $\mathbf{Q}_i$ and $\mathbf{Q}_o$, we conduct experiments using only partial data of the training set. As shown in Table 36, OLinear exhibits strong robustness, with minimal performance variation across different data proportions. This property not only enables more efficient preprocessing before training, but also ensures robust performance in few-shot learning scenarios (see Appendix I.4).

## J  More ablation studies

### J.1  Ablations on NormLin design

In the NormLin layer, the Softplus function followed by L1 row-wise normalization is applied to the learnable weight matrix $\mathbf{W}$ to produce attention-like matrices, i.e., $\mathrm{RowNorm}_{\mathrm{L1}}(\mathrm{Softplus}(\mathbf{W}))$. To assess the effects of different transformations and normalization strategies, we consider transformation

Table 17: Full results for the long-term forecasting task. The lookback length $T$ is set to 96 for all baselines. *Avg* means the average results across four prediction lengths. This table presents the full version of Table 2.

| Category | | Linear-Based | | | | | | | | | | | | Transformer-Based | | | | | | | | | | TCN-Based | |
| Model / Metric | | OLinear (Ours) | | TimeMixer 2024a | | FilterNet 2024a | | FITS 2024 | | DLinear 2023 | | TimeMixer++ 2025a | | Leddam 2024 | | CARD 2024b | | Fredformer 2024 | | iTrans. 2024a | | PatchTST 2023 | | TimesNet 2023b | |
| Data | Len | MSE | MAE | MSE | MAE | MSE | MAE | MSE | MAE | MSE | MAE | MSE | MAE | MSE | MAE | MSE | MAE | MSE | MAE | MSE | MAE | MSE | MAE | MSE | MAE |
|---|---|---|---|---|---|---|---|---|---|---|---|---|---|---|---|---|---|---|---|---|---|---|---|---|---|
| ETTm1 | 96 | **0.302** | **0.334** | 0.320 | 0.357 | 0.321 | 0.361 | 0.353 | 0.375 | 0.345 | 0.372 | 0.310 | 0.334 | 0.319 | 0.359 | 0.316 | 0.347 | 0.326 | 0.361 | 0.334 | 0.368 | 0.329 | 0.367 | 0.338 | 0.375 |
| | 192 | 0.357 | 0.363 | 0.361 | 0.381 | 0.367 | 0.387 | 0.486 | 0.445 | 0.380 | 0.389 | 0.348 | 0.362 | 0.369 | 0.383 | 0.363 | 0.370 | 0.363 | 0.380 | 0.377 | 0.391 | 0.367 | 0.385 | 0.374 | 0.387 |
| | 336 | 0.387 | 0.385 | 0.390 | 0.404 | 0.401 | 0.409 | 0.531 | 0.475 | 0.413 | 0.413 | 0.376 | 0.391 | 0.394 | 0.402 | 0.392 | 0.390 | 0.395 | 0.403 | 0.426 | 0.420 | 0.399 | 0.410 | 0.410 | 0.411 |
| | 720 | 0.452 | 0.426 | 0.454 | 0.441 | 0.477 | 0.448 | 0.600 | 0.513 | 0.474 | 0.453 | 0.440 | 0.423 | 0.460 | 0.442 | 0.458 | 0.425 | 0.453 | 0.438 | 0.491 | 0.459 | 0.454 | 0.439 | 0.478 | 0.450 |
| | Avg | 0.374 | **0.377** | 0.381 | 0.395 | 0.392 | 0.401 | 0.493 | 0.452 | 0.403 | 0.407 | **0.369** | 0.378 | 0.386 | 0.397 | 0.383 | 0.384 | 0.384 | 0.395 | 0.407 | 0.410 | 0.387 | 0.400 | 0.400 | 0.406 |
| ETTm2 | 96 | **0.169** | 0.249 | 0.175 | 0.258 | 0.175 | 0.258 | 0.182 | 0.266 | 0.193 | 0.292 | 0.170 | 0.245 | 0.176 | 0.257 | 0.169 | 0.248 | 0.177 | 0.259 | 0.180 | 0.264 | 0.175 | 0.259 | 0.187 | 0.267 |
| | 192 | 0.232 | 0.290 | 0.237 | 0.299 | 0.240 | 0.301 | 0.253 | 0.312 | 0.284 | 0.362 | 0.229 | 0.291 | 0.243 | 0.303 | 0.234 | 0.292 | 0.243 | 0.301 | 0.250 | 0.309 | 0.241 | 0.302 | 0.249 | 0.309 |
| | 336 | 0.291 | 0.328 | 0.298 | 0.340 | 0.311 | 0.347 | 0.313 | 0.349 | 0.369 | 0.427 | 0.303 | 0.343 | 0.303 | 0.341 | 0.294 | 0.339 | 0.302 | 0.340 | 0.311 | 0.348 | 0.305 | 0.343 | 0.321 | 0.351 |
| | 720 | 0.389 | 0.387 | 0.391 | 0.396 | 0.414 | 0.405 | 0.416 | 0.406 | 0.554 | 0.522 | 0.373 | 0.399 | 0.400 | 0.398 | 0.390 | 0.388 | 0.397 | 0.396 | 0.412 | 0.407 | 0.402 | 0.400 | 0.408 | 0.403 |
| | Avg | 0.270 | **0.313** | 0.275 | 0.323 | 0.285 | 0.328 | 0.291 | 0.333 | 0.350 | 0.401 | **0.269** | 0.320 | 0.281 | 0.325 | 0.272 | 0.317 | 0.279 | 0.324 | 0.288 | 0.332 | 0.281 | 0.326 | 0.291 | 0.333 |
| ETTh1 | 96 | **0.360** | **0.382** | 0.375 | 0.400 | 0.382 | 0.402 | 0.385 | 0.394 | 0.386 | 0.400 | 0.361 | 0.403 | 0.377 | 0.394 | 0.383 | 0.391 | 0.373 | 0.392 | 0.386 | 0.405 | 0.414 | 0.419 | 0.384 | 0.402 |
| | 192 | 0.416 | 0.414 | 0.429 | 0.421 | 0.430 | 0.429 | 0.434 | 0.422 | 0.437 | 0.432 | 0.416 | 0.441 | 0.424 | 0.422 | 0.435 | 0.420 | 0.433 | 0.420 | 0.441 | 0.436 | 0.460 | 0.445 | 0.436 | 0.429 |
| | 336 | 0.457 | 0.438 | 0.484 | 0.458 | 0.472 | 0.451 | 0.476 | 0.444 | 0.481 | 0.459 | 0.430 | 0.434 | 0.459 | 0.442 | 0.479 | 0.442 | 0.470 | 0.437 | 0.487 | 0.458 | 0.501 | 0.466 | 0.491 | 0.469 |
| | 720 | 0.463 | 0.462 | 0.498 | 0.482 | 0.481 | 0.473 | 0.465 | 0.462 | 0.519 | 0.516 | 0.467 | 0.451 | 0.463 | 0.459 | 0.471 | 0.461 | 0.467 | 0.456 | 0.503 | 0.491 | 0.500 | 0.488 | 0.521 | 0.500 |
| | Avg | 0.424 | **0.424** | 0.447 | 0.440 | 0.441 | 0.439 | 0.440 | 0.431 | 0.456 | 0.452 | **0.419** | 0.432 | 0.431 | 0.429 | 0.442 | 0.429 | 0.435 | 0.426 | 0.454 | 0.447 | 0.469 | 0.454 | 0.458 | 0.450 |
| ETTh2 | 96 | 0.284 | 0.329 | 0.289 | 0.341 | 0.293 | 0.343 | 0.292 | 0.340 | 0.333 | 0.387 | **0.276** | **0.328** | 0.292 | 0.343 | 0.281 | 0.330 | 0.293 | 0.342 | 0.297 | 0.349 | 0.292 | 0.342 | 0.340 | 0.374 |
| | 192 | 0.360 | 0.379 | 0.372 | 0.392 | 0.374 | 0.396 | 0.377 | 0.391 | 0.477 | 0.476 | 0.342 | 0.379 | 0.367 | 0.389 | 0.363 | 0.381 | 0.371 | 0.389 | 0.380 | 0.400 | 0.387 | 0.400 | 0.402 | 0.414 |
| | 336 | 0.409 | 0.415 | 0.386 | 0.414 | 0.417 | 0.430 | 0.416 | 0.425 | 0.594 | 0.541 | 0.346 | 0.398 | 0.412 | 0.424 | 0.411 | 0.418 | 0.382 | 0.409 | 0.428 | 0.432 | 0.426 | 0.433 | 0.452 | 0.452 |
| | 720 | 0.415 | 0.431 | 0.412 | 0.434 | 0.449 | 0.460 | 0.418 | 0.437 | 0.831 | 0.657 | 0.392 | 0.415 | 0.419 | 0.438 | 0.416 | 0.431 | 0.415 | 0.434 | 0.427 | 0.445 | 0.431 | 0.446 | 0.462 | 0.468 |
| | Avg | 0.367 | 0.388 | 0.365 | 0.395 | 0.383 | 0.407 | 0.376 | 0.398 | 0.559 | 0.515 | **0.339** | **0.380** | 0.373 | 0.399 | 0.368 | 0.390 | 0.365 | 0.393 | 0.383 | 0.407 | 0.384 | 0.405 | 0.414 | 0.427 |
| ECL | 96 | **0.131** | **0.221** | 0.153 | 0.247 | 0.147 | 0.245 | 0.198 | 0.274 | 0.197 | 0.282 | 0.135 | 0.222 | 0.141 | 0.235 | 0.141 | 0.233 | 0.147 | 0.241 | 0.148 | 0.240 | 0.161 | 0.250 | 0.168 | 0.272 |
| | 192 | 0.150 | 0.238 | 0.166 | 0.256 | 0.160 | 0.250 | 0.363 | 0.422 | 0.196 | 0.285 | **0.147** | **0.235** | 0.159 | 0.252 | 0.160 | 0.250 | 0.165 | 0.258 | 0.162 | 0.253 | 0.199 | 0.289 | 0.184 | 0.289 |
| | 336 | 0.165 | 0.254 | 0.185 | 0.277 | 0.173 | 0.267 | 0.444 | 0.490 | 0.209 | 0.301 | 0.164 | 0.245 | 0.173 | 0.268 | 0.173 | 0.263 | 0.177 | 0.273 | 0.178 | 0.269 | 0.215 | 0.305 | 0.198 | 0.300 |
| | 720 | **0.191** | **0.279** | 0.225 | 0.310 | 0.210 | 0.309 | 0.532 | 0.551 | 0.245 | 0.333 | 0.212 | 0.310 | 0.201 | 0.295 | 0.197 | 0.284 | 0.213 | 0.304 | 0.225 | 0.317 | 0.256 | 0.337 | 0.220 | 0.320 |
| | Avg | **0.159** | **0.248** | 0.182 | 0.273 | 0.173 | 0.268 | 0.384 | 0.434 | 0.212 | 0.300 | 0.165 | 0.253 | 0.169 | 0.263 | 0.168 | 0.258 | 0.176 | 0.269 | 0.178 | 0.270 | 0.208 | 0.295 | 0.192 | 0.295 |
| Exchange | 96 | 0.082 | 0.200 | 0.086 | 0.205 | 0.091 | 0.211 | 0.087 | 0.208 | 0.088 | 0.218 | 0.085 | 0.214 | 0.086 | 0.207 | 0.084 | 0.202 | 0.084 | 0.202 | 0.086 | 0.206 | 0.088 | 0.205 | 0.107 | 0.234 |
| | 192 | **0.171** | **0.293** | 0.193 | 0.312 | 0.186 | 0.305 | 0.185 | 0.306 | 0.176 | 0.315 | 0.175 | 0.313 | 0.175 | 0.301 | 0.179 | 0.298 | 0.178 | 0.302 | 0.177 | 0.299 | 0.176 | 0.299 | 0.226 | 0.344 |
| | 336 | 0.331 | 0.414 | 0.356 | 0.433 | 0.380 | 0.449 | 0.342 | 0.425 | 0.313 | 0.427 | 0.316 | 0.420 | 0.325 | 0.415 | 0.333 | 0.418 | 0.319 | 0.408 | 0.331 | 0.417 | **0.301** | **0.397** | 0.367 | 0.448 |
| | 720 | 0.837 | 0.688 | 0.912 | 0.712 | 0.896 | 0.712 | 0.846 | 0.694 | 0.839 | 0.695 | 0.851 | 0.689 | 0.831 | 0.686 | 0.851 | 0.691 | 0.749 | 0.651 | 0.847 | 0.691 | 0.901 | 0.714 | 0.964 | 0.746 |
| | Avg | 0.355 | 0.399 | 0.387 | 0.416 | 0.388 | 0.419 | 0.365 | 0.408 | 0.354 | 0.414 | 0.357 | 0.409 | 0.354 | 0.402 | 0.362 | 0.402 | **0.333** | **0.391** | 0.360 | 0.403 | 0.367 | 0.404 | 0.416 | 0.443 |
| Traffic | 96 | 0.398 | **0.226** | 0.462 | 0.285 | 0.430 | 0.294 | 0.601 | 0.361 | 0.650 | 0.396 | **0.392** | 0.253 | 0.419 | 0.269 | 0.406 | 0.277 | 0.395 | 0.268 | | | 0.446 | 0.283 | 0.593 | 0.321 |
| | 192 | 0.439 | **0.241** | 0.473 | 0.296 | 0.452 | 0.307 | 0.603 | 0.365 | 0.598 | 0.370 | **0.402** | 0.258 | 0.458 | 0.289 | 0.443 | 0.276 | 0.426 | 0.290 | 0.417 | 0.276 | 0.540 | 0.354 | 0.617 | 0.336 |
| | 336 | 0.464 | **0.250** | 0.498 | 0.296 | 0.470 | 0.316 | 0.609 | 0.366 | 0.605 | 0.373 | **0.428** | 0.263 | 0.486 | 0.297 | 0.460 | 0.283 | 0.437 | 0.292 | 0.433 | 0.283 | 0.551 | 0.358 | 0.629 | 0.336 |
| | 720 | 0.502 | **0.270** | 0.506 | 0.313 | 0.498 | 0.323 | 0.648 | 0.387 | 0.645 | 0.394 | **0.441** | 0.282 | 0.498 | 0.313 | 0.490 | 0.299 | 0.462 | 0.305 | 0.467 | 0.302 | 0.586 | 0.375 | 0.640 | 0.350 |
| | Avg | 0.451 | **0.247** | 0.485 | 0.298 | 0.463 | 0.310 | 0.615 | 0.370 | 0.625 | 0.383 | **0.416** | 0.264 | 0.467 | 0.294 | 0.453 | 0.282 | 0.433 | 0.291 | 0.428 | 0.282 | 0.531 | 0.343 | 0.620 | 0.336 |
| Weather | 96 | 0.153 | 0.190 | 0.163 | 0.209 | 0.162 | 0.207 | 0.196 | 0.236 | 0.196 | 0.255 | 0.155 | 0.205 | 0.156 | 0.202 | 0.150 | **0.188** | 0.163 | 0.207 | 0.174 | 0.214 | 0.177 | 0.218 | 0.172 | 0.220 |
| | 192 | **0.200** | **0.235** | 0.208 | 0.250 | 0.210 | 0.250 | 0.240 | 0.271 | 0.237 | 0.296 | 0.201 | 0.245 | 0.207 | 0.250 | 0.202 | 0.238 | 0.211 | 0.251 | 0.221 | 0.254 | 0.225 | 0.259 | 0.219 | 0.261 |
| | 336 | 0.258 | 0.280 | 0.251 | 0.287 | 0.265 | 0.290 | 0.292 | 0.307 | 0.283 | 0.335 | **0.237** | **0.265** | 0.262 | 0.291 | 0.260 | 0.282 | 0.267 | 0.292 | 0.278 | 0.296 | 0.278 | 0.297 | 0.280 | 0.306 |
| | 720 | 0.337 | 0.333 | 0.339 | 0.341 | 0.342 | 0.340 | 0.365 | 0.354 | 0.345 | 0.381 | **0.312** | **0.334** | 0.343 | 0.343 | 0.343 | 0.353 | 0.343 | 0.341 | 0.358 | 0.349 | 0.354 | 0.348 | 0.365 | 0.359 |
| | Avg | 0.237 | 0.260 | 0.240 | 0.272 | 0.245 | 0.272 | 0.273 | 0.292 | 0.265 | 0.317 | **0.226** | **0.262** | 0.242 | 0.272 | 0.239 | 0.265 | 0.246 | 0.272 | 0.258 | 0.279 | 0.259 | 0.281 | 0.259 | 0.287 |
| Solar-Energy | 96 | 0.179 | **0.191** | 0.189 | 0.259 | 0.205 | 0.242 | 0.319 | 0.353 | 0.290 | 0.378 | **0.171** | 0.231 | 0.197 | 0.241 | 0.197 | 0.211 | 0.185 | 0.233 | 0.203 | 0.237 | 0.234 | 0.286 | 0.250 | 0.292 |
| | 192 | **0.209** | **0.213** | 0.222 | 0.283 | 0.233 | 0.265 | 0.367 | 0.387 | 0.320 | 0.398 | 0.218 | 0.263 | 0.231 | 0.264 | 0.234 | 0.234 | 0.227 | 0.253 | 0.233 | 0.261 | 0.267 | 0.310 | 0.296 | 0.318 |
| | 336 | 0.231 | 0.229 | 0.231 | 0.292 | 0.249 | 0.278 | 0.408 | 0.403 | 0.353 | 0.415 | **0.212** | 0.269 | 0.241 | 0.268 | 0.256 | 0.250 | 0.246 | 0.284 | 0.248 | 0.273 | 0.290 | 0.315 | 0.319 | 0.330 |
| | 720 | 0.241 | **0.236** | 0.223 | 0.285 | 0.253 | 0.281 | 0.411 | 0.395 | 0.356 | 0.413 | **0.212** | 0.270 | 0.250 | 0.281 | 0.260 | 0.254 | 0.247 | 0.276 | 0.249 | 0.275 | 0.289 | 0.317 | 0.338 | 0.337 |
| | Avg | 0.215 | **0.217** | 0.216 | 0.280 | 0.235 | 0.266 | 0.376 | 0.384 | 0.330 | 0.401 | **0.203** | 0.258 | 0.230 | 0.264 | 0.237 | 0.237 | 0.226 | 0.262 | 0.233 | 0.262 | 0.270 | 0.307 | 0.301 | 0.319 |
| PEMS03 | 12 | **0.060** | **0.159** | 0.076 | 0.188 | 0.071 | 0.177 | 0.117 | 0.226 | 0.122 | 0.243 | 0.097 | 0.208 | 0.063 | 0.164 | 0.072 | 0.177 | 0.068 | 0.174 | 0.071 | 0.174 | 0.099 | 0.216 | 0.085 | 0.192 |
| | 24 | **0.078** | **0.179** | 0.113 | 0.226 | 0.102 | 0.213 | 0.235 | 0.324 | 0.201 | 0.317 | 0.120 | 0.230 | 0.080 | 0.185 | 0.107 | 0.217 | 0.093 | 0.202 | 0.093 | 0.201 | 0.142 | 0.259 | 0.118 | 0.223 |
| | 48 | **0.104** | **0.210** | 0.191 | 0.292 | 0.162 | 0.272 | 0.541 | 0.521 | 0.333 | 0.425 | 0.170 | 0.272 | 0.124 | 0.226 | 0.194 | 0.302 | 0.146 | 0.258 | 0.125 | 0.236 | 0.211 | 0.319 | 0.155 | 0.260 |
| | 96 | **0.140** | **0.247** | 0.288 | 0.363 | 0.244 | 0.340 | 1.062 | 0.790 | 0.457 | 0.515 | 0.274 | 0.342 | 0.160 | 0.266 | 0.228 | 0.330 | 0.164 | 0.275 | | | 0.269 | 0.370 | 0.228 | 0.317 |
| | Avg | **0.095** | **0.199** | 0.167 | 0.267 | 0.145 | 0.251 | 0.489 | 0.465 | 0.278 | 0.375 | 0.165 | 0.263 | 0.107 | 0.210 | 0.174 | 0.275 | 0.135 | 0.243 | 0.113 | 0.221 | 0.180 | 0.291 | 0.147 | 0.248 |
| PEMS04 | 12 | **0.068** | **0.163** | 0.092 | 0.204 | 0.082 | 0.190 | 0.129 | 0.239 | 0.148 | 0.272 | 0.099 | 0.214 | 0.071 | 0.172 | 0.089 | 0.194 | 0.085 | 0.189 | 0.078 | 0.183 | 0.105 | 0.224 | 0.087 | 0.195 |
| | 24 | **0.079** | **0.176** | 0.128 | 0.243 | 0.110 | 0.224 | 0.246 | 0.337 | 0.224 | 0.340 | 0.115 | 0.231 | 0.087 | 0.193 | 0.128 | 0.234 | 0.117 | 0.224 | 0.095 | 0.205 | 0.153 | 0.275 | 0.103 | 0.215 |
| | 48 | **0.095** | **0.197** | 0.213 | 0.315 | 0.160 | 0.276 | 0.568 | 0.539 | 0.355 | 0.437 | 0.144 | 0.261 | 0.113 | 0.222 | 0.224 | 0.321 | 0.174 | 0.276 | 0.120 | 0.233 | 0.229 | 0.339 | 0.136 | 0.250 |
| | 96 | **0.122** | **0.226** | 0.307 | 0.384 | 0.234 | 0.343 | 1.181 | 0.843 | 0.452 | 0.504 | 0.185 | 0.297 | 0.141 | 0.252 | 0.382 | 0.445 | 0.273 | 0.354 | 0.150 | 0.262 | 0.291 | 0.389 | 0.190 | 0.303 |
| | Avg | **0.091** | **0.190** | 0.185 | 0.287 | 0.146 | 0.258 | 0.531 | 0.489 | 0.295 | 0.388 | 0.136 | 0.251 | 0.103 | 0.210 | 0.206 | 0.299 | 0.162 | 0.261 | 0.111 | 0.221 | 0.195 | 0.307 | 0.129 | 0.241 |
| PEMS07 | 12 | **0.052** | **0.138** | 0.073 | 0.184 | 0.064 | 0.163 | 0.109 | 0.222 | 0.115 | 0.242 | 0.090 | 0.197 | 0.055 | 0.145 | 0.068 | 0.166 | 0.063 | 0.158 | 0.067 | 0.165 | 0.095 | 0.207 | 0.082 | 0.181 |
| | 24 | **0.065** | **0.151** | 0.111 | 0.219 | 0.093 | 0.200 | 0.230 | 0.327 | 0.210 | 0.329 | 0.110 | 0.219 | 0.070 | 0.164 | 0.103 | 0.206 | 0.089 | 0.192 | 0.088 | 0.190 | 0.150 | 0.262 | 0.101 | 0.204 |
| | 48 | **0.084** | **0.171** | 0.237 | 0.328 | 0.137 | 0.248 | 0.551 | 0.531 | 0.398 | 0.458 | 0.149 | 0.256 | 0.094 | 0.192 | 0.165 | 0.268 | 0.136 | 0.241 | 0.110 | 0.215 | 0.253 | 0.340 | 0.134 | 0.238 |
| | 96 | **0.108** | **0.196** | 0.303 | 0.354 | 0.198 | 0.306 | 1.112 | 0.809 | 0.594 | 0.553 | 0.258 | 0.359 | 0.117 | 0.217 | 0.258 | 0.346 | 0.197 | 0.298 | 0.139 | 0.245 | 0.346 | 0.404 | 0.181 | 0.279 |
| | Avg | **0.077** | **0.164** | 0.181 | 0.271 | 0.123 | 0.229 | 0.500 | 0.472 | 0.329 | 0.395 | 0.152 | 0.258 | 0.084 | 0.180 | 0.149 | 0.247 | 0.121 | 0.222 | 0.101 | 0.204 | 0.211 | 0.303 | 0.124 | 0.225 |
| PEMS08 | 12 | **0.068** | **0.159** | 0.091 | 0.201 | 0.080 | 0.182 | 0.122 | 0.233 | 0.154 | 0.276 | 0.119 | 0.222 | 0.071 | 0.171 | 0.080 | 0.181 | 0.081 | 0.185 | 0.079 | 0.182 | 0.168 | 0.232 | 0.112 | 0.212 |
| | 24 | **0.089** | **0.178** | 0.137 | 0.246 | 0.114 | 0.219 | 0.236 | 0.330 | 0.248 | 0.353 | 0.149 | 0.249 | 0.091 | 0.189 | 0.118 | 0.220 | 0.112 | 0.214 | 0.115 | 0.219 | 0.224 | 0.281 | 0.141 | 0.238 |
| | 48 | **0.123** | **0.204** | 0.265 | 0.343 | 0.184 | 0.284 | 0.562 | 0.540 | 0.440 | 0.470 | 0.206 | 0.292 | 0.128 | 0.219 | 0.199 | 0.289 | 0.174 | 0.267 | 0.186 | 0.235 | 0.321 | 0.354 | 0.198 | 0.283 |
| | 96 | **0.173** | **0.236** | 0.410 | 0.407 | 0.309 | 0.356 | 1.216 | 0.846 | 0.674 | 0.565 | 0.329 | 0.355 | 0.198 | 0.266 | 0.405 | 0.431 | 0.277 | 0.335 | 0.221 | 0.267 | 0.408 | 0.417 | 0.320 | 0.351 |
| | Avg | **0.113** | **0.194** | 0.226 | 0.299 | 0.172 | 0.260 | 0.534 | 0.487 | 0.379 | 0.416 | 0.200 | 0.279 | 0.122 | 0.211 | 0.201 | 0.280 | 0.161 | 0.250 | 0.150 | 0.226 | 0.280 | 0.321 | 0.193 | 0.271 |
| 1st Count | | 33 | 49 | 0 | 0 | 0 | 0 | 0 | 0 | 0 | 0 | 29 | 14 | 1 | 0 | 2 | 1 | 2 | 2 | 0 | 0 | 1 | 1 | 0 | 0 |

Table 18: Full results for the short-term forecasting task (Part 1). For prediction lengths $\tau \in \{3, 6, 9, 12\}$, the lookback horizon is $T = 12$; for $\tau \in \{24, 36, 48, 60\}$, it is $T = 36$. The best results are highlighted in **bold**, and the second-best results are underlined. This table presents the detailed version of Table 3.

| | Model | OrthoLienar (Ours) | | TimeMix. 2024a | | FilterNet 2024a | | FITS 2024 | | DLinear 2023 | | TimeMix.++ 2025a | | Leddam 2024 | | CARD 2024b | | Fredformer 2024 | | iTrans. 2024a | | PatchTST 2023 | | TimesNet 2023b | |
|---|---|---|---|---|---|---|---|---|---|---|---|---|---|---|---|---|---|---|---|---|---|---|---|---|---|
| | Metric | MSE | MAE | MSE | MAE | MSE | MAE | MSE | MAE | MSE | MAE | MSE | MAE | MSE | MAE | MSE | MAE | MSE | MAE | MSE | MAE | MSE | MAE | MSE | MAE |
| ILI | 3 | **0.468** | **0.349** | 0.659 | 0.435 | 0.660 | 0.437 | 1.461 | 0.743 | 1.280 | 0.747 | 0.658 | 0.430 | 0.551 | _0.388_ | 0.597 | 0.392 | _0.528_ | 0.403 | 0.555 | 0.395 | 0.646 | 0.417 | 0.627 | 0.420 |
| | 6 | **0.923** | **0.516** | 1.306 | 0.643 | 1.140 | 0.606 | 2.337 | 0.974 | 2.054 | 0.967 | 1.273 | 0.645 | _1.021_ | _0.569_ | 1.246 | 0.610 | 1.128 | 0.604 | 1.124 | 0.586 | 1.269 | 0.629 | 1.147 | 0.610 |
| | 9 | **1.289** | **0.655** | 2.070 | 0.842 | 1.815 | 0.798 | 3.397 | 1.197 | 2.771 | 1.138 | 2.009 | 0.840 | 1.881 | 0.795 | 2.041 | 0.829 | 1.804 | 0.786 | _1.794_ | _0.772_ | 2.021 | 0.830 | 1.796 | 0.785 |
| | 12 | **1.698** | **0.791** | 2.792 | 1.018 | 2.435 | 0.953 | 4.244 | 1.361 | 3.497 | 1.284 | 2.683 | 1.004 | 2.421 | 0.964 | 2.746 | 0.996 | 2.610 | 0.989 | _2.273_ | _0.884_ | 2.788 | 1.016 | 2.349 | 0.920 |
| | Avg | **1.094** | **0.578** | 1.707 | 0.734 | 1.512 | 0.698 | 2.860 | 1.069 | 2.400 | 1.034 | 1.656 | 0.730 | 1.468 | 0.679 | 1.658 | 0.707 | 1.518 | 0.696 | _1.437_ | _0.659_ | 1.681 | 0.723 | 1.480 | 0.684 |
| | 24 | **1.737** | **0.800** | 2.110 | 0.879 | 2.190 | 0.870 | 4.265 | 1.523 | 3.158 | 1.243 | _1.877_ | _0.826_ | 2.085 | 0.883 | 2.407 | 0.970 | 2.098 | 0.894 | 2.004 | 0.860 | 2.046 | 0.849 | 2.317 | 0.934 |
| | 36 | _1.714_ | **0.795** | 2.084 | 0.890 | 1.902 | _0.862_ | 3.718 | 1.363 | 3.009 | 1.200 | 2.276 | 0.912 | 2.017 | 0.892 | 2.324 | 0.948 | **1.712** | 0.867 | 1.910 | 0.880 | 2.344 | 0.912 | 1.972 | 0.920 |
| | 48 | **1.821** | **0.804** | 1.961 | 0.866 | 2.051 | 0.882 | 3.994 | 1.422 | 2.994 | 1.194 | 1.921 | 0.850 | _1.860_ | _0.847_ | 2.133 | 0.911 | 2.054 | 0.922 | 2.036 | 0.891 | 2.123 | 0.883 | 2.238 | 0.940 |
| | 60 | _1.785_ | **0.810** | 1.926 | 0.878 | 2.151 | 0.925 | 4.543 | 1.554 | 3.172 | 1.232 | **1.745** | _0.838_ | 1.967 | 0.879 | 2.177 | 0.921 | 1.925 | 0.913 | 2.022 | 0.919 | 2.001 | 0.895 | 2.027 | 0.928 |
| | Avg | **1.764** | **0.802** | 2.020 | 0.878 | 2.073 | 0.885 | 4.130 | 1.465 | 3.083 | 1.217 | 1.955 | _0.857_ | 1.982 | 0.875 | 2.260 | 0.938 | _1.947_ | 0.899 | 1.993 | 0.887 | 2.128 | 0.885 | 2.139 | 0.931 |
| COVID-19 | 3 | **1.100** | **0.487** | 1.237 | 0.547 | 1.195 | 0.555 | 2.039 | 0.790 | 2.386 | 0.909 | 1.298 | 0.584 | 1.216 | 0.570 | _1.103_ | _0.521_ | 1.165 | 0.548 | 1.193 | 0.561 | 1.220 | 0.573 | 2.021 | 0.704 |
| | 6 | _1.750_ | **0.619** | 2.003 | 0.739 | 1.839 | 0.711 | 2.683 | 0.919 | 3.220 | 1.053 | 1.833 | _0.682_ | 1.782 | 0.689 | 1.919 | 0.735 | **1.465** | 0.685 | 1.933 | 0.755 | 1.982 | 0.762 | 2.405 | 0.808 |
| | 9 | _2.239_ | **0.734** | 2.594 | 0.860 | 2.537 | 0.897 | 3.147 | 1.050 | 3.803 | 1.160 | 2.472 | _0.822_ | 2.407 | 0.866 | 2.358 | 0.841 | **2.145** | 0.845 | 2.441 | 0.879 | 2.633 | 0.916 | 2.858 | 0.969 |
| | 12 | **2.538** | **0.831** | 3.103 | 0.981 | _2.782_ | 0.956 | 3.630 | 1.156 | 4.524 | 1.288 | 3.273 | 1.084 | 2.851 | 0.991 | 2.857 | 0.971 | 2.833 | 0.984 | 2.819 | 0.984 | 3.050 | 1.030 | 2.993 | _0.964_ |
| | Avg | _1.907_ | **0.668** | 2.234 | 0.782 | 2.088 | 0.780 | 2.875 | 0.979 | 3.483 | 1.102 | 2.219 | 0.793 | 2.064 | 0.779 | 2.059 | 0.767 | **1.902** | _0.765_ | 2.096 | 0.795 | 2.221 | 0.820 | 2.569 | 0.861 |
| | 24 | **4.474** | **1.180** | 6.335 | 1.554 | 5.926 | 1.517 | 8.248 | 1.829 | 9.780 | 1.851 | 6.539 | 1.618 | 4.860 | 1.342 | 5.133 | 1.394 | 4.799 | 1.347 | _4.715_ | _1.321_ | 5.528 | 1.450 | 5.634 | 1.442 |
| | 36 | **7.241** | **1.670** | 8.222 | 1.787 | 7.696 | 1.733 | 10.345 | 2.050 | 12.804 | 2.083 | 7.986 | 1.770 | 7.378 | 1.708 | 7.377 | 1.725 | 7.536 | 1.727 | _7.299_ | _1.681_ | 8.351 | 1.830 | 9.114 | 1.848 |
| | 48 | 10.076 | _1.985_ | 11.669 | 2.157 | 11.572 | 2.141 | 11.999 | 2.197 | 14.244 | 2.189 | 11.655 | 2.156 | _10.051_ | 1.999 | 11.013 | 2.103 | **9.833** | **1.951** | 10.141 | 2.012 | 11.259 | 2.114 | 10.940 | 2.033 |
| | 60 | 12.079 | 2.182 | 12.188 | 2.173 | **11.311** | **2.066** | 13.185 | 2.282 | 15.472 | 2.275 | 12.734 | 2.235 | _11.467_ | _2.119_ | 12.528 | 2.227 | 12.455 | 2.209 | 11.871 | 2.156 | 12.666 | 2.225 | 12.888 | 2.186 |
| | Avg | _8.467_ | **1.754** | 9.604 | 1.918 | 9.126 | 1.864 | 10.944 | 2.089 | 13.075 | 2.099 | 9.728 | 1.945 | **8.439** | _1.792_ | 9.013 | 1.862 | 8.656 | 1.808 | 8.506 | _1.792_ | 9.451 | 1.905 | 9.644 | 1.877 |
| METR-LA | 3 | 0.207 | **0.171** | _0.205_ | 0.192 | _0.204_ | 0.189 | 0.211 | 0.198 | 0.218 | 0.231 | **0.204** | 0.189 | **0.204** | 0.191 | 0.210 | _0.180_ | _0.205_ | 0.188 | _0.205_ | 0.188 | **0.204** | 0.190 | 0.221 | 0.204 |
| | 6 | 0.301 | **0.207** | 0.297 | 0.230 | _0.296_ | 0.228 | 0.306 | 0.237 | 0.307 | 0.278 | 0.298 | 0.230 | **0.293** | 0.227 | 0.311 | _0.219_ | 0.298 | 0.227 | 0.300 | 0.229 | 0.298 | 0.227 | 0.308 | 0.238 |
| | 9 | 0.382 | **0.238** | 0.381 | 0.264 | _0.377_ | 0.261 | 0.392 | 0.271 | 0.386 | 0.316 | 0.384 | 0.266 | **0.369** | 0.264 | 0.401 | _0.253_ | 0.385 | 0.263 | 0.386 | 0.265 | 0.382 | 0.263 | 0.387 | 0.273 |
| | 12 | 0.452 | **0.263** | 0.455 | 0.295 | _0.449_ | 0.290 | 0.467 | 0.300 | 0.452 | 0.353 | 0.456 | 0.294 | **0.442** | 0.292 | 0.474 | _0.281_ | 0.457 | 0.292 | 0.460 | 0.295 | 0.456 | 0.292 | 0.462 | 0.298 |
| | Avg | 0.335 | **0.220** | 0.334 | 0.245 | _0.331_ | 0.242 | 0.344 | 0.251 | 0.341 | 0.294 | 0.335 | 0.245 | **0.327** | 0.243 | 0.349 | _0.233_ | 0.336 | 0.242 | 0.338 | 0.244 | 0.335 | 0.243 | 0.344 | 0.253 |
| | 24 | 0.650 | **0.337** | 0.671 | 0.413 | 0.670 | 0.402 | 0.698 | 0.416 | _0.645_ | 0.458 | **0.617** | 0.394 | 0.680 | 0.405 | 0.700 | _0.378_ | 0.676 | 0.408 | 0.700 | 0.413 | 0.679 | 0.410 | 0.698 | 0.415 |
| | 36 | 0.800 | **0.388** | 0.841 | 0.480 | 0.824 | 0.471 | 0.874 | 0.490 | _0.785_ | 0.533 | **0.781** | 0.457 | 0.781 | 0.471 | 0.852 | _0.448_ | 0.852 | 0.477 | 0.867 | 0.480 | 0.845 | 0.484 | 0.856 | 0.475 |
| | 48 | 0.905 | **0.427** | 0.964 | 0.531 | 0.955 | 0.521 | 1.013 | 0.546 | _0.885_ | 0.585 | **0.842** | 0.520 | 0.963 | 0.528 | 1.017 | _0.498_ | 0.982 | 0.526 | 1.017 | 0.539 | 0.972 | 0.536 | 0.972 | 0.518 |
| | 60 | 0.999 | **0.457** | 1.047 | 0.573 | 1.050 | 0.563 | 1.122 | 0.589 | _0.959_ | 0.623 | **0.958** | 0.551 | 1.029 | 0.556 | 1.126 | _0.541_ | 1.084 | 0.569 | 1.079 | 0.572 | 1.077 | 0.578 | 1.033 | 0.543 |
| | Avg | 0.838 | **0.402** | 0.881 | 0.499 | 0.875 | 0.489 | 0.927 | 0.510 | _0.819_ | 0.550 | **0.799** | 0.480 | 0.878 | 0.490 | 0.929 | _0.466_ | 0.898 | 0.495 | 0.916 | 0.501 | 0.893 | 0.502 | 0.890 | 0.488 |
| NASDAQ | 3 | _0.036_ | **0.092** | **0.035** | _0.093_ | 0.038 | 0.100 | 0.042 | 0.111 | 0.044 | 0.123 | 0.036 | 0.098 | 0.040 | 0.103 | 0.037 | 0.096 | 0.039 | 0.102 | 0.040 | 0.105 | 0.038 | 0.099 | 0.049 | 0.123 |
| | 6 | **0.049** | **0.117** | **0.049** | _0.118_ | 0.052 | 0.126 | 0.056 | 0.134 | 0.062 | 0.155 | 0.050 | 0.121 | 0.054 | 0.128 | _0.052_ | 0.123 | 0.053 | 0.126 | 0.054 | 0.129 | 0.053 | 0.124 | 0.061 | 0.142 |
| | 9 | **0.062** | **0.137** | _0.062_ | _0.139_ | 0.065 | 0.145 | 0.069 | 0.154 | 0.082 | 0.189 | 0.063 | 0.144 | 0.066 | 0.147 | _0.063_ | 0.141 | 0.067 | 0.147 | 0.068 | 0.150 | 0.065 | 0.145 | 0.073 | 0.161 |
| | 12 | **0.073** | **0.154** | _0.073_ | _0.156_ | 0.076 | 0.162 | 0.081 | 0.172 | 0.100 | 0.215 | 0.075 | 0.161 | 0.078 | 0.164 | _0.075_ | 0.158 | 0.079 | 0.165 | 0.078 | 0.165 | 0.077 | 0.161 | 0.088 | 0.179 |
| | Avg | **0.055** | **0.125** | **0.055** | _0.126_ | 0.058 | 0.133 | 0.062 | 0.142 | 0.072 | 0.170 | 0.056 | 0.131 | 0.059 | 0.135 | _0.057_ | 0.130 | 0.059 | 0.135 | 0.060 | 0.137 | 0.058 | 0.132 | 0.068 | 0.151 |
| | 24 | **0.121** | **0.216** | _0.122_ | 0.221 | 0.130 | 0.230 | 0.140 | 0.244 | 0.155 | 0.274 | 0.132 | 0.233 | 0.125 | 0.222 | 0.124 | _0.220_ | 0.128 | 0.226 | 0.137 | 0.237 | 0.127 | 0.224 | 0.198 | 0.299 |
| | 36 | **0.163** | **0.261** | 0.183 | 0.279 | 0.175 | 0.273 | 0.184 | 0.284 | 0.196 | 0.306 | 0.177 | 0.278 | 0.174 | 0.271 | _0.167_ | _0.266_ | 0.170 | 0.268 | 0.184 | 0.280 | 0.174 | 0.269 | 0.229 | 0.326 |
| | 48 | _0.205_ | **0.296** | **0.200** | _0.298_ | 0.224 | 0.314 | 0.234 | 0.324 | 0.244 | 0.344 | 0.216 | 0.311 | 0.222 | 0.312 | 0.218 | 0.307 | 0.218 | 0.306 | 0.229 | 0.318 | 0.225 | 0.314 | 0.267 | 0.352 |
| | 60 | 0.259 | **0.336** | **0.238** | **0.328** | 0.259 | 0.340 | 0.282 | 0.357 | 0.318 | 0.401 | _0.249_ | 0.337 | 0.264 | 0.341 | 0.264 | 0.341 | 0.262 | 0.339 | 0.279 | 0.352 | 0.265 | 0.339 | 0.327 | 0.394 |
| | Avg | _0.187_ | **0.277** | **0.186** | _0.281_ | 0.197 | 0.289 | 0.210 | 0.302 | 0.228 | 0.331 | 0.193 | 0.290 | 0.196 | 0.286 | 0.193 | 0.284 | 0.194 | 0.285 | 0.207 | 0.297 | 0.198 | 0.286 | 0.255 | 0.343 |
| Wiki | 3 | 6.161 | **0.368** | 6.209 | 0.392 | 6.234 | 0.402 | 7.470 | 0.496 | 6.254 | 0.438 | 6.149 | 0.389 | _6.148_ | 0.383 | 6.183 | _0.378_ | 6.190 | 0.387 | 6.237 | 0.393 | **6.112** | 0.380 | 7.597 | 0.510 |
| | 6 | 6.453 | **0.385** | 6.475 | 0.402 | 6.460 | 0.401 | 8.326 | 0.544 | 6.579 | 0.467 | _6.436_ | 0.401 | 6.455 | 0.397 | 6.465 | _0.393_ | 6.696 | 0.404 | 6.484 | 0.400 | **6.425** | 0.395 | 7.962 | 0.515 |
| | 9 | **6.666** | **0.398** | 6.702 | 0.418 | 6.697 | 0.416 | 8.869 | 0.564 | 6.776 | 0.508 | 6.714 | 0.420 | _6.687_ | 0.412 | 6.714 | 0.415 | 6.768 | _0.411_ | 6.689 | _0.411_ | 6.743 | 0.426 | 8.150 | 0.524 |
| | 12 | _6.834_ | **0.406** | 6.902 | 0.426 | 6.899 | 0.426 | 9.394 | 0.608 | 6.927 | 0.513 | 6.852 | 0.421 | 6.899 | 0.424 | 6.852 | 0.415 | 7.168 | 0.424 | 6.868 | 0.419 | **6.814** | _0.414_ | 8.117 | 0.533 |
| | Avg | _6.528_ | **0.389** | 6.572 | 0.409 | 6.572 | 0.411 | 8.515 | 0.553 | 6.634 | 0.481 | 6.538 | 0.408 | 6.547 | 0.404 | 6.553 | _0.400_ | 6.705 | 0.406 | 6.569 | 0.405 | **6.523** | 0.404 | 7.956 | 0.520 |
| | 24 | 6.894 | **0.423** | 6.900 | 0.446 | 6.907 | 0.443 | 7.887 | 0.613 | 6.883 | 0.520 | 6.902 | 0.460 | 6.919 | 0.450 | 6.925 | 0.440 | **6.531** | 0.432 | 6.886 | 0.437 | _6.858_ | _0.430_ | 8.023 | 0.612 |
| | 36 | 6.446 | **0.439** | 6.520 | 0.467 | 6.514 | 0.467 | 7.774 | 0.660 | _6.393_ | 0.538 | 6.539 | 0.473 | 6.456 | 0.457 | 6.463 | 0.451 | **5.935** | 0.453 | 6.431 | 0.452 | 6.400 | _0.445_ | 7.229 | 0.595 |
| | 48 | 6.004 | **0.446** | 6.108 | 0.484 | 6.135 | 0.478 | 7.737 | 0.706 | _5.940_ | 0.547 | 6.115 | 0.487 | 6.031 | 0.468 | 6.031 | 0.460 | **5.871** | 0.464 | 6.101 | 0.483 | 5.959 | _0.449_ | 7.184 | 0.641 |
| | 60 | 5.705 | _0.454_ | 5.732 | 0.476 | 5.811 | 0.482 | 7.855 | 0.746 | _5.605_ | 0.552 | 5.736 | 0.497 | 5.740 | 0.478 | 5.723 | 0.463 | **5.389** | 0.463 | 5.681 | 0.462 | 5.633 | **0.452** | 6.805 | 0.645 |
| | Avg | 6.262 | **0.440** | 6.315 | 0.468 | 6.342 | 0.467 | 7.813 | 0.681 | _6.205_ | 0.539 | 6.323 | 0.479 | 6.286 | 0.463 | 6.285 | 0.453 | **5.931** | 0.453 | 6.275 | 0.458 | 6.212 | _0.444_ | 7.310 | 0.623 |
| | 1st Count | 19 | 47 | 8 | 1 | 2 | 1 | 0 | 0 | 0 | 0 | 7 | 0 | 6 | 0 | 0 | 0 | 10 | 1 | 0 | 0 | 5 | 1 | 0 | 0 |

Table 19: Full results for the short-term forecasting task (Part 2). For prediction lengths $\tau \in \{3, 6, 9, 12\}$, the lookback horizon is $T = 12$; for $\tau \in \{24, 36, 48, 60\}$, it is $T = 36$. The best and second-best results are highlighted in **bold**, and underlined, respectively. This table presents the detailed version of Table 3.

| Model | OrthoLienar (Ours) | | TimeMix. 2024a | | FilterNet 2024a | | FITS 2024 | | DLinear 2023 | | TimeMix.++ 2025a | | Leddam 2024 | | CARD 2024b | | Fredformer 2024 | | iTrans. 2024a | | PatchTST 2023 | | TimesNet 2023b | |
|---|---|---|---|---|---|---|---|---|---|---|---|---|---|---|---|---|---|---|---|---|---|---|---|---|
| Metric | MSE | MAE | MSE | MAE | MSE | MAE | MSE | MAE | MSE | MAE | MSE | MAE | MSE | MAE | MSE | MAE | MSE | MAE | MSE | MAE | MSE | MAE | MSE | MAE |
| SP500 3 | 0.035 | 0.126 | 0.038 | 0.137 | 0.046 | 0.154 | 0.046 | 0.157 | 0.047 | 0.152 | 0.040 | 0.141 | 0.047 | 0.155 | 0.036 | 0.130 | 0.045 | 0.153 | 0.046 | 0.155 | 0.042 | 0.146 | 0.037 | 0.133 |
| SP500 6 | 0.053 | 0.158 | 0.056 | 0.167 | 0.071 | 0.193 | 0.067 | 0.190 | 0.070 | 0.197 | 0.057 | 0.168 | 0.064 | 0.181 | 0.054 | 0.162 | 0.066 | 0.187 | 0.066 | 0.187 | 0.063 | 0.183 | 0.055 | 0.165 |
| SP500 9 | 0.070 | 0.181 | 0.074 | 0.192 | 0.080 | 0.203 | 0.087 | 0.218 | 0.091 | 0.220 | 0.076 | 0.196 | 0.081 | 0.206 | 0.072 | 0.184 | 0.080 | 0.205 | 0.080 | 0.203 | 0.081 | 0.207 | 0.071 | 0.187 |
| SP500 12 | 0.088 | 0.204 | 0.092 | 0.213 | 0.097 | 0.224 | 0.106 | 0.242 | 0.113 | 0.251 | 0.093 | 0.213 | 0.104 | 0.236 | 0.089 | 0.206 | 0.103 | 0.234 | 0.099 | 0.228 | 0.101 | 0.233 | 0.089 | 0.210 |
| SP500 Avg | 0.061 | 0.167 | 0.065 | 0.177 | 0.073 | 0.193 | 0.076 | 0.202 | 0.080 | 0.205 | 0.066 | 0.179 | 0.074 | 0.194 | 0.063 | 0.170 | 0.073 | 0.195 | 0.073 | 0.193 | 0.072 | 0.192 | 0.063 | 0.174 |
| SP500 24 | 0.155 | 0.271 | 0.159 | 0.288 | 0.181 | 0.317 | 0.193 | 0.334 | 0.189 | 0.330 | 0.172 | 0.305 | 0.175 | 0.308 | 0.156 | 0.276 | 0.181 | 0.315 | 0.180 | 0.309 | 0.164 | 0.298 | 0.162 | 0.291 |
| SP500 36 | 0.209 | 0.317 | 0.218 | 0.343 | 0.224 | 0.341 | 0.259 | 0.389 | 0.250 | 0.363 | 0.227 | 0.344 | 0.232 | 0.358 | 0.206 | 0.319 | 0.239 | 0.365 | 0.225 | 0.346 | 0.221 | 0.341 | 0.219 | 0.344 |
| SP500 48 | 0.258 | 0.358 | 0.264 | 0.367 | 0.280 | 0.384 | 0.324 | 0.439 | 0.291 | 0.398 | 0.272 | 0.383 | 0.276 | 0.388 | 0.258 | 0.354 | 0.283 | 0.394 | 0.275 | 0.383 | 0.278 | 0.397 | 0.262 | 0.371 |
| SP500 60 | 0.305 | 0.387 | 0.322 | 0.416 | 0.332 | 0.416 | 0.391 | 0.486 | 0.377 | 0.475 | 0.319 | 0.413 | 0.325 | 0.423 | 0.303 | 0.385 | 0.341 | 0.438 | 0.322 | 0.418 | 0.321 | 0.409 | 0.305 | 0.399 |
| SP500 Avg | 0.231 | 0.333 | 0.241 | 0.353 | 0.254 | 0.365 | 0.291 | 0.412 | 0.277 | 0.391 | 0.247 | 0.361 | 0.252 | 0.369 | 0.231 | 0.333 | 0.261 | 0.378 | 0.250 | 0.364 | 0.246 | 0.361 | 0.237 | 0.351 |
| DowJones 3 | 1.550 | 0.276 | 1.568 | 0.276 | 1.587 | 0.278 | 1.611 | 0.283 | 1.574 | 0.280 | 1.555 | 0.278 | 1.538 | 0.277 | 1.563 | 0.276 | 1.563 | 0.277 | 1.551 | 0.276 | 1.556 | 0.276 | 2.269 | 0.325 |
| DowJones 6 | 2.569 | 0.361 | 2.593 | 0.362 | 2.634 | 0.364 | 2.661 | 0.370 | 2.613 | 0.367 | 2.591 | 0.364 | 2.588 | 0.364 | 2.579 | 0.362 | 2.619 | 0.364 | 2.589 | 0.363 | 2.588 | 0.362 | 3.934 | 0.430 |
| DowJones 9 | 3.565 | 0.430 | 3.564 | 0.431 | 3.651 | 0.434 | 3.704 | 0.443 | 3.621 | 0.436 | 3.572 | 0.433 | 3.644 | 0.440 | 3.569 | 0.430 | 3.575 | 0.432 | 3.599 | 0.433 | 3.585 | 0.432 | 4.715 | 0.492 |
| DowJones 12 | 4.517 | 0.490 | 4.472 | 0.490 | 4.634 | 0.496 | 4.666 | 0.503 | 4.586 | 0.497 | 4.630 | 0.496 | 4.622 | 0.498 | 4.528 | 0.490 | 4.605 | 0.493 | 4.579 | 0.493 | 4.533 | 0.492 | 5.999 | 0.561 |
| DowJones Avg | 3.050 | 0.389 | 3.049 | 0.390 | 3.126 | 0.393 | 3.160 | 0.400 | 3.099 | 0.395 | 3.087 | 0.393 | 3.098 | 0.395 | 3.060 | 0.389 | 3.090 | 0.391 | 3.079 | 0.391 | 3.065 | 0.390 | 4.229 | 0.452 |
| DowJones 24 | 7.432 | 0.664 | 8.327 | 0.683 | 8.000 | 0.683 | 7.974 | 0.690 | 7.590 | 0.670 | 8.283 | 0.689 | 8.029 | 0.679 | 7.416 | 0.665 | 7.758 | 0.672 | 7.925 | 0.677 | 7.641 | 0.670 | 11.535 | 0.834 |
| DowJones 36 | 10.848 | 0.799 | 11.192 | 0.813 | 12.011 | 0.823 | 11.907 | 0.837 | 10.986 | 0.803 | 14.754 | 0.856 | 11.962 | 0.828 | 10.799 | 0.798 | 11.456 | 0.808 | 12.087 | 0.827 | 11.210 | 0.807 | 16.922 | 0.982 |
| DowJones 48 | 14.045 | 0.914 | 15.278 | 0.945 | 14.814 | 0.933 | 15.821 | 0.969 | 14.157 | 0.922 | 16.893 | 0.970 | 15.266 | 0.937 | 13.881 | 0.912 | 14.696 | 0.921 | 14.787 | 0.930 | 14.866 | 0.935 | 19.501 | 1.093 |
| DowJones 60 | 16.959 | 1.017 | 20.997 | 1.067 | 18.932 | 1.054 | 19.320 | 1.077 | 18.018 | 1.035 | 18.881 | 1.059 | 18.407 | 1.045 | 17.257 | 1.021 | 18.058 | 1.032 | 18.298 | 1.041 | 17.947 | 1.036 | 22.804 | 1.177 |
| DowJones Avg | 12.321 | 0.848 | 13.948 | 0.877 | 13.439 | 0.873 | 13.755 | 0.893 | 12.688 | 0.857 | 14.703 | 0.893 | 13.416 | 0.872 | 12.338 | 0.849 | 12.992 | 0.858 | 13.274 | 0.869 | 12.916 | 0.862 | 17.690 | 1.021 |
| CarSales 3 | 0.303 | 0.277 | 0.307 | 0.296 | 0.304 | 0.291 | 0.411 | 0.375 | 0.396 | 0.383 | 0.309 | 0.289 | 0.316 | 0.300 | 0.318 | 0.295 | 0.311 | 0.300 | 0.281 | 0.276 | 0.300 | 0.290 | 0.315 | 0.303 |
| CarSales 6 | 0.315 | 0.285 | 0.327 | 0.307 | 0.317 | 0.300 | 0.401 | 0.367 | 0.390 | 0.376 | 0.324 | 0.300 | 0.320 | 0.303 | 0.335 | 0.307 | 0.326 | 0.309 | 0.294 | 0.285 | 0.311 | 0.295 | 0.319 | 0.306 |
| CarSales 9 | 0.327 | 0.293 | 0.336 | 0.315 | 0.329 | 0.308 | 0.418 | 0.380 | 0.406 | 0.390 | 0.331 | 0.306 | 0.332 | 0.311 | 0.351 | 0.319 | 0.339 | 0.318 | 0.305 | 0.292 | 0.324 | 0.306 | 0.338 | 0.318 |
| CarSales 12 | 0.336 | 0.301 | 0.343 | 0.319 | 0.335 | 0.312 | 0.424 | 0.385 | 0.410 | 0.390 | 0.343 | 0.314 | 0.336 | 0.313 | 0.357 | 0.324 | 0.348 | 0.324 | 0.314 | 0.298 | 0.331 | 0.310 | 0.338 | 0.320 |
| CarSales Avg | 0.320 | 0.289 | 0.328 | 0.309 | 0.321 | 0.303 | 0.413 | 0.377 | 0.400 | 0.385 | 0.327 | 0.302 | 0.326 | 0.306 | 0.340 | 0.311 | 0.331 | 0.313 | 0.298 | 0.288 | 0.316 | 0.300 | 0.327 | 0.312 |
| CarSales 24 | 0.320 | 0.302 | 0.320 | 0.318 | 0.318 | 0.319 | 0.359 | 0.347 | 0.354 | 0.350 | 0.323 | 0.320 | 0.325 | 0.322 | 0.337 | 0.321 | 0.319 | 0.326 | 0.303 | 0.312 | 0.319 | 0.319 | 0.316 | 0.328 |
| CarSales 36 | 0.334 | 0.315 | 0.332 | 0.331 | 0.331 | 0.330 | 0.373 | 0.360 | 0.368 | 0.365 | 0.351 | 0.348 | 0.337 | 0.333 | 0.348 | 0.333 | 0.333 | 0.335 | 0.318 | 0.323 | 0.332 | 0.330 | 0.338 | 0.343 |
| CarSales 48 | 0.347 | 0.327 | 0.345 | 0.343 | 0.342 | 0.341 | 0.385 | 0.370 | 0.382 | 0.379 | 0.351 | 0.342 | 0.351 | 0.346 | 0.362 | 0.345 | 0.349 | 0.344 | 0.331 | 0.332 | 0.347 | 0.344 | 0.349 | 0.351 |
| CarSales 60 | 0.358 | 0.337 | 0.355 | 0.351 | 0.352 | 0.349 | 0.399 | 0.385 | 0.388 | 0.380 | 0.363 | 0.352 | 0.361 | 0.353 | 0.372 | 0.353 | 0.359 | 0.349 | 0.344 | 0.342 | 0.355 | 0.348 | 0.359 | 0.359 |
| CarSales Avg | 0.340 | 0.320 | 0.338 | 0.336 | 0.336 | 0.335 | 0.379 | 0.365 | 0.373 | 0.368 | 0.347 | 0.340 | 0.343 | 0.338 | 0.355 | 0.338 | 0.340 | 0.338 | 0.324 | 0.327 | 0.338 | 0.335 | 0.340 | 0.345 |
| Power 3 | 0.864 | 0.688 | 0.850 | 0.683 | 0.842 | 0.685 | 0.899 | 0.712 | 0.876 | 0.708 | 0.843 | 0.685 | 0.859 | 0.696 | 0.896 | 0.691 | 0.865 | 0.697 | 0.863 | 0.696 | 0.830 | 0.687 | 0.929 | 0.716 |
| Power 6 | 0.991 | 0.742 | 0.971 | 0.743 | 0.988 | 0.747 | 1.079 | 0.785 | 0.991 | 0.761 | 0.997 | 0.748 | 1.023 | 0.761 | 1.013 | 0.750 | 0.994 | 0.751 | 0.990 | 0.748 | 0.948 | 0.741 | 1.070 | 0.773 |
| Power 9 | 1.062 | 0.770 | 1.024 | 0.763 | 1.063 | 0.778 | 1.150 | 0.804 | 1.051 | 0.790 | 1.050 | 0.767 | 1.071 | 0.783 | 1.085 | 0.781 | 1.073 | 0.784 | 1.065 | 0.780 | 1.027 | 0.773 | 1.104 | 0.788 |
| Power 12 | 1.119 | 0.789 | 1.087 | 0.788 | 1.125 | 0.803 | 1.266 | 0.851 | 1.110 | 0.814 | 1.107 | 0.792 | 1.130 | 0.804 | 1.124 | 0.792 | 1.135 | 0.807 | 1.142 | 0.808 | 1.083 | 0.797 | 1.212 | 0.825 |
| Power Avg | 1.009 | 0.747 | 0.983 | 0.744 | 1.004 | 0.753 | 1.098 | 0.788 | 1.007 | 0.768 | 0.999 | 0.748 | 1.021 | 0.761 | 1.029 | 0.753 | 1.017 | 0.760 | 1.015 | 0.758 | 0.972 | 0.749 | 1.079 | 0.775 |
| Power 24 | 1.343 | 0.870 | 1.341 | 0.881 | 1.410 | 0.916 | 1.491 | 0.944 | 1.390 | 0.916 | 1.340 | 0.877 | 1.397 | 0.909 | 1.406 | 0.886 | 1.410 | 0.913 | 1.462 | 0.924 | 1.468 | 0.935 | 1.494 | 0.924 |
| Power 36 | 1.445 | 0.903 | 1.420 | 0.914 | 1.590 | 0.968 | 1.621 | 0.994 | 1.518 | 0.957 | 1.446 | 0.920 | 1.509 | 0.951 | 1.506 | 0.921 | 1.538 | 0.953 | 1.582 | 0.964 | 1.593 | 0.972 | 1.526 | 0.950 |
| Power 48 | 1.559 | 0.946 | 1.567 | 0.963 | 1.680 | 1.009 | 1.775 | 1.052 | 1.610 | 0.995 | 1.467 | 0.933 | 1.646 | 0.999 | 1.583 | 0.957 | 1.652 | 1.008 | 1.696 | 1.011 | 1.710 | 1.020 | 1.581 | 0.981 |
| Power 60 | 1.602 | 0.971 | 1.609 | 0.988 | 1.776 | 1.053 | 1.958 | 1.122 | 1.679 | 1.020 | 1.626 | 1.006 | 1.727 | 1.043 | 1.693 | 1.003 | 1.752 | 1.049 | 1.796 | 1.061 | 1.829 | 1.064 | 1.625 | 1.010 |
| Power Avg | 1.487 | 0.922 | 1.484 | 0.937 | 1.614 | 0.986 | 1.711 | 1.028 | 1.549 | 0.972 | 1.470 | 0.934 | 1.570 | 0.975 | 1.547 | 0.942 | 1.588 | 0.981 | 1.634 | 0.990 | 1.650 | 0.998 | 1.556 | 0.966 |
| Website 3 | 0.077 | 0.199 | 0.086 | 0.215 | 0.084 | 0.213 | 0.191 | 0.320 | 0.159 | 0.288 | 0.083 | 0.208 | 0.082 | 0.210 | 0.099 | 0.225 | 0.080 | 0.207 | 0.072 | 0.198 | 0.089 | 0.217 | 0.075 | 0.200 |
| Website 6 | 0.103 | 0.227 | 0.124 | 0.248 | 0.116 | 0.242 | 0.235 | 0.356 | 0.192 | 0.302 | 0.115 | 0.237 | 0.114 | 0.239 | 0.135 | 0.255 | 0.116 | 0.241 | 0.098 | 0.227 | 0.121 | 0.246 | 0.105 | 0.232 |
| Website 9 | 0.135 | 0.251 | 0.159 | 0.275 | 0.151 | 0.269 | 0.276 | 0.372 | 0.220 | 0.330 | 0.156 | 0.273 | 0.147 | 0.266 | 0.171 | 0.282 | 0.150 | 0.269 | 0.124 | 0.252 | 0.157 | 0.273 | 0.138 | 0.262 |
| Website 12 | 0.176 | 0.281 | 0.204 | 0.306 | 0.194 | 0.297 | 0.409 | 0.484 | 0.255 | 0.355 | 0.193 | 0.296 | 0.196 | 0.298 | 0.213 | 0.307 | 0.196 | 0.301 | 0.144 | 0.269 | 0.200 | 0.302 | 0.169 | 0.285 |
| Website Avg | 0.123 | 0.240 | 0.143 | 0.261 | 0.136 | 0.255 | 0.278 | 0.383 | 0.204 | 0.319 | 0.137 | 0.253 | 0.135 | 0.253 | 0.154 | 0.267 | 0.135 | 0.254 | 0.109 | 0.237 | 0.141 | 0.259 | 0.122 | 0.244 |
| Website 24 | 0.186 | 0.306 | 0.229 | 0.335 | 0.273 | 0.357 | 0.431 | 0.469 | 0.315 | 0.393 | 0.231 | 0.349 | 0.240 | 0.345 | 0.325 | 0.370 | 0.216 | 0.335 | 0.181 | 0.305 | 0.245 | 0.350 | 0.276 | 0.358 |
| Website 36 | 0.272 | 0.356 | 0.361 | 0.420 | 0.401 | 0.441 | 0.554 | 0.552 | 0.385 | 0.447 | 0.328 | 0.397 | 0.327 | 0.405 | 0.428 | 0.442 | 0.331 | 0.411 | 0.226 | 0.342 | 0.370 | 0.429 | 0.377 | 0.436 |
| Website 48 | 0.365 | 0.391 | 0.501 | 0.507 | 0.530 | 0.522 | 0.694 | 0.647 | 0.436 | 0.486 | 0.450 | 0.473 | 0.446 | 0.475 | 0.457 | 0.478 | 0.483 | 0.496 | 0.263 | 0.370 | 0.504 | 0.513 | 0.389 | 0.446 |
| Website 60 | 0.486 | 0.481 | 0.571 | 0.562 | 0.630 | 0.592 | 0.736 | 0.673 | 0.468 | 0.510 | 0.525 | 0.517 | 0.561 | 0.549 | 0.596 | 0.566 | 0.556 | 0.547 | 0.323 | 0.410 | 0.565 | 0.478 | 0.587 | 0.507 |
| Website Avg | 0.327 | 0.383 | 0.415 | 0.456 | 0.458 | 0.478 | 0.604 | 0.585 | 0.401 | 0.459 | 0.384 | 0.434 | 0.393 | 0.443 | 0.451 | 0.464 | 0.396 | 0.447 | 0.248 | 0.357 | 0.426 | 0.464 | 0.380 | 0.437 |
| Unemp 3 | 0.012 | 0.047 | 0.015 | 0.074 | 0.012 | 0.062 | 0.161 | 0.289 | 0.072 | 0.200 | 0.014 | 0.069 | 0.013 | 0.061 | 0.014 | 0.070 | 0.013 | 0.068 | 0.012 | 0.061 | 0.012 | 0.060 | 0.062 | 0.171 |
| Unemp 6 | 0.041 | 0.108 | 0.057 | 0.154 | 0.043 | 0.130 | 0.229 | 0.345 | 0.115 | 0.255 | 0.079 | 0.194 | 0.040 | 0.122 | 0.042 | 0.127 | 0.046 | 0.139 | 0.047 | 0.136 | 0.043 | 0.127 | 0.111 | 0.227 |
| Unemp 9 | 0.084 | 0.170 | 0.109 | 0.213 | 0.107 | 0.216 | 0.369 | 0.443 | 0.191 | 0.329 | 0.101 | 0.220 | 0.119 | 0.226 | 0.093 | 0.194 | 0.095 | 0.198 | 0.098 | 0.199 | 0.093 | 0.190 | 0.176 | 0.281 |
| Unemp 12 | 0.131 | 0.220 | 0.195 | 0.293 | 0.155 | 0.255 | 0.475 | 0.500 | 0.240 | 0.386 | 0.196 | 0.291 | 0.141 | 0.239 | 0.155 | 0.250 | 0.148 | 0.250 | 0.158 | 0.256 | 0.164 | 0.261 | 0.240 | 0.326 |
| Unemp Avg | 0.067 | 0.136 | 0.094 | 0.183 | 0.079 | 0.166 | 0.308 | 0.394 | 0.154 | 0.292 | 0.097 | 0.188 | 0.078 | 0.162 | 0.076 | 0.162 | 0.075 | 0.163 | 0.079 | 0.163 | 0.078 | 0.160 | 0.147 | 0.251 |
| Unemp 24 | 0.458 | 0.447 | 0.668 | 0.549 | 0.655 | 0.548 | 2.520 | 1.255 | 0.951 | 0.645 | 0.779 | 0.578 | 0.861 | 0.645 | 0.583 | 0.522 | 0.545 | 0.496 | 0.719 | 0.595 | 0.620 | 0.536 | 1.798 | 0.967 |
| Unemp 36 | 0.870 | 0.619 | 1.580 | 0.866 | 1.247 | 0.769 | 4.478 | 1.703 | 0.906 | 0.717 | 1.318 | 0.777 | 1.741 | 0.938 | 1.205 | 0.762 | 1.107 | 0.725 | 2.212 | 1.103 | 2.245 | 1.120 | 2.820 | 1.171 |
| Unemp 48 | 1.643 | 0.900 | 3.702 | 1.483 | 2.700 | 1.209 | 6.385 | 2.052 | 0.974 | 0.747 | 2.977 | 1.258 | 3.526 | 1.427 | 2.737 | 1.218 | 6.192 | 2.055 | 2.577 | 1.178 | 2.644 | 1.191 | 5.909 | 1.620 |
| Unemp 60 | 2.593 | 1.182 | 6.320 | 2.030 | 5.370 | 1.830 | 8.460 | 2.392 | 1.071 | 0.798 | 6.582 | 2.060 | 5.574 | 1.857 | 4.475 | 1.624 | 8.236 | 2.384 | 5.441 | 1.799 | 4.074 | 1.511 | 7.511 | 2.053 |
| Unemp Avg | 1.391 | 0.787 | 3.068 | 1.232 | 2.493 | 1.089 | 5.461 | 1.850 | 0.975 | 0.727 | 2.914 | 1.168 | 2.925 | 1.217 | 2.250 | 1.031 | 4.020 | 1.415 | 2.737 | 1.169 | 2.396 | 1.089 | 4.510 | 1.452 |
| 1st Count | 18 | 35 | 5 | 6 | 1 | 0 | 0 | 0 | 3 | 3 | 3 | 1 | 2 | 0 | 7 | 9 | 0 | 0 | 21 | 14 | 5 | 2 | 0 | 0 |

Table 20: Comparison of different transformation bases. For *S1* and *S2*, the prediction lengths are $\tau \in \{3, 6, 9, 12\}$ and $\tau \in \{24, 36, 48, 60\}$, respectively. *Wavelet1* and *Wavelet2* denote the classic Haar and discrete Meyer wavelets, respectively. *Identity* means that no transformation is applied. This is a complete version of Table 4.

| Base | | ECL | | Solar-Energy | | PEMS03 | | Power (S2) | | ILI (S1) | | COVID-19 (S2) | | METR-LA (S2) | |
|---|---|---|---|---|---|---|---|---|---|---|---|---|---|---|---|
| Metric | | MSE | MAE | MSE | MAE | MSE | MAE | MSE | MAE | MSE | MAE | MSE | MAE | MSE | MAE |
| Ours | H1 | **0.131** | **0.221** | **0.179** | **0.191** | **0.060** | **0.159** | 1.343 | 0.870 | 0.468 | 0.349 | **4.474** | **1.180** | 0.650 | 0.337 |
| | H2 | **0.150** | **0.238** | **0.209** | **0.213** | **0.078** | **0.179** | 1.445 | 0.903 | 0.923 | 0.516 | 7.241 | 1.670 | 0.800 | **0.388** |
| | H3 | **0.165** | 0.254 | **0.231** | **0.229** | **0.104** | **0.210** | 1.559 | 0.946 | 1.289 | 0.655 | 10.076 | 1.985 | 0.905 | **0.427** |
| | H4 | 0.191 | 0.279 | **0.241** | **0.236** | **0.140** | **0.247** | 1.602 | 0.971 | 1.698 | 0.791 | 12.079 | 2.182 | 0.999 | 0.457 |
| | Avg | **0.159** | **0.248** | **0.215** | **0.217** | **0.095** | **0.199** | 1.487 | 0.922 | 1.094 | 0.578 | 8.467 | 1.754 | 0.838 | 0.402 |
| Fourier | H1 | 0.132 | 0.222 | 0.181 | 0.193 | 0.061 | 0.161 | 1.469 | 0.917 | 0.486 | 0.350 | 5.192 | 1.378 | 0.654 | 0.337 |
| | H2 | 0.153 | 0.241 | 0.214 | 0.215 | 0.078 | 0.182 | 1.584 | 0.962 | **0.885** | **0.500** | 7.553 | 1.699 | **0.799** | 0.390 |
| | H3 | **0.165** | **0.253** | 0.236 | 0.232 | 0.110 | 0.216 | 1.625 | 0.970 | 1.438 | 0.664 | 11.096 | 2.070 | 0.910 | **0.427** |
| | H4 | 0.195 | 0.283 | 0.247 | 0.238 | 0.155 | 0.258 | 1.778 | 1.020 | 2.264 | 0.822 | 12.819 | 2.209 | 1.007 | 0.459 |
| | Avg | 0.161 | 0.250 | 0.219 | 0.219 | 0.101 | 0.204 | 1.614 | 0.967 | 1.268 | 0.584 | 9.165 | 1.839 | 0.843 | 0.403 |
| Wavelet1 | H1 | 0.134 | 0.223 | 0.182 | 0.194 | 0.061 | 0.161 | 1.476 | 0.917 | 0.483 | 0.351 | 4.826 | 1.317 | **0.647** | **0.336** |
| | H2 | 0.153 | 0.241 | 0.217 | 0.216 | 0.079 | 0.183 | 1.586 | 0.959 | 0.943 | 0.520 | **7.117** | **1.652** | 0.801 | 0.390 |
| | H3 | 0.166 | 0.255 | 0.238 | 0.234 | 0.114 | 0.219 | 1.711 | 1.011 | **1.251** | **0.653** | 10.133 | 1.995 | 0.903 | 0.428 |
| | H4 | **0.186** | **0.276** | 0.250 | 0.240 | 0.176 | 0.278 | 1.880 | 1.060 | 1.786 | 0.795 | 12.588 | 2.233 | 1.011 | 0.464 |
| | Avg | 0.160 | 0.249 | 0.221 | 0.221 | 0.107 | 0.210 | 1.663 | 0.987 | 1.116 | 0.580 | 8.666 | 1.799 | 0.840 | 0.404 |
| Wavelet2 | H1 | 0.133 | 0.222 | 0.186 | 0.198 | 0.062 | 0.162 | 1.511 | 0.929 | 0.461 | 0.349 | 5.398 | 1.413 | 0.653 | 0.340 |
| | H2 | 0.154 | 0.243 | 0.222 | 0.220 | 0.080 | 0.184 | 1.570 | 0.953 | 0.935 | 0.526 | 7.199 | 1.663 | 0.803 | 0.392 |
| | H3 | **0.165** | 0.255 | 0.242 | 0.236 | 0.115 | 0.221 | 1.701 | 1.005 | 1.393 | 0.683 | 10.616 | 2.051 | 0.909 | 0.432 |
| | H4 | 0.196 | 0.283 | 0.254 | 0.243 | 0.175 | 0.272 | 1.876 | 1.060 | 1.921 | 0.818 | 12.583 | 2.232 | 1.007 | 0.462 |
| | Avg | 0.162 | 0.251 | 0.226 | 0.224 | 0.108 | 0.210 | 1.664 | 0.987 | 1.177 | 0.594 | 8.949 | 1.840 | 0.843 | 0.406 |
| Chebyshev | H1 | 0.134 | 0.223 | 0.184 | 0.198 | 0.061 | 0.160 | **1.338** | **0.864** | **0.456** | **0.341** | 5.381 | 1.391 | 0.661 | 0.339 |
| | H2 | 0.156 | 0.244 | 0.220 | 0.222 | 0.079 | 0.182 | 1.564 | 0.969 | 0.945 | 0.524 | 8.001 | 1.767 | 0.803 | 0.391 |
| | H3 | 0.182 | 0.270 | 0.244 | 0.240 | 0.118 | 0.220 | 1.604 | 0.950 | 1.527 | 0.700 | 11.244 | 2.108 | 0.923 | 0.433 |
| | H4 | 0.399 | 0.444 | 0.256 | 0.246 | 0.163 | 0.266 | 1.774 | 1.076 | 1.940 | 0.824 | 12.695 | 2.233 | 1.028 | 0.466 |
| | Avg | 0.218 | 0.295 | 0.226 | 0.226 | 0.105 | 0.207 | 1.570 | 0.965 | 1.217 | 0.597 | 9.330 | 1.875 | 0.854 | 0.407 |
| Laguerre | H1 | 0.139 | 0.227 | 0.192 | 0.204 | 0.065 | 0.167 | 1.425 | 0.899 | 0.595 | 0.395 | 5.632 | 1.456 | 0.659 | 0.346 |
| | H2 | 0.156 | 0.243 | 0.227 | 0.227 | 0.085 | 0.191 | 1.658 | 0.985 | 1.083 | 0.577 | 8.729 | 1.859 | 0.829 | 0.407 |
| | H3 | 0.173 | 0.262 | 0.251 | 0.244 | 0.122 | 0.226 | 1.774 | 1.013 | 1.581 | 0.737 | 10.381 | 2.024 | 0.933 | 0.444 |
| | H4 | 0.199 | 0.286 | 0.261 | 0.248 | 0.171 | 0.271 | 1.780 | 1.039 | 2.155 | 0.896 | 12.467 | 2.221 | 1.053 | 0.484 |
| | Avg | 0.167 | 0.255 | 0.233 | 0.230 | 0.111 | 0.214 | 1.659 | 0.984 | 1.353 | 0.651 | 9.302 | 1.890 | 0.868 | 0.420 |
| Legendre | H1 | 0.134 | 0.224 | 0.200 | 0.208 | 0.074 | 0.181 | 1.502 | 0.926 | 0.501 | 0.366 | 4.965 | 1.350 | 0.652 | 0.337 |
| | H2 | 0.152 | 0.239 | 0.238 | 0.231 | 0.080 | 0.183 | 1.603 | 0.962 | 0.964 | 0.529 | 7.396 | 1.703 | **0.799** | 0.389 |
| | H3 | **0.165** | 0.256 | 0.262 | 0.247 | 0.113 | 0.219 | 1.790 | 1.031 | 1.446 | 0.695 | 9.481 | 1.935 | 0.900 | 0.427 |
| | H4 | 0.193 | 0.282 | 0.274 | 0.253 | 0.168 | 0.269 | 1.846 | 1.060 | 1.799 | 0.822 | 12.358 | 2.203 | 1.015 | 0.463 |
| | Avg | 0.161 | 0.250 | 0.243 | 0.235 | 0.109 | 0.213 | 1.685 | 0.995 | 1.177 | 0.603 | 8.550 | 1.798 | 0.841 | 0.404 |
| Identity | H1 | 0.134 | 0.224 | 0.186 | 0.199 | 0.063 | 0.163 | 1.419 | 0.900 | 0.477 | 0.352 | 4.841 | 1.284 | 0.659 | 0.342 |
| | H2 | 0.156 | 0.245 | 0.221 | 0.220 | 0.080 | 0.184 | 1.453 | 0.903 | 0.956 | 0.523 | 7.828 | 1.757 | 0.807 | 0.395 |
| | H3 | 0.167 | 0.257 | 0.244 | 0.237 | 0.117 | 0.222 | 1.582 | 0.956 | 1.307 | 0.659 | 10.153 | 2.003 | 0.921 | 0.433 |
| | H4 | 0.194 | 0.282 | 0.256 | 0.244 | 0.165 | 0.266 | 1.715 | 1.023 | 1.874 | 0.812 | 12.603 | 2.232 | 1.006 | 0.461 |
| | Avg | 0.163 | 0.252 | 0.227 | 0.225 | 0.106 | 0.209 | 1.542 | 0.945 | 1.153 | 0.587 | 8.856 | 1.819 | 0.848 | 0.408 |

functions from the set $\{\text{Softplus}, \text{Identity}, \text{Softmax}, \text{Sigmoid}, \text{ReLU}\}$ and normalization methods from $\{\text{L1}, \text{L2}\}$ (all applied row-wise). Here, Identity denotes no transformation.

As shown in Table 37, the combination 'Softplus + L1' yields the best performance among the 10 candidates. On average, L1 normalization leads to a 10% performance improvement over L2, while Softplus slightly outperforms Softmax and other transformation functions.

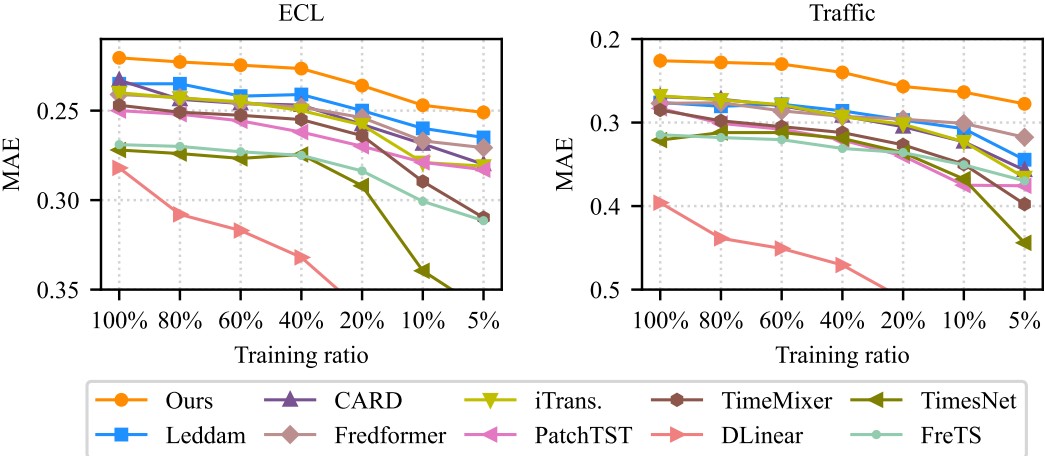

Figure 16: Performance change with less training data. The lookback and prediction lengths are set to 96. The Y-axis is inverted for clarity.

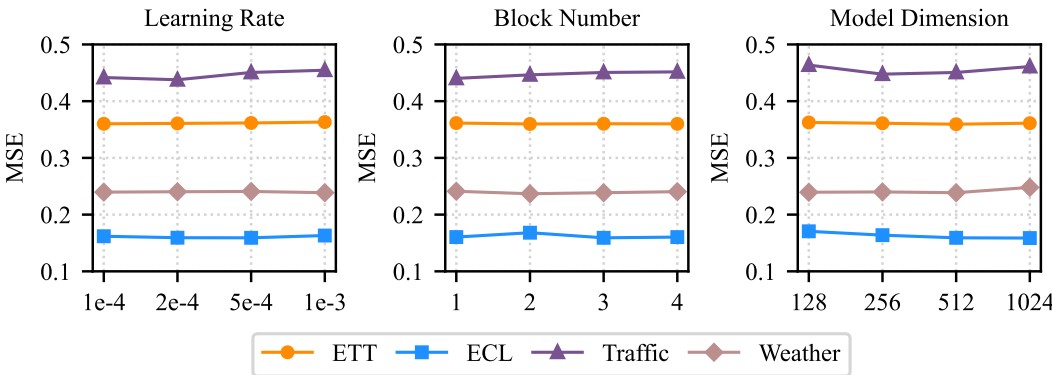

Figure 17: Hyperparameter sensitivity with respect to the learning rate $lr$, the number of blocks $L$, and the model dimension $D$. Average MSEs are reported across four prediction lengths $\tau \in \{96, 192, 336, 720\}$. ETT denotes the average performance over four subsets: ETT $\{h1, h2, m1, m2\}$. The lookback length $T$ is uniformly set as 96.

## J.2 Pre- and post-linear layers in NormLin module

In CSL, the pre- and post-linear layers, named based on their positions in the execution sequence, are applied around the NormLin layer (see Equation 4): $\text{Post\_Lin}\,(\text{NormLin}\,(\text{Pre\_Lin}(\cdot)))$.

We conduct ablation studies to evaluate the contributions of these two layers. As shown in Table 38, incorporating both pre- and post-linear layers yields an average performance gain of 6% compared to the variant without them, highlighting their effectiveness in refining inputs for multivariate correlation modeling and downstream series representation learning.

## J.3 Ablations on Q matrices

The effectiveness of OrthoTrans has been demonstrated in Tables 4 and 5. To further analyze the individual contributions of $\mathbf{Q}_i$ and $\mathbf{Q}_o$, we conduct ablation studies on these transformation matrices.

As shown in Table 39, using both $\mathbf{Q}_i$ and $\mathbf{Q}_o$ achieves the best performance—reducing the average MSE by 5.3% compared to the baseline without transformations, by 4.4% compared to using $\mathbf{Q}_o$ only, and by 0.8% compared to using $\mathbf{Q}_i$ only. Notably, the variant using only $\mathbf{Q}_i$ outperforms the one using only $\mathbf{Q}_o$, highlighting the critical role of temporal decorrelation in effective encoding and representation learning.

### J.4 Swapping Q matrices

Table 40 reports the forecasting performance when both training and testing are conducted using $\mathbf{Q}$ matrices ($\mathbf{Q}_i$ and $\mathbf{Q}_o$) derived from different datasets. As shown, using the $\mathbf{Q}$ matrices from the same dataset yields the best performance, highlighting the importance of incorporating dataset-specific temporal correlation information in time series forecasting.

## K  Model efficiency

As a linear model, OLinear achieves remarkable training and inference efficiency while delivering state-of-the-art forecasting performance. Figure 3 shows that OLinear achieves better efficiency than Transformer-based forecasters, benefiting from its relatively simple architecture. Detailed resource consumption for both training and inference is summarized in Table 41. Moreover, Table 42 presents the efficiency differences before and after applying the NormLin module to Transformer-based forecasters.

## L  Limitations

While this work includes extensive experiments, it primarily focuses on time series forecasting, and our claims are confined to this setting. Future work includes larger-scale pre-training and a wider range of time series analysis tasks (e.g., imputation and anomaly detection), in order to further validate the robustness of OLinear and its core components, OrthoTrans and NormLin. In addition, although NormLin is linear-based, it still incurs $\mathcal{O}(N^2)$ computational and memory complexity with respect to the number of variates $N$. Developing simpler yet effective alternatives for modeling multivariate correlations—and more generally, for learning general-purpose token dependencies—remains a promising direction.

## M  Broader impacts

Multivariate time series forecasting is a fundamental field with a wide range of real-world applications across domains such as energy, finance, healthcare, and transportation. While this work does not target a specific application domain, OLinear could have general real-world applicability (e.g., weather forecasting and traffic planning). We contribute two key insights: (1) the OrthoTrans scheme, which reformulates temporally correlated forecasting as inter-independent feature prediction; and (2) the linear-based NormLin module, which effectively models multivariate correlations, enhances representation learning among temporal and frequency tokens, and remains compatible with decoder architectures and large-scale pre-training. As a step toward simpler and more efficient token dependency learners, the NormLin module shows potential for broader use in deep learning and may inspire future research in this direction.

Table 21: Full results of applying OrthoTrans to iTransformer, PatchTST and RLinear. 'Imp.' denotes the improvement over the corresponding vanilla model. Note that RLinear consists of only a single linear layer that projects the lookback length $T$ to the prediction horizon $\tau$. Nevertheless, OrthoTrans consistently improves RLinear's performance, demonstrating its ability to enhance model capacity. This table is the complete version of Table 5.

| Model | | iTransformer 2024a | | | | PatchTST 2023 | | | | RLinear 2023 | | | |
|---|---|---|---|---|---|---|---|---|---|---|---|---|---|
| | | Vanilla | | +OrthoTrans | | Vanilla | | +OrthoTrans | | Vanilla | | +OrthoTrans | |
| Metric | | MSE | MAE | MSE | MAE | MSE | MAE | MSE | MAE | MSE | MAE | MSE | MAE |
| ETTm1 | 96 | **0.334** | 0.368 | **0.334** | **0.367** | 0.329 | 0.367 | **0.322** | **0.359** | 0.355 | 0.376 | **0.348** | **0.366** |
| | 192 | **0.377** | 0.391 | **0.377** | **0.389** | 0.367 | **0.385** | **0.363** | **0.383** | 0.391 | 0.392 | **0.386** | **0.385** |
| | 336 | 0.426 | 0.420 | **0.414** | **0.413** | 0.399 | 0.410 | **0.394** | **0.405** | 0.424 | 0.415 | **0.419** | **0.406** |
| | 720 | 0.491 | 0.459 | **0.490** | **0.452** | **0.454** | **0.439** | 0.457 | 0.445 | 0.487 | 0.450 | **0.478** | **0.439** |
| | Avg | 0.407 | 0.410 | **0.404** | **0.405** | 0.387 | 0.400 | **0.384** | **0.398** | 0.414 | 0.407 | **0.408** | **0.399** |
| | Imp. | – | – | 0.80% | 1.16% | – | – | 0.82% | 0.53% | – | – | 1.51% | 1.97% |
| ECL | 96 | 0.148 | 0.240 | **0.136** | **0.233** | 0.161 | 0.250 | **0.153** | **0.243** | 0.201 | 0.281 | **0.197** | **0.273** |
| | 192 | 0.162 | 0.253 | **0.155** | **0.250** | 0.199 | 0.289 | **0.165** | **0.254** | 0.201 | 0.283 | **0.196** | **0.276** |
| | 336 | 0.178 | 0.269 | **0.169** | **0.265** | 0.215 | 0.305 | **0.182** | **0.272** | 0.215 | 0.298 | **0.211** | **0.291** |
| | 720 | **0.225** | 0.317 | **0.225** | **0.311** | 0.256 | 0.337 | **0.222** | **0.306** | 0.257 | 0.331 | **0.253** | **0.324** |
| | Avg | 0.178 | 0.270 | **0.171** | **0.265** | 0.208 | 0.295 | **0.181** | **0.269** | 0.219 | 0.298 | **0.214** | **0.291** |
| | Imp. | – | – | 3.97% | 1.87% | – | – | 13.12% | 8.98% | – | – | 2.17% | 2.35% |
| PEMS03 | 12 | 0.071 | 0.174 | **0.062** | **0.164** | 0.099 | 0.216 | **0.073** | **0.176** | 0.126 | 0.236 | **0.117** | **0.226** |
| | 24 | 0.093 | 0.201 | **0.080** | **0.185** | 0.142 | 0.259 | **0.104** | **0.210** | 0.246 | 0.334 | **0.233** | **0.322** |
| | 48 | 0.125 | 0.236 | **0.112** | **0.222** | 0.211 | 0.319 | **0.181** | **0.268** | 0.551 | 0.529 | **0.532** | **0.514** |
| | 96 | 0.164 | 0.275 | **0.160** | **0.269** | 0.269 | 0.370 | **0.293** | **0.338** | 1.057 | 0.787 | **1.024** | **0.760** |
| | Avg | 0.113 | 0.221 | **0.103** | **0.210** | 0.180 | 0.291 | **0.163** | **0.248** | 0.495 | 0.472 | **0.477** | **0.456** |
| | Imp. | – | – | 8.45% | 5.02% | – | – | 9.69% | 14.81% | – | – | 3.74% | 3.50% |
| PEMS07 | 12 | 0.067 | 0.165 | **0.055** | **0.147** | 0.095 | 0.207 | **0.065** | **0.160** | 0.118 | 0.235 | **0.109** | **0.221** |
| | 24 | 0.088 | 0.190 | **0.071** | **0.165** | 0.150 | 0.262 | **0.097** | **0.192** | 0.242 | 0.341 | **0.228** | **0.324** |
| | 48 | 0.110 | 0.215 | **0.093** | **0.188** | 0.253 | 0.340 | **0.164** | **0.247** | 0.562 | 0.541 | **0.539** | **0.522** |
| | 96 | 0.139 | 0.245 | **0.122** | **0.218** | 0.346 | 0.404 | **0.264** | **0.311** | 1.096 | 0.795 | **1.065** | **0.773** |
| | Avg | 0.101 | 0.204 | **0.085** | **0.179** | 0.211 | 0.303 | **0.147** | **0.227** | 0.504 | 0.478 | **0.485** | **0.460** |
| | Imp. | – | – | 15.59% | 11.94% | – | – | 30.17% | 24.94% | – | – | 3.76% | 3.77% |
| Solar-Energy | 96 | 0.203 | **0.237** | **0.194** | **0.237** | 0.234 | 0.286 | **0.208** | **0.248** | 0.322 | 0.339 | **0.302** | **0.327** |
| | 192 | 0.233 | **0.261** | **0.231** | 0.265 | 0.267 | 0.310 | **0.237** | **0.268** | 0.359 | 0.356 | **0.342** | **0.346** |
| | 336 | 0.248 | 0.273 | **0.242** | **0.272** | 0.290 | 0.315 | **0.255** | **0.281** | 0.397 | 0.369 | **0.384** | **0.362** |
| | 720 | 0.249 | **0.275** | **0.245** | **0.275** | 0.289 | 0.317 | **0.255** | **0.281** | 0.397 | **0.356** | **0.387** | **0.356** |
| | Avg | 0.233 | **0.262** | **0.228** | **0.262** | 0.270 | 0.307 | **0.239** | **0.269** | 0.369 | 0.356 | **0.354** | **0.348** |
| | Imp. | – | – | 2.16% | 0.00% | – | – | 11.57% | 12.22% | – | – | 4.14% | 2.37% |
| Weather | 96 | 0.174 | 0.214 | **0.164** | **0.208** | 0.177 | 0.218 | **0.164** | **0.209** | **0.192** | **0.232** | **0.192** | **0.232** |
| | 192 | **0.221** | 0.254 | **0.221** | **0.259** | 0.225 | 0.259 | **0.211** | **0.251** | 0.240 | 0.271 | **0.236** | **0.268** |
| | 336 | 0.278 | 0.296 | **0.271** | **0.295** | 0.278 | 0.297 | **0.266** | **0.292** | 0.292 | 0.307 | **0.288** | **0.304** |
| | 720 | 0.358 | 0.349 | **0.351** | **0.347** | 0.354 | 0.348 | **0.345** | **0.343** | 0.364 | 0.353 | **0.359** | **0.350** |
| | Avg | 0.258 | 0.279 | **0.252** | **0.277** | 0.259 | 0.281 | **0.246** | **0.274** | 0.272 | 0.291 | **0.269** | **0.289** |
| | Imp. | – | – | 2.42% | 0.63% | – | – | 4.86% | 2.59% | – | – | 1.22% | 0.86% |
| METR-LA | 3 | **0.205** | **0.188** | **0.205** | 0.189 | 0.204 | 0.190 | **0.203** | **0.187** | 0.210 | 0.195 | **0.208** | **0.194** |
| | 6 | 0.300 | 0.229 | **0.292** | **0.227** | 0.298 | **0.227** | **0.296** | 0.228 | 0.304 | **0.234** | **0.303** | 0.235 |
| | 9 | 0.386 | 0.265 | **0.375** | **0.260** | 0.382 | **0.263** | **0.380** | **0.263** | 0.389 | 0.268 | **0.387** | **0.266** |
| | 12 | 0.460 | 0.295 | **0.444** | **0.291** | 0.456 | **0.292** | **0.452** | 0.294 | 0.465 | 0.298 | **0.464** | **0.297** |
| | Avg | 0.338 | 0.244 | **0.329** | **0.242** | 0.335 | **0.243** | **0.333** | **0.243** | 0.342 | 0.249 | **0.341** | **0.248** |
| | Imp. | – | – | 2.55% | 1.02% | – | – | 0.67% | 0.00% | – | – | 0.44% | 0.28% |

Table 22: Ablation results for replacing or removing components along the variate and temporal dimensions. 'Hor.' denotes the prediction horizon. The simplified results are presented in Table 6.

| Var. | Temp. | Hor. | ECL MSE | ECL MAE | Traffic MSE | Traffic MAE | Solar MSE | Solar MAE | PEMS03 MSE | PEMS03 MAE | Weather MSE | Weather MAE | ETTm1 MSE | ETTm1 MAE | NASDAQ (S1) MSE | NASDAQ (S1) MAE | ILI (S2) MSE | ILI (S2) MAE |
|---|---|---|---|---|---|---|---|---|---|---|---|---|---|---|---|---|---|---|
| NormLin | Linear | H1 | 0.131 | 0.221 | 0.398 | 0.226 | 0.179 | 0.191 | 0.060 | 0.159 | 0.153 | 0.190 | 0.302 | 0.334 | 0.036 | 0.092 | 1.737 | 0.800 |
|  |  | H2 | 0.150 | 0.238 | 0.439 | 0.241 | 0.209 | 0.213 | 0.078 | 0.179 | 0.200 | 0.235 | 0.357 | 0.363 | 0.049 | 0.117 | 1.714 | 0.795 |
|  |  | H3 | 0.165 | 0.254 | 0.464 | 0.250 | 0.231 | 0.229 | 0.104 | 0.210 | 0.258 | 0.280 | 0.387 | 0.385 | 0.062 | 0.137 | 1.821 | 0.804 |
|  |  | H4 | 0.191 | 0.279 | 0.502 | 0.270 | 0.241 | 0.236 | 0.140 | 0.247 | 0.337 | 0.333 | 0.452 | 0.426 | 0.073 | 0.154 | 1.785 | 0.810 |
|  |  | Avg | 0.159 | 0.248 | 0.451 | 0.247 | 0.215 | 0.217 | 0.095 | 0.199 | 0.237 | 0.260 | 0.374 | 0.377 | 0.055 | 0.125 | 1.764 | 0.802 |
| Linear | Linear | H1 | 0.147 | 0.244 | 0.564 | 0.308 | 0.207 | 0.213 | 0.069 | 0.173 | 0.149 | 0.187 | 0.307 | 0.339 | 0.038 | 0.101 | 2.015 | 0.828 |
|  |  | H2 | 0.168 | 0.262 | 0.585 | 0.311 | 0.246 | 0.238 | 0.090 | 0.197 | 0.200 | 0.236 | 0.358 | 0.365 | 0.051 | 0.123 | 1.938 | 0.865 |
|  |  | H3 | 0.184 | 0.279 | 0.607 | 0.319 | 0.265 | 0.251 | 0.135 | 0.241 | 0.264 | 0.286 | 0.394 | 0.390 | 0.064 | 0.143 | 1.856 | 0.810 |
|  |  | H4 | 0.213 | 0.304 | 0.668 | 0.342 | 0.268 | 0.250 | 0.189 | 0.292 | 0.338 | 0.334 | 0.451 | 0.425 | 0.075 | 0.160 | 1.942 | 0.847 |
|  |  | Avg | 0.178 | 0.272 | 0.606 | 0.320 | 0.246 | 0.238 | 0.121 | 0.226 | 0.238 | 0.261 | 0.377 | 0.380 | 0.057 | 0.132 | 1.938 | 0.837 |
| w/o | Linear | H1 | 0.147 | 0.230 | 0.441 | 0.241 | 0.201 | 0.208 | 0.069 | 0.170 | 0.164 | 0.199 | 0.304 | 0.336 | 0.035 | 0.091 | 1.767 | 0.805 |
|  |  | H2 | 0.165 | 0.246 | 0.471 | 0.251 | 0.236 | 0.229 | 0.099 | 0.200 | 0.210 | 0.243 | 0.362 | 0.366 | 0.049 | 0.116 | 1.905 | 0.826 |
|  |  | H3 | 0.181 | 0.263 | 0.486 | 0.259 | 0.260 | 0.244 | 0.168 | 0.254 | 0.264 | 0.283 | 0.392 | 0.388 | 0.061 | 0.136 | 1.909 | 0.826 |
|  |  | H4 | 0.219 | 0.297 | 0.529 | 0.278 | 0.267 | 0.248 | 0.254 | 0.314 | 0.349 | 0.339 | 0.456 | 0.428 | 0.073 | 0.153 | 1.876 | 0.834 |
|  |  | Avg | 0.178 | 0.259 | 0.482 | 0.257 | 0.241 | 0.232 | 0.147 | 0.234 | 0.247 | 0.266 | 0.378 | 0.379 | 0.054 | 0.124 | 1.864 | 0.823 |
| NormLin | NormLin | H1 | 0.138 | 0.229 | 0.427 | 0.260 | 0.205 | 0.213 | 0.060 | 0.160 | 0.152 | 0.190 | 0.312 | 0.342 | 0.035 | 0.093 | 2.093 | 0.850 |
|  |  | H2 | 0.155 | 0.243 | 0.447 | 0.269 | 0.245 | 0.235 | 0.079 | 0.184 | 0.202 | 0.238 | 0.361 | 0.368 | 0.049 | 0.118 | 1.818 | 0.822 |
|  |  | H3 | 0.173 | 0.261 | 0.466 | 0.276 | 0.277 | 0.255 | 0.120 | 0.227 | 0.259 | 0.280 | 0.393 | 0.388 | 0.062 | 0.138 | 1.974 | 0.842 |
|  |  | H4 | 0.212 | 0.294 | 0.500 | 0.294 | 0.283 | 0.258 | 0.189 | 0.286 | 0.342 | 0.335 | 0.457 | 0.430 | 0.073 | 0.154 | 1.905 | 0.832 |
|  |  | Avg | 0.169 | 0.257 | 0.460 | 0.275 | 0.252 | 0.240 | 0.112 | 0.214 | 0.239 | 0.261 | 0.381 | 0.382 | 0.055 | 0.126 | 1.947 | 0.836 |
| Linear | NormLin | H1 | 0.156 | 0.252 | 0.557 | 0.329 | 0.214 | 0.226 | 0.072 | 0.178 | 0.155 | 0.191 | 0.313 | 0.345 | 0.038 | 0.101 | 2.135 | 0.885 |
|  |  | H2 | 0.171 | 0.264 | 0.555 | 0.333 | 0.255 | 0.250 | 0.100 | 0.209 | 0.202 | 0.238 | 0.364 | 0.369 | 0.051 | 0.122 | 1.986 | 0.865 |
|  |  | H3 | 0.187 | 0.282 | 0.567 | 0.336 | 0.287 | 0.270 | 0.154 | 0.264 | 0.263 | 0.283 | 0.396 | 0.392 | 0.064 | 0.144 | 1.968 | 0.848 |
|  |  | H4 | 0.220 | 0.306 | 0.632 | 0.359 | 0.294 | 0.272 | 0.245 | 0.334 | 0.340 | 0.334 | 0.461 | 0.431 | 0.077 | 0.164 | 2.059 | 0.870 |
|  |  | Avg | 0.183 | 0.276 | 0.578 | 0.339 | 0.262 | 0.254 | 0.143 | 0.246 | 0.240 | 0.261 | 0.383 | 0.384 | 0.057 | 0.133 | 2.037 | 0.867 |
| w/o | NormLin | H1 | 0.158 | 0.242 | 0.474 | 0.282 | 0.237 | 0.237 | 0.073 | 0.178 | 0.163 | 0.199 | 0.314 | 0.346 | 0.035 | 0.092 | 2.154 | 0.886 |
|  |  | H2 | 0.168 | 0.251 | 0.479 | 0.284 | 0.273 | 0.257 | 0.111 | 0.218 | 0.210 | 0.242 | 0.365 | 0.370 | 0.049 | 0.117 | 2.074 | 0.870 |
|  |  | H3 | 0.185 | 0.268 | 0.493 | 0.289 | 0.310 | 0.277 | 0.202 | 0.293 | 0.264 | 0.282 | 0.397 | 0.392 | 0.062 | 0.138 | 2.105 | 0.874 |
|  |  | H4 | 0.228 | 0.304 | 0.526 | 0.307 | 0.313 | 0.276 | 0.344 | 0.387 | 0.346 | 0.336 | 0.462 | 0.433 | 0.073 | 0.154 | 2.039 | 0.866 |
|  |  | Avg | 0.185 | 0.266 | 0.493 | 0.290 | 0.283 | 0.262 | 0.182 | 0.269 | 0.246 | 0.265 | 0.384 | 0.385 | 0.055 | 0.125 | 2.093 | 0.874 |
| NormLin | w/o | H1 | 0.138 | 0.228 | 0.427 | 0.260 | 0.207 | 0.214 | 0.060 | 0.160 | 0.151 | 0.189 | 0.311 | 0.342 | 0.036 | 0.093 | 2.117 | 0.852 |
|  |  | H2 | 0.155 | 0.243 | 0.446 | 0.269 | 0.246 | 0.236 | 0.079 | 0.184 | 0.202 | 0.238 | 0.360 | 0.367 | 0.049 | 0.118 | 1.767 | 0.812 |
|  |  | H3 | 0.171 | 0.260 | 0.467 | 0.277 | 0.280 | 0.257 | 0.121 | 0.227 | 0.260 | 0.282 | 0.392 | 0.388 | 0.063 | 0.139 | 1.960 | 0.846 |
|  |  | H4 | 0.214 | 0.296 | 0.500 | 0.295 | 0.281 | 0.258 | 0.195 | 0.291 | 0.343 | 0.336 | 0.459 | 0.431 | 0.073 | 0.155 | 1.916 | 0.837 |
|  |  | Avg | 0.169 | 0.257 | 0.460 | 0.275 | 0.253 | 0.241 | 0.114 | 0.215 | 0.239 | 0.261 | 0.380 | 0.382 | 0.055 | 0.126 | 1.940 | 0.837 |
| Linear | w/o | H1 | 0.155 | 0.252 | 0.575 | 0.331 | 0.216 | 0.228 | 0.073 | 0.179 | 0.152 | 0.190 | 0.315 | 0.346 | 0.037 | 0.099 | 2.190 | 0.891 |
|  |  | H2 | 0.171 | 0.266 | 0.575 | 0.333 | 0.254 | 0.249 | 0.100 | 0.210 | 0.204 | 0.238 | 0.362 | 0.369 | 0.052 | 0.123 | 1.978 | 0.861 |
|  |  | H3 | 0.189 | 0.282 | 0.572 | 0.334 | 0.286 | 0.269 | 0.154 | 0.264 | 0.262 | 0.283 | 0.396 | 0.392 | 0.064 | 0.145 | 2.074 | 0.874 |
|  |  | H4 | 0.217 | 0.306 | 0.642 | 0.368 | 0.292 | 0.270 | 0.242 | 0.332 | 0.342 | 0.337 | 0.462 | 0.431 | 0.076 | 0.162 | 2.051 | 0.872 |
|  |  | Avg | 0.183 | 0.276 | 0.591 | 0.341 | 0.262 | 0.254 | 0.142 | 0.246 | 0.240 | 0.262 | 0.384 | 0.384 | 0.057 | 0.132 | 2.073 | 0.874 |
| Attn. | Linear | H1 | 0.138 | 0.226 | 0.407 | 0.228 | 0.185 | 0.194 | 0.060 | 0.160 | 0.155 | 0.192 | 0.317 | 0.346 | 0.036 | 0.094 | 2.404 | 0.888 |
|  |  | H2 | 0.157 | 0.245 | 0.445 | 0.243 | 0.218 | 0.218 | 0.075 | 0.179 | 0.206 | 0.240 | 0.369 | 0.375 | 0.050 | 0.118 | 1.979 | 0.857 |
|  |  | H3 | 0.180 | 0.267 | 0.461 | 0.252 | 0.235 | 0.234 | 0.105 | 0.212 | 0.267 | 0.288 | 0.416 | 0.403 | 0.063 | 0.138 | 1.895 | 0.827 |
|  |  | H4 | 0.190 | 0.281 | 0.516 | 0.282 | 0.240 | 0.237 | 0.149 | 0.256 | 0.350 | 0.339 | 0.465 | 0.433 | 0.074 | 0.155 | 1.811 | 0.818 |
|  |  | Avg | 0.166 | 0.255 | 0.457 | 0.251 | 0.220 | 0.221 | 0.097 | 0.202 | 0.244 | 0.265 | 0.391 | 0.389 | 0.056 | 0.126 | 2.022 | 0.847 |

Table 23: Comparison of NormLin with the attention variants using the OLinear architecture. This is a complete version of Table 8.

| Model | OLinear (Ours) | | Trans. 2017 | | Reformer 2020 | | Flowformer 2022 | | Flashformer 2022 | | FLatten 2023 | | Mamba 2023 | | Lin.Attn. 2024 | | Enh.Attn. 2025 | |
|---|---|---|---|---|---|---|---|---|---|---|---|---|---|---|---|---|---|---|
| Metric | MSE | MAE | MSE | MAE | MSE | MAE | MSE | MAE | MSE | MAE | MSE | MAE | MSE | MAE | MSE | MAE | MSE | MAE |
| **ECL** 96 | 0.131 | 0.221 | 0.138 | 0.226 | 0.134 | 0.223 | 0.135 | 0.225 | 0.138 | 0.227 | 0.135 | 0.225 | 0.142 | 0.231 | 0.134 | 0.222 | 0.131 | 0.221 |
| 192 | 0.150 | 0.238 | 0.157 | 0.245 | 0.153 | 0.240 | 0.157 | 0.245 | 0.155 | 0.243 | 0.156 | 0.245 | 0.164 | 0.250 | 0.155 | 0.243 | 0.154 | 0.242 |
| 336 | 0.165 | 0.254 | 0.180 | 0.267 | 0.169 | 0.257 | 0.171 | 0.261 | 0.177 | 0.266 | 0.172 | 0.262 | 0.180 | 0.267 | 0.174 | 0.261 | 0.166 | 0.255 |
| 720 | 0.191 | 0.279 | 0.190 | 0.281 | 0.211 | 0.293 | 0.196 | 0.283 | 0.197 | 0.285 | 0.194 | 0.283 | 0.219 | 0.302 | 0.190 | 0.279 | 0.187 | 0.277 |
| Avg | 0.159 | 0.248 | 0.166 | 0.255 | 0.167 | 0.253 | 0.165 | 0.253 | 0.167 | 0.255 | 0.164 | 0.254 | 0.176 | 0.262 | 0.163 | 0.251 | 0.159 | 0.249 |
| **Traffic** 96 | 0.398 | 0.226 | 0.407 | 0.228 | 0.413 | 0.226 | 0.411 | 0.229 | 0.407 | 0.229 | 0.417 | 0.229 | 0.420 | 0.250 | 0.409 | 0.228 | 0.403 | 0.229 |
| 192 | 0.439 | 0.241 | 0.445 | 0.243 | 0.440 | 0.241 | 0.446 | 0.246 | 0.441 | 0.243 | 0.441 | 0.242 | 0.445 | 0.263 | 0.441 | 0.243 | 0.424 | 0.243 |
| 336 | 0.464 | 0.250 | 0.461 | 0.252 | 0.470 | 0.250 | 0.468 | 0.255 | 0.464 | 0.250 | 0.474 | 0.252 | 0.463 | 0.271 | 0.469 | 0.252 | 0.447 | 0.252 |
| 720 | 0.502 | 0.270 | 0.516 | 0.282 | 0.514 | 0.270 | 0.516 | 0.275 | 0.528 | 0.283 | 0.526 | 0.273 | 0.496 | 0.290 | 0.513 | 0.274 | 0.482 | 0.271 |
| Avg | 0.451 | 0.247 | 0.457 | 0.251 | 0.459 | 0.247 | 0.460 | 0.251 | 0.460 | 0.251 | 0.464 | 0.249 | 0.456 | 0.269 | 0.458 | 0.249 | 0.439 | 0.249 |
| **PEMS03** 12 | 0.060 | 0.159 | 0.060 | 0.160 | 0.060 | 0.159 | 0.060 | 0.160 | 0.061 | 0.160 | 0.060 | 0.160 | 0.062 | 0.161 | 0.062 | 0.162 | 0.061 | 0.160 |
| 24 | 0.078 | 0.179 | 0.075 | 0.179 | 0.077 | 0.178 | 0.076 | 0.179 | 0.075 | 0.178 | 0.077 | 0.181 | 0.078 | 0.181 | 0.078 | 0.182 | 0.078 | 0.181 |
| 48 | 0.104 | 0.210 | 0.105 | 0.212 | 0.103 | 0.208 | 0.109 | 0.214 | 0.104 | 0.210 | 0.112 | 0.216 | 0.115 | 0.220 | 0.111 | 0.215 | 0.108 | 0.213 |
| 96 | 0.140 | 0.247 | 0.149 | 0.256 | 0.144 | 0.248 | 0.149 | 0.255 | 0.147 | 0.255 | 0.155 | 0.259 | 0.163 | 0.266 | 0.156 | 0.261 | 0.142 | 0.251 |
| Avg | 0.095 | 0.199 | 0.097 | 0.202 | 0.096 | 0.198 | 0.099 | 0.202 | 0.097 | 0.201 | 0.101 | 0.204 | 0.104 | 0.207 | 0.102 | 0.205 | 0.097 | 0.201 |
| **Weather** 96 | 0.153 | 0.190 | 0.155 | 0.192 | 0.153 | 0.190 | 0.156 | 0.191 | 0.157 | 0.194 | 0.156 | 0.193 | 0.155 | 0.193 | 0.156 | 0.194 | 0.155 | 0.191 |
| 192 | 0.200 | 0.235 | 0.206 | 0.240 | 0.204 | 0.237 | 0.209 | 0.242 | 0.205 | 0.240 | 0.211 | 0.245 | 0.205 | 0.241 | 0.211 | 0.245 | 0.202 | 0.239 |
| 336 | 0.258 | 0.280 | 0.267 | 0.288 | 0.267 | 0.285 | 0.264 | 0.284 | 0.264 | 0.285 | 0.268 | 0.288 | 0.263 | 0.285 | 0.266 | 0.285 | 0.267 | 0.282 |
| 720 | 0.337 | 0.333 | 0.350 | 0.339 | 0.340 | 0.332 | 0.341 | 0.336 | 0.341 | 0.337 | 0.349 | 0.340 | 0.345 | 0.339 | 0.361 | 0.347 | 0.341 | 0.336 |
| Avg | 0.237 | 0.260 | 0.244 | 0.265 | 0.241 | 0.261 | 0.242 | 0.263 | 0.242 | 0.264 | 0.246 | 0.266 | 0.242 | 0.264 | 0.248 | 0.267 | 0.241 | 0.262 |
| **PEMS08** 12 | 0.068 | 0.159 | 0.070 | 0.164 | 0.069 | 0.157 | 0.068 | 0.161 | 0.070 | 0.164 | 0.071 | 0.164 | 0.072 | 0.167 | 0.070 | 0.164 | 0.068 | 0.159 |
| 24 | 0.089 | 0.178 | 0.095 | 0.187 | 0.087 | 0.175 | 0.090 | 0.181 | 0.096 | 0.188 | 0.094 | 0.185 | 0.095 | 0.190 | 0.094 | 0.184 | 0.084 | 0.178 |
| 48 | 0.123 | 0.204 | 0.130 | 0.215 | 0.118 | 0.201 | 0.130 | 0.212 | 0.128 | 0.216 | 0.141 | 0.220 | 0.139 | 0.228 | 0.133 | 0.217 | 0.117 | 0.204 |
| 96 | 0.173 | 0.236 | 0.207 | 0.265 | 0.174 | 0.234 | 0.195 | 0.252 | 0.206 | 0.263 | 0.213 | 0.260 | 0.226 | 0.276 | 0.192 | 0.255 | 0.183 | 0.240 |
| Avg | 0.113 | 0.194 | 0.125 | 0.208 | 0.112 | 0.192 | 0.121 | 0.201 | 0.125 | 0.208 | 0.130 | 0.207 | 0.133 | 0.215 | 0.122 | 0.205 | 0.113 | 0.195 |
| **Solar-Energy** 96 | 0.179 | 0.191 | 0.192 | 0.198 | 0.179 | 0.192 | 0.180 | 0.190 | 0.193 | 0.198 | 0.200 | 0.198 | 0.188 | 0.202 | 0.193 | 0.195 | 0.180 | 0.192 |
| 192 | 0.209 | 0.213 | 0.220 | 0.219 | 0.211 | 0.214 | 0.218 | 0.220 | 0.215 | 0.215 | 0.230 | 0.222 | 0.221 | 0.224 | 0.230 | 0.223 | 0.211 | 0.213 |
| 336 | 0.231 | 0.229 | 0.237 | 0.236 | 0.232 | 0.230 | 0.237 | 0.234 | 0.240 | 0.237 | 0.246 | 0.238 | 0.247 | 0.241 | 0.244 | 0.237 | 0.234 | 0.230 |
| 720 | 0.241 | 0.236 | 0.243 | 0.240 | 0.243 | 0.236 | 0.252 | 0.242 | 0.246 | 0.241 | 0.251 | 0.242 | 0.257 | 0.245 | 0.249 | 0.242 | 0.244 | 0.236 |
| Avg | 0.215 | 0.217 | 0.223 | 0.223 | 0.216 | 0.218 | 0.222 | 0.221 | 0.223 | 0.223 | 0.231 | 0.225 | 0.228 | 0.228 | 0.229 | 0.224 | 0.217 | 0.218 |
| **ILI** 24 | 1.737 | 0.800 | 2.404 | 0.888 | 1.776 | 0.805 | 2.015 | 0.842 | 2.565 | 0.895 | 2.723 | 0.909 | 1.867 | 0.795 | 2.580 | 0.903 | 1.846 | 0.824 |
| 36 | 1.714 | 0.795 | 1.979 | 0.857 | 1.769 | 0.806 | 1.820 | 0.832 | 1.973 | 0.856 | 1.944 | 0.864 | 2.002 | 0.846 | 1.832 | 0.835 | 1.794 | 0.829 |
| 48 | 1.821 | 0.804 | 1.895 | 0.827 | 1.911 | 0.820 | 1.867 | 0.824 | 1.891 | 0.831 | 1.884 | 0.839 | 1.996 | 0.837 | 1.854 | 0.829 | 1.890 | 0.824 |
| 60 | 1.785 | 0.810 | 1.811 | 0.818 | 1.827 | 0.814 | 1.823 | 0.815 | 1.837 | 0.816 | 1.987 | 0.872 | 1.935 | 0.835 | 1.972 | 0.868 | 1.982 | 0.868 |
| Avg | 1.764 | 0.802 | 2.022 | 0.847 | 1.821 | 0.811 | 1.881 | 0.828 | 2.066 | 0.849 | 2.134 | 0.871 | 1.950 | 0.828 | 2.059 | 0.859 | 1.878 | 0.836 |
| **NASDAQ** 3 | 0.036 | 0.092 | 0.036 | 0.094 | 0.035 | 0.092 | 0.036 | 0.093 | 0.036 | 0.094 | 0.037 | 0.096 | 0.036 | 0.093 | 0.036 | 0.094 | 0.036 | 0.093 |
| 6 | 0.049 | 0.117 | 0.050 | 0.118 | 0.049 | 0.118 | 0.049 | 0.117 | 0.050 | 0.119 | 0.052 | 0.122 | 0.050 | 0.118 | 0.051 | 0.119 | 0.049 | 0.118 |
| 9 | 0.062 | 0.137 | 0.063 | 0.138 | 0.062 | 0.137 | 0.062 | 0.137 | 0.063 | 0.138 | 0.064 | 0.140 | 0.062 | 0.138 | 0.062 | 0.139 | 0.062 | 0.138 |
| 12 | 0.073 | 0.154 | 0.074 | 0.155 | 0.074 | 0.155 | 0.073 | 0.154 | 0.074 | 0.155 | 0.075 | 0.157 | 0.074 | 0.155 | 0.074 | 0.156 | 0.074 | 0.155 |
| Avg | 0.055 | 0.125 | 0.056 | 0.126 | 0.055 | 0.125 | 0.055 | 0.125 | 0.056 | 0.126 | 0.057 | 0.129 | 0.055 | 0.126 | 0.056 | 0.127 | 0.055 | 0.126 |
| 1st Count | 27 | 26 | 3 | 0 | 10 | 21 | 6 | 5 | 1 | 2 | 1 | 0 | 2 | 1 | 1 | 0 | 15 | 5 |

Table 24: Applying the NormLin module to Transformer-based forecasters: iTransformer, PatchTST, Leddam, and Fredformer. For a fair comparison, model hyperparameters, loss functions, and training strategies are kept unchanged. In the case of Leddam, only the *cross-channel attention* module is updated. This table presents the complete results corresponding to Table 9 in the main text.

| | | iTransformer 2024a | | | | PatchTST 2023 | | | | Leddam 2024 | | | | Fredformer 2024 | | | |
| | | Vanilla | | + NormLin | | Vanilla | | + NormLin | | Vanilla | | + NormLin | | Vanilla | | + NormLin | |
| Model | Metric | MSE | MAE | MSE | MAE | MSE | MAE | MSE | MAE | MSE | MAE | MSE | MAE | MSE | MAE | MSE | MAE |
|---|---|---|---|---|---|---|---|---|---|---|---|---|---|---|---|---|---|
| ETTm1 | 96 | 0.334 | 0.368 | **0.320** | **0.359** | 0.329 | 0.367 | **0.323** | **0.361** | **0.319** | 0.359 | **0.319** | **0.357** | 0.326 | 0.361 | **0.318** | **0.356** |
| | 192 | 0.377 | 0.391 | **0.365** | **0.382** | 0.367 | 0.385 | **0.357** | **0.384** | 0.369 | 0.383 | **0.360** | **0.380** | 0.363 | **0.380** | **0.362** | 0.383 |
| | 336 | 0.426 | 0.420 | **0.400** | **0.405** | 0.399 | 0.410 | **0.390** | **0.408** | 0.394 | 0.402 | **0.391** | **0.401** | 0.395 | **0.403** | **0.391** | 0.404 |
| | 720 | 0.491 | 0.459 | **0.467** | **0.443** | 0.454 | **0.439** | **0.448** | 0.443 | 0.460 | 0.442 | **0.455** | **0.439** | 0.453 | **0.438** | 0.454 | 0.440 |
| | Avg | 0.407 | 0.410 | **0.388** | **0.397** | 0.387 | 0.400 | **0.379** | **0.399** | 0.386 | 0.397 | **0.381** | **0.394** | 0.384 | **0.396** | **0.381** | 0.396 |
| ECL | 96 | 0.148 | 0.240 | **0.138** | **0.232** | 0.161 | 0.250 | **0.155** | **0.246** | 0.141 | 0.235 | **0.139** | **0.234** | 0.147 | 0.241 | **0.138** | **0.235** |
| | 192 | 0.162 | 0.253 | **0.154** | **0.248** | 0.199 | 0.289 | **0.167** | **0.256** | 0.159 | 0.252 | **0.155** | **0.248** | 0.165 | 0.258 | **0.155** | **0.251** |
| | 336 | 0.178 | 0.269 | **0.171** | **0.265** | 0.215 | 0.305 | **0.183** | **0.273** | 0.173 | 0.268 | **0.171** | **0.265** | 0.177 | 0.273 | **0.172** | **0.271** |
| | 720 | 0.225 | 0.317 | **0.201** | **0.297** | 0.256 | 0.337 | **0.221** | **0.307** | 0.201 | 0.295 | **0.195** | **0.293** | 0.213 | **0.304** | **0.210** | **0.304** |
| | Avg | 0.178 | 0.270 | **0.166** | **0.260** | 0.208 | 0.295 | **0.181** | **0.270** | 0.169 | 0.263 | **0.165** | **0.260** | 0.176 | 0.269 | **0.169** | **0.265** |
| PEMS03 | 12 | 0.071 | 0.174 | **0.062** | **0.164** | 0.099 | 0.216 | **0.071** | **0.176** | **0.063** | **0.164** | 0.063 | 0.164 | 0.068 | 0.174 | **0.064** | **0.168** |
| | 24 | 0.093 | 0.201 | **0.078** | **0.184** | 0.142 | 0.259 | **0.103** | **0.211** | 0.080 | 0.185 | **0.079** | **0.182** | 0.093 | 0.202 | **0.081** | **0.188** |
| | 48 | 0.125 | 0.236 | **0.110** | **0.219** | 0.211 | 0.319 | **0.160** | **0.248** | 0.124 | 0.226 | **0.110** | **0.217** | 0.146 | 0.258 | **0.119** | **0.228** |
| | 96 | 0.164 | 0.275 | **0.159** | **0.269** | 0.269 | 0.370 | **0.251** | **0.331** | 0.160 | **0.266** | 0.159 | 0.266 | 0.228 | 0.330 | **0.170** | **0.284** |
| | Avg | 0.113 | 0.221 | **0.102** | **0.209** | 0.180 | 0.291 | **0.146** | **0.241** | 0.107 | 0.210 | **0.103** | **0.207** | 0.134 | 0.241 | **0.108** | **0.217** |
| PEMS07 | 12 | 0.067 | 0.165 | **0.055** | **0.145** | 0.095 | 0.207 | **0.084** | **0.199** | 0.055 | **0.145** | 0.054 | 0.145 | 0.063 | 0.158 | **0.057** | **0.150** |
| | 24 | 0.088 | 0.190 | **0.070** | **0.162** | 0.150 | 0.262 | **0.098** | **0.198** | 0.070 | 0.164 | **0.067** | **0.160** | 0.089 | 0.190 | **0.073** | **0.168** |
| | 48 | 0.110 | 0.215 | **0.091** | **0.187** | 0.253 | **0.340** | **0.243** | 0.341 | 0.094 | 0.192 | **0.087** | **0.184** | 0.135 | 0.239 | **0.099** | **0.201** |
| | 96 | 0.139 | 0.245 | **0.128** | **0.220** | 0.346 | 0.404 | **0.249** | **0.310** | **0.117** | **0.217** | 0.119 | 0.222 | 0.196 | 0.294 | **0.157** | **0.267** |
| | Avg | 0.101 | 0.204 | **0.086** | **0.178** | 0.211 | 0.303 | **0.168** | **0.262** | 0.084 | 0.180 | **0.082** | **0.178** | 0.121 | 0.220 | **0.096** | **0.196** |
| Solar-Energy | 96 | 0.203 | 0.237 | **0.194** | **0.233** | 0.234 | 0.286 | **0.208** | **0.249** | 0.197 | 0.241 | **0.186** | **0.227** | **0.185** | 0.233 | 0.190 | **0.231** |
| | 192 | 0.233 | 0.261 | **0.225** | **0.258** | 0.267 | 0.310 | **0.234** | **0.266** | 0.231 | 0.264 | **0.221** | **0.258** | 0.227 | **0.253** | **0.222** | 0.254 |
| | 336 | 0.248 | **0.273** | **0.241** | **0.273** | 0.290 | 0.315 | **0.252** | **0.278** | **0.241** | 0.268 | 0.242 | 0.270 | 0.246 | 0.284 | **0.245** | **0.275** |
| | 720 | 0.249 | **0.275** | **0.244** | 0.277 | 0.289 | 0.317 | **0.253** | **0.277** | 0.250 | 0.281 | **0.239** | **0.271** | **0.247** | **0.276** | 0.249 | 0.281 |
| | Avg | 0.233 | 0.262 | **0.226** | **0.260** | 0.270 | 0.307 | **0.237** | **0.267** | 0.230 | 0.264 | **0.222** | **0.257** | **0.226** | 0.262 | **0.226** | **0.260** |
| Weather | 96 | 0.174 | 0.214 | **0.160** | **0.205** | 0.177 | 0.218 | **0.161** | **0.206** | **0.156** | 0.202 | **0.156** | **0.201** | 0.163 | 0.207 | **0.156** | **0.202** |
| | 192 | 0.221 | 0.254 | **0.209** | **0.250** | 0.225 | 0.259 | **0.207** | **0.249** | 0.207 | 0.250 | **0.206** | **0.249** | 0.211 | 0.251 | **0.205** | **0.248** |
| | 336 | 0.278 | 0.296 | **0.268** | **0.294** | 0.278 | 0.297 | **0.266** | **0.291** | 0.262 | 0.291 | **0.262** | **0.291** | 0.267 | 0.292 | **0.261** | **0.289** |
| | 720 | 0.358 | 0.349 | **0.344** | **0.344** | 0.354 | 0.348 | **0.345** | **0.344** | 0.343 | 0.343 | 0.345 | 0.344 | 0.343 | **0.341** | **0.340** | **0.341** |
| | Avg | 0.258 | 0.279 | **0.245** | **0.273** | 0.259 | 0.281 | **0.245** | **0.272** | 0.242 | 0.272 | **0.242** | **0.271** | 0.246 | 0.273 | **0.240** | **0.270** |
| METR-LA | 3 | 0.205 | **0.188** | **0.202** | 0.189 | **0.204** | 0.190 | 0.205 | **0.188** | 0.204 | 0.191 | **0.202** | **0.189** | 0.205 | 0.188 | **0.200** | **0.186** |
| | 6 | 0.300 | 0.229 | **0.293** | **0.226** | 0.298 | **0.227** | 0.297 | 0.229 | 0.293 | 0.227 | **0.289** | **0.226** | 0.298 | **0.227** | **0.296** | **0.227** |
| | 9 | 0.386 | 0.265 | **0.375** | **0.259** | 0.382 | 0.263 | **0.379** | **0.262** | 0.369 | 0.264 | **0.366** | **0.259** | 0.385 | **0.263** | **0.369** | **0.263** |
| | 12 | 0.460 | 0.295 | **0.442** | **0.289** | 0.456 | **0.292** | 0.453 | 0.293 | 0.442 | 0.292 | **0.423** | **0.287** | 0.457 | 0.292 | **0.451** | **0.291** |
| | Avg | 0.338 | 0.244 | **0.328** | **0.241** | 0.335 | **0.243** | 0.333 | 0.243 | 0.327 | 0.243 | **0.320** | **0.240** | 0.336 | **0.242** | **0.329** | **0.242** |

Table 25: Full evaluation of OLinear-C. *S1* and *S2* correspond to 'Input-12, Predict-$\{3, 6, 9, 12\}$' and 'Input-36, Predict-$\{24, 36, 48, 60\}$', respectively. This table presents the full results of Table 7.

| | Dataset | ETTm1 | | ETTm2 | | ETTh1 | | ETTh2 | | ECL | | Exchange | | Traffic | |
|---|---|---|---|---|---|---|---|---|---|---|---|---|---|---|---|
| | Metric | MSE | MAE | MSE | MAE | MSE | MAE | MSE | MAE | MSE | MAE | MSE | MAE | MSE | MAE |
| OLinear | 96 | 0.302 | 0.334 | 0.169 | 0.249 | 0.360 | 0.382 | 0.284 | 0.329 | 0.131 | 0.221 | 0.082 | 0.200 | 0.398 | 0.226 |
| | 192 | 0.357 | 0.363 | 0.232 | 0.290 | 0.416 | 0.414 | 0.360 | 0.379 | 0.150 | 0.238 | 0.171 | 0.293 | 0.439 | 0.241 |
| | 336 | 0.387 | 0.385 | 0.291 | 0.328 | 0.457 | 0.438 | 0.409 | 0.415 | 0.165 | 0.254 | 0.331 | 0.414 | 0.464 | 0.250 |
| | 720 | 0.452 | 0.426 | 0.389 | 0.387 | 0.463 | 0.462 | 0.415 | 0.431 | 0.191 | 0.279 | 0.837 | 0.688 | 0.502 | 0.270 |
| | Avg | **0.374** | **0.377** | **0.270** | **0.313** | **0.424** | **0.424** | **0.367** | **0.388** | **0.159** | **0.248** | **0.355** | 0.399 | **0.451** | **0.247** |
| OLinear-C | 96 | 0.303 | 0.335 | 0.169 | 0.249 | 0.362 | 0.383 | 0.285 | 0.329 | 0.130 | 0.220 | 0.081 | 0.198 | 0.404 | 0.227 |
| | 192 | 0.357 | 0.364 | 0.232 | 0.290 | 0.415 | 0.414 | 0.358 | 0.377 | 0.156 | 0.243 | 0.171 | 0.293 | 0.435 | 0.241 |
| | 336 | 0.389 | 0.386 | 0.291 | 0.328 | 0.460 | 0.440 | 0.407 | 0.415 | 0.167 | 0.256 | 0.325 | 0.410 | 0.465 | 0.250 |
| | 720 | 0.452 | 0.426 | 0.389 | 0.387 | 0.461 | 0.460 | 0.423 | 0.436 | 0.192 | 0.279 | 0.845 | 0.692 | 0.502 | 0.271 |
| | Avg | 0.375 | 0.378 | **0.270** | **0.313** | **0.424** | **0.424** | 0.368 | 0.389 | 0.161 | 0.249 | **0.355** | **0.398** | **0.451** | **0.247** |

| | Dataset | Weather | | Solar-Energy | | PEMS03 | | PEMS04 | | PEMS07 | | PEMS08 | | ILI (S1) | |
|---|---|---|---|---|---|---|---|---|---|---|---|---|---|---|---|
| | Metric | MSE | MAE | MSE | MAE | MSE | MAE | MSE | MAE | MSE | MAE | MSE | MAE | MSE | MAE |
| OLinear | H1 | 0.153 | 0.190 | 0.179 | 0.191 | 0.060 | 0.159 | 0.068 | 0.163 | 0.052 | 0.138 | 0.068 | 0.159 | 0.468 | 0.349 |
| | H2 | 0.200 | 0.235 | 0.209 | 0.213 | 0.078 | 0.179 | 0.079 | 0.176 | 0.065 | 0.151 | 0.089 | 0.178 | 0.923 | 0.516 |
| | H3 | 0.258 | 0.280 | 0.231 | 0.229 | 0.104 | 0.210 | 0.095 | 0.197 | 0.084 | 0.171 | 0.123 | 0.204 | 1.289 | 0.655 |
| | H4 | 0.337 | 0.333 | 0.241 | 0.236 | 0.140 | 0.247 | 0.122 | 0.226 | 0.108 | 0.196 | 0.173 | 0.236 | 1.698 | 0.791 |
| | Avg | **0.237** | 0.260 | **0.215** | **0.217** | **0.095** | **0.199** | **0.091** | **0.190** | **0.077** | **0.164** | **0.113** | **0.194** | 1.094 | 0.578 |
| OLinear-C | H1 | 0.152 | 0.189 | 0.178 | 0.191 | 0.061 | 0.160 | 0.068 | 0.163 | 0.052 | 0.138 | 0.068 | 0.159 | 0.485 | 0.352 |
| | H2 | 0.202 | 0.237 | 0.209 | 0.213 | 0.076 | 0.178 | 0.078 | 0.176 | 0.064 | 0.151 | 0.089 | 0.177 | 0.951 | 0.523 |
| | H3 | 0.258 | 0.279 | 0.231 | 0.229 | 0.105 | 0.211 | 0.097 | 0.197 | 0.083 | 0.171 | 0.120 | 0.203 | 1.336 | 0.667 |
| | H4 | 0.340 | 0.333 | 0.241 | 0.236 | 0.142 | 0.248 | 0.121 | 0.227 | 0.108 | 0.195 | 0.177 | 0.237 | 1.693 | 0.790 |
| | Avg | 0.238 | **0.259** | **0.215** | **0.217** | 0.096 | **0.199** | **0.091** | 0.191 | **0.077** | **0.164** | **0.113** | **0.194** | 1.116 | 0.583 |

| | Dataset | ILI (S2) | | COVID-19 (S1) | | COVID-19 (S2) | | METR-LA (S1) | | METR-LA (S2) | | NASDAQ (S1) | | NASDAQ (S2) | |
|---|---|---|---|---|---|---|---|---|---|---|---|---|---|---|---|
| | Metric | MSE | MAE | MSE | MAE | MSE | MAE | MSE | MAE | MSE | MAE | MSE | MAE | MSE | MAE |
| OLinear | H1 | 1.737 | 0.800 | 1.100 | 0.487 | 4.474 | 1.180 | 0.207 | 0.171 | 0.650 | 0.337 | 0.036 | 0.092 | 0.121 | 0.216 |
| | H2 | 1.714 | 0.795 | 1.750 | 0.619 | 7.241 | 1.670 | 0.301 | 0.207 | 0.800 | 0.388 | 0.049 | 0.117 | 0.163 | 0.261 |
| | H3 | 1.821 | 0.804 | 2.239 | 0.734 | 10.076 | 1.985 | 0.382 | 0.238 | 0.905 | 0.427 | 0.062 | 0.137 | 0.205 | 0.296 |
| | H4 | 1.785 | 0.810 | 2.538 | 0.831 | 12.079 | 2.182 | 0.452 | 0.263 | 0.999 | 0.457 | 0.073 | 0.154 | 0.259 | 0.336 |
| | Avg | **1.764** | **0.802** | 1.907 | **0.668** | **8.467** | **1.754** | **0.335** | **0.220** | **0.838** | **0.402** | 0.055 | **0.125** | 0.187 | 0.277 |
| OLinear-C | H1 | 1.850 | 0.835 | 1.107 | 0.493 | 4.980 | 1.336 | 0.207 | 0.172 | 0.654 | 0.337 | 0.035 | 0.092 | 0.119 | 0.214 |
| | H2 | 1.749 | 0.801 | 1.693 | 0.630 | 7.605 | 1.719 | 0.301 | 0.207 | 0.804 | 0.388 | 0.049 | 0.117 | 0.162 | 0.259 |
| | H3 | 1.847 | 0.808 | 2.251 | 0.740 | 10.336 | 2.015 | 0.382 | 0.238 | 0.916 | 0.428 | 0.061 | 0.137 | 0.207 | 0.298 |
| | H4 | 1.792 | 0.808 | 2.545 | 0.843 | 12.251 | 2.204 | 0.450 | 0.263 | 1.016 | 0.462 | 0.073 | 0.153 | 0.255 | 0.334 |
| | Avg | 1.809 | 0.813 | **1.899** | 0.676 | 8.793 | 1.819 | **0.335** | **0.220** | 0.847 | 0.404 | **0.054** | **0.125** | **0.186** | **0.276** |

| | Dataset | Wiki (S1) | | Wiki (S2) | | SP500 (S1) | | SP500 (S2) | | DowJones (S1) | | DowJones (S2) | | CarSales (S1) | |
|---|---|---|---|---|---|---|---|---|---|---|---|---|---|---|---|
| | Metric | MSE | MAE | MSE | MAE | MSE | MAE | MSE | MAE | MSE | MAE | MSE | MAE | MSE | MAE |
| OLinear | H1 | 6.161 | 0.368 | 6.894 | 0.423 | 0.035 | 0.126 | 0.155 | 0.271 | 1.550 | 0.276 | 7.432 | 0.664 | 0.303 | 0.277 |
| | H2 | 6.453 | 0.385 | 6.446 | 0.439 | 0.053 | 0.158 | 0.209 | 0.317 | 2.569 | 0.361 | 10.848 | 0.799 | 0.315 | 0.285 |
| | H3 | 6.666 | 0.398 | 6.004 | 0.446 | 0.070 | 0.181 | 0.258 | 0.358 | 3.565 | 0.430 | 14.045 | 0.914 | 0.327 | 0.293 |
| | H4 | 6.834 | 0.406 | 5.705 | 0.454 | 0.088 | 0.204 | 0.305 | 0.387 | 4.517 | 0.490 | 16.959 | 1.017 | 0.336 | 0.301 |
| | Avg | **6.528** | 0.389 | 6.262 | **0.440** | **0.061** | **0.167** | **0.231** | 0.333 | **3.050** | **0.389** | **12.321** | **0.848** | **0.320** | **0.289** |
| OLinear-C | H1 | 6.187 | 0.368 | 6.888 | 0.422 | 0.035 | 0.126 | 0.156 | 0.271 | 1.553 | 0.276 | 7.496 | 0.665 | 0.303 | 0.276 |
| | H2 | 6.457 | 0.384 | 6.437 | 0.438 | 0.054 | 0.159 | 0.210 | 0.319 | 2.570 | 0.361 | 10.965 | 0.802 | 0.317 | 0.287 |
| | H3 | 6.652 | 0.396 | 6.003 | 0.446 | 0.071 | 0.182 | 0.262 | 0.358 | 3.573 | 0.431 | 14.161 | 0.915 | 0.328 | 0.294 |
| | H4 | 6.835 | 0.406 | 5.717 | 0.456 | 0.088 | 0.205 | 0.296 | 0.378 | 4.538 | 0.490 | 17.084 | 1.018 | 0.337 | 0.300 |
| | Avg | 6.533 | **0.388** | **6.261** | **0.440** | 0.062 | 0.168 | **0.231** | **0.331** | 3.058 | **0.389** | 12.427 | 0.850 | 0.321 | **0.289** |

| | Dataset | CarSales (S2) | | Power (S1) | | Power (S2) | | Website (S1) | | Website (S2) | | Unemp (S1) | | Unemp (S2) | |
|---|---|---|---|---|---|---|---|---|---|---|---|---|---|---|---|
| | Metric | MSE | MAE | MSE | MAE | MSE | MAE | MSE | MAE | MSE | MAE | MSE | MAE | MSE | MAE |
| OLinear | H1 | 0.320 | 0.302 | 0.864 | 0.688 | 1.343 | 0.870 | 0.077 | 0.199 | 0.186 | 0.306 | 0.012 | 0.047 | 0.458 | 0.447 |
| | H2 | 0.334 | 0.315 | 0.991 | 0.742 | 1.445 | 0.903 | 0.103 | 0.227 | 0.272 | 0.356 | 0.041 | 0.108 | 0.870 | 0.619 |
| | H3 | 0.347 | 0.327 | 1.062 | 0.770 | 1.559 | 0.946 | 0.135 | 0.251 | 0.365 | 0.391 | 0.084 | 0.170 | 1.643 | 0.900 |
| | H4 | 0.358 | 0.337 | 1.119 | 0.789 | 1.602 | 0.971 | 0.176 | 0.281 | 0.486 | 0.481 | 0.131 | 0.220 | 2.593 | 1.182 |
| | Avg | **0.340** | **0.320** | **1.009** | **0.747** | **1.487** | **0.922** | **0.123** | **0.240** | **0.327** | **0.383** | 0.067 | 0.136 | **1.391** | **0.787** |
| OLinear-C | H1 | 0.320 | 0.301 | 0.873 | 0.696 | 1.460 | 0.922 | 0.092 | 0.222 | 0.176 | 0.304 | 0.012 | 0.047 | 0.437 | 0.434 |
| | H2 | 0.334 | 0.315 | 0.997 | 0.744 | 1.527 | 0.939 | 0.110 | 0.232 | 0.283 | 0.364 | 0.039 | 0.105 | 0.899 | 0.633 |
| | H3 | 0.347 | 0.327 | 1.060 | 0.768 | 1.721 | 1.016 | 0.144 | 0.257 | 0.402 | 0.421 | 0.079 | 0.163 | 1.963 | 1.026 |
| | H4 | 0.358 | 0.337 | 1.128 | 0.796 | 1.869 | 1.077 | 0.188 | 0.286 | 0.517 | 0.490 | 0.132 | 0.222 | 2.571 | 1.162 |
| | Avg | **0.340** | **0.320** | 1.014 | 0.751 | 1.644 | 0.988 | 0.133 | 0.249 | 0.344 | 0.395 | **0.065** | **0.134** | 1.467 | 0.814 |

Table 26: Robustness of OLinear-C performance. Standard deviations are calculated over seven random seeds.

| Dataset | | ECL | | Traffic | | ETTm1 | | Solar-Energy | |
|---|---|---|---|---|---|---|---|---|---|
| Metric | | MSE | MAE | MSE | MAE | MSE | MAE | MSE | MAE |
| Horizon | 96 | 0.130±3e-4 | 0.220±4e-4 | 0.404±4e-3 | 0.227±7e-4 | 0.303±7e-4 | 0.335±7e-4 | 0.178±1e-3 | 0.191±7e-4 |
| | 192 | 0.156±1e-3 | 0.243±9e-4 | 0.435±2e-3 | 0.241±4e-4 | 0.357±2e-4 | 0.364±2e-4 | 0.209±8e-4 | 0.213±3e-4 |
| | 336 | 0.167±1e-3 | 0.256±8e-4 | 0.465±5e-3 | 0.250±2e-4 | 0.389±9e-4 | 0.386±5e-4 | 0.231±5e-4 | 0.229±4e-5 |
| | 720 | 0.192±9e-3 | 0.279±6e-3 | 0.502±5e-3 | 0.271±6e-4 | 0.452±6e-4 | 0.426±4e-4 | 0.241±4e-4 | 0.236±4e-4 |

| Dataset | | Weather | | PEMS03 | | NASDAQ (S1) | | Wiki (S1) | |
|---|---|---|---|---|---|---|---|---|---|
| Metric | | MSE | MAE | MSE | MAE | MSE | MAE | MSE | MAE |
| Horizon | H1 | 0.154±1e-3 | 0.191±8e-4 | 0.061±4e-4 | 0.160±4e-4 | 0.035±2e-4 | 0.092±5e-4 | 6.187±9e-3 | 0.368±7e-4 |
| | H2 | 0.204±3-3 | 0.238±2e-3 | 0.076±8e-4 | 0.178±8e-4 | 0.049±8e-5 | 0.117±4e-4 | 6.457±7e-3 | 0.384±6e-4 |
| | H3 | 0.259±4e-3 | 0.280±4e-3 | 0.105±9e-4 | 0.211±6e-4 | 0.061±0.000 | 0.137±3e-4 | 6.652±4e-3 | 0.396±6e-4 |
| | H4 | 0.343±6e-3 | 0.334±3e-3 | 0.142±2e-3 | 0.248±1e-3 | 0.073±5e-5 | 0.153±2e-4 | 6.835±3e-3 | 0.406±2e-4 |

| Dataset | | DowJones (S2) | | SP500 (S2) | | CarSales (S1) | | Power (S2) | |
|---|---|---|---|---|---|---|---|---|---|
| Metric | | MSE | MAE | MSE | MAE | MSE | MAE | MSE | MAE |
| Horizon | H1 | 7.496±4e-2 | 0.665±9e-4 | 0.156±2e-3 | 0.271±2e-3 | 0.303±1e-3 | 0.276±1e-3 | 1.460±4e-2 | 0.922±1e-2 |
| | H2 | 10.965±7e-2 | 0.802±1e-3 | 0.210±1e-3 | 0.319±1e-3 | 0.317±7e-4 | 0.287±6e-4 | 1.527±3e-2 | 0.939±2e-2 |
| | H3 | 14.161±5e-2 | 0.915±7e-4 | 0.262±2e-3 | 0.358±2e-3 | 0.328±7e-4 | 0.294±6e-4 | 1.721±7e-2 | 1.016±3e-2 |
| | H4 | 17.084±7e-2 | 1.018±1e-3 | 0.296±3e-3 | 0.378±3e-3 | 0.337±4e-4 | 0.300±4e-4 | 1.869±1e-2 | 1.077±5e-3 |

Table 27: Ablation study on different transformations of $\mathrm{CorrMat}_v$

| Dataset | | ECL | | Traffic | | Solar | | PEMS03 | | Weather | | NASDAQ S1 | | ILI S2 | | Web_S1 | | Cars_S2 | |
|---|---|---|---|---|---|---|---|---|---|---|---|---|---|---|---|---|---|---|---|
| Metric | | MSE | MAE | MSE | MAE | MSE | MAE | MSE | MAE | MSE | MAE | MSE | MAE | MSE | MAE | MSE | MAE | MSE | MAE |
| Softmax | H1 | 0.130 | 0.220 | 0.404 | 0.227 | 0.178 | 0.191 | 0.061 | 0.160 | 0.152 | 0.189 | 0.035 | 0.092 | 1.850 | 0.835 | 0.078 | 0.200 | 0.320 | 0.302 |
| | H2 | 0.156 | 0.243 | 0.435 | 0.241 | 0.209 | 0.213 | 0.076 | 0.178 | 0.202 | 0.237 | 0.049 | 0.117 | 1.749 | 0.801 | 0.108 | 0.230 | 0.334 | 0.315 |
| | H3 | 0.167 | 0.256 | 0.465 | 0.250 | 0.231 | 0.229 | 0.105 | 0.211 | 0.258 | 0.279 | 0.061 | 0.137 | 1.847 | 0.808 | 0.143 | 0.256 | 0.347 | 0.327 |
| | H4 | 0.192 | 0.279 | 0.502 | 0.271 | 0.241 | 0.236 | 0.142 | 0.248 | 0.340 | 0.333 | 0.073 | 0.153 | 1.792 | 0.808 | 0.185 | 0.285 | 0.358 | 0.337 |
| | Avg | 0.161 | **0.249** | **0.451** | **0.247** | **0.215** | **0.217** | 0.096 | **0.199** | **0.238** | **0.259** | **0.054** | 0.125 | 1.809 | 0.813 | **0.128** | **0.243** | **0.340** | **0.320** |
| Softplus | H1 | 0.131 | 0.221 | 0.401 | 0.227 | 0.178 | 0.191 | 0.061 | 0.160 | 0.154 | 0.191 | 0.035 | 0.092 | 1.854 | 0.838 | 0.078 | 0.200 | 0.320 | 0.302 |
| | H2 | 0.153 | 0.241 | 0.436 | 0.241 | 0.209 | 0.213 | 0.078 | 0.180 | 0.204 | 0.240 | 0.049 | 0.117 | 1.741 | 0.801 | 0.108 | 0.230 | 0.334 | 0.315 |
| | H3 | 0.166 | 0.255 | 0.461 | 0.249 | 0.231 | 0.229 | 0.106 | 0.211 | 0.262 | 0.281 | 0.061 | 0.137 | 1.843 | 0.807 | 0.143 | 0.256 | 0.347 | 0.327 |
| | H4 | 0.191 | 0.280 | 0.511 | 0.271 | 0.242 | 0.236 | 0.140 | 0.247 | 0.342 | 0.336 | 0.073 | 0.154 | 1.793 | 0.808 | 0.185 | 0.285 | 0.358 | 0.337 |
| | Avg | **0.160** | **0.249** | 0.452 | **0.247** | **0.215** | **0.217** | 0.096 | **0.199** | 0.240 | 0.262 | **0.054** | 0.125 | **1.808** | 0.813 | **0.128** | **0.243** | **0.340** | **0.320** |
| Sigmoid | H1 | 0.131 | 0.221 | 0.398 | 0.227 | 0.178 | 0.192 | 0.060 | 0.159 | 0.152 | 0.189 | 0.036 | 0.093 | 1.727 | 0.796 | 0.078 | 0.200 | 0.320 | 0.302 |
| | H2 | 0.152 | 0.240 | 0.441 | 0.241 | 0.213 | 0.214 | 0.077 | 0.179 | 0.206 | 0.241 | 0.049 | 0.117 | 1.908 | 0.832 | 0.108 | 0.230 | 0.334 | 0.315 |
| | H3 | 0.166 | 0.256 | 0.460 | 0.250 | 0.230 | 0.229 | 0.104 | 0.210 | 0.266 | 0.285 | 0.061 | 0.137 | 1.845 | 0.807 | 0.143 | 0.256 | 0.347 | 0.327 |
| | H4 | 0.191 | 0.280 | 0.509 | 0.270 | 0.241 | 0.236 | 0.140 | 0.247 | 0.345 | 0.337 | 0.073 | 0.154 | 1.795 | 0.809 | 0.185 | 0.285 | 0.358 | 0.337 |
| | Avg | **0.160** | **0.249** | 0.452 | **0.247** | 0.216 | 0.218 | **0.095** | **0.199** | 0.242 | 0.263 | 0.055 | 0.125 | 1.819 | **0.811** | **0.128** | **0.243** | **0.340** | **0.320** |
| ReLU | H1 | 0.132 | 0.222 | 0.403 | 0.227 | 0.179 | 0.192 | 0.060 | 0.159 | 0.154 | 0.190 | 0.035 | 0.092 | 1.844 | 0.832 | 0.078 | 0.201 | 0.320 | 0.302 |
| | H2 | 0.153 | 0.241 | 0.446 | 0.241 | 0.213 | 0.214 | 0.076 | 0.178 | 0.205 | 0.241 | 0.049 | 0.116 | 1.757 | 0.802 | 0.108 | 0.229 | 0.334 | 0.315 |
| | H3 | 0.171 | 0.259 | 0.464 | 0.250 | 0.230 | 0.229 | 0.106 | 0.211 | 0.258 | 0.280 | 0.061 | 0.136 | 1.841 | 0.807 | 0.143 | 0.256 | 0.347 | 0.327 |
| | H4 | 0.193 | 0.282 | 0.519 | 0.271 | 0.242 | 0.236 | 0.141 | 0.248 | 0.343 | 0.335 | 0.073 | 0.153 | 1.790 | 0.808 | 0.185 | 0.285 | 0.358 | 0.337 |
| | Avg | 0.162 | 0.251 | 0.458 | **0.247** | 0.216 | 0.218 | 0.096 | **0.199** | 0.240 | 0.261 | **0.054** | **0.124** | **1.808** | 0.812 | **0.128** | **0.243** | **0.340** | **0.320** |
| Identity | H1 | 0.132 | 0.222 | 0.402 | 0.226 | 0.179 | 0.192 | 0.060 | 0.159 | 0.150 | 0.188 | 0.035 | 0.092 | 1.843 | 0.832 | 0.078 | 0.201 | 0.320 | 0.301 |
| | H2 | 0.152 | 0.240 | 0.445 | 0.241 | 0.213 | 0.214 | 0.076 | 0.178 | 0.204 | 0.240 | 0.049 | 0.116 | 1.757 | 0.802 | 0.108 | 0.229 | 0.334 | 0.315 |
| | H3 | 0.170 | 0.259 | 0.464 | 0.250 | 0.230 | 0.229 | 0.106 | 0.211 | 0.262 | 0.282 | 0.061 | 0.136 | 1.842 | 0.807 | 0.144 | 0.257 | 0.347 | 0.327 |
| | H4 | 0.194 | 0.284 | 0.520 | 0.271 | 0.242 | 0.236 | 0.141 | 0.248 | 0.341 | 0.334 | 0.073 | 0.153 | 1.791 | 0.808 | 0.188 | 0.285 | 0.358 | 0.337 |
| | Avg | 0.162 | 0.251 | 0.458 | **0.247** | 0.216 | 0.218 | 0.096 | **0.199** | 0.239 | 0.261 | **0.054** | **0.124** | **1.808** | 0.812 | 0.129 | **0.243** | **0.340** | **0.320** |

Table 28: Temporal decorrelation comparison of OrthoTrans, DFT and Wavelet transforms. We report the off-diagonal Frobenius norm of the temporal correlation matrix, divided by the number of off-diagonal elements.

| Dataset | Training Set | | | | Test Set | | | |
|---|---|---|---|---|---|---|---|---|
| | Original | OrthoTrans | DFT | Wavelet | Original | OrthoTrans | DFT | Wavelet |
| ETTh1 | 6.7E-03 | **1.6E-04** | 5.8E-04 | 3.7E-03 | 5.2E-03 | **2.1E-04** | 6.0E-04 | 3.1E-03 |
| ETTm2 | 9.1E-03 | **1.4E-04** | 1.1E-03 | 4.6E-03 | 7.2E-03 | **2.2E-04** | 1.3E-03 | 3.7E-03 |
| ECL | 5.0E-03 | **1.9E-04** | 4.7E-04 | 3.6E-03 | 5.0E-03 | **2.2E-04** | 4.9E-04 | 3.6E-03 |
| Traffic | 3.8E-03 | **1.9E-04** | 5.5E-04 | 2.6E-03 | 3.8E-03 | **2.3E-04** | 5.7E-04 | 2.6E-03 |
| Weather | 6.7E-03 | **3.1E-04** | 4.1E-03 | 3.6E-03 | 7.2E-03 | **4.0E-04** | 4.6E-03 | 3.7E-03 |
| Avg | 6.3E-03 | **2.0E-04** | 1.4E-03 | 3.6E-03 | 5.7E-03 | **2.6E-04** | 1.5E-03 | 3.3E-03 |

Table 29: Performance with varying lookback horizons. The best and second-best results are highlighted in **bold** and underlined, respectively. *OLinear-C* refers to the OLinear variant where the weight matrix in NormLin is replaced by $\mathrm{Softmax}(\mathrm{CorrMat}_v)$, with $\mathrm{CorrMat}_v$ being the cross-variate correlation matrix.

| | Model | OLinear (Ours) | | OLinear-C (Ours) | | Leddam 2024 | | CARD 2024b | | Fredformer 2024 | | iTrans. 2024a | | TimeMixer 2024a | | PatchTST 2023 | | TimesNet 2023b | | DLinear 2023 | |
|---|---|---|---|---|---|---|---|---|---|---|---|---|---|---|---|---|---|---|---|---|---|
| | Metric | MSE | MAE | MSE | MAE | MSE | MAE | MSE | MAE | MSE | MAE | MSE | MAE | MSE | MAE | MSE | MAE | MSE | MAE | MSE | MAE |
| ECL | 96 | **0.123** | **0.212** | **0.123** | **0.212** | 0.134 | 0.227 | 0.129 | 0.223 | 0.129 | 0.226 | 0.132 | 0.227 | 0.129 | 0.224 | 0.129 | 0.222 | 0.168 | 0.272 | 0.140 | 0.237 |
| | 192 | 0.143 | 0.232 | 0.144 | 0.233 | 0.156 | 0.248 | 0.154 | 0.245 | 0.148 | 0.244 | 0.154 | 0.251 | **0.140** | **0.220** | 0.147 | 0.240 | 0.184 | 0.322 | 0.153 | 0.249 |
| | 336 | 0.157 | **0.247** | **0.156** | **0.247** | 0.166 | 0.264 | 0.161 | 0.257 | 0.165 | 0.262 | 0.170 | 0.268 | 0.161 | 0.255 | 0.163 | 0.259 | 0.198 | 0.300 | 0.169 | 0.267 |
| | 720 | 0.181 | **0.270** | **0.180** | **0.270** | 0.195 | 0.291 | 0.185 | 0.278 | 0.193 | 0.286 | 0.193 | 0.288 | 0.194 | 0.287 | 0.197 | 0.290 | 0.220 | 0.320 | 0.203 | 0.301 |
| | Avg | **0.151** | **0.240** | **0.151** | **0.240** | 0.163 | 0.257 | 0.157 | 0.251 | 0.159 | 0.254 | 0.162 | 0.258 | 0.156 | 0.247 | 0.159 | 0.253 | 0.193 | 0.304 | 0.166 | 0.264 |
| Traffic | 96 | **0.338** | **0.221** | 0.340 | **0.221** | 0.366 | 0.260 | 0.341 | 0.229 | 0.358 | 0.257 | 0.359 | 0.262 | 0.360 | 0.249 | 0.360 | 0.249 | 0.593 | 0.321 | 0.410 | 0.282 |
| | 192 | **0.361** | **0.233** | 0.369 | 0.235 | 0.394 | 0.270 | 0.367 | 0.243 | 0.381 | 0.272 | 0.376 | 0.270 | 0.375 | 0.250 | 0.379 | 0.256 | 0.617 | 0.336 | 0.423 | 0.287 |
| | 336 | 0.386 | **0.241** | 0.390 | **0.241** | 0.400 | 0.283 | 0.388 | 0.254 | 0.396 | 0.277 | 0.393 | 0.279 | **0.385** | 0.270 | 0.392 | 0.264 | 0.629 | 0.336 | 0.436 | 0.296 |
| | 720 | 0.444 | **0.264** | 0.451 | 0.266 | 0.442 | 0.297 | 0.427 | 0.276 | **0.424** | 0.296 | 0.434 | 0.293 | 0.430 | 0.281 | 0.432 | 0.286 | 0.640 | 0.350 | 0.466 | 0.315 |
| | Avg | 0.382 | **0.240** | 0.387 | 0.241 | 0.400 | 0.278 | **0.381** | 0.251 | 0.390 | 0.275 | 0.390 | 0.276 | 0.388 | 0.263 | 0.391 | 0.264 | 0.620 | 0.336 | 0.434 | 0.295 |
| Weather | 96 | **0.144** | 0.184 | **0.144** | **0.183** | 0.149 | 0.199 | 0.145 | 0.186 | 0.150 | 0.203 | 0.165 | 0.214 | 0.147 | 0.197 | 0.149 | 0.198 | 0.172 | 0.220 | 0.176 | 0.237 |
| | 192 | 0.190 | 0.230 | 0.189 | 0.228 | 0.196 | 0.243 | **0.187** | **0.227** | 0.194 | 0.246 | 0.208 | 0.253 | 0.189 | 0.239 | 0.194 | 0.241 | 0.219 | 0.261 | 0.220 | 0.282 |
| | 336 | **0.235** | 0.268 | 0.236 | 0.268 | 0.243 | 0.280 | 0.238 | **0.258** | 0.243 | 0.284 | 0.257 | 0.292 | 0.241 | 0.280 | 0.306 | 0.282 | 0.246 | 0.337 | 0.265 | 0.319 |
| | 720 | 0.316 | 0.323 | 0.316 | 0.323 | 0.321 | 0.334 | **0.308** | **0.321** | **0.308** | 0.333 | 0.331 | 0.343 | 0.310 | 0.330 | 0.314 | 0.334 | 0.365 | 0.359 | 0.323 | 0.362 |
| | Avg | 0.221 | 0.251 | 0.221 | 0.250 | 0.227 | 0.264 | **0.220** | **0.248** | 0.224 | 0.266 | 0.240 | 0.275 | 0.222 | 0.262 | 0.241 | 0.264 | 0.251 | 0.294 | 0.246 | 0.300 |
| Solar-Energy | 96 | 0.169 | **0.194** | 0.195 | 0.207 | 0.186 | 0.242 | 0.170 | 0.207 | 0.187 | 0.236 | 0.190 | 0.241 | **0.167** | 0.220 | 0.224 | 0.278 | 0.219 | 0.314 | 0.289 | 0.377 |
| | 192 | 0.190 | **0.212** | 0.189 | **0.212** | 0.208 | 0.262 | 0.192 | 0.219 | 0.196 | 0.251 | 0.233 | 0.261 | **0.187** | 0.249 | 0.253 | 0.298 | 0.231 | 0.322 | 0.319 | 0.397 |
| | 336 | **0.192** | 0.215 | 0.194 | 0.219 | 0.218 | 0.265 | 0.226 | 0.233 | 0.208 | 0.265 | 0.226 | 0.275 | 0.200 | 0.258 | 0.273 | 0.306 | 0.246 | 0.337 | 0.352 | 0.415 |
| | 720 | 0.204 | **0.225** | 0.203 | 0.226 | 0.208 | 0.273 | 0.217 | 0.243 | 0.209 | 0.272 | 0.220 | 0.282 | 0.215 | 0.250 | 0.272 | 0.308 | 0.280 | 0.363 | 0.356 | 0.412 |
| | Avg | **0.189** | **0.211** | 0.195 | 0.216 | 0.205 | 0.261 | 0.201 | 0.225 | 0.200 | 0.256 | 0.217 | 0.265 | 0.192 | 0.244 | 0.256 | 0.298 | 0.244 | 0.334 | 0.329 | 0.400 |
| ETTm1 | 96 | **0.275** | **0.326** | **0.275** | **0.326** | 0.294 | 0.347 | 0.288 | 0.332 | 0.284 | 0.338 | 0.309 | 0.357 | 0.291 | 0.340 | 0.293 | 0.346 | 0.338 | 0.375 | 0.299 | 0.343 |
| | 192 | 0.318 | 0.352 | **0.317** | 0.351 | 0.336 | 0.369 | 0.332 | 0.357 | 0.323 | 0.364 | 0.346 | 0.383 | 0.327 | 0.365 | 0.333 | 0.370 | 0.374 | 0.387 | 0.335 | 0.365 |
| | 336 | **0.352** | **0.373** | **0.352** | **0.373** | 0.364 | 0.389 | 0.364 | 0.376 | 0.358 | 0.387 | 0.385 | 0.410 | 0.360 | 0.381 | 0.369 | 0.392 | 0.410 | 0.411 | 0.369 | 0.386 |
| | 720 | **0.406** | 0.408 | **0.406** | 0.408 | 0.421 | 0.419 | 0.414 | **0.407** | 0.420 | 0.417 | 0.440 | 0.442 | 0.415 | 0.417 | 0.416 | 0.420 | 0.478 | 0.450 | 0.425 | 0.421 |
| | Avg | **0.338** | 0.365 | **0.338** | 0.364 | 0.354 | 0.381 | 0.350 | 0.368 | 0.346 | 0.376 | 0.370 | 0.398 | 0.348 | 0.376 | 0.353 | 0.382 | 0.400 | 0.406 | 0.357 | 0.379 |
| 1st Count | | 12 | 16 | 11 | 12 | 0 | 0 | 4 | 4 | 2 | 0 | 0 | 0 | 4 | 1 | 0 | 0 | 0 | 0 | 0 | 0 |

Table 30: Comparison of OLinear with more baselines. MAEs are reported.

| Model | | OLinear (Ours) | SimpleTM 2025 | TQNet 2025 | TimePro 2025g | TimeBase 2025c |
|---|---|---|---|---|---|---|
| ETTm1 | 96 | **0.334** | 0.361 | 0.353 | 0.364 | 0.388 |
| | 192 | **0.363** | 0.380 | 0.378 | 0.383 | 0.409 |
| | 336 | **0.385** | 0.404 | 0.401 | 0.409 | 0.421 |
| | 720 | **0.426** | 0.438 | 0.440 | 0.446 | 0.461 |
| | Avg | **0.377** | 0.396 | 0.393 | 0.400 | 0.420 |
| ETTm2 | 96 | **0.249** | 0.257 | 0.256 | 0.260 | 0.271 |
| | 192 | **0.290** | 0.299 | 0.298 | 0.303 | 0.309 |
| | 336 | **0.328** | 0.338 | 0.340 | 0.342 | 0.346 |
| | 720 | **0.387** | 0.395 | 0.396 | 0.399 | 0.401 |
| | Avg | **0.313** | 0.322 | 0.323 | 0.326 | 0.332 |
| ETTh1 | 96 | **0.382** | 0.392 | 0.393 | 0.398 | 0.392 |
| | 192 | **0.414** | 0.421 | 0.426 | 0.429 | 0.423 |
| | 336 | **0.438** | **0.438** | 0.446 | 0.450 | 0.443 |
| | 720 | 0.462 | 0.462 | 0.470 | 0.474 | **0.458** |
| | Avg | **0.424** | 0.428 | 0.434 | 0.438 | 0.429 |
| ETTh2 | 96 | **0.329** | 0.338 | 0.343 | 0.345 | 0.376 |
| | 192 | **0.379** | 0.387 | 0.393 | 0.394 | 0.405 |
| | 336 | 0.415 | **0.401** | 0.427 | 0.431 | 0.440 |
| | 720 | **0.431** | 0.436 | 0.446 | 0.445 | 0.477 |
| | Avg | **0.388** | 0.391 | 0.402 | 0.403 | 0.424 |
| ECL | 96 | **0.221** | 0.235 | 0.229 | 0.234 | 0.279 |
| | 192 | **0.238** | 0.247 | 0.247 | 0.249 | 0.281 |
| | 336 | **0.254** | 0.267 | 0.264 | 0.267 | 0.295 |
| | 720 | **0.279** | 0.293 | 0.294 | 0.299 | 0.327 |
| | Avg | **0.248** | 0.260 | 0.259 | 0.262 | 0.295 |
| Traffic | 96 | **0.226** | 0.274 | 0.261 | 0.269 | 0.384 |
| | 192 | **0.241** | 0.280 | 0.271 | 0.276 | 0.362 |
| | 336 | **0.250** | 0.290 | 0.277 | 0.287 | 0.365 |
| | 720 | **0.270** | 0.309 | 0.295 | 0.312 | 0.386 |
| | Avg | **0.247** | 0.289 | 0.276 | 0.286 | 0.374 |
| Weather | 96 | **0.190** | 0.207 | 0.200 | 0.207 | 0.215 |
| | 192 | **0.235** | 0.248 | 0.245 | 0.254 | 0.256 |
| | 336 | **0.280** | 0.290 | 0.287 | 0.296 | 0.297 |
| | 720 | **0.333** | 0.341 | 0.342 | 0.346 | 0.348 |
| | Avg | **0.260** | 0.271 | 0.269 | 0.276 | 0.279 |
| Solar-Energy | 96 | **0.191** | 0.232 | 0.233 | 0.237 | 0.363 |
| | 192 | **0.213** | 0.247 | 0.257 | 0.263 | 0.404 |
| | 336 | **0.229** | 0.257 | 0.263 | 0.281 | 0.398 |
| | 720 | **0.236** | 0.252 | 0.270 | 0.285 | 0.388 |
| | Avg | **0.217** | 0.247 | 0.256 | 0.266 | 0.388 |

Table 31: Few-shot forecasting performance with 10% training data. Results are averaged over four prediction lengths {96, 192, 336, 720}.

| Model | OLinear (Ours) | | TimeMixer++ 2025a | | TimeMixer 2024a | | iTrans. 2024a | | PatchTST 2023 | | TimesNet 2023b | | DLinear 2023 | | TiDE 2023 | | Crossfm. 2023 | | FEDformer 2022 | |
|---|---|---|---|---|---|---|---|---|---|---|---|---|---|---|---|---|---|---|---|---|
| Metric | MSE | MAE | MSE | MAE | MSE | MAE | MSE | MAE | MSE | MAE | MSE | MAE | MSE | MAE | MSE | MAE | MSE | MAE | MSE | MAE |
| ETTh1 | **0.478** | **0.456** | 0.517 | 0.512 | 0.613 | 0.520 | 0.510 | 0.597 | 0.633 | 0.542 | 0.869 | 0.628 | 0.691 | 0.600 | 0.589 | 0.535 | 0.645 | 0.558 | 0.639 | 0.561 |
| ETTh2 | 0.390 | **0.407** | **0.379** | **0.391** | 0.402 | 0.433 | 0.455 | 0.461 | 0.415 | 0.431 | 0.479 | 0.465 | 0.605 | 0.538 | 0.395 | 0.412 | 0.428 | 0.447 | 0.466 | 0.475 |
| ETTm1 | 0.405 | **0.400** | **0.398** | 0.431 | 0.487 | 0.461 | 0.491 | 0.516 | 0.501 | 0.466 | 0.677 | 0.537 | 0.411 | 0.429 | 0.425 | 0.458 | 0.462 | 0.489 | 0.722 | 0.605 |
| ETTm2 | **0.287** | **0.330** | 0.291 | 0.351 | 0.311 | 0.367 | 0.375 | 0.412 | 0.296 | 0.343 | 0.320 | 0.353 | 0.316 | 0.368 | 0.317 | 0.371 | 0.343 | 0.389 | 0.463 | 0.488 |
| Weather | 0.245 | **0.265** | **0.241** | 0.271 | 0.242 | 0.281 | 0.291 | 0.331 | 0.242 | 0.279 | 0.279 | 0.301 | 0.241 | 0.283 | 0.249 | 0.291 | 0.267 | 0.306 | 0.284 | 0.324 |

Table 32: Zero-shot forecasting performance. Results are averaged over four prediction lengths {96, 192, 336, 720}. $\mathcal{A} \to \mathcal{B}$ denotes that the model is trained on dataset $\mathcal{A}$ but evaluated on $\mathcal{B}$.

| Model | OLinear (Ours) | | TimeMixer++ 2025a | | TimeMixer 2024a | | LLMTime 2023 | | DLinear 2023 | | PatchTST 2023 | | TimesNet 2023b | | iTrans. 2024a | | Crossfm. 2023 | | FEDformer 2022 | |
|---|---|---|---|---|---|---|---|---|---|---|---|---|---|---|---|---|---|---|---|---|
| Metric | MSE | MAE | MSE | MAE | MSE | MAE | MSE | MAE | MSE | MAE | MSE | MAE | MSE | MAE | MSE | MAE | MSE | MAE | MSE | MAE |
| ETTh1→ETTh2 | **0.367** | **0.391** | **0.367** | **0.391** | 0.427 | 0.424 | 0.992 | 0.708 | 0.493 | 0.488 | 0.380 | 0.405 | 0.421 | 0.431 | 0.481 | 0.474 | 0.555 | 0.574 | 0.712 | 0.693 |
| ETTh1→ETTm2 | 0.311 | **0.352** | **0.301** | 0.357 | 0.361 | 0.397 | 1.867 | 0.869 | 0.415 | 0.452 | 0.314 | 0.360 | 0.327 | 0.361 | 0.311 | 0.361 | 0.613 | 0.629 | 0.681 | 0.588 |
| ETTh2→ETTh1 | **0.507** | **0.479** | 0.511 | 0.498 | 0.679 | 0.577 | 1.961 | 0.981 | 0.703 | 0.574 | 0.565 | 0.513 | 0.865 | 0.621 | 0.552 | 0.511 | 0.587 | 0.518 | 0.612 | 0.624 |
| ETTm1→ETTh2 | **0.413** | **0.425** | 0.417 | 0.422 | 0.452 | 0.441 | 0.992 | 0.708 | 0.464 | 0.475 | 0.439 | 0.438 | 0.457 | 0.454 | 0.434 | 0.438 | 0.624 | 0.541 | 0.533 | 0.594 |
| ETTm1→ETTm2 | **0.291** | **0.327** | **0.291** | 0.331 | 0.329 | 0.357 | 1.867 | 0.869 | 0.335 | 0.389 | 0.296 | 0.334 | 0.322 | 0.354 | 0.324 | 0.331 | 0.595 | 0.572 | 0.612 | 0.611 |
| ETTm2→ETTm1 | 0.480 | **0.445** | **0.427** | 0.448 | 0.554 | 0.478 | 1.933 | 0.984 | 0.649 | 0.537 | 0.568 | 0.492 | 0.769 | 0.567 | 0.559 | 0.491 | 0.611 | 0.593 | 0.577 | 0.601 |

Table 33: Performance comparison of OLinear and large time series models on fine-tuning and zero-shot scenarios. MSEs are reported, with a prediction length of 96. Better results are highlighted in **bold**. For OLinar's zero-shot learning, we employ a cross-dataset strategy: training on ETTh1 and evaluating on ETTh2 (and vice versa), and similarly for ETTm1 and ETTm2. For Time-MoE [Shi et al., 2025], Moirai [Woo et al., 2024], and Chronos [Ansari et al., 2024], we use the largest model variants from their respective families. Baseline results are sourced from [Liu et al., 2024b] and [Liu et al., 2025].

| Scenario | 100% Target | | 20% Target | | 5% Target | | 0% Target (Zero-shot) | | | | | | |
|---|---|---|---|---|---|---|---|---|---|---|---|---|---|
| Model | OLinear (Ours) | Timer 2024b | OLinear (Ours) | Timer 2024b | OLinear (Ours) | Timer 2024b | OLinear (Ours) | Timer-28B 2024b | Time-MoE 2025 | Moirai 2024 | Moment 2024 | TimesFM 2024 | Chronos 2024 |
| Pre-trained | No | UTSD | No | UTSD | No | UTSD | ETT | UTSD+LOTSA | Time-300B | LOTSA | TS Pile | O(100B) Timepoints | 13 Public Datasets |
| ETTh1 | 0.363 | **0.358** | 0.396 | **0.359** | 0.413 | **0.362** | 0.416 | 0.393 | **0.349** | 0.381 | 0.674 | 0.414 | 0.441 |
| ETTh2 | 0.276 | – | 0.289 | **0.284** | 0.303 | **0.280** | **0.287** | 0.308 | 0.292 | 0.296 | 0.330 | 0.315 | 0.320 |
| ETTm1 | 0.275 | – | **0.300** | 0.321 | **0.321** | 0.321 | 0.355 | 0.420 | **0.281** | 0.380 | 0.670 | 0.361 | 0.457 |
| ETTm2 | 0.161 | – | **0.167** | 0.187 | **0.170** | 0.176 | **0.176** | 0.247 | 0.198 | 0.211 | 0.257 | 0.202 | 0.197 |
| ECL | **0.123** | 0.136 | **0.131** | 0.134 | 0.153 | **0.132** | - | **0.147** | - | 0.153 | 0.744 | - | - |
| Traffic | **0.338** | 0.351 | 0.371 | **0.352** | 0.419 | **0.361** | - | 0.414 | - | **0.399** | 1.293 | - | - |
| Weather | **0.144** | 0.154 | 0.156 | **0.151** | 0.168 | **0.151** | - | 0.243 | **0.157** | 0.199 | 0.255 | - | - |
| PEMS03 | **0.103** | 0.118 | **0.116** | 0.116 | 0.226 | **0.125** | - | - | - | - | - | - | - |
| PEMS04 | **0.086** | 0.107 | 0.195 | **0.120** | 0.542 | **0.135** | - | - | - | - | - | - | - |

Table 34: Performance with various embedding size $d$ settings. In our main experiments, we set $d = 16$ without specific statements.

| Emb. size | | 1 | | 4 | | 8 | | 16 | | 32 | | 64 | |
|---|---|---|---|---|---|---|---|---|---|---|---|---|---|
| | Metric | MSE | MAE | MSE | MAE | MSE | MAE | MSE | MAE | MSE | MAE | MSE | MAE |
| ECL | 96 | 0.133 | 0.223 | 0.132 | **0.221** | 0.132 | **0.221** | **0.131** | 0.221 | **0.131** | **0.221** | **0.131** | **0.221** |
| | 192 | 0.152 | 0.240 | 0.153 | 0.241 | 0.152 | 0.240 | **0.150** | **0.238** | 0.153 | 0.241 | 0.152 | 0.240 |
| | 336 | 0.166 | 0.256 | 0.169 | 0.257 | **0.165** | 0.255 | **0.165** | **0.254** | **0.165** | 0.255 | 0.168 | 0.258 |
| | 720 | **0.188** | **0.277** | 0.198 | 0.282 | 0.207 | 0.290 | 0.191 | 0.279 | 0.193 | 0.280 | 0.200 | 0.286 |
| | Avg | **0.159** | 0.249 | 0.163 | 0.250 | 0.164 | 0.252 | **0.159** | **0.248** | 0.161 | 0.249 | 0.163 | 0.251 |
| Traffic | 96 | **0.398** | 0.230 | 0.411 | 0.227 | 0.403 | **0.226** | **0.398** | **0.226** | 0.402 | **0.226** | 0.404 | **0.226** |
| | 192 | **0.425** | 0.243 | 0.433 | **0.241** | 0.443 | **0.241** | 0.439 | **0.241** | 0.436 | **0.241** | 0.435 | 0.242 |
| | 336 | **0.450** | 0.251 | 0.462 | 0.250 | 0.465 | **0.249** | 0.464 | 0.250 | 0.460 | 0.251 | 0.459 | 0.254 |
| | 720 | **0.496** | **0.269** | 0.518 | **0.269** | 0.520 | 0.270 | 0.502 | 0.270 | 0.507 | 0.274 | 0.515 | 0.275 |
| | Avg | **0.442** | 0.248 | 0.456 | 0.247 | 0.458 | **0.246** | 0.451 | 0.247 | 0.451 | 0.248 | 0.453 | 0.249 |
| Solar-Energy | 96 | 0.180 | 0.194 | 0.179 | 0.193 | 0.180 | 0.191 | 0.179 | 0.191 | **0.177** | **0.190** | 0.182 | 0.192 |
| | 192 | 0.212 | 0.216 | 0.211 | 0.214 | **0.209** | **0.213** | **0.209** | **0.213** | 0.212 | **0.213** | 0.211 | 0.214 |
| | 336 | 0.234 | 0.233 | 0.232 | 0.230 | **0.230** | 0.230 | 0.231 | **0.229** | 0.231 | 0.230 | 0.233 | 0.231 |
| | 720 | 0.244 | 0.238 | **0.241** | 0.236 | **0.241** | **0.235** | **0.241** | 0.236 | 0.242 | 0.236 | 0.243 | 0.238 |
| | Avg | 0.217 | 0.220 | 0.216 | 0.218 | **0.215** | **0.217** | **0.215** | **0.217** | **0.215** | **0.217** | 0.217 | 0.219 |
| Weather | 96 | 0.152 | 0.189 | 0.158 | 0.192 | **0.150** | **0.187** | 0.152 | 0.189 | 0.152 | 0.190 | 0.157 | 0.194 |
| | 192 | 0.203 | 0.238 | **0.200** | **0.234** | 0.207 | 0.242 | 0.207 | 0.243 | 0.202 | 0.238 | 0.205 | 0.239 |
| | 336 | 0.261 | 0.282 | **0.259** | **0.280** | **0.259** | **0.280** | 0.261 | 0.282 | 0.263 | 0.282 | 0.269 | 0.287 |
| | 720 | **0.342** | 0.337 | 0.343 | 0.338 | 0.343 | 0.337 | 0.344 | 0.337 | 0.350 | 0.339 | 0.344 | **0.336** |
| | Avg | **0.240** | **0.261** | **0.240** | **0.261** | **0.240** | **0.261** | 0.241 | 0.262 | 0.242 | 0.262 | 0.244 | 0.264 |
| ILI | 3 | 0.487 | 0.355 | 0.545 | 0.380 | 0.500 | 0.353 | **0.468** | **0.349** | 0.515 | 0.362 | 0.486 | 0.352 |
| | 6 | 0.915 | 0.515 | 1.021 | 0.533 | **0.890** | **0.506** | 0.923 | 0.516 | 0.914 | 0.515 | 1.059 | 0.558 |
| | 9 | 1.301 | 0.662 | 1.363 | 0.666 | 1.376 | 0.667 | 1.289 | 0.655 | 1.265 | **0.644** | 1.264 | 0.645 |
| | 12 | 1.638 | 0.783 | 1.693 | 0.788 | 1.810 | 0.799 | 1.698 | 0.791 | **1.619** | **0.769** | 1.747 | 0.773 |
| | Avg | 1.085 | 0.579 | 1.155 | 0.592 | 1.144 | 0.581 | 1.094 | 0.578 | **1.078** | **0.572** | 1.139 | 0.582 |
| PEMS03 | 12 | 0.060 | 0.159 | 0.060 | 0.159 | 0.060 | 0.159 | 0.060 | 0.159 | 0.061 | 0.160 | **0.059** | **0.158** |
| | 24 | 0.077 | **0.179** | 0.078 | **0.179** | 0.078 | 0.180 | 0.078 | **0.179** | **0.077** | **0.179** | **0.077** | **0.179** |
| | 48 | 0.107 | 0.213 | 0.106 | 0.211 | 0.106 | 0.211 | **0.104** | **0.210** | 0.106 | 0.211 | 0.105 | **0.210** |
| | 96 | 0.146 | 0.253 | 0.143 | 0.248 | 0.141 | 0.248 | **0.140** | **0.247** | **0.140** | **0.247** | 0.143 | 0.248 |
| | Avg | 0.097 | 0.201 | 0.097 | **0.199** | 0.096 | 0.200 | **0.095** | **0.199** | 0.096 | **0.199** | 0.096 | **0.199** |
| 1st Count | | 9 | 4 | 4 | 8 | 9 | 12 | 12 | 14 | 8 | 12 | 4 | 7 |

Table 35: Forecasting performance on the scale-free metrics: Coefficient of Determination ($R^2$), Pearson Correlation Coefficient ($r$), and MASE. The symbols ↓ and ↑ indicate that lower and higher values are better, respectively. The best and second-best results are highlighted in **bold** and underlined, respectively. The lookback length $T$ is uniformly set as 96.

| Model | | OLinear (Ours) | | | Leddam 2024 | | | CARD 2024b | | | iTrans. 2024a | | | TimeMixer 2024a | |
|---|---|---|---|---|---|---|---|---|---|---|---|---|---|---|---|---|
| Metric | $R^2$ (↑) | $r$ (↑) | MASE (↓) | $R^2$ (↑) | $r$ (↑) | MASE (↓) | $R^2$ (↑) | $r$ (↑) | MASE (↓) | $R^2$ (↑) | $r$ (↑) | MASE (↓) | $R^2$ (↑) | $r$ (↑) | MASE (↓) |
| ECL 96 | **0.640** | **0.916** | **0.902** | 0.527 | 0.911 | 0.964 | 0.533 | 0.905 | 0.975 | 0.550 | 0.907 | 0.990 | 0.456 | 0.897 | 1.022 |
| ECL 192 | **0.638** | **0.908** | **0.983** | 0.593 | 0.902 | 1.041 | 0.547 | 0.899 | 1.028 | 0.590 | 0.900 | 1.057 | 0.575 | 0.893 | 1.076 |
| ECL 336 | **0.710** | **0.901** | **1.061** | 0.670 | 0.893 | 1.134 | 0.678 | 0.891 | 1.113 | 0.678 | 0.893 | 1.133 | 0.640 | 0.886 | 1.155 |
| ECL 720 | **0.701** | **0.889** | **1.174** | 0.661 | 0.882 | 1.251 | 0.653 | 0.877 | 1.273 | 0.638 | 0.872 | 1.315 | 0.634 | 0.871 | 1.314 |
| ECL Avg | **0.672** | **0.903** | **1.030** | 0.613 | 0.897 | 1.098 | 0.603 | 0.893 | 1.097 | 0.614 | 0.893 | 1.124 | 0.576 | 0.887 | 1.142 |
| Traffic 96 | **0.737** | **0.899** | **0.729** | 0.681 | 0.885 | 0.909 | 0.690 | 0.882 | 0.855 | 0.690 | 0.889 | 0.860 | 0.640 | 0.888 | 0.979 |
| Traffic 192 | **0.719** | **0.884** | **0.756** | 0.666 | 0.865 | 0.949 | 0.683 | 0.869 | 0.856 | 0.689 | 0.877 | 0.879 | 0.660 | 0.862 | 0.959 |
| Traffic 336 | **0.730** | **0.875** | **0.758** | 0.707 | 0.867 | 0.897 | 0.702 | 0.862 | 0.843 | 0.713 | 0.867 | 0.883 | 0.685 | 0.852 | 0.960 |
| Traffic 720 | **0.703** | **0.860** | **0.815** | 0.687 | 0.853 | 0.947 | 0.679 | 0.848 | 0.894 | 0.693 | 0.857 | 0.913 | 0.672 | 0.842 | 0.986 |
| Traffic Avg | **0.722** | **0.879** | **0.764** | 0.685 | 0.868 | 0.926 | 0.689 | 0.865 | 0.862 | 0.696 | 0.873 | 0.884 | 0.664 | 0.861 | 0.971 |

| Model | | FilterNet 2024a | | | FITS 2024 | | | DLinear 2023 | | | PatchTST 2023 | | | TimesNet 2023b | |
|---|---|---|---|---|---|---|---|---|---|---|---|---|---|---|---|---|
| Metric | $R^2$ (↑) | $r$ (↑) | MASE (↓) | $R^2$ (↑) | $r$ (↑) | MASE (↓) | $R^2$ (↑) | $r$ (↑) | MASE (↓) | $R^2$ (↑) | $r$ (↑) | MASE (↓) | $R^2$ (↑) | $r$ (↑) | MASE (↓) |
| ECL 96 | 0.537 | 0.905 | 0.996 | 0.352 | 0.883 | 1.144 | 0.120 | 0.870 | 1.300 | 0.426 | 0.890 | 1.119 | 0.405 | 0.892 | 1.118 |
| ECL 192 | 0.576 | 0.898 | 1.052 | 0.446 | 0.880 | 1.153 | 0.215 | 0.867 | 1.312 | 0.509 | 0.887 | 1.140 | 0.503 | 0.885 | 1.195 |
| ECL 336 | 0.670 | 0.891 | 1.139 | 0.588 | 0.873 | 1.229 | 0.247 | 0.858 | 1.387 | 0.604 | 0.880 | 1.216 | 0.592 | 0.875 | 1.262 |
| ECL 720 | 0.628 | 0.875 | 1.328 | 0.593 | 0.858 | 1.376 | 0.148 | 0.844 | 1.536 | 0.605 | 0.864 | 1.371 | 0.603 | 0.864 | 1.388 |
| ECL Avg | 0.603 | 0.892 | 1.129 | 0.495 | 0.873 | 1.225 | 0.182 | 0.860 | 1.384 | 0.536 | 0.880 | 1.211 | 0.526 | 0.879 | 1.241 |
| Traffic 96 | 0.637 | 0.876 | 0.987 | 0.458 | 0.811 | 1.284 | 0.441 | 0.811 | 1.346 | 0.668 | 0.880 | 0.903 | 0.625 | 0.877 | 1.023 |
| Traffic 192 | 0.665 | 0.867 | 0.950 | 0.542 | 0.814 | 1.151 | 0.551 | 0.815 | 1.189 | 0.677 | 0.871 | 0.887 | 0.635 | 0.867 | 1.009 |
| Traffic 336 | 0.696 | 0.861 | 0.930 | 0.582 | 0.810 | 1.119 | 0.582 | 0.810 | 1.167 | 0.700 | 0.863 | 0.874 | 0.654 | 0.852 | 1.079 |
| Traffic 720 | 0.678 | 0.847 | 0.982 | 0.568 | 0.794 | 1.169 | 0.564 | 0.794 | 1.224 | 0.681 | 0.850 | 0.925 | 0.661 | 0.850 | 1.047 |
| Traffic Avg | 0.669 | 0.862 | 0.962 | 0.537 | 0.807 | 1.181 | 0.535 | 0.808 | 1.232 | 0.682 | 0.866 | 0.897 | 0.644 | 0.862 | 1.040 |

Table 36: Forecasting performance when orthogonal matrices are computed from a subset of the training data ('Q_ratio'), while model training still uses the full trainset.

| Q_ratio | | 100% | | 80% | | 60% | | 40% | | 20% | | 10% | |
|---|---|---|---|---|---|---|---|---|---|---|---|---|---|
| Metric | | MSE | MAE | MSE | MAE | MSE | MAE | MSE | MAE | MSE | MAE | MSE | MAE |
| ECL | 96 | 0.131 | 0.221 | 0.131 | 0.221 | 0.131 | 0.221 | 0.131 | 0.221 | 0.130 | 0.221 | 0.131 | 0.221 |
| | 192 | 0.150 | 0.238 | 0.155 | 0.243 | 0.152 | 0.239 | 0.153 | 0.240 | 0.151 | 0.240 | 0.152 | 0.240 |
| | 336 | 0.165 | 0.254 | 0.169 | 0.257 | 0.164 | 0.253 | 0.166 | 0.255 | 0.165 | 0.254 | 0.166 | 0.255 |
| | 720 | 0.191 | 0.279 | 0.193 | 0.284 | 0.188 | 0.277 | 0.193 | 0.284 | 0.193 | 0.280 | 0.189 | 0.278 |
| | Avg | **0.159** | 0.248 | 0.162 | 0.251 | **0.159** | **0.247** | 0.161 | 0.250 | 0.160 | 0.248 | **0.159** | 0.248 |
| Traffic | 96 | 0.398 | 0.226 | 0.403 | 0.226 | 0.403 | 0.227 | 0.401 | 0.226 | 0.404 | 0.226 | 0.402 | 0.226 |
| | 192 | 0.439 | 0.241 | 0.436 | 0.240 | 0.434 | 0.240 | 0.432 | 0.240 | 0.434 | 0.240 | 0.431 | 0.240 |
| | 336 | 0.464 | 0.250 | 0.461 | 0.250 | 0.462 | 0.250 | 0.463 | 0.250 | 0.459 | 0.250 | 0.465 | 0.250 |
| | 720 | 0.502 | 0.270 | 0.499 | 0.271 | 0.499 | 0.271 | 0.501 | 0.271 | 0.502 | 0.271 | 0.512 | 0.271 |
| | Avg | 0.451 | **0.247** | 0.450 | **0.247** | **0.449** | **0.247** | **0.449** | **0.247** | 0.450 | **0.247** | 0.453 | **0.247** |
| Weather | 96 | 0.153 | 0.190 | 0.150 | 0.188 | 0.153 | 0.191 | 0.153 | 0.190 | 0.154 | 0.191 | 0.149 | 0.187 |
| | 192 | 0.200 | 0.235 | 0.202 | 0.237 | 0.204 | 0.239 | 0.203 | 0.238 | 0.202 | 0.237 | 0.203 | 0.239 |
| | 336 | 0.258 | 0.280 | 0.260 | 0.280 | 0.260 | 0.281 | 0.264 | 0.284 | 0.260 | 0.281 | 0.259 | 0.281 |
| | 720 | 0.337 | 0.333 | 0.339 | 0.333 | 0.338 | 0.332 | 0.337 | 0.332 | 0.335 | 0.330 | 0.339 | 0.332 |
| | Avg | **0.237** | 0.260 | 0.238 | **0.259** | 0.239 | 0.261 | 0.239 | 0.261 | 0.238 | 0.260 | **0.237** | 0.260 |
| PEMS03 | 12 | 0.060 | 0.159 | 0.060 | 0.159 | 0.060 | 0.159 | 0.060 | 0.159 | 0.061 | 0.160 | 0.060 | 0.159 |
| | 24 | 0.078 | 0.179 | 0.077 | 0.179 | 0.077 | 0.180 | 0.077 | 0.179 | 0.077 | 0.180 | 0.077 | 0.180 |
| | 48 | 0.104 | 0.210 | 0.105 | 0.210 | 0.104 | 0.210 | 0.104 | 0.210 | 0.104 | 0.210 | 0.105 | 0.210 |
| | 96 | 0.140 | 0.247 | 0.141 | 0.248 | 0.142 | 0.248 | 0.141 | 0.248 | 0.143 | 0.249 | 0.140 | 0.249 |
| | Avg | **0.095** | **0.199** | 0.096 | **0.199** | 0.096 | **0.199** | **0.095** | **0.199** | 0.096 | **0.199** | **0.095** | **0.199** |
| NASDAQ | 3 | 0.036 | 0.092 | 0.036 | 0.092 | 0.036 | 0.092 | 0.036 | 0.093 | 0.036 | 0.093 | 0.036 | 0.093 |
| | 6 | 0.049 | 0.117 | 0.049 | 0.117 | 0.049 | 0.117 | 0.049 | 0.117 | 0.049 | 0.117 | 0.049 | 0.118 |
| | 9 | 0.062 | 0.137 | 0.062 | 0.137 | 0.062 | 0.137 | 0.062 | 0.137 | 0.062 | 0.137 | 0.062 | 0.137 |
| | 12 | 0.073 | 0.154 | 0.073 | 0.154 | 0.073 | 0.154 | 0.073 | 0.154 | 0.073 | 0.154 | 0.073 | 0.154 |
| | Avg | **0.055** | **0.125** | **0.055** | **0.125** | **0.055** | **0.125** | **0.055** | **0.125** | **0.055** | **0.125** | **0.055** | **0.125** |
| CarSales | 24 | 0.320 | 0.302 | 0.319 | 0.302 | 0.319 | 0.301 | 0.319 | 0.302 | 0.320 | 0.302 | 0.320 | 0.302 |
| | 36 | 0.334 | 0.315 | 0.334 | 0.315 | 0.334 | 0.315 | 0.334 | 0.315 | 0.334 | 0.315 | 0.334 | 0.316 |
| | 48 | 0.347 | 0.327 | 0.347 | 0.327 | 0.347 | 0.327 | 0.347 | 0.327 | 0.347 | 0.327 | 0.347 | 0.328 |
| | 60 | 0.358 | 0.337 | 0.358 | 0.337 | 0.358 | 0.337 | 0.358 | 0.337 | 0.358 | 0.337 | 0.358 | 0.337 |
| | Avg | 0.340 | **0.320** | **0.339** | **0.320** | **0.339** | **0.320** | 0.340 | **0.320** | 0.340 | **0.320** | 0.340 | **0.320** |

Table 37: Ablation study on transformation functions and normalization methods in NormLin. For *S1* and *S2*, the prediction lengths are $\{3, 6, 9, 12\}$ and $\{24, 36, 48, 60\}$, respectively.

| Fun. | Norm | Hor. | ECL | | Traffic | | Solar | | PEMS03 | | Weather | | NASDAQ (S1) | | ILI (S2) | |
|---|---|---|---|---|---|---|---|---|---|---|---|---|---|---|---|---|
| Softplus | L1 | H1 | **0.131** | **0.221** | **0.398** | **0.226** | 0.179 | **0.191** | **0.060** | **0.159** | 0.153 | 0.190 | 0.036 | **0.092** | 1.737 | 0.800 |
| | | H2 | **0.150** | **0.238** | **0.439** | **0.241** | **0.209** | **0.213** | 0.078 | **0.179** | 0.200 | **0.235** | **0.049** | **0.117** | **1.714** | **0.795** |
| | | H3 | **0.165** | **0.254** | **0.464** | **0.250** | **0.231** | **0.229** | 0.104 | **0.210** | **0.258** | **0.280** | 0.062 | 0.137 | 1.821 | 0.804 |
| | | H4 | **0.191** | **0.279** | 0.502 | **0.270** | **0.241** | **0.236** | 0.140 | **0.247** | 0.337 | 0.333 | **0.073** | **0.154** | 1.785 | 0.810 |
| | | Avg | **0.159** | **0.248** | 0.451 | **0.247** | **0.215** | **0.217** | **0.095** | **0.199** | 0.237 | 0.260 | **0.055** | **0.125** | 1.764 | **0.802** |
| Identity | L1 | H1 | 0.140 | 0.229 | 0.455 | 0.233 | 0.182 | 0.193 | 0.062 | 0.161 | 0.154 | 0.191 | 0.036 | 0.093 | 1.783 | 0.807 |
| | | H2 | 0.157 | 0.245 | 0.486 | 0.246 | 0.216 | 0.215 | 0.078 | **0.179** | 0.203 | 0.240 | **0.049** | **0.117** | 1.844 | 0.810 |
| | | H3 | 0.170 | 0.260 | 0.506 | 0.255 | 0.239 | 0.233 | **0.103** | **0.208** | 0.267 | 0.285 | 0.061 | 0.137 | 1.909 | 0.826 |
| | | H4 | 0.200 | 0.287 | 0.551 | 0.276 | 0.506 | 0.255 | 0.143 | 0.250 | 0.350 | 0.339 | **0.073** | **0.154** | 1.841 | 0.830 |
| | | Avg | 0.167 | 0.255 | 0.499 | 0.252 | 0.285 | 0.224 | 0.096 | **0.199** | 0.243 | 0.264 | **0.055** | **0.125** | 1.844 | 0.818 |
| Softmax | L1 | H1 | **0.131** | **0.221** | **0.398** | **0.226** | 0.179 | 0.192 | 0.061 | 0.160 | 0.150 | **0.187** | 0.036 | **0.092** | **1.707** | **0.800** |
| | | H2 | 0.155 | 0.242 | 0.441 | **0.241** | **0.209** | **0.213** | 0.077 | **0.179** | 0.202 | 0.237 | **0.049** | **0.117** | 1.886 | 0.828 |
| | | H3 | 0.167 | 0.255 | 0.467 | **0.250** | 0.232 | **0.229** | 0.107 | 0.211 | 0.259 | **0.280** | 0.062 | 0.138 | 1.820 | 0.803 |
| | | H4 | 0.199 | 0.286 | 0.504 | **0.270** | **0.241** | **0.236** | 0.139 | **0.247** | **0.335** | **0.332** | **0.073** | **0.154** | 1.795 | 0.814 |
| | | Avg | 0.163 | 0.251 | 0.452 | **0.247** | **0.215** | **0.217** | 0.096 | **0.199** | **0.236** | **0.259** | **0.055** | **0.125** | 1.802 | 0.811 |
| Sigmoid | L1 | H1 | **0.131** | **0.221** | **0.398** | **0.226** | **0.178** | **0.191** | 0.061 | **0.159** | 0.152 | 0.189 | **0.035** | 0.093 | 1.742 | 0.799 |
| | | H2 | **0.150** | **0.238** | **0.439** | **0.241** | **0.209** | **0.213** | 0.077 | **0.179** | 0.203 | 0.238 | **0.049** | **0.117** | 1.719 | 0.797 |
| | | H3 | 0.167 | 0.256 | **0.464** | **0.250** | **0.231** | **0.229** | 0.106 | 0.211 | 0.262 | 0.283 | 0.062 | **0.137** | 1.826 | 0.804 |
| | | H4 | 0.199 | 0.285 | **0.500** | **0.270** | **0.241** | **0.236** | 0.139 | 0.248 | 0.338 | **0.332** | **0.073** | **0.154** | **1.780** | **0.808** |
| | | Avg | 0.161 | 0.250 | **0.450** | **0.247** | **0.215** | **0.217** | 0.096 | **0.199** | 0.239 | 0.260 | **0.055** | **0.125** | 1.767 | **0.802** |
| ReLU | L1 | H1 | 0.132 | 0.222 | **0.398** | **0.226** | 0.179 | **0.191** | 0.061 | 0.160 | 0.153 | 0.189 | 0.036 | **0.092** | 1.711 | 0.799 |
| | | H2 | 0.153 | 0.241 | 0.443 | **0.241** | 0.210 | **0.213** | 0.078 | **0.179** | 0.202 | 0.238 | **0.049** | **0.117** | 1.721 | **0.795** |
| | | H3 | 0.170 | 0.259 | 0.467 | **0.250** | **0.231** | **0.229** | 0.106 | 0.211 | 0.264 | 0.285 | 0.062 | 0.138 | 1.816 | 0.804 |
| | | H4 | 0.197 | 0.285 | 0.507 | 0.271 | **0.241** | **0.236** | 0.143 | 0.248 | 0.338 | 0.335 | **0.073** | **0.154** | 1.790 | 0.812 |
| | | Avg | 0.163 | 0.251 | 0.454 | **0.247** | **0.215** | **0.217** | 0.097 | **0.199** | 0.239 | 0.262 | **0.055** | **0.125** | **1.759** | **0.802** |
| Softplus | L2 | H1 | 0.192 | 0.285 | 0.613 | 0.313 | 0.234 | 0.244 | 0.095 | 0.199 | 0.151 | 0.188 | 0.036 | 0.093 | 1.928 | 0.852 |
| | | H2 | 0.194 | 0.288 | 0.640 | 0.325 | 0.280 | 0.267 | 0.121 | 0.221 | 0.203 | 0.238 | 0.050 | 0.119 | 1.929 | 0.837 |
| | | H3 | 0.202 | 0.297 | 0.653 | 0.329 | 0.323 | 0.288 | 0.174 | 0.265 | 0.259 | 0.281 | 0.063 | 0.139 | 1.845 | 0.806 |
| | | H4 | 0.236 | 0.322 | 0.683 | 0.344 | 0.337 | 0.292 | 0.264 | 0.331 | 0.339 | 0.335 | **0.073** | **0.154** | 1.794 | 0.813 |
| | | Avg | 0.206 | 0.298 | 0.647 | 0.328 | 0.293 | 0.273 | 0.163 | 0.254 | 0.238 | 0.260 | **0.055** | 0.126 | 1.874 | 0.827 |
| Identity | L2 | H1 | 0.139 | 0.230 | 0.456 | 0.238 | 0.187 | 0.198 | 0.063 | 0.163 | 0.151 | 0.189 | 0.036 | **0.092** | 1.718 | **0.795** |
| | | H2 | 0.159 | 0.249 | 0.516 | 0.251 | 0.221 | 0.219 | 0.080 | 0.184 | 0.201 | 0.238 | 0.050 | 0.118 | 1.855 | 0.812 |
| | | H3 | 0.171 | 0.261 | 0.529 | 0.261 | 0.239 | 0.234 | 0.112 | 0.217 | 0.266 | 0.284 | 0.062 | **0.137** | 1.947 | 0.831 |
| | | H4 | 0.215 | 0.298 | 0.573 | 0.288 | 0.248 | 0.239 | 0.153 | 0.258 | 0.347 | 0.339 | **0.073** | **0.154** | 1.832 | 0.829 |
| | | Avg | 0.171 | 0.259 | 0.518 | 0.259 | 0.224 | 0.222 | 0.102 | 0.205 | 0.241 | 0.262 | **0.055** | **0.125** | 1.838 | 0.817 |
| Softmax | L2 | H1 | 0.183 | 0.278 | 0.612 | 0.312 | 0.233 | 0.242 | 0.096 | 0.199 | 0.153 | 0.190 | 0.036 | 0.093 | 1.876 | 0.837 |
| | | H2 | 0.192 | 0.286 | 0.641 | 0.325 | 0.266 | 0.256 | 0.118 | 0.219 | **0.199** | **0.235** | 0.050 | 0.119 | 1.875 | 0.825 |
| | | H3 | 0.202 | 0.297 | 0.653 | 0.330 | 0.313 | 0.280 | 0.174 | 0.265 | 0.259 | 0.282 | 0.063 | 0.139 | 1.823 | 0.803 |
| | | H4 | 0.226 | 0.316 | 0.686 | 0.346 | 0.336 | 0.292 | 0.268 | 0.334 | 0.337 | 0.333 | **0.073** | **0.154** | 1.798 | 0.814 |
| | | Avg | 0.201 | 0.294 | 0.648 | 0.328 | 0.287 | 0.267 | 0.164 | 0.254 | 0.237 | 0.260 | **0.055** | 0.126 | 1.843 | 0.820 |
| Sigmoid | L2 | H1 | 0.193 | 0.285 | 0.613 | 0.313 | 0.234 | 0.244 | 0.096 | 0.200 | **0.149** | **0.187** | 0.036 | 0.094 | 1.993 | 0.861 |
| | | H2 | 0.206 | 0.296 | 0.640 | 0.324 | 0.279 | 0.267 | 0.119 | 0.220 | 0.204 | 0.239 | 0.050 | 0.119 | 1.762 | 0.807 |
| | | H3 | 0.201 | 0.296 | 0.653 | 0.329 | 0.324 | 0.288 | 0.175 | 0.267 | 0.262 | 0.284 | 0.063 | 0.139 | 1.866 | 0.811 |
| | | H4 | 0.256 | 0.334 | 0.684 | 0.345 | 0.336 | 0.291 | 0.265 | 0.332 | 0.337 | 0.333 | **0.073** | 0.155 | 1.797 | 0.813 |
| | | Avg | 0.214 | 0.302 | 0.647 | 0.328 | 0.293 | 0.272 | 0.164 | 0.255 | 0.238 | 0.261 | 0.056 | 0.127 | 1.854 | 0.823 |
| ReLU | L2 | H1 | 0.181 | 0.277 | 0.612 | 0.312 | 0.231 | 0.241 | 0.095 | 0.197 | 0.153 | 0.190 | 0.036 | 0.093 | 1.860 | 0.838 |
| | | H2 | 0.190 | 0.284 | 0.640 | 0.325 | 0.265 | 0.255 | 0.118 | 0.219 | 0.201 | 0.237 | 0.050 | 0.119 | 1.900 | 0.833 |
| | | H3 | 0.200 | 0.295 | 0.653 | 0.329 | 0.309 | 0.278 | 0.176 | 0.268 | 0.260 | 0.282 | 0.063 | 0.139 | **1.803** | **0.800** |
| | | H4 | 0.232 | 0.320 | 0.686 | 0.346 | 0.335 | 0.291 | 0.263 | 0.330 | 0.336 | **0.332** | **0.073** | **0.154** | 1.798 | 0.813 |
| | | Avg | 0.201 | 0.294 | 0.648 | 0.328 | 0.285 | 0.266 | 0.163 | 0.254 | 0.237 | 0.260 | **0.055** | 0.126 | 1.840 | 0.821 |

Table 38: Ablation study on pre- and post-linear layers in the NormLin module. 'Hor.' denotes prediction horizons. For *S1* and *S2*, the prediction horizons are $\{3, 6, 9, 12\}$ and $\{24, 36, 48, 60\}$, respectively.

| PreLin | PostLin | Hor. | ECL MSE | ECL MAE | Traffic MSE | Traffic MAE | Solar MSE | Solar MAE | PEMS03 MSE | PEMS03 MAE | Weather MSE | Weather MAE | NASDAQ (S1) MSE | NASDAQ (S1) MAE | ILI (S2) MSE | ILI (S2) MAE | Wiki (S1) MSE | Wiki (S1) MAE | ETTm1 MSE | ETTm1 MAE |
|---|---|---|---|---|---|---|---|---|---|---|---|---|---|---|---|---|---|---|---|---|
| ✗ | ✗ | H1 | 0.135 | 0.229 | 0.474 | 0.253 | 0.192 | 0.206 | 0.065 | 0.167 | 0.159 | 0.195 | 0.040 | 0.104 | 1.744 | 0.861 | 6.184 | 0.372 | 0.320 | 0.352 |
| | | H2 | 0.153 | 0.246 | 0.492 | 0.264 | 0.225 | 0.228 | 0.085 | 0.191 | 0.207 | 0.241 | 0.053 | 0.126 | 1.970 | 0.891 | 6.471 | 0.387 | 0.373 | 0.384 |
| | | H3 | 0.168 | 0.264 | 0.511 | 0.273 | 0.247 | 0.244 | 0.125 | 0.233 | 0.264 | 0.285 | 0.065 | 0.146 | 2.153 | 0.892 | 6.673 | 0.403 | 0.401 | 0.400 |
| | | H4 | 0.204 | 0.297 | 0.555 | 0.294 | 0.255 | 0.247 | 0.178 | 0.281 | 0.342 | 0.336 | 0.077 | 0.162 | 2.091 | 0.890 | 6.846 | 0.410 | 0.461 | 0.435 |
| | | Avg | 0.165 | 0.259 | 0.508 | 0.271 | 0.230 | 0.231 | 0.113 | 0.218 | 0.243 | 0.264 | 0.058 | 0.134 | 1.989 | 0.883 | 6.543 | 0.393 | 0.389 | 0.393 |
| ✗ | ✔ | H1 | 0.131 | 0.222 | 0.403 | 0.228 | 0.179 | 0.192 | 0.061 | 0.161 | 0.155 | 0.192 | 0.036 | 0.094 | 1.926 | 0.858 | 6.176 | 0.368 | 0.303 | 0.334 |
| | | H2 | 0.151 | 0.240 | 0.434 | 0.242 | 0.210 | 0.213 | 0.077 | 0.180 | 0.205 | 0.240 | 0.049 | 0.118 | 1.796 | 0.827 | 6.447 | 0.384 | 0.356 | 0.364 |
| | | H3 | 0.166 | 0.256 | 0.457 | 0.251 | 0.231 | 0.230 | 0.107 | 0.212 | 0.262 | 0.282 | 0.062 | 0.137 | 1.960 | 0.833 | 6.650 | 0.399 | 0.389 | 0.387 |
| | | H4 | 0.211 | 0.298 | 0.500 | 0.272 | 0.243 | 0.236 | 0.146 | 0.253 | 0.352 | 0.340 | 0.073 | 0.155 | 1.799 | 0.811 | 6.839 | 0.406 | 0.452 | 0.426 |
| | | Avg | 0.165 | 0.254 | 0.448 | 0.248 | 0.216 | 0.218 | 0.098 | 0.201 | 0.243 | 0.263 | 0.055 | 0.126 | 1.870 | 0.832 | 6.528 | 0.389 | 0.375 | 0.378 |
| ✔ | ✗ | H1 | 0.131 | 0.222 | 0.404 | 0.228 | 0.178 | 0.192 | 0.061 | 0.160 | 0.154 | 0.192 | 0.036 | 0.094 | 1.924 | 0.858 | 6.175 | 0.368 | 0.303 | 0.334 |
| | | H2 | 0.155 | 0.243 | 0.434 | 0.242 | 0.211 | 0.214 | 0.077 | 0.180 | 0.204 | 0.239 | 0.049 | 0.118 | 1.796 | 0.828 | 6.448 | 0.385 | 0.356 | 0.364 |
| | | H3 | 0.165 | 0.255 | 0.457 | 0.251 | 0.231 | 0.230 | 0.107 | 0.212 | 0.263 | 0.282 | 0.062 | 0.137 | 1.959 | 0.833 | 6.649 | 0.399 | 0.389 | 0.387 |
| | | H4 | 0.209 | 0.293 | 0.499 | 0.272 | 0.242 | 0.236 | 0.144 | 0.252 | 0.344 | 0.339 | 0.073 | 0.155 | 1.799 | 0.811 | 6.839 | 0.406 | 0.452 | 0.426 |
| | | Avg | 0.165 | 0.253 | 0.448 | 0.248 | 0.215 | 0.218 | 0.097 | 0.201 | 0.241 | 0.263 | 0.055 | 0.126 | 1.869 | 0.832 | 6.528 | 0.389 | 0.375 | 0.378 |
| ✔ | ✔ | H1 | 0.131 | 0.221 | 0.398 | 0.226 | 0.179 | 0.191 | 0.060 | 0.159 | 0.152 | 0.189 | 0.036 | 0.092 | 1.737 | 0.800 | 6.161 | 0.368 | 0.302 | 0.334 |
| | | H2 | 0.150 | 0.238 | 0.439 | 0.241 | 0.209 | 0.213 | 0.078 | 0.179 | 0.207 | 0.243 | 0.049 | 0.117 | 1.714 | 0.795 | 6.453 | 0.385 | 0.357 | 0.364 |
| | | H3 | 0.165 | 0.254 | 0.464 | 0.250 | 0.231 | 0.229 | 0.104 | 0.210 | 0.261 | 0.282 | 0.062 | 0.137 | 1.821 | 0.804 | 6.666 | 0.398 | 0.387 | 0.385 |
| | | H4 | 0.191 | 0.279 | 0.502 | 0.270 | 0.241 | 0.236 | 0.140 | 0.247 | 0.344 | 0.337 | 0.073 | 0.154 | 1.785 | 0.810 | 6.834 | 0.406 | 0.452 | 0.426 |
| | | Avg | 0.159 | 0.248 | 0.451 | 0.247 | 0.215 | 0.217 | 0.095 | 0.199 | 0.241 | 0.263 | 0.055 | 0.125 | 1.764 | 0.802 | 6.528 | 0.389 | 0.374 | 0.377 |

Table 39: Ablation study on $\mathbf{Q}_i$ and $\mathbf{Q}_o$. 'Hor.' denotes prediction horizons. For *S1* and *S2*, the prediction lengths are $\{3, 6, 9, 12\}$ and $\{24, 36, 48, 60\}$, respectively.

| $\mathbf{Q}_i$ | $\mathbf{Q}_o$ | Hor. | ECL MSE | ECL MAE | Solar-Energy MSE | Solar-Energy MAE | PEMS03 MSE | PEMS03 MAE | PEMS08 MSE | PEMS08 MAE | NASDAQ (S1) MSE | NASDAQ (S1) MAE | ILI (S2) MSE | ILI (S2) MAE | COVID-19 (S2) MSE | COVID-19 (S2) MAE | Exchange MSE | Exchange MAE |
|---|---|---|---|---|---|---|---|---|---|---|---|---|---|---|---|---|---|---|
| ✗ | ✗ | H1 | 0.134 | 0.224 | 0.186 | 0.199 | 0.063 | 0.163 | 0.070 | 0.162 | 0.036 | 0.093 | 1.891 | 0.847 | 4.841 | 1.284 | 0.083 | 0.203 |
| | | H2 | 0.156 | 0.245 | 0.221 | 0.220 | 0.080 | 0.184 | 0.091 | 0.183 | 0.050 | 0.119 | 1.889 | 0.844 | 7.828 | 1.757 | 0.173 | 0.295 |
| | | H3 | 0.167 | 0.257 | 0.244 | 0.237 | 0.117 | 0.222 | 0.128 | 0.216 | 0.063 | 0.139 | 1.854 | 0.811 | 10.153 | 2.003 | 0.335 | 0.416 |
| | | H4 | 0.194 | 0.282 | 0.256 | 0.244 | 0.165 | 0.266 | 0.231 | 0.262 | 0.074 | 0.156 | 1.802 | 0.816 | 12.603 | 2.232 | 0.846 | 0.692 |
| | | Avg | 0.163 | 0.252 | 0.227 | 0.225 | 0.106 | 0.209 | 0.130 | 0.206 | 0.056 | 0.127 | 1.859 | 0.829 | 8.856 | 1.819 | 0.359 | 0.401 |
| ✗ | ✔ | H1 | 0.132 | 0.223 | 0.186 | 0.198 | 0.062 | 0.162 | 0.069 | 0.162 | 0.036 | 0.093 | 1.844 | 0.830 | 4.645 | 1.260 | 0.083 | 0.203 |
| | | H2 | 0.153 | 0.243 | 0.219 | 0.219 | 0.079 | 0.183 | 0.092 | 0.183 | 0.050 | 0.119 | 1.779 | 0.811 | 8.137 | 1.798 | 0.173 | 0.295 |
| | | H3 | 0.167 | 0.258 | 0.246 | 0.238 | 0.115 | 0.221 | 0.132 | 0.217 | 0.063 | 0.139 | 1.854 | 0.809 | 10.054 | 1.990 | 0.334 | 0.416 |
| | | H4 | 0.194 | 0.285 | 0.255 | 0.244 | 0.164 | 0.267 | 0.216 | 0.261 | 0.074 | 0.156 | 1.787 | 0.813 | 12.000 | 2.189 | 0.848 | 0.693 |
| | | Avg | 0.161 | 0.252 | 0.226 | 0.225 | 0.105 | 0.208 | 0.127 | 0.206 | 0.056 | 0.127 | 1.816 | 0.816 | 8.709 | 1.809 | 0.359 | 0.401 |
| ✔ | ✗ | H1 | 0.131 | 0.221 | 0.179 | 0.191 | 0.061 | 0.159 | 0.068 | 0.159 | 0.036 | 0.092 | 1.897 | 0.847 | 4.957 | 1.330 | 0.082 | 0.200 |
| | | H2 | 0.154 | 0.241 | 0.210 | 0.212 | 0.076 | 0.178 | 0.087 | 0.177 | 0.049 | 0.117 | 1.886 | 0.836 | 7.100 | 1.654 | 0.172 | 0.293 |
| | | H3 | 0.165 | 0.254 | 0.230 | 0.229 | 0.105 | 0.211 | 0.121 | 0.204 | 0.062 | 0.137 | 1.809 | 0.802 | 10.224 | 2.009 | 0.332 | 0.414 |
| | | H4 | 0.188 | 0.279 | 0.241 | 0.235 | 0.142 | 0.249 | 0.182 | 0.238 | 0.073 | 0.154 | 1.784 | 0.811 | 11.779 | 2.160 | 0.837 | 0.688 |
| | | Avg | 0.159 | 0.249 | 0.215 | 0.217 | 0.096 | 0.199 | 0.114 | 0.194 | 0.055 | 0.125 | 1.844 | 0.824 | 8.515 | 1.788 | 0.355 | 0.399 |
| ✔ | ✔ | H1 | 0.131 | 0.221 | 0.179 | 0.191 | 0.060 | 0.159 | 0.068 | 0.159 | 0.036 | 0.092 | 1.737 | 0.800 | 4.474 | 1.180 | 0.082 | 0.200 |
| | | H2 | 0.150 | 0.238 | 0.209 | 0.213 | 0.078 | 0.179 | 0.089 | 0.178 | 0.049 | 0.117 | 1.714 | 0.795 | 7.241 | 1.670 | 0.171 | 0.293 |
| | | H3 | 0.165 | 0.254 | 0.231 | 0.229 | 0.104 | 0.210 | 0.123 | 0.204 | 0.062 | 0.137 | 1.821 | 0.804 | 10.076 | 1.985 | 0.331 | 0.414 |
| | | H4 | 0.191 | 0.279 | 0.241 | 0.236 | 0.140 | 0.247 | 0.173 | 0.236 | 0.073 | 0.154 | 1.785 | 0.810 | 12.079 | 2.182 | 0.837 | 0.688 |
| | | Avg | 0.159 | 0.248 | 0.215 | 0.217 | 0.095 | 0.199 | 0.113 | 0.194 | 0.055 | 0.125 | 1.764 | 0.802 | 8.467 | 1.754 | 0.355 | 0.399 |

Table 40: Forecasting performance when both training and testing use Q matrices $\mathbf{Q}_i$ and $\mathbf{Q}_o$ from different datasets. $\mathcal{A} \rightarrow \mathcal{B}$ indicates that the Q matrices computed from dataset $\mathcal{A}$ are applied in experiments on dataset $\mathcal{B}$.

| Setting | ILI→ILI | | Power→ILI | | Web→ILI | |
|---|---|---|---|---|---|---|
| Metric | MSE | MAE | MSE | MAE | MSE | MAE |
| 24 | 1.737 | 0.800 | 1.846 | 0.830 | 1.887 | 0.842 |
| 36 | 1.714 | 0.795 | 1.721 | 0.801 | 1.743 | 0.807 |
| 48 | 1.821 | 0.804 | 1.849 | 0.810 | 1.875 | 0.817 |
| 60 | 1.785 | 0.810 | 1.824 | 0.814 | 1.815 | 0.813 |
| Avg | **1.764** | **0.802** | 1.810 | 0.814 | 1.830 | 0.820 |
| Setting | ILI→Power | | Power→Power | | Web→Power | |
| Metric | MSE | MAE | MSE | MAE | MSE | MAE |
| 24 | 1.391 | 0.895 | 1.343 | 0.870 | 1.393 | 0.889 |
| 36 | 1.457 | 0.902 | 1.445 | 0.903 | 1.493 | 0.915 |
| 48 | 1.612 | 0.972 | 1.559 | 0.946 | 1.680 | 0.997 |
| 60 | 1.733 | 1.035 | 1.602 | 0.971 | 1.773 | 1.048 |
| Avg | 1.548 | 0.951 | **1.487** | **0.922** | 1.585 | 0.962 |
| Setting | ILI→Web | | Power→Web | | Web→Web | |
| Metric | MSE | MAE | MSE | MAE | MSE | MAE |
| 24 | 0.208 | 0.319 | 0.185 | 0.306 | 0.186 | 0.306 |
| 36 | 0.285 | 0.369 | 0.289 | 0.364 | 0.272 | 0.356 |
| 48 | 0.377 | 0.403 | 0.400 | 0.427 | 0.365 | 0.391 |
| 60 | 0.480 | 0.476 | 0.467 | 0.466 | 0.486 | 0.481 |
| Avg | 0.337 | 0.391 | 0.335 | 0.391 | **0.327** | **0.383** |

Table 41: Training and inference resource footprint for OLinear and baseline forecasters. 'T.T.', 'T.M.', 'I.T.' and 'I.M.' denote training time, training GPU memory usage, inference time, and inference GPU memory usage, respectively. All experiments use a lookback/prediction of 96 and a batch size of 16. To ensure a fair comparison, the number of layers (or blocks) is fixed at 2, except for DLinear, which employs only 1 linear layer. 'OL' refers to OLinear. 'OOM' indicates that the experiment runs out of memory on a 24 GB GPU.

| | Model | OL (Ours) | OL-C (Ours) | Leddam 2024 | CARD 2024b | Fredformer 2024 | iTrans. 2024a | TimeMix.++ 2025a | TimeMix. 2024a | Dlinear 2023 | TimesNet 2023b | PatchTST 2023 | FreTS 2023 |
|---|---|---|---|---|---|---|---|---|---|---|---|---|---|
| ETT | Params(M) | 4.52 | 4.52 | 8.55 | 0.03 | 8.59 | 4.83 | 1.19 | 0.08 | 0.02 | 299.94 | 3.76 | 0.42 |
| | FLOPs(M) | 33.74 | 33.74 | 111.54 | 1.41 | 135.01 | 33.99 | 1396.90 | 10.72 | 0.13 | 289708 | 272.20 | 2.94 |
| | T.T. (ms/iter) | 8.09 | 7.26 | 25.02 | 29.10 | 26.48 | 10.11 | 74.74 | 27.99 | 1.42 | 501.98 | 8.37 | 3.63 |
| | T.M.(GB) | 0.20 | 0.2 | 0.18 | 0.03 | 0.24 | 0.21 | 0.27 | 0.05 | 0.02 | 5.80 | 0.14 | 0.03 |
| | I.T.(ms/iter) | 1.19 | 1.09 | 2.62 | 4.05 | 3.73 | 1.72 | 28.15 | 3.86 | 0.21 | 155.97 | 1.38 | 0.49 |
| | I.M. (MB) | 156.41 | 156.41 | 51.04 | 11.02 | 78.04 | 156.00 | 149.58 | 18.06 | 8.48 | 1396.24 | 51.83 | 14.47 |
| ECL | Params(M) | 4.79 | 4.58 | 8.56 | 1.39 | 12.12 | 4.83 | 1.19 | 0.12 | 0.02 | 300.58 | 3.76 | 0.42 |
| | FLOPs(G) | 1.65 | 1.65 | 5.32 | 5.07 | 5.55 | 1.87 | 64.02 | 1.74 | 0.01 | 293.98 | 12.48 | 0.13 |
| | T.T. (ms/iter) | 7.75 | 6.94 | 65.19 | 78.25 | 26.59 | 10.66 | 106.06 | 57.09 | 1.37 | 509.17 | 65.83 | 3.78 |
| | T.M.(GB) | 0.45 | 0.42 | 1.44 | 5.08 | 1.24 | 0.70 | 1.02 | 4.24 | 0.04 | 5.82 | 3.02 | 0.30 |
| | I.T.(ms/iter) | 2.11 | 2.09 | 14.78 | 28.21 | 7.08 | 3.46 | 34.10 | 22.79 | 0.20 | 157.13 | 20.52 | 0.48 |
| | I.M. (GB) | 0.17 | 0.17 | 0.49 | 0.63 | 0.26 | 0.23 | 0.36 | 0.82 | 0.02 | 1.41 | 0.93 | 0.23 |
| Exchange | Params(M) | 1.74 | 1.74 | 8.56 | 1.39 | 8.59 | 4.83 | 1.19 | 0.08 | 0.02 | 299.94 | 3.76 | 0.42 |
| | FLOPs(M) | 16.27 | 16.27 | 127.49 | 114.83 | 135.01 | 38.88 | 1595.72 | 12.25 | 0.15 | 287909 | 311.08 | 3.36 |
| | T.T. (ms/iter) | 7.33 | 7.53 | 24.81 | 35.57 | 28.56 | 10.14 | 97.05 | 27.46 | 1.47 | 505.39 | 9.75 | 3.51 |
| | T.M.(GB) | 0.17 | 0.17 | 0.18 | 0.17 | 0.24 | 0.21 | 0.28 | 0.06 | 0.02 | 5.80 | 0.15 | 0.03 |
| | I.T.(ms/iter) | 1.18 | 1.89 | 2.62 | 6.02 | 6.86 | 1.70 | 28.46 | 3.81 | 0.21 | 156.33 | 2.47 | 0.48 |
| | I.M. (GB) | 0.02 | 0.02 | 0.05 | 0.03 | 0.08 | 0.16 | 0.15 | 0.02 | 0.01 | 1.40 | 0.05 | 0.02 |
| Traffic | Params(M) | 6.17 | 4.69 | 8.56 | 0.98 | 11.09 | 4.83 | 4.73 | 0.12 | 0.02 | 301.69 | 3.76 | 0.42 |
| | FLOPs(G) | 4.91 | 4.91 | 15.24 | 10.47 | 14.78 | 6.45 | 588.47 | 4.66 | 0.02 | 288.72 | 33.52 | 0.36 |
| | T.T. (s/iter) | 0.02 | 0.02 | 0.19 | 0.18 | 0.10 | 0.04 | OOM | 0.17 | 0.001 | 0.50 | 0.17 | 0.01 |
| | T.M.(GB) | 1.01 | 0.95 | 3.74 | 9.50 | 5.76 | 2.95 | OOM | 11.33 | 0.07 | 5.86 | 8.01 | 0.78 |
| | I.T.(ms/iter) | 5.71 | 5.67 | 45.68 | 61.21 | 31.60 | 13.45 | 1610.66 | 67.21 | 0.20 | 15.64 | 55.10 | 3.48 |
| | I.M. (GB) | 0.43 | 0.43 | 1.25 | 1.69 | 1.84 | 1.27 | 9.35 | 2.20 | 0.04 | 1.42 | 2.44 | 0.59 |
| Weather | Params(M) | 4.52 | 4.52 | 8.56 | 0.03 | 0.50 | 4.83 | 2.37 | 0.10 | 0.02 | 299.97 | 3.76 | 0.42 |
| | FLOPs(G) | 0.10 | 0.10 | 0.34 | 0.004 | 0.01 | 0.10 | 8.36 | 0.05 | 0.000 | 291.21 | 0.82 | 0.01 |
| | T.T. (ms/iter) | 7.47 | 6.90 | 27.55 | 28.69 | 37.85 | 9.45 | 102.04 | 33.30 | 1.38 | 505.88 | 9.89 | 3.53 |
| | T.M.(GB) | 0.21 | 0.21 | 0.21 | 0.05 | 0.05 | 0.22 | 0.86 | 0.16 | 0.02 | 5.80 | 0.27 | 0.04 |
| | I.T.(ms/iter) | 1.18 | 1.05 | 3.15 | 4.11 | 7.19 | 1.66 | 30.18 | 5.10 | 0.20 | 156.27 | 2.96 | 0.48 |
| | I.M. (MB) | 39.26 | 39.26 | 70.06 | 14.45 | 18.28 | 34.88 | 316.49 | 36.03 | 9.06 | 1397.18 | 90.78 | 24.37 |
| Solar-Energy | Params(M) | 4.58 | 4.54 | 8.56 | 1.39 | 4.61 | 4.83 | 2.37 | 0.12 | 0.02 | 300.21 | 3.76 | 0.42 |
| | FLOPs(G) | 0.68 | 0.68 | 2.22 | 2.05 | 1.20 | 0.72 | 46.81 | 0.74 | 0.003 | 290.64 | 5.33 | 0.06 |
| | T.T. (ms/iter) | 9.12 | 7.46 | 48.64 | 36.28 | 31.77 | 9.97 | 515.44 | 35.90 | 1.44 | 506.58 | 28.89 | 3.60 |
| | T.M.(GB) | 0.30 | 0.30 | 0.68 | 2.22 | 0.61 | 0.37 | 5.11 | 1.83 | 0.03 | 5.81 | 1.34 | 0.15 |
| | I.T.(ms/iter) | 1.20 | 1.08 | 7.65 | 10.44 | 5.16 | 1.96 | 189.20 | 8.54 | 0.20 | 156.66 | 8.49 | 0.48 |
| | I.M. (GB) | 0.21 | 0.21 | 0.23 | 0.28 | 0.16 | 0.18 | 1.94 | 0.36 | 0.01 | 1.40 | 0.42 | 0.11 |
| PEMS07 | Params(M) | 4.75 | 3.19 | 8.56 | 1.39 | 13.44 | 4.83 | 2.38 | 0.12 | 0.02 | 301.73 | 3.76 | 0.42 |
| | FLOPs(G) | 3.67 | 3.67 | 15.65 | 16.10 | 20.53 | 6.66 | 301.92 | 4.77 | 0.02 | 290.97 | 34.34 | 0.37 |
| | T.T. (s/iter) | 0.01 | 0.01 | 0.19 | 0.28 | 0.11 | 0.04 | OOM | 0.17 | 0.002 | 0.51 | 0.19 | 0.01 |
| | T.M.(GB) | 0.83 | 0.76 | 3.85 | 13.99 | 6.42 | 3.07 | OOM | 11.62 | 0.07 | 5.86 | 9.18 | 0.79 |
| | I.T.(ms/iter) | 1.21 | 1.10 | 47.06 | 94.01 | 36.54 | 13.92 | 1370.91 | 68.96 | 0.20 | 156.78 | 56.49 | 0.50 |
| | I.M. (GB) | 0.43 | 0.43 | 1.28 | 1.74 | 2.03 | 1.32 | 12.37 | 2.25 | 0.04 | 1.43 | 2.50 | 0.61 |

Table 42: Efficiency comparison of Leddam, iTransformer, PatchTST, and Fredformer before and after replacing multi-head self-attention with the NormLin module. In Leddam, only the *cross-channel attention* module is updated. Metrics include number of learnable parameters (Params), FLOPs, training time (T.T.), training memory (T.M.), inference time (I.T.), and inference memory (I.M.). All experiments use a lookback/prediction length of 96 and a batch size of 16. Notably, in its official implementation, Fredformer employs Nystromformer [Xiong et al., 2021]—an approximate yet more efficient self-attention mechanism—for the ETTm1, ECL, and Weather datasets.

| Model | | Leddam 2024 | | iTrans. 2024a | | PatchTST 2023 | | Fredformer 2024 | |
|---|---|---|---|---|---|---|---|---|---|
| | | Attn. | NormLin | Attn. | NormLin | Attn. | NormLin | Attn. | NormLin |
| ETTm1 | Params(M) | 8.55 | 7.51 | 4.83 | 2.21 | 3.76 | 2.71 | 8.59 | 8.39 |
| | FLOPs(M) | 111.54 | 104.15 | 33.99 | 15.51 | 272.20 | 182.20 | 135.01 | 79.10 |
| | T.T. (ms/iter) | 25.02 | 23.67 | 10.11 | 7.10 | 8.37 | 6.82 | 26.48 | 10.06 |
| | T.M.(GB) | 0.18 | 0.16 | 0.21 | 0.17 | 0.14 | 0.12 | 0.24 | 0.18 |
| | I.T.(ms/iter) | 2.62 | 2.50 | 1.72 | 1.25 | 1.38 | 1.14 | 3.73 | 1.36 |
| | I.M. (MB) | 51.04 | 47.03 | 156.00 | 146.20 | 51.83 | 50.45 | 78.04 | 52.44 |
| ECL | Params(M) | 8.56 | 7.71 | 4.83 | 2.41 | 3.76 | 2.71 | 12.12 | 12.92 |
| | FLOPs(G) | 5.32 | 4.88 | 1.87 | 0.81 | 12.48 | 8.35 | 5.55 | 6.04 |
| | T.T. (ms/iter) | 65.19 | 64.93 | 10.66 | 7.39 | 65.83 | 50.82 | 26.59 | 21.03 |
| | T.M.(GB) | 1.44 | 1.41 | 0.70 | 0.41 | 3.02 | 2.78 | 1.24 | 1.27 |
| | I.T.(ms/iter) | 14.78 | 14.52 | 3.46 | 1.01 | 20.52 | 14.08 | 7.08 | 6.71 |
| | I.M. (GB) | 0.49 | 0.48 | 0.23 | 0.10 | 0.93 | 0.99 | 0.26 | 0.26 |
| Weather | Params(M) | 8.56 | 7.52 | 4.83 | 2.21 | 3.76 | 2.71 | 0.50 | 0.48 |
| | FLOPs(G) | 0.34 | 0.31 | 0.10 | 0.05 | 0.82 | 0.55 | 0.01 | 0.003 |
| | T.T. (ms/iter) | 27.55 | 27.21 | 9.45 | 7.49 | 9.89 | 7.47 | 37.85 | 28.19 |
| | T.M.(GB) | 0.21 | 0.19 | 0.22 | 0.18 | 0.27 | 0.24 | 0.05 | 0.03 |
| | I.T.(ms/iter) | 3.15 | 3.08 | 1.66 | 0.99 | 2.96 | 1.12 | 7.19 | 3.55 |
| | I.M. (MB) | 70.06 | 66.05 | 34.88 | 25.51 | 90.78 | 92.69 | 18.28 | 13.77 |
| Solar-Energy | Params(M) | 8.56 | 7.54 | 4.83 | 2.24 | 3.76 | 2.71 | 4.61 | 4.32 |
| | FLOPs(G) | 2.22 | 2.06 | 0.72 | 0.32 | 5.33 | 3.56 | 1.20 | 0.91 |
| | T.T. (ms/iter) | 48.64 | 48.12 | 9.97 | 7.40 | 28.89 | 22.31 | 31.77 | 12.32 |
| | T.M.(GB) | 0.68 | 0.67 | 0.37 | 0.26 | 1.34 | 1.23 | 0.61 | 0.46 |
| | I.T.(ms/iter) | 7.65 | 7.59 | 1.96 | 1.01 | 8.49 | 5.83 | 5.16 | 2.89 |
| | I.M. (GB) | 0.23 | 0.22 | 0.18 | 0.18 | 0.42 | 0.44 | 0.16 | 0.09 |
| PEMS07 | Params(M) | 8.56 | 9.07 | 4.83 | 3.77 | 3.76 | 2.71 | 13.44 | 15.29 |
| | FLOPs(G) | 15.65 | 13.93 | 6.66 | 2.75 | 34.34 | 22.96 | 20.53 | 17.57 |
| | T.T. (s/iter) | 0.19 | 0.18 | 0.04 | 0.01 | 0.19 | 0.14 | 0.11 | 0.06 |
| | T.M.(GB) | 3.85 | 3.74 | 3.07 | 0.82 | 9.18 | 7.57 | 6.42 | 3.16 |
| | I.T.(ms/iter) | 47.06 | 45.16 | 13.92 | 3.02 | 56.49 | 38.63 | 36.54 | 19.16 |
| | I.M. (GB) | 1.28 | 1.28 | 1.32 | 0.27 | 2.50 | 2.70 | 2.03 | 0.60 |

