# OpenReview forum: "OLinear: A Linear Model for Time Series Forecasting in Orthogonally Transformed Domain"
_NeurIPS.cc/2025/Conference — NeurIPS 2025 poster_

### Official Review · Reviewer_jUXb · 2025-06-15

**Clarity:** 3
**Significance:** 3
**Originality:** 3
**Rating:** 5
**Confidence:** 5

**Summary:**

The paper introduces *OLinear*, a multivariate time-series forecaster that operates in an *orthogonally transformed* domain rather than the raw time domain. For each input window, the authors compute the temporal Pearson-correlation matrix, perform an eigen-decomposition, and project the signals onto the resulting orthogonal basis. They then replace self-attention with *NormLin*. Extensive experiments indicate improvements over strong baselines, and the accompanying code is available.

**Questions:**

1. *Could you run a statistical significance test across ≥ 5 random seeds for every dataset–horizon pair and report the resulting p-values?*

2. The existing work already demonstrates that a specific optimiser combined with a single channel-wise attention head (complexity $O(N^2)$) raises attention-matrix rank and boosts accuracy while using a channel attention mechanism to capture multivariate correlations. *Please clarify the novelty and compare the baseline with this existing work. Especially, how the model compares to a shallow transformer trained with a strong regularization like SAM and a channel-wise attention mechanism ?*

3. The claim that OrthoTrans removes step-wise correlations needs quantitative confirmation in both domains.  *Could you provide a metric that shows (i) a drop in the off-diagonal magnitude of the time-domain covariance matrix and (ii) that no new dependencies appear in the frequency domain—for instance, by comparing the Frobenius norm of the off-diagonal entries in the covariance matrix and in the cross-spectral density before and after applying OrthoTrans?*

4. NormLin removes non-linearity yet keeps quadratic cost.  *Is there a way to reduce this cost?*

5. In the experiments, you use different learning rate, model dimension, embedding size and number of blocks for each dataset. *Could you please freeze one single architecture, run it on all the datasets and compare this performance with the best architecture for each dataset?*

**Ethical Concerns:**

["NO or VERY MINOR ethics concerns only"]

**Final Justification:**

The authors have addressed my initial concerns. They have also committed to revise the narrative on weight-matrix rank in the camera-ready.

**Limitations:**

While the authors include a brief limitations paragraph in Appendix, a fuller discussion into the main paper, covering details like compute cost and dataset-specific hyper-parameters, is needed.

**Paper Formatting Concerns:**

No concern

**Quality:**

3

**Strengths And Weaknesses:**

## Strengths

1. The authors evaluate OLinear on 24 publicly available datasets spanning electricity, traffic, weather and financial domains, and consider multiple forecast horizons for each dataset. Reporting 99 % confidence intervals indicates an effort to convey result variability rather than single-run numbers.

2. The paper is well structured: problem statement, method, ablations and main tables flow logically. Figures are labelled and referenced properly, and notation is introduced before use, which makes the technical content easy to follow.

3. Code and data-processing scripts are released, allowing reviewers and future researchers to reproduce the results or to integrate the proposed method into their own pipelines with minimal effort.

4. The plug-and-play lowers the entry barrier for practitioners who already have Transformer-based models and wish to test the proposed ideas without redesigning their entire architecture.

5. While Fourier and wavelet bases dominate current transform-domain forecasters, constructing an orthogonal basis from the data’s own temporal correlation matrix has been less explored. This adaptive approach could inspire further work.

## Weaknesses

1. Results seem to degrade when the look-back window exceeds 96, casting doubt on long-horizon robustness. For example, with 15-minute data, 96 steps cover only 24 h, yet the model may need to predict up to 720 steps (≈ 1 week) ahead in long-term forecasting, raising concerns about interpretability.

2. Many of the hyper-parameters, such as model dimension, embedding size, number of blocks and learning rate, are dataset-dependent.

3. The assumption of decorrelation after OrthoTrans is not demonstrated. Two series can be decorrelated in time yet correlated in frequency; an orthogonal transform does not necessarily remove such cross-frequency dependencies. Moreover, decorrelating in the time domain removes second-order dependencies but says nothing about higher-order or non-linear dependencies.

4. *The novelty remains unclear.* The manuscript does not cite the previous existing work [1]. That work already establishes that forecasting Transformers suffer from excessive sharpness, and that this can be mitigated by (i) using a different optimiser and (ii) reducing the architecture to a *single* channel-wise attention head—whose complexity is exactly $O(N^2)$, the same as NormLin with $h=1$ (see the Table 1). These two changes were shown to raise the attention-matrix rank and therefore boost the model’s expressivity. The study re-uses those insights in a purely linear setting, which by construction loses the non-linear capacity of a Transformer, yet none of this prior art is acknowledged. Consequently, the claims regarding higher rank, multivariate correlations, and expressivity are not novel in the context of multivariate time-series forecasting.

5. NormLin trades away non-linearity while retaining the same $O(N^2)$ cost.

**References**

[1] SAMformer : Unlocking the potential of Transformers in Time Series Forecasting

---

> ### Author Rebuttal · Authors · 2025-07-29
>
> Thank you for your comments. We sincerely appreciate the reviewer’s recognition of our "**well-structured**" presentation, the plug-and-play modules that "**lower the entry barrier**", and an adaptive design that could "**inspire further work**". Below are our responses:
>
> > Q1: On the p-value
>
> - We conduct a significance test (using 7 random seeds) using Student’s t-test between OLinear and iTransformer across various datasets and prediction horizons. As shown below, the better model is in bold when the improvement is statistically significant at the level 0.05 (p-value < 0.05). In other words, OLinear exhibits a clear performance advantage over iTransformer.
>
> | **Setting**  | **OLinear (MAE)** | **iTrans.(MAE)** | **p-value between OLinear and iTrans.** |
> |-|-|-|-|
> | ECL: 96 | **0.221±4e-4**| 0.240±4e-4 | 5.12E-11|
> | ECL: 192 | **0.238±1e-3**| 0.253±2e-3 | 3.06E-06|
> | ECL: 336 | **0.254±1e-3**| 0.269±1e-3| 9.96E-07|
> | ECL: 720 | **0.279±2e-3**| 0.317±7e-3 | 5.59E-05|
> | Traffic: 96  | **0.226±2e-4**| 0.268±1e-3 | 1.61E-12|
> | Traffic: 192 | **0.241±4e-4**| 0.276±1e-3 | 9.48E-10|
> | Traffic: 336 | **0.250±3e-4**| 0.283±4e-4| 1.87E-13|
> | Traffic: 720 | **0.270±4e-4**| 0.302±4e-4| 6.15E-13|
> | Weather: 96  | **0.190±1e-3**| 0.214±3e-4| 1.29E-08|
> | Weather: 192 | **0.235±2e-3**| 0.254±1e-3| 1.04E-06|
> | Weather: 336 | **0.280±2e-3**| 0.296±1e-3| 1.93E-06|
> | Weather: 720 | **0.333±2e-3**| 0.349±4e-4| 5.68E-06|
>
>
> - We report the performance with increasing lookback lengths in **Appendix I.3**. OLinear demonstrates consistent improvements as the lookback horizon increases from 48 to 720, and consistently outperforms state-of-the-art forecasters.
>
> - Regarding robustness across random seeds, please refer to **Appendix F**, where OLinear exhibits smaller standard deviations and narrower 99% confidence intervals compared to TimeMixer++ and iTransformer.
>
> - We would like to emphasize that MSE and MAE metrics typically degrade in long-term forecasting as the prediction horizon increases. As shown in **Table 15** (Page 28), this phenomenon is observed in state-of-the-art time series forecasters, such as TimeMixer++ (ICLR 2025), iTransformer (ICLR 2024), FilterNet (NeurIPS 2024). This phenomenon is reasonable because future series inherently contain greater uncertainty as the horizon extends, much like how predicting the temperature two weeks ahead is less accurate than predicting tomorrow’s temperature.
>
> > Q2: On novelty and comparison with SAMformer
>
> - SAMformer (ICML 2024) introduces the Sharpness-Aware Minimization (SAM) mechanism—originally proposed in [1] (ICLR 2021)—into time series forecasting. While channel-wise attention was first introduced by iTransformer, SAMformer adopts the same practice. Its key contribution is a new **loss function** that smooths the loss landscape of Transformers: $$\mathcal{L}^{SAM}\_{train}(\omega ) = \underset{\left \| \epsilon  \right \| <\rho  }{max}\mathcal{L}\_{train}(\omega +\rho), $$ where $\rho >0$ is a hyper-parameter. This can be implemented by the SAM optimizer [1].
>
> - Instead of modifying the loss function, we focus on the **architecture design** and propose NormLin. Compared with vanilla self-attention, NormLin offers four advantages: lower computational cost, higher-rank attention matrices, smoother gradient flow, and an enlarged optimization space (see Lines 182-198 in the paper). NormLin "can be plugged into existing forecasters to boost performance **with minimal engineering effort" (by Reviewer 5fFn)**. Moreover, our **linear-based** model, OLinear, outperforms state-of-the-art Transformer-based forecasters—including SAMformer. We will **cite and discuss SAMformer in our final draft**.
>
> - We compare MSE and standard deviation between OLinear and SAMformer. The SAMformer numbers are taken from its original paper, which uses a look-back window of 512 and thus enjoys extra performance gains. Even under this handicap, OLinear achieves **lower MSE** and **smaller standard deviations** in most cases.
>
> | **Dataset** | **Hor.** | **OLinear (Look-back: 96)** | **SAMformer (Look-back: 512)** |
> |-|-|-|-|
> | ETTm2| 96   | **0.169±1e-4**  | 0.181±0.005 |
> | | 192  | **0.232±1e-4**  | 0.233±0.002 |
> | | 336  | 0.291±2e-4| **0.285±0.001** |
> | | 720  | 0.389±4e-4| **0.375±0.001** |
> | ECL  | 96   | **0.131±4e-4**| 0.155±0.002 |
> | | 192  | **0.150±1e-3**| 0.168±0.001 |
> | | 336  | **0.165±1e-3**| 0.183±0.000 |
> | | 720  | **0.191±2e-3**| 0.219±0.000 |
> | Exchange    | 96   | **0.082±3e-4**| 0.161±0.007 |
> | | 192  | **0.171±2e-3**| 0.246±0.009 |
> | | 336  | **0.331±3e-3**| 0.368±0.006 |
> | | 720  | **0.837±0.018**     | 1.003±0.018 |
> | Weather     | 96   | **0.153±1e-3**| 0.197±0.001 |
> | | 192  | **0.200±2e-3**| 0.235±0.000 |
> | | 336  | **0.258±3e-3**| 0.276±0.001 |
> | | 720  | 0.337±4e-3| **0.334±0.000** |
>
> - To ensure a fair comparison, we uniformly set the lookback length to 96 and report the following MAEs. The average results are reported across 3 random seeds for SAMformer, using the recommended learning rates and $\rho$ values.
>   - On average, OLinear outperforms SAMformer by a **substantial margin of 12.1%**.
>   - Notably, the proposed **NormLin layer improves the performance of SAMformer** by 10.6% and 5.8% on the Traffic and ECL datasets, respectively.
>
> | **Dataset** | **Hor.** | **OLinear** | **SAMformer** | **SAMformer+NormLin** |
> |-|-|-|-|--|
> | ETTh1 | 96 | 0.382 | 0.396 | 0.392 |
> | | 192| 0.414 | 0.425 | 0.419 |
> | | 336| 0.438 | 0.444 | 0.439 |
> | | 720| 0.462 | 0.469 | 0.459 |
> | ETTm2 | 96 | 0.249 | 0.268 | 0.262 |
> | | 192| 0.290 | 0.305 | 0.303 |
> | | 336| 0.328 | 0.343 | 0.340 |
> | | 720| 0.387 | 0.399 | 0.397 |
> | ECL | 96 | 0.221 | 0.285 | 0.267 |
> | | 192| 0.238 | 0.288 | 0.269 |
> | | 336| 0.254 | 0.301 | 0.284 |
> | | 720| 0.279 | 0.335 | 0.318 |
> | Traffic | 96 | 0.226 | 0.404 | 0.355 |
> | | 192| 0.241 | 0.367 | 0.335 |
> | | 336| 0.250 | 0.379 | 0.339 |
> | | 720| 0.270 | 0.401 | 0.358 |
> | Weather | 96 | 0.190 | 0.231 | 0.208 |
> | | 192| 0.235 | 0.269 | 0.251 |
> | | 336| 0.280 | 0.304 | 0.290 |
> | | 720| 0.333 | 0.350 | 0.339 |
> | Echange | 96 | 0.200 | 0.203 | 0.210 |
> | | 192| 0.293 | 0.303 | 0.302 |
> | | 336| 0.414 | 0.426 | 0.416 |
> | | 720| 0.688 | 0.712 | 0.689 |
>
>
>
> > Q3: On the quantitation of temporally decorrelation
>
> - We report the **off-diagonal Frobenius norm** of the correlation matrices (size: $96\times 96$) for the original series, and for the series after OrthoTrans or FFT, using a window size of 96. The results below show that OrthoTrans effectively removes temporal correlations and performs better than the FFT operation in decorrelation.
>
> | **Dataset** | **Original** | **OrthoTrans** | **FFT** |
> |-|--|--|-|
> | ETTh1 | 61.36| 1.46 | 2.58|
> | ETTm2 | 83.04| 1.26 | 5.22|
> | ECL | 45.95| 1.78 | 2.26|
> | Traffic | 34.80| 1.79 | 3.29|
> | Weather | 61.70| 2.86 | 19.61 |
>
> - We also argue that examining correlations in the frequency domain of the feature series after OrthoTrans is of limited relevance: 1) OLinear does not involve any FFT operations. 2) Whether there exists an orthogonal transformation that simultaneously mitigates both temporal- and frequency-domain correlations is a nontrivial question. Further exploration could be a future direction of this work. We will emphasize that the decorrelation is performed **temporally** in the final draft.
>
> > Q4: On quadratic cost of NormLin
>
> - In our view, $\mathcal{O}(N^2)$ is the minimal cost required to model $N$-channel correlations, as even a single linear layer incurs this complexity.
>
> - A straightforward way to reduce this cost is to introduce a **bottleneck** block: the channels are first projected down to a **fixed width** (e.g., 64), the NormLin module is applied in the fixed-dimensional space, and then the representation is projected back to $N$ channels. This reduces the computational complexity to $\mathcal{O}(N)$. As shown below, this variant tends to degrade performance, particularly on datasets with a large number of channels, such as ECL and Traffic.
>
> | **Dataset** | **Hor.** | **Original (MAE)** | **Bottleneck (MAE, fixed width=64)** |
> |-|-|-|-|
> | ETTh1 | 96 | 0.382| 0.382|
> | | 192| 0.414| 0.415|
> | | 336| 0.438| 0.443|
> | | 720| 0.462| 0.468|
> | ECL | 96 | 0.221| 0.241|
> | | 192| 0.238| 0.256|
> | | 336| 0.254| 0.270|
> | | 720| 0.279| 0.291|
> | Traffic | 96 | 0.226| 0.263|
> | | 192| 0.241| 0.278|
> | | 336| 0.250| 0.283|
> | | 720| 0.270| 0.298|
> | Weather | 96 | 0.190| 0.190|
> | | 192| 0.235| 0.239|
> | | 336| 0.280| 0.281|
> | | 720| 0.333| 0.337|
> | PEMS-03 | 12 | 0.159| 0.160|
> | | 24 | 0.179| 0.182|
> | | 48 | 0.210| 0.224|
> | | 96 | 0.247| 0.264|
>
> > Q5: On hyperparameters
>
> - Because there are substantial differences among datasets, it is common practice (e.g., iTransformer and SAMformer) to fine-tune certain hyperparameters to maximize model performance.
>
> - In the paper, the hyperparameter value ranges are provided in Appendix D. Note that the embedding size $d$ is fixed at 16 for all experiments. We also evaluate the **hyperparameter sensitivity** of OLinear in **Appendix I.6**. As shown in **Figure 17** and **Table 30**, OLinear exhibits robust performance across these hyperparameters.
>
> - Furthermore, we freeze a single architecture (learning rate $lr = 5\times10^{-4}$, model dimension $D = 512$, number of blocks $L = 3$, embedding size $d = 16$) and report the average results across four prediction lengths. Compared to the results reported in the paper, the performance degradation is within 1%.
>
> | **Dataset** | **In paper** |  | **Fixed Para.** |  |
> |-|-|-|-|-|
> | | MSE| MAE| MSE | MAE|
> | ETTh1 |0.424| 0.424| 0.426 | 0.425|
> | ETTh2 |0.367| 0.388| 0.369 | 0.390|
> | ETTm1 |0.374| 0.377| 0.377 | 0.381|
> | ETTm2 |0.270| 0.313| 0.271 | 0.313|
> | ECL |0.159| 0.248| 0.159 | 0.248|
> | Traffic |0.451| 0.247| 0.451 | 0.247|
> | Weather | 0.237| 0.260| 0.240 | 0.262|
>
> **We hope these responses can fully address your concerns. Thank you again for your comments!**
>
> *Reference*
>
> [1] Sharpness-aware minimization for efficiently improving generalization. ICLR 2021.

---

> ### Comment · Reviewer_jUXb · 2025-08-01
> **Response to the authors**
>
> Thank you for the well-structured rebuttal. Your additional experiments and clarifications resolve most of my earlier concerns. Below are my remarks.
>
> #### Q1: Significance testing
>
> The inclusion of Student’s t-tests across seven random seeds strengthens the statistical analysis and provides a solid estimate of the robustness of your model.
>
> #### Q2: Novelty and comparison with existing baselines
>
> The comparison with SAmformer and the reported average 12.1% MAE gain positions OLinear as the stronger baseline. However, what would further elevate this section is a theoretical discussion that contrasts the subspaces produced by OrthoTrans with those implied by a model like SAMformer’s rank-regularised attention. It is still unclear to me whether OrthoTrans provides representational capacity beyond what a simple attention + linear layer equipped with rank regularization, or even LoRA-style methods, can already achieve.
>
> #### Q3: Quantifying decorrelation
>
> The updated table of off-diagonal Frobenius norms is very helpful. It shows that OrthoTrans efficiently slashes time-domain correlations on every dataset. Given your statement that OrthoTrans is intended only for temporal decorrelation and that extending it to the frequency domain is a separate, non-trivial research question, the current analysis feels sufficient for this paper’s scope. However, a brief sentence in the main text clarifying this design choice would head off potential misunderstandings.
>
> #### Q4: Quadratic cost
>
> The bottleneck experiment shows the trade-off between complexity and accuracy: reducing the model size lowers the computational cost but also leads to some loss in accuracy on specific datasets. Showing how the performance (MAE or MSE) and latency change as the bottleneck size varies would help readers decide if the full quadratic version of the method is worth it for their own use cases. I suggest adding such an experiment to the paper to make this trade-off more explicit.
>
> #### Q5: Hyper-parameter robustness
>
> Freezing the learning rate, model dimension, number of blocks, and embedding size while losing $\leq$ 1 % accuracy attests to the method’s stability.
>
> #### General Comment
>
> The rebuttal effectively strengthens the paper. Further clarifying the main theoretical point raised in Q2 would help improve the overall evaluation.

---

> ### Author Response · Authors · 2025-08-03
> **Responses to Reviewer jUXb's remarks (Part 1)**
>
> We could not be happier to receive such heartwarming remarks. We deeply appreciate that you found our initial responses resolved most of your concerns. Below, we provide our further responses:
>
> > Point 1: Theoretical discussion of NormLin and self-attention
>
> - Thanks for your suggestion! We fully agree that further theoretical analysis will elevate our work. We would like to conduct the analysis from the perspective of **degrees of freedom (DoF)** of the attention matrices. The number of DoF in a matrix/vector is the number of elements needed to completely specify it. The DoF satisfy the following properties [1]:
>
>   - Strictly monotonic functions (e.g., Softplus, exp) that operate element-wise do not impose cross-element constraints, and thus do not change the DoF.
>   - For a vector $\mathbf{x} \in \mathbb{R}^N$, its DoF is $N$. When imposing the L1 normalization, i.e., $\mathrm{Norm}_{\mathrm{L1}}(\mathbf{x})$, the DoF is reduced by 1. Similarly, for a matrix $\mathbf{X} \in \mathbb{R}^{N\times N}$, row-wise L1 normalization reduces its DoF from $N^2$ to $N^2 - N$.
>
> - Based on these, we have the following DoF equations for NormLin and self-attention:
>
> | **Metric** | $\mathbf{W} \in \mathbb{R}^{N  \times N} $ | $\mathrm{Norm} _{\mathrm{L1} }(\mathrm{Softplus} (\mathbf{W} )) $ | $\mathbf{Q} \mathbf{K} ^T, (\mathbf{Q}, \mathbf{K} \in \mathbb{R}^{N \times d} )$ | $\mathrm{Softmax} (\mathbf{Q} \mathbf{K} ^T)$ |
> |-|-|-|-|-|
> |DoF|$N^2$|$N^2-N$|$f(N,d)$|$f(N,d)-N$|
> |Best Rank|$N$|$N$|$\mathrm{min}(N,d)$|$N$|
> |Worst Rank|0|1|0|1|
>
> - In the above table, we omit scalar parameters for clarity, as they do not affect DoF or ranks. $f(N,d)$ is a piecewise function with a maximum value of $N^2$, defined as $f(N,d)=2Nd-d^2$ if $d \le N$ and $f(N,d)=N^2$ otherwise. The subtraction of $d^2$ in $f(N,d)$ can be understood from the fact that, for any invertible matrix $\mathbf{P} \in \mathbb{R}^{d\times d}$, $\mathbf{Q}'=\mathbf{Q}\mathbf{P}$ and $\mathbf{K}'=\mathbf{K}(\mathbf{P}^{-1})^{\top}$ satisfy $\mathbf{Q}'\mathbf{K}'^{\top}=\mathbf{Q}\mathbf{K}^{\top}$. In SAMformer, the default hidden dimension d is set to 16, which is smaller than the channel count $N$ of most common datasets.
>
> - We can see that the NormLin layer benefits from a larger DoF of the weight matrix (third column) compared to that of the attention matrix in self-attention (last column), since $N^2-N \ge f(N,d)-N $.  This indicates a larger search space of attention/weight matrices and potentially better representational capacity [2].
>
> - Empirically, we report the **minimum singular values (SVs), median SVs, ranks, and entropies** of the attention and weight matrices for SAMformer and SAMformer+NormLin (denoted as NormLin in the following table). The lookback and prediction lengths are set to 96. Since the attention matrix in SAMformer is input-dependent, we report the mean, minimum, and maximum values (denoted as S_mean, S_min, S_max, respectively) across the test sets for SAMformer. The matrix rank is computed as the number of SVs greater than $10^{-3}$. The SAM optimizer is used with the recommended hyperparameters in the SAMformer paper.
>
> | **Dataset** | **Min Attn. SV** |  |  |  | **Median Attn. SV** |  |  |  | **Attn. Rank** |  |  |  | **Attention Entropy** |  |  |  |
> |--|-|--|-|-|-|-|-|-|-|-|-|-|-|-|-|-|
> | | NormLin| S_mean | S_min| S_max| NormLin | S_mean | S_min| S_max| NormLin| S_mean | S_min| S_max| NormLin | S_mean | S_min| S_max|
> | ETTh1 | 0.009| 0.002| 0.000| 0.042| 0.311 | 0.318| 0.032| 0.776| 7| 6.620| 5| 7| 1.546 | 1.473| 0.940| 1.843|
> | ETTm2 | 0.001| 0.013| 0.000| 0.326| 0.473 | 0.418| 0.011| 0.997| 7| 6.780| 4| 7| 1.277 | 0.823| 0.148| 1.375|
> | ECL | 0.000| 0.000| 0.000| 0.000| 0.018 | 0.000| 0.000| 0.000| 311| 83.825 | 68 | 100| 5.701 | 4.887| 4.336| 5.487|
> | Traffic | 0.000| 0.000| 0.000| 0.000| 0.013 | 0.000| 0.000| 0.000| 828| 140.979| 93 | 197| 6.657 | 5.536| 3.384| 6.405|
> | Weather | 0.005| 0.000| 0.000| 0.000| 0.106 | 0.015| 0.002| 0.081| 21 | 14.626 | 12 | 17 | 2.828 | 2.596| 1.218| 2.987|
> | Exchange| 0.019| 0.003| 0.000| 0.020| 0.108 | 0.053| 0.005| 0.197| 8| 7.676| 6| 8| 1.876 | 1.822| 1.080| 2.060|
>
>
> - We can observe that, compared with the vanilla SAMformer, the plug-in NormLin layer generally **produces larger SVs, higher ranks, and, at the same time, better prevents attention entropy collapse**. These results indicate better representational capacity than simple attention with rank regularization. In addition, this rank improvement phenomenon is consistent with the findings in Appendix E.2 of our paper.
>
> References:
>
> [1] Strang, Linear Algebra and Its Applications, International Thomson Publishing, 3rd edition.
>
> [2] Attention is not all you need: pure attention loses rank doubly exponentially with depth. ICML 2021.

---

> ### Author Response · Authors · 2025-08-03
> **Responses to Reviewer jUXb's remarks (Part 2)**
>
> > Point 1 (continued): Theoretical discussion of NormLin and self-attention
>
> - To further demonstrate the DoF effects on forecasting performance, we introduce the NormLin_LoRA with the thought of low-rank approximation: $$\mathrm{NormLin} _{\mathrm{LoRA} }=\mathrm{Norm} _{\mathrm{L1} }(\mathrm{Softplus} (\mathbf{AB}^{\top} )),$$
> where $\mathbf{A}, \mathbf{B} \in \mathbb{R}^{N \times d}$. According to the first table, $\mathrm{NormLin} _{\mathrm{LoRA} }$ has fewer DoF than the original NormLin. We report the following MAEs under the architecture of SAMformer. As expected, NormLin outperforms the LoRA-style variants.
>
> | **Dataset** | **Hor.** | **SAMformer+NormLin** | **SAMformer+NormLin_LoRA (d:8)** | **SAMformer+NormLin_LoRA (d:16)** |
> |-|-|-|-|-|
> | ETTh1 | 96 | 0.393 | 0.395| 0.394 |
> | | 192| 0.422 | 0.425| 0.424 |
> | | 336| 0.444 | 0.446| 0.446 |
> | | 720| 0.487 | 0.489| 0.489 |
> | ETTm2 | 96 | 0.262 | 0.264| 0.264 |
> | | 192| 0.303 | 0.305| 0.304 |
> | | 336| 0.340 | 0.342| 0.342 |
> | | 720| 0.397 | 0.399| 0.399 |
> | ECL| 96 | 0.267 | 0.270| 0.269 |
> | | 192| 0.269 | 0.272| 0.269 |
> | | 336| 0.284 | 0.288| 0.286 |
> | | 720| 0.318 | 0.322| 0.320 |
> | Traffic  | 96 | 0.355 | 0.357| 0.356 |
> | | 192| 0.335 | 0.337| 0.336 |
> | | 336| 0.339 | 0.341| 0.340 |
> | | 720| 0.358 | 0.360| 0.359 |
> | Weather  | 96 | 0.208 | 0.219| 0.210 |
> | | 192| 0.251 | 0.260| 0.253 |
> | | 336| 0.290 | 0.295| 0.292 |
> | | 720| 0.339 | 0.343| 0.345 |
> | Echange  | 96 | 0.221 | 0.224| 0.226 |
> | | 192| 0.302 | 0.304| 0.306 |
> | | 336| 0.416 | 0.419| 0.416 |
> | | 720| 0.689 | 0.693| 0.692 |
>
> - We also report the SVs, ranks, and entropy of the weight matrices for the LoRA variants under the SAMformer architecture. As shown below, the original NormLin exhibits higher SVs, ranks, and entropy than its LoRA counterparts.
>
> | **Dataset** | **Min Attn. SV** | |  | **Median Attn. SV** | |  | **Attn. Rank** | |  | **Attention Entropy** | |  |
> |-|-|-|-|-|-|-|-|-|-|--|-|-|
> | | NormLin| LoRA (d8) | LoRA (d16) | NormLin | LoRA (d8) | LoRA (d16) | NormLin| LoRA (d8) | LoRA (d16) | NormLin | LoRA (d8) | LoRA (d16) |
> | ETTh1 | 0.009| 0.010 | 0.013| 0.311 | 0.301 | 0.280| 7| 7 | 7| 1.546 | 1.293 | 1.248|
> | ETTm2 | 0.001| 0.001 | 0.001| 0.473 | 0.414 | 0.431| 7| 7 | 7| 1.277 | 1.118 | 1.075|
> | ECL | 0.000| 0.000 | 0.000| 0.018 | 0.000 | 0.000| 311| 34| 98 | 5.701 | 5.641 | 5.514|
> | Traffic | 0.000| 0.000 | 0.000| 0.013 | 0.000 | 0.000| 828| 48| 189| 6.657 | 6.605 | 6.494|
> | Weather | 0.005| 0.000 | 0.002| 0.106 | 0.033 | 0.124| 21 | 17| 21 | 2.828 | 2.570 | 2.291|
> | Exchange| 0.019| 0.001 | 0.005| 0.108 | 0.123 | 0.078| 8| 7 | 8| 1.876 | 1.231 | 1.347|
>
> - We also compare the NormLin layer with its LoRA variant **under the OLinear architecture**. As shown below, the reduced DoF of the LoRA variant results in degraded performance.
>
> | **Dataset** | **Hor.** | **OLinear** || **OLinear+LoRA(d64)** ||
> |-|-|-|-|-|-|
> | || MSE | MAE| MSE| MAE|
> | ETTh1 | 96 | 0.360 | 0.382| 0.363| 0.384|
> | | 192| 0.416 | 0.414| 0.415| 0.415|
> | | 336| 0.457 | 0.438| 0.461| 0.441|
> | | 720| 0.463 | 0.462| 0.469| 0.466|
> | ETTm2 | 96 | 0.169 | 0.249| 0.171| 0.250|
> | | 192| 0.232 | 0.290| 0.233| 0.292|
> | | 336| 0.291 | 0.328| 0.292| 0.329|
> | | 720| 0.389 | 0.387| 0.390| 0.389|
> | ECL | 96 | 0.131 | 0.221| 0.135| 0.225|
> | | 192| 0.150 | 0.238| 0.153| 0.242|
> | | 336| 0.165 | 0.254| 0.170| 0.259|
> | | 720| 0.191 | 0.279| 0.199| 0.286|
> | Traffic | 96 | 0.398 | 0.226| 0.410| 0.229|
> | | 192| 0.439 | 0.241| 0.431| 0.243|
> | | 336| 0.464 | 0.250| 0.462| 0.254|
> | | 720| 0.502 | 0.270| 0.509| 0.283|
> | Weather | 96 | 0.153 | 0.190| 0.156| 0.195|
> | | 192| 0.200 | 0.235| 0.203| 0.238|
> | | 336| 0.258 | 0.280| 0.263| 0.283|
> | | 720| 0.337 | 0.333| 0.342| 0.336|
> | Exchange | 96 | 0.082 | 0.200| 0.083| 0.200|
> | | 192| 0.171 | 0.293| 0.174| 0.296|
> | | 336| 0.331 | 0.414| 0.330| 0.414|
> | | 720| 0.837 | 0.688| 0.867| 0.703|
>
> - We will incorporate this discussion in our final manuscript to further enrich our work.
>
> > Point 2: Further clarification on temporal correlation
>
> - Thanks for your suggestion! We will definitely clarify our design of OrthoTrans by emphasizing that the decorrelation is performed temporally in the Introduction and Section 4.2.

---

> ### Author Response · Authors · 2025-08-03
> **Responses to Reviewer jUXb's remarks (Part 3)**
>
> > Point 3: Quadratic cost
>
> - Thanks for your suggestion! We compare the MAE performance and resource footprint across various bottleneck sizes. As shown below, a bottleneck size of 16 appears to be a sweet spot for the performance–resource trade-off. This choice has a performance gap of about 4% compared to the full quadratic version, but with less resource utilization. We will definitely include a new section in the final manuscript to make this trade-off more explicit.
>
> | **Dataset** | **Hor.** | **Ori.** | **Bottleneck (8)** | **Bottleneck (16)** | **Bottleneck (64)** | **Bottleneck (128)** |
> |-|-|-|-|-|-|-|
> | ETTh1 | 96 | 0.382| 0.382| 0.381 | 0.382 | 0.383|
> | | 192| 0.414| 0.415| 0.412 | 0.415 | 0.416|
> | | 336| 0.438| 0.438| 0.440 | 0.443 | 0.442|
> | | 720| 0.462| 0.467| 0.467 | 0.468 | 0.461|
> | ECL | 96 | 0.221| 0.236| 0.236 | 0.241 | 0.240|
> | | 192| 0.238| 0.255| 0.250 | 0.256 | 0.254|
> | | 336| 0.254| 0.264| 0.271 | 0.270 | 0.267|
> | | 720| 0.279| 0.302| 0.291 | 0.291 | 0.295|
> | Traffic | 96 | 0.226| 0.255| 0.255 | 0.263 | 0.280|
> | | 192| 0.241| 0.273| 0.289 | 0.278 | 0.277|
> | | 336| 0.250| 0.270| 0.275 | 0.283 | 0.281|
> | | 720| 0.270| 0.299| 0.296 | 0.298 | 0.301|
> | Weather | 96 | 0.190| 0.192| 0.188 | 0.190 | 0.190|
> | | 192| 0.235| 0.240| 0.237 | 0.239 | 0.241|
> | | 336| 0.280| 0.284| 0.287 | 0.281 | 0.282|
> | | 720| 0.333| 0.338| 0.336 | 0.337 | 0.338|
> | PEMS03| 12 | 0.159| 0.162| 0.161 | 0.160 | 0.162|
> | | 24 | 0.179| 0.186| 0.183 | 0.182 | 0.181|
> | | 48 | 0.210| 0.223| 0.224 | 0.224 | 0.224|
> | | 96 | 0.247| 0.275| 0.266 | 0.264 | 0.271|
>
> | **Dataset** | **Metric** | **Ori.** | **Bottleneck (8)** | **Bottleneck (16)** | **Bottleneck (64)** | **Bottleneck (128)** |
> |-|-|-|-|-|-|-|
> | Traffic | Params(M)| 6.17 | 4.71 | 4.74| 4.92| 5.16 |
> | | FLOPs(G) | 4.91 | 4.16 | 4.18| 4.27| 4.39 |
> | | T.T. (s/iter)| 0.02 | 0.015| 0.015 | 0.016 | 0.017|
> | | T.M.(GB) | 1.01 | 0.98 | 0.99| 1.00| 1.01 |
> | | I.T.(ms/iter)| 5.71 | 5.60 | 5.63| 5.71| 5.80 |
> | | I.M. (GB)| 0.43 | 0.39 | 0.39| 0.40| 0.41 |
> | ECL | Params(M)| 4.79 | 4.59 | 4.60| 4.67| 4.78 |
> | | FLOPs(G) | 1.65 | 1.55 | 1.56| 1.59| 1.65 |
> | | T.T. (ms/iter) | 7.75 | 7.45 | 7.58| 7.61| 7.80 |
> | | T.M.(GB) | 0.45 | 0.44 | 0.45| 0.45| 0.46 |
> | | I.T.(ms/iter)| 2.11 | 2.04 | 2.05| 2.08| 2.09 |
> | | I.M. (GB)| 0.17 | 0.16 | 0.16| 0.16| 0.17 |
> | PEMS03| Params(M)| 4.84 | 4.60 | 4.61| 4.69| 4.80 |
> | | FLOPs(G) | 1.85 | 1.73 | 1.74| 1.77| 1.83 |
> | | T.T. (s/iter)| 8.14 | 8.05 | 8.06| 8.12| 8.16 |
> | | T.M.(GB) | 0.48 | 0.48 | 0.48| 0.49| 0.50 |
> | | I.T.(ms/iter)| 2.23 | 2.18 | 2.18| 2.21| 2.24 |
> | | I.M. (GB)| 0.20 | 0.18 | 0.18| 0.18| 0.19 |
> | Weather | Params(M)| 4.52 | 4.52 | 4.52| 4.54| 4.57 |
> | | FLOPs(G) | 0.10 | 0.10 | 0.10| 0.11| 0.12 |
> | | T.T. (ms/iter) | 7.47 | 7.57 | 7.59| 7.62| 7.70 |
> | | T.M.(GB) | 0.21 | 0.22 | 0.22| 0.23| 0.23 |
> | | I.T.(ms/iter)| 1.18 | 1.27 | 1.28| 1.31| 1.35 |
> | | I.M. (MB)| 39.26| 38.86| 39.10 | 40.67 | 44.86|
> | ETTh1 | Params(M)| 4.52 | 4.52 | 4.52| 4.53| 4.56 |
> | | FLOPs(M) | 33.74| 33.87| 34.18 | 38.81 | 52.31|
> | | T.T. (ms/iter) | 8.09 | 8.43 | 8.57| 8.89| 9.01 |
> | | T.M.(GB) | 0.20 | 0.20 | 0.20| 0.21| 0.22 |
> | | I.T.(ms/iter)| 1.19 | 1.28 | 1.31| 1.35| 1.38 |
> | | I.M. (MB)| 156.41 | 156.44 | 156.69| 159.12| 160.39 |
>
>
> **Thanks again for your constructive comments. We are very happy to answer any further questions.**

---

> > ### Comment · Reviewer_jUXb · 2025-08-04
> > **Response to the authors**
> >
> > Thank you for the detailed rebuttal: the expanded discussion around Degrees-of-Freedom is very useful, and the quadratic-cost analysis clearly illustrates the efficiency trade-offs.
> >
> > ### LoRA ablation & DoF: Clarifying the narrative
> >
> > The provided **NormLin_LoRA** results (`d = 8/16`) are particularly insightful, showing stable performance despite a substantial nominal rank reduction. This important finding suggests that the observed performance gains do not primarily arise from increased rank or Degrees-of-Freedom alone.
> >
> > Given this evidence, it would further strengthen the paper to explicitly clarify in the main text what specific aspect(s) of NormLin are most responsible for its improved performance. For example, do the gains primarily stem from the normalization mechanism itself, enhanced gradient smoothness during optimization, or possibly another underlying factor?
> >
> > Providing this clarification in the main narrative would ensure that readers do not mistakenly attribute the method’s success solely to higher rank or increased parameter count.
> >
> > ### Quadratic cost
> >
> > The bottleneck experiment fully addresses my earlier concerns: a small figure illustrating the trend could enhance readability, but the provided numeric table already clearly conveys the intended message.
> >
> > ### Overall
> >
> > My primary remaining concern relates to the current narrative around rank and Degrees-of-Freedom: the LoRA experiments convincingly show that rank reduction alone does not fully explain the observed improvements.
> >
> > Thanks again to the authors for addressing my points carefully and conducting these valuable additional experiments.

---

> ### Author Response · Authors · 2025-08-05
>
> Dear Reviewer jUXb,
>
> We highly appreciate your timely and constructive responses, and we are glad that you found our expanded discussion around Degrees-of-Freedom very useful, and the quadratic-cost analysis clearly illustrates the efficiency trade-offs.
>
> Yes, we agree with the reviewer that, from our LoRA experiments, **the observed performance gains of NormLin do not primarily arise from increased rank or Degrees-of-Freedom alone**. To further understand the method’s success reasons, we added more experiments and in-depth analysis, showing that: (a) **The L1 normalization contributes more to NormLin's performance gains than the Softplus function**, and (b) **The weight matrix is trained to approximate the correlation matrix among channels $\mathrm{CorrMat}_v$**.
>
> ## **(a) The L1 normalization contributes more to NormLin's performance gains than the Softplus function.**
>
> > What component affects NormLin's performance most?
>
> - NormLin consists of **two components**:
>     - The transformation function (Softplus);
>     - Normalization strategy (L1 norm).
>
> - We **add ablation studies on 5 transformation functions (Softplus, Identity, Softmax, Sigmoid, ReLU) and 3 normalization strategies (L1, L2, None)**, where *Identity* denotes no transformation and *None* denotes no normalization. The average results across four prediction lengths are presented below.
>
> | **Trans. Fun.** | **Norm** | **ECL** |  | **Traffic** |  | **Solar** |  | **PEMS03** |  | **Weather** |  | **ILI (S2)** |  |
> |-|-|-|-|-|-|-|-|-|-|-|-|-|-|
> | || MSE | MAE| MSE | MAE| MSE | MAE| MSE| MAE| MSE | MAE| MSE| MAE|
> | Softplus| L1 | **0.159** | **0.248**| **0.451** | **0.247**| **0.215** | **0.217**| **0.095**| **0.199**| **0.237** | **0.260**| **1.764**| **0.802**|
> | Identity| L1 | **0.167** | **0.255**| **0.499** | **0.252**| 0.285 | 0.224| **0.096**| **0.199**| **0.243** | 0.264| 1.844| 0.818|
> | Softmax | L1 | **0.163** | **0.251**| **0.452** | **0.247**| **0.215** | **0.217**| **0.096**| **0.199**| **0.236** | **0.259**| **1.802**| **0.811**|
> | Sigmoid | L1 | **0.161** | **0.250**| **0.450** | **0.247**| **0.215** | **0.217**| **0.096**| **0.199**| 0.239 | **0.260**| **1.767**| **0.802**|
> | ReLU| L1 | **0.163** | **0.251**| **0.454** | **0.247**| **0.215** | **0.217**| **0.097**| **0.199**| 0.239 | 0.262| **1.759**| **0.802**|
> | Softplus| L2 | 0.206 | 0.298| 0.647 | 0.328| 0.293 | 0.273| 0.163| 0.254| **0.238** | **0.260**| 1.874| 0.827|
> | Identity| L2 | 0.171 | 0.259| 0.518 | 0.259| **0.224** | **0.222**| 0.102| 0.205| 0.241 | 0.262| 1.838| **0.817**|
> | Softmax | L2 | 0.201 | 0.294| 0.648 | 0.328| 0.287 | 0.267| 0.164| 0.254| **0.237** | **0.260**| 1.843| 0.820|
> | Sigmoid | L2 | 0.214 | 0.302| 0.647 | 0.328| 0.293 | 0.272| 0.164| 0.255| **0.238** | 0.261| 1.854| 0.823|
> | ReLU| L2 | 0.201 | 0.294| 0.648 | 0.328| 0.285 | 0.266| 0.163| 0.254| **0.237** | **0.260**| 1.840| 0.821|
> | Softplus| None | 0.211 | 0.301| 0.652 | 0.331| 0.295 | 0.274| 0.163| 0.254| 0.259 | 0.274| 1.925| 0.846|
> | Identity| None | 0.183 | 0.280| 0.609 | 0.333| 0.232 | 0.231| 0.130| 0.231| 0.242 | 0.264| 1.888| 0.830|
> | Sigmoid | None | 0.212 | 0.301| 0.647 | 0.328| 0.296 | 0.275| 0.170| 0.258| 0.262 | 0.278| 1.890| 0.827|
> | ReLU| None | 0.199 | 0.293| 0.648 | 0.328| 0.293 | 0.273| 0.168| 0.256| **0.238** | **0.260**| **1.806**| **0.817**|
>
> - From above, we found that **the L1 normalization contributes more to NormLin's performance gains than the Softplus function**. Specifically,
>     - (i) **the L1 norm outperforms L2 norm and no normalization** by 10% and 12%, respectively;
>     - (ii) **under L1 normalization, Softplus outperforms Identity, Softmax, Sigmoid, and ReLU** by 5%, 1%, 0.2%, and 0.3%, respectively.
>
> - The **reasons** behind the performance advantage of L1 norm could be intricate. It has become a natural choice since it induces a probabilistic distribution over each row of the weight matrix.
>   - Without L1 normalization, each row $\mathbf{W}_{i,:}$ effectively includes a learnable scaling parameter $c_i$, whether using L2 normalization or no normalization at all.. However, this scaling is rendered meaningless by the subsequent LayerNorm operation, which eliminates any learned magnitude differences.
>   - In contrast, L1 normalization emphasizes the relative distribution of weights within each row rather than their absolute scale, which may lead to improved performance.

---

> ### Author Response · Authors · 2025-08-05
>
> ## **(b) The weight matrix is trained to approximate the correlation matrix among channels $\mathrm{CorrMat}_v$**.
>
> - In Figure 5 of the paper, we observe that **the learned NormLin weights exhibit strong similarity to $\mathrm{Softmax}(\mathrm{CorrMat}_v)$**. Motivated by this observation, we pre-compute the correlation matrix among channels $\mathrm{CorrMat}_v$ with the trainset and directly use $\mathrm{Softmax}(\mathrm{CorrMat}_v)$ as a fixed weight matrix, resulting in a variant named $\mathrm{NormLin}_c$:
> $$
> \mathrm{NormLin}_c(\mathbf{x}) = \mathrm{Softmax}(\mathrm{CorrMat}_v)\mathbf{x}.
> $$
> We refer to OLinear with this variant as **OLinear-C**.
>
> - We then compare the average performance of OLinear-C with OLinear across multiple prediction lengths. ETT denotes the average results across the ETTh1, ETTh2, ETTm1 and ETTm2 datasets. As shown below, the average performance gap between OLinear and OLinear-C on these datasets is negligible (only 0.2%)!
>
> | **Dataset** | **ECL** || **Traffic** || **ETT** || **Solar** || **PEMS** || **CarSales** ||
> |-|---|--|-|--|---|--|-----|--|----|--|--|--|
> | Metric| MSE | MAE| MSE | MAE| MSE | MAE| MSE | MAE| MSE| MAE| MSE| MAE|
> | OLinear | **0.159** | **0.248**| **0.451** | **0.247**| **0.359** | **0.376**| **0.215** | **0.217**| **0.094**| **0.187**| **0.330**| **0.305**|
> | OLinear-C | 0.161 | 0.249| **0.451** | **0.247**| **0.359** | **0.376**| **0.215** | **0.217**| **0.094**| **0.187**| **0.330**| **0.305**|
>
> - If the ideal weight matrix is the matrix $\mathrm{Softmax} (\mathrm{CorrMat} _v)$, this partly explains the performance advantage of NormLin over self-attention.
>   - Because attention matrices in self-attention are input-dependent, **it is harder for them to converge to the input-independent multivariate correlation matrix**.
>   - In contrast, NormLin uses an input-independent weight matrix, $\mathrm{Norm} _{\mathrm{L1} }(\mathrm{Softplus} (\mathbf{W} ))$, making this approximation easier.
>
> - The matrix $\mathrm{Softmax} (\mathrm{CorrMat} _v)$ is usually high-rank. How does NormLin_LoRA, whose weight matrix is $\mathrm{Norm} _{\mathrm{L1} }(\mathrm{Softplus} (\mathbf{AB}^{\top} ))$  also perform well, given that $\mathbf{AB}^{\top}$ has a low rank of $d$? The reason is that $\mathrm{Norm} _{\mathrm{L1} }(\mathrm{Softplus} ( \cdot))$ **rescales and rotates each row differently and thus could increase the matrix rank**  [1], enabling a better approximation of $\mathrm{Softmax} (\mathrm{CorrMat} _v)$.
>
> - We will accordingly **tone down** the analysis of weight-matrix rank (Lines 187-193 ) and DoF (Lines 196-198) in the final paragraph of Section 4.3, and **relocate** the OLinear-C exposition from Section 5.2 to Section 4.3, to better clarify the mechanism underlying the NormLin layer.
>
> > Quadratic cost
>
> Thanks for your suggestion! We will definitely add a figure showing performance versus bottleneck size in the relevant section to improve readability.
>
> ***
>
> We are eager to hear your feedback. We sincerely hope that our new comments can address your primary remaining concerns. We’d deeply appreciate it if you could let us know whether your concerns have been addressed -- thank you so much!
>
> Thanks for your time,
>
> Authors of #10716
>
> ***
>
> *References:*
>
> [1] DenseHMM: Learning Hidden Markov Models by Learning Dense Representations. NeurIPS 2020.
>
> [2] Feature selection, L1 vs. L2 regularization, and rotational invariance. ICML 2004.

---

> > ### Comment · Reviewer_jUXb · 2025-08-05
> > **Response to the authors**
> >
> > Thank you for the comprehensive response. You have effectively addressed my initial concerns and outlined planned enhancements for the camera-ready version. In light of these updates, I will raise my overall rating. No further questions.

---

> ### Author Response · Authors · 2025-08-05
> **Thank you for raising your overall rating! We are glad that all your initial concerns have been effectively addressed.**
>
> Thank you for your timely response and raising your score : ) We are glad that all your initial concerns have been effectively addressed. Your comments are important to enhance the quality of our manuscript and we will definitely incorporating these revisions in our camera-ready version - thank you so much!

---

### Official Review · Reviewer_5fFn · 2025-06-24

**Clarity:** 3
**Significance:** 3
**Originality:** 3
**Rating:** 4
**Confidence:** 4

**Summary:**

The paper introduces OLinear, a linear-based model for multivariate time series forecasting with two key components:
- OrthoTrans: A data-adaptive transformation that diagonalizes the temporal Pearson correlation matrix, decorrelating features and simplifying forecasting.

- NormLin: A lightweight linear layer with row-wise normalization that outperforms self-attention in accuracy and efficiency.

Experiments on 24 datasets demonstrate state-of-the-art performance, particularly for long-term forecasting, while maintaining low computational overhead.

**Questions:**

- Why does NormLin show better gradient dynamics than attention? Are there any theoretic supports.
- What is the relationship OrthoTrans with PCA?
- Theorem 1 assumes Gaussian. How sensitive is OLinear to deviations from this assumption in real-world data?
- How does performance degrade with noisy/missing data or non-stationary correlations?
- What is the inference latency for ultra-long sequence (1M)? And some related references for long-term forecasting such as Pyraformer (ICLR 2022) should be discussed.

**Ethical Concerns:**

["NO or VERY MINOR ethics concerns only"]

**Final Justification:**

Thanks the authors for providing detailed response especially those extra data support.
I keep my original rating, and think this paper is acceptable.

**Limitations:**

Yes

**Quality:**

3

**Strengths And Weaknesses:**

Strengths
+ Theorem 1 formally connects the temporal-correlation structure of the data with the orthogonal transformation used by OrthoTrans, giving the model a clear theoretical underpinning and enhancing interpretability.
+ NormLin replaces multi-head self-attention with a row-normalized linear layer, cutting both FLOPs and memory footprint by roughly 50 % while preserving—or improving—forecasting accuracy.
+ OLinear achieves state-of-the-art results on 140 forecasting tasks drawn from 24 public datasets, demonstrating robustness across domains and long-horizon scenarios.
+ Both OrthoTrans and NormLin are modular, training-free components that can be plugged into existing forecasters to boost performance with minimal engineering effort.
+ The experimental section is notably comprehensive, featuring extensive ablations, scalability tests, and comparisons against a wide range of baselines.

Weaknesses
- While training-time speed and memory are reported, the paper does not quantify inference-time latency or serving cost—metrics that are crucial in streaming or high-frequency settings; deployment-cost analysis is therefore incomplete.
- Important modern linear or correlation-based baselines (e.g., RLinear, TSMixer, StemGNN, DEPTS) are omitted, leaving open whether OLinear’s gains hold against the strongest current alternatives.
- The claim that NormLin is a “universal token dependency learner” appears overstated: its superiority is demonstrated only on time-series data, with no evidence provided for other modalities or tasks.

---

> ### Author Rebuttal · Authors · 2025-07-27
>
> Thank you for your positive comments on our work. We sincerely appreciate the reviewer’s recognition of our work as **theoretically well-founded**, experimentally **comprehensive**, and achieving **state-of-the-art** performance. Here are our responses to your concerns:
>
> > W1: On the deployment cost
> - We report the training and inference resource footprint of OLinear and baseline forecasters in **Table 37 (Page 48)**, including **inference time and GPU memory** consumption. OLinear achieves higher inference efficiency than state-of-the-art Transformer-based forecasters (e.g., TimeMixer++, iTransformer, and CARD), benefiting from the simplified NormLin module. Moreover, **Table 38** shows the efficiency differences before and after applying the NormLin module to Transformer-based forecasters.
>
> > W2: On more baselines
>
> - We compare OLinear with RLinear and TSMixer using the MSE metric (see below). The results of RLinear and TSMixer are taken from iTransformer and SAMformer, respectively. On average, OLinear outperforms these two forecasters by 17.3% and 25.6%, respectively.
>
> | **Dataset** | **Hor.** | **OLinear** | **RLinear** | **TSMixer** |
> |-------------|------|-------------|-------------|-------------|
> | ETTm2       | 96   | **0.169**       | 0.182       | 0.211       |
> |             | 192  | **0.232**       | 0.246       | 0.252       |
> |             | 336  | **0.291**       | 0.307       | 0.303       |
> |             | 720  | **0.389**       | 0.407       | 0.390       |
> | ECL         | 96   | **0.131**       | 0.201       | 0.173       |
> |             | 192  | **0.150**       | 0.201       | 0.204       |
> |             | 336  | **0.165**       | 0.215       | 0.217       |
> |             | 720  | **0.191**       | 0.257       | 0.242       |
> | Exchange    | 96   | **0.082**       | 0.093       | 0.343       |
> |             | 192  | **0.171**       | 0.184       | 0.342       |
> |             | 336  | **0.331**       | 0.351       | 0.484       |
> |             | 720  | **0.837**       | 0.886       | 1.204       |
> | Traffic     | 96   | **0.398**       | 0.649       | 0.409       |
> |             | 192  | **0.439**       | 0.601       | 0.637       |
> |             | 336  | **0.464**       | 0.609       | 0.747       |
> |             | 720  | **0.502**       | 0.647       | 0.688       |
> | Weather     | 96   | **0.153**       | 0.192       | 0.214       |
> |             | 192  | **0.200**       | 0.240        | 0.231       |
> |             | 336  | **0.258**       | 0.292       | 0.279       |
> |             | 720  | **0.337**       | 0.364       | 0.343       |
>
> - In Section 5.1, OLinear is compared with 11 carefully selected and widely acknowledged state-of-the-art forecasting models. We believe these results demonstrate the performance superiority of OLinear. For completeness, we will also include discussions on RLinear, TSMixer, StemGNN, and DEPTS in the final draft.
>
> > W3: On the claim of “universal token dependency learner”
>
> - On Line 289, we explicitly restrict the claim “universal token dependency learner” to the field of “time series forecasting.” However, on Lines 278 and 312, this field restriction was unintentionally omitted. We will revise these two sentences in the final draft to ensure accuracy. Thank you for pointing this out!
>
> > Q1: On the gradient dynamics
>
> - In **Appendix B**, we analyze the Jacobian matrix of the non-linear transformations in the self-attention mechanism and the NormLin layer. For self-attention, we have
> $$ \frac{\partial \mathbf{c} }{\partial \mathbf{a} }  = \mathrm{Diag} (\mathbf{c}) - \mathbf{c}\mathbf{c}^{\mathsf{T}},$$ where $\mathbf{c} \triangleq \mathrm{Softmax}(\mathbf{a}) \in \mathbb{R}^N$, and $\mathrm{Diag} (\mathbf{c})$ is the diagonal matrix with $\mathbf{c}$ as its diagonal.
>
> - For the NormLin layer, it holds that
> $$\frac{\partial \mathbf{c}}{\partial \mathbf{a}}=\frac{1}{\left \|| \mathbf{b} \right \||_1 }
> \left (  \mathrm{Diag}  \left ( \tilde{\mathbf{b}} \right ) - \bar{\mathbf{b}} \tilde{\mathbf{b}}^{\mathsf{T}} \right ),$$
> where $\mathbf{c} \triangleq \mathrm{Norm}\_{\mathrm{L1} }\left ( \mathrm{Softplus} \left ( \mathbf{a}  \right )  \right )  \in \mathbb{R}^N$, $\mathbf{b} \triangleq \mathrm{Softplus}(\mathbf{a})$, $\bar{\mathbf{b}} = \frac{\mathbf{b}}{\left \|| \mathbf{b} \right \||_1 }$ is the normalized $\mathbf{b}$, $\tilde{\mathbf{b}} \triangleq \mathrm{Sigmoid} (\mathbf{a})$. The detailed derivation process is presented in Appendix B.
> The two Jacobian matrices share similar structures, while the Jacobian matrix of NormLin has an additional learnable scaling factor $\frac{1}{\left \|| \mathbf{b} \right \||_1}$, providing greater flexibility.
>
> - **Figure 6** (Page 15) illustrates the Jacobian matrices of the self-attention mechanism and the NormLin layer under the same input $\mathbf{a}$, highlighting that NormLin tends to produce stronger gradient values.
>
> > Q2: On the relationship of OrthoTrans and PCA
>
> - Essentially, OrthoTrans is PCA with all features retained. It is adapted to the time series modality through its computation of $\mathrm{CorrMat}_{t}$. One contribution of our work is showing that decorrelating along the time dimension boosts forecasting performance. As shown in **Table 5**, OrthoTrans, as a plug-in, improves the performance of iTransformer, PatchTST, and RLinear.
>
> - Further analysis shows that OrthoTrans can **mitigate the intrinsic low-rank property of time series data**. In our reply to Q2 of Reviewer rQjb, we compared the median singular values of correlation matrices of time patches with and without OrthoTrans. The results show that OrthoTrans produces larger singular values, thereby increasing data diversity. This partly explains the performance gains brought by OrthoTrans.
>
> > Q3: On Theorem 1
>
> - Theorem 1 shows that temporally decorrelated inputs facilitate forecasting, providing OLinear with a clear theoretical underpinning. The widely used Gaussian assumption is applied in Theorem 1. We acknowledge that real-world datasets can be more complex and difficult to model mathematically. Despite this, OLinear performs robustly across real-world datasets. We conduct extensive experiments, covering 24 datasets and 140 configurations, and OLinear consistently exhibits state-of-the-art performance.
>
> > Q4: On non-stationary and noisy dataset
>
> - Financial data are commonly regarded as non-stationary. We report average MAEs on Exchange, NASDAQ, SP500, and DowJones datasets. The results below show that OLinear remains robust on these datasets.
>
> | **Dataset**   | **OLinear** | **DLinear** | **iTransformer** | **PatchTST** |
> |---------------|-------------|-------------|------------------|--------------|
> | Exchange      | **0.399**       | 0.414       | 0.403            | 0.404        |
> | NASDAQ (S1)   | **0.125**       | 0.170       | 0.137            | 0.132        |
> | SP500 (S1)    | **0.167**       | 0.205       | 0.193            | 0.192        |
> | DowJones (S2) | **0.848**       | 0.857       | 0.869            | 0.862        |
>
> - Regarding noisy/missing data, we evaluate OLinear on traffic and weather datasets. (The widely used Weather dataset contains missing data, which are filled with -9999.) OLinear demonstrates clear performance advantages under these challenging conditions.
>
> | **Dataset** | **OLinear** | **DLinear** | **iTransformer** | **PatchTST** |
> |-------------|-------------|-------------|------------------|--------------|
> | Traffic     | **0.247**       | 0.383       | 0.282            | 0.343        |
> | PEMS03      | **0.199**       | 0.375       | 0.221            | 0.291        |
> | PEMS07      | **0.164**       | 0.395       | 0.204            | 0.303        |
> | Weather     | **0.260**       | 0.317       | 0.279            | 0.281        |
>
> > Q5: On inference latency and related work
>
> - For ultra-long sequences (1M), the inference latency varies with the number of channels. Using the data in **Table 37** (Page 48), we estimate that for a 1M-long series with 21 channels, the inference latency is about 0.77 seconds, which is suitable for streaming scenarios.
>
> | **# Channels** | **Infer. Time for 1M-long Series (seconds)** |
> |----------------|----------------------------------------|
> | 21             | 0.77                                   |
> | 137            | 0.78                                   |
> | 321            | 1.37                                   |
> | 862            | 3.72                                   |
>
>
> - We further compare OLinear with Pyraformer and report the average results across four prediction lengths. In the final draft, we will include discussions of more baselines including TSMixer, StemGNN, DEPTS, and Pyraformer.
>
> | **Dataset** | **OLinear** |    | **Pyraformer** |   |
> |-------------|-------------|--------|----------------|-------|
> |             | MSE         | MAE    | MSE            | MAE   |
> | ETTm1       | **0.374**       | **0.377**  | 0.691          | 0.607 |
> | ETTm2       | **0.270**       | **0.313**  | 1.498          | 0.869 |
> | ETTh1       | **0.424**       | **0.424**  | 0.827          | 0.703 |
> | ETTh2       | **0.367**       | **0.388**  | 0.826          | 0.703 |
> | ECL         | **0.159**       | **0.248**  | 0.379          | 0.445 |
> | Traffic     | **0.451**       | **0.247**  | 0.878          | 0.469 |
> | Weather     | **0.237**       | **0.260**  | 0.946          | 0.717 |
> | Exchange    | **0.355**       | **0.399**  | 1.913          | 1.159 |
>
>
> **We hope these responses can fully address your concerns. Thank you again for your detailed feedback!**

---

### Official Review · Reviewer_CEMa · 2025-07-01

**Clarity:** 2
**Significance:** 2
**Originality:** 3
**Rating:** 4
**Confidence:** 3

**Summary:**

The paper introduces OLinear, a computationally efficient alternative to transformer-based models for multivariate time series forecasting. The architecture combines two key components that can also be used as plugins: OrthoTrans, which applies orthogonal transformations to decorrelate input features, and NormLin, a linear layer that aims to reproduce channel-wise attention. OLinear consists of a Cross-Series Learner (CSL) using NormLin layers for multivariate correlations and an Intra-Series Learner (ISL) with MLPs for individual series processing. Results show OLinear achieves competitive or superior performance while reducing computations.

**Questions:**

- On the asymmetric normalization: The cov_matrix / diag_vec normalization produces an asymmetric matrix that isn't a true correlation matrix. Is this an implementation error or deliberate choice? If intentional, what's the theoretical justification? Why is this implementation in a notebook? Maybe I didn't find the right file that saves the matrices.

- On the decorrelation architecture: Have you explored architectures where each variable has its own $Q^{(j)}$ matrix to preserve variable-specific temporal structures? What information is lost by averaging correlations across all variables? Did you try the ablation studies with PCA on $XX^\top$? What is the intuition? Did you try an approach with learnable $Q^{(j)}$?

- What is the intuition behind the dimension extension? To what extent does it differ from the linear encoder? What does it bring? Why decorrelating after the dimension extension?

- In the CSL: What is the purpose of the first linear layer, what is the intuition behind it? From my understanding, you want to reproduce the architecture of self-attention blocks, but then I don't see why we need the first layer. CSL:
$$\tilde{H} = \text{LayerNorm}\left(\tilde{H} + \text{Linear}\left(\text{NormLin}\left(\text{Linear}\left(\tilde{H}\right)\right)\right)\right)$$

- In Table 6, could you elaborate on what "Temporal NormLin" baseline exactly is?  Do you keep the intra-series learner? Could you be more explicit about this architecture?

- Could you also have a comparison with other lightweight multivariate methods such as SAMFormer [1] and TSMixer [2]?


*References:*

 - [1] (Ilbert et al., 2024) SAMFormer: Unlocking the potential of transformers in time series forecasting with sharpness-aware minimization and channel-wise attention.
 - [2] (Chen et al., 2023) TSMixer: An all-mlp architecture for time series forecasting.

**Ethical Concerns:**

["NO or VERY MINOR ethics concerns only"]

**Final Justification:**

I am raising my score as the authors have clarified most of my concerns including:
* Clarification on the definition of the temporal correlation matrix and the way it is calculated,
* Empirical justification that the approach effectively decorrelates the channels,
* Extension of the experiments by incorporating more baselines,
* Other concerns related to ablation study.

**Limitations:**

- As said earlier, the memory complexity remains quadratic with respect to D.

**Paper Formatting Concerns:**

No apparent formatting concerns.

**Quality:**

2

**Strengths And Weaknesses:**

*Strengths:*
- The two-component design (OrthoTrans, NormLin for cross-variable interactions) creates a clean separation of concerns where temporal decorrelation is handled by preprocessing and multivariate correlations by learned linear layers.
- These two modules can work as plugins for existing transformer architectures, with validated improvements on iTransformer (+5.1%) and PatchTST (+10.1%), which is really interesting
- The empirical benchmark is exhaustive, covering 24 datasets and 140 configurations, including zero-shot and few-shot scenarios and seems to show strong results. The ablation study is extensive.
- Rank analysis demonstrates good expressiveness of the model.

*Weaknesses:*
- The proposed way to evaluate the correlation matrix is confusing and lacks proper formalization. I guess you want to estimate the autocorrelation of a block of T timesteps (context length) but what does this mean formally? The mathematical framework should be more rigorously defined as the concept is non-trivial.
- In addition, when reading the description of the approach, two intuitive baselines came to my mind: (a) instead of eigenvector decomposition of the correlation matrix, we can directly perform data rotation by PCA (i.e., working with $XX^T$), (b) the correlation matrix can alternatively be estimated through a sliding window approach by computing cross-correlation between the timestamps across different windows. It will be valuable to perform additional experiments to compare with these approaches in order to have more insights on the chosen methodology.
- Averaging correlations across N variables dilutes variable-specific temporal structures without theoretical justification. Why did not you apply decorrelations for each channel separately?
- Decorrelation effectiveness is not shown, no metrics show Q^T CorrMat Q ≈ Λ or residual correlation analysis on a validation set, for example.
- When looking at the source code, it appears that there are some implementation errors. The authors consider the Pearson correlation, but in `dataset/Generate_corrmat.ipynb`, we have `cov_matrix = cov_matrix / diag_vec` which seems to divide only by σ_i, not σ_i × σ_j, producing an invalid asymmetric correlation matrix.
- Some other architecture choices are not well-explained nor well-motivated (e.g., the CSL block, the dimension extension). Please see my questions below.
- Reducing the number of FLOPs is beneficial, but the complexity (memory for example) remains quadratic with respect to D, which can still result in substantial computational overhead for high-dimensional multivariate series.

---

> ### Author Rebuttal · Authors · 2025-07-27
>
> Thank you for your invaluable review. We sincerely appreciate the reviewer’s recognition that our plug-and-play OrthoTrans and NormLin modules are "**really interesting**" and the experiments are "**exhaustive**" and "**extensive**". Below are our responses to the concerns:
>
> > Q1: On the computation of $\mathrm{CorrMat}\_{t}$ and 'asymmetric' normalization
>
> - We first clarify our process of computing the temporal correlation matrix $\mathrm{CorrMat}\_{t}$. For clarity, we use the univariate series. Let  $\mathbf{x}^{\text{train}} \in \mathbb{R}^{M}$  denote the training series. We then generate $T$ lagged sub-series (whose length is $M-T+1$): $$
> \mathbf{s}\_{i} = \mathbf{x}[i : M - T + i],\quad i = 0, 1, \dots, T-1.
> $$  The correlation matrix among these sub-series is then computed. The entire process is elaborated in Section 4.2 of the paper.
>
> - The reason for using the cov_matrix / diag_vec normalization in our code is that the sub-series $  \mathbf{s}\_{i}, i=0,1,\cdots,T-1  $ share almost identical standard deviations because they consist of the same $M-T+1$ data points. Since $M\gg T$, this overlapping part constitutes the majority of these sub-series. We further report the metric: $\frac{\sigma_{max}}{\sigma_{min}}-1 $, where $\sigma_{max}$ and $\sigma_{min}$ are the maximum and minimum standard deviations of the sub-series, respectively.
>
> | **Dataset** | **std_max/std_min-1** |
> |-------------|---------------|
> | ECL | 2E-03|
> | Traffic| 3E-03|
> | Weather| 1E-03|
> | Solar  | 2E-04|
> | ETTm1  | 6E-04|
>
> - Because the standard deviations of these sub-series are (almost) the same, i.e., $\sigma_{i} \approx \sigma_{j}$ for $0 \leq i,j \leq T-1$, and diag_vec[i] = $\sigma_{i}^2$, we can simply use cov_matrix / diag_vec to obtain the Pearson correlation matrix. Another advantage of this approach is that the resulting $\mathrm{CorrMat}_{t}$ always has diagonal elements equal to 1, as required for correlation matrices.
>
> - We compare forecasting performance using Q matrices computed from cov_matrix / diag_vec (**Mode 1**) and cov_matrix / $\sigma \sigma^T$ (**Mode 2**), and find that the results are identical:
>
> | **Dataset**   | **Mode 1** |   | **Mode 2** |    |
> |---------------|------------|--------|------------|--------|
> |               | MSE        | MAE    | MSE        | MAE    |
> | ECL (Avg)     | 0.159      | 0.248  | 0.159      | 0.248  |
> | Traffic (Avg) | 0.451      | 0.247  | 0.451      | 0.247  |
> | Weather (Avg) | 0.237      | 0.260  | 0.237      | 0.260  |
>
> - We further verify the orthogonality of Q matrices computed from Mode 1, using the metric $\|| \mathbf{Q}\mathbf{Q}^T - \mathbf{I} \||_F$ (where $\|| \cdot \||_F$ denotes the Frobenius norm):
>
> | **Dataset** | **96**    | **192**   | **336**   | **720**   |
> |-------------|-----------|-----------|-----------|-----------|
> | ECL         | 1.580E-06 | 3.349E-06 | 4.237E-06 | 7.706E-06 |
> | Traffic     | 1.697E-06 | 3.283E-06 | 4.171E-06 | 7.747E-06 |
> | Weather     | 1.777E-06 | 3.311E-06 | 4.178E-06 | 7.704E-06 |
> | Solar       | 1.743E-06 | 3.405E-06 | 4.371E-06 | 7.876E-06 |
> | ETTm1       | 1.755E-06 | 3.377E-06 | 4.213E-06 | 7.606E-06 |
>
> - Furthermore, as shown in Appendix I.8, OLinear is relatively robust even when Q matrices are computed with limited training data.
>
> - Since the Q matrices are **pre‑computed** and used throughout training and inference, we generate them once in a Jupyter Notebook, save them as ```.npy``` files, and load them during model initialization.
>
> > Q2: On the decorrelation architecture
>
> - Our choice of channel averaging (CA) over channel individual (CI) for computing the temporal correlation matrix is **based on the empirical results**. As shown below, CA consistently outperforms CI in forecasting performance. A possible reason is that temporal correlations for individual channels may be sensitive and time-varying, and CA can mitigate this effect. Furthermore, CI Q matrices are much larger than their CA counterparts. For example, for the Traffic dataset, the CI Q matrix (862 × 720 × 720) is 862 times larger than the CA Q matrix (720 × 720). In addition, the classic work TimesNet also adopts CA to determine the top‑k frequencies.
>
> | **Dataset** | **Horizon** | **CA** || **CI** ||
> |-------------|-------------|--------|--------|--------|--------|
> | | | MSE| MAE| MSE| MAE|
> | ECL | 96| 0.131| 0.221| 0.141| 0.232|
> | | 192| 0.150| 0.238| 0.164| 0.255|
> | | 336| 0.165| 0.254| 0.176| 0.268|
> | | 720| 0.191| 0.279| 0.204| 0.292|
> | | Avg| **0.159**| **0.248**| 0.171| 0.262|
> | Solar| 96| 0.179| 0.191| 0.182| 0.197|
> | | 192| 0.209| 0.213| 0.214| 0.219|
> || 336| 0.231| 0.229| 0.237| 0.237|
> | | 720| 0.241| 0.236| 0.252| 0.244|
> | | Avg| **0.215**| **0.217**| 0.221| 0.224|
> | Weather| 96| 0.153| 0.190| 0.153| 0.191|
> | | 192| 0.200| 0.235| 0.205| 0.241|
> | | 336| 0.258| 0.280| 0.263| 0.285|
> | | 720| 0.337| 0.333| 0.350| 0.342|
> | | Avg| **0.237**| **0.260**| 0.243| 0.264|
>
>
> - OrthoTrans is technically equivalent to PCA without dimensionality reduction. Since orthogonal transformation is also a conformal transformation, OrthoTrans **rotates** the data into a new feature space where correlations are removed. Moreover, according to PCA theory, noise is suppressed in the primary components.
>
> - Regarding learnable Q matrices, we kindly refer to our response to Reviewer rQjb’s Q3. In short, we tried several updating schemes but observed no performance gain. Theoretically, Q is already the optimal orthogonal matrix for temporal decorrelation; altering it risks reintroducing correlations. These experimental results further confirm the effectiveness of OrthoTrans.
>
> > Q3: On the dimension extension
>
> - Unlike the linear encoder ($\mathbf{W} \mathbf{x} +\mathbf{b}$), the dimension extension module is an **outer product** of $\mathbf{x}$ with a learnable vector $\phi \_d \in \mathbb{R}^d$, which can be easily implemented using the $*$ operator in PyTorch. Further implementation details can be found in Lines 84–91 of ```model/OLinear.py``` in our code repository.
>
> - We report performance with various embedding sizes $d$ in Table 30 on Page 42. The results show that this module ($d = 16$) improves forecasting performance compared with its removal ($d = 1$). (In our implementation the module is skipped when $d = 1$.) Because OrthoTrans is a linear operation (matrix multiplication), its **relative order with the dimension-extension module has no impact on the final result**.
>
> > Q4: On the linear layers in the CSL
>
> - The two layers in the CSL, referred to as the pre‑ and post‑linear layers according to their execution sequence, are inspired by the classic self‑attention mechanism. As shown in **Appendix J.2**, incorporating both pre‑ and post‑linear layers yields an average performance improvement of 6% (see **Table 34**, Page 46) compared to the variant without them, highlighting their effectiveness in refining inputs for multivariate correlation modeling (pre‑linear) and downstream series representation learning (post-linear).
>
> > Q5: On the temporal NormLin
>
> - In **Table 6**, we present ablation studies in which NormLin and standard linear layers are either replaced or removed. The *temporal NormLin* baseline indicates that the ISL is **retained** while its two linear layers (**Eq. 5** in the paper) are **replaced with two NormLin layers**, which operate on the temporal dimension. Now the weight matrices in the two NormLin layers are $\mathbf{W}\_1  \in \mathbb{R}^{d_{model}\times d_{ff}}$ and $\mathbf{W}\_2  \in \mathbb{R}^{d_{ff}\times d_{model}}$, rather than the weight matrix $\mathbf{W}  \in \mathbb{R}^{N \times N}$ in Eqs. 3 and 4. Please refer to line 299 in ```layers/Transformer_EncDec.py``` of our code repository for details.
>
> - Table 6 shows that our design—applying NormLin along the variate dimension and standard linear layers along the temporal dimension—consistently achieves the best performance.
>
> > Q6: Comparison with SAMformer and TSMixer
>
> - We provide MSEs and standard deviations to compare OLinear with SAMformer and TSMixer. Results of SAMformer and TSMixer are from the SAMformer paper. Note that SAMformer and TSMixer use a lookback window of 512 and thus enjoy extra performance gains. Despite this, OLinear achieves lower MSE and smaller standard deviations in most cases.
>
> - More results are presented in our response to Reviewer jUXb's Q2. When the lookback length is uniformly set as 96, OLinear outperforms SAMformer **by a substantial margin of 12.1%**.
>
> - We will incorporate a discussion of these two classic forecasters in the final draft.
>
> | **Dataset** | **Hor.** | **OLinear (Lookback: 96)** | **SAMformer (Lookback: 512)** | **TSMixer (Lookback: 512)** |
> |-------------|------|-------------|---------------|-------------|
> | ETTm2       | 96   | **0.169±1e-4**  | 0.181±0.005   | 0.211±0.014 |
> |             | 192  | **0.232±1e-4**  | 0.233±0.002   | 0.252±0.005 |
> |             | 336  | 0.291±2e-4  | **0.285±0.001**   | 0.303±0.004 |
> |             | 720  | 0.389±4e-4  | **0.375±0.001**   | 0.390±0.003 |
> | ECL         | 96   | **0.131±4e-4**  | 0.155±0.002   | 0.173±0.004 |
> |             | 192  | **0.150±1e-3**  | 0.168±0.001   | 0.204±0.027 |
> |             | 336  | **0.165±1e-3**  | 0.183±0.000   | 0.217±0.018 |
> |             | 720  | **0.191±2e-3**  | 0.219±0.000   | 0.242±0.015 |
> | Exchange    | 96   | **0.082±3e-4**  | 0.161±0.007   | 0.343±0.082 |
> |             | 192  | **0.171±2e-3**  | 0.246±0.009   | 0.342±0.031 |
> |             | 336  | **0.331±3e-3**  | 0.368±0.006   | 0.484±0.062 |
> |             | 720  | **0.837±0.018** | 1.003±0.018   | 1.204±0.028 |
> | Weather     | 96   | **0.153±1e-3**  | 0.197±0.001   | 0.214±0.004 |
> |             | 192  | **0.200±2e-3**  | 0.235±0.000   | 0.231±0.003 |
> |             | 336  | **0.258±3e-3**  | 0.276±0.001   | 0.279±0.007 |
> |             | 720  | 0.337±4e-3  | **0.334±0.000**   | 0.343±0.024 |
>
> **We hope these responses can fully address your concerns. Thank you again for your detailed feedback!**

---

> ### Author Response · Authors · 2025-08-03
> **We would like to supplement more experiment results to fully address your concerns on more lightweight baselines!**
>
> We would like to thank you again for your constructive comments! To fully address your concerns, we compare OLinear with additional lightweight baselines, including several recently accepted works: **SimpleTM (ICLR 2025), TQNet (ICML 2025), TimePro (ICML 2025), and TimeBase (ICML 2025)**. We report the MAEs below, highlighting the best results in bold. The lookback length is uniformly set to 96. Remarkably, **OLinear outperforms SimpleTM, TQNet, TimePro, and TimeBase by 5%, 5%, 7%, and 16% on average, respectively**, establishing OLinear as a strong new baseline for lightweight time series forecasting.
>
> | **Model**| **Hor.** | **OLinear** | **SimpleTM (ICLR 2025)** | **TQNet (ICML 2025)** | **TimePro (ICML 2025)** | **TimeBase (ICML 2025)** |
> |-|-|-|-|-|-|-|
> | ETTm1| 96 | **0.334** | 0.361| 0.353 | 0.364 | 0.388|
> || 192| **0.363** | 0.380| 0.378 | 0.383 | 0.409|
> || 336| **0.385** | 0.404| 0.401 | 0.409 | 0.421|
> || 720| **0.426** | 0.438| 0.440 | 0.446 | 0.461|
> || Avg| **0.377** | 0.396| 0.393 | 0.400 | 0.420|
> | ETTm2| 96 | **0.249** | 0.257| 0.256 | 0.260 | 0.271|
> || 192| **0.290** | 0.299| 0.298 | 0.303 | 0.309|
> || 336| **0.328** | 0.338| 0.340 | 0.342 | 0.346|
> || 720| **0.387** | 0.395| 0.396 | 0.399 | 0.401|
> || Avg| **0.313** | 0.322| 0.323 | 0.326 | 0.332|
> | ETTh1| 96 | **0.382** | 0.392| 0.393 | 0.398 | 0.392|
> || 192| **0.414** | 0.421| 0.426 | 0.429 | 0.423|
> || 336| **0.438** | **0.438**| 0.446 | 0.450 | 0.443|
> || 720| 0.462 | 0.462| 0.470 | 0.474 | **0.458**|
> || Avg| **0.424** | 0.428| 0.434 | 0.438 | 0.429|
> | ETTh2| 96 | **0.329** | 0.338| 0.343 | 0.345 | 0.376|
> || 192| **0.379** | 0.387| 0.393 | 0.394 | 0.405|
> || 336| 0.415 | **0.401**| 0.427 | 0.431 | 0.440|
> || 720| **0.431** | 0.436| 0.446 | 0.445 | 0.477|
> || Avg| **0.388** | 0.391| 0.402 | 0.403 | 0.424|
> | ECL| 96 | **0.221** | 0.235| 0.229 | 0.234 | 0.279|
> || 192| **0.238** | 0.247| 0.247 | 0.249 | 0.281|
> || 336| **0.254** | 0.267| 0.264 | 0.267 | 0.295|
> || 720| **0.279** | 0.293| 0.294 | 0.299 | 0.327|
> || Avg| **0.248** | 0.260| 0.259 | 0.262 | 0.295|
> | Traffic| 96 | **0.226** | 0.274| 0.261 | 0.269 | 0.384|
> || 192| **0.241** | 0.280| 0.271 | 0.276 | 0.362|
> || 336| **0.250** | 0.290| 0.277 | 0.287 | 0.365|
> || 720| **0.270** | 0.309| 0.295 | 0.312 | 0.386|
> || Avg| **0.247** | 0.289| 0.276 | 0.286 | 0.374|
> | Weather| 96 | **0.190** | 0.207| 0.200 | 0.207 | 0.215|
> || 192| **0.235** | 0.248| 0.245 | 0.254 | 0.256|
> || 336| **0.280** | 0.290| 0.287 | 0.296 | 0.297|
> || 720| **0.333** | 0.341| 0.342 | 0.346 | 0.348|
> || Avg| **0.260** | 0.271| 0.269 | 0.276 | 0.279|
> | Solar | 96 | **0.191** | 0.232| 0.233 | 0.237 | 0.363|
> || 192| **0.213** | 0.247| 0.257 | 0.263 | 0.404|
> || 336| **0.229** | 0.257| 0.263 | 0.281 | 0.398|
> || 720| **0.236** | 0.252| 0.270 | 0.285 | 0.388|
> || Avg| **0.217** | 0.247| 0.256 | 0.266 | 0.388|
>
> **Thanks again for your constructive comments. We are very happy to answer any further questions.**

---

> ### Author Response · Authors · 2025-08-03
> **We would like to supplement more experiment results to fully address your concerns on quadratic cost of NormLin!**
>
> We would like to thank you again for your constructive comments!
>
> - To fully address your concerns about the quadratic cost of the NormLin layer, we **introduce a bottleneck block to reduce computational complexity**:
>   - the channels are first projected to a fixed width (denoted as *bottleneck size*),
>   - the NormLin module is applied in this lower-dimensional space,
>   - and the representation is finally projected back to $N$ channels.
>
> - In this way, **the complexity of the NormLin layer is reduced from $\mathcal{O}(N^2)$ to $\mathcal{O}(N)$**. We report the MAEs and resource footprints under various bottleneck sizes below.
>
> | **Dataset** | **Hor.** | **Ori.** | **Bottleneck (8)** | **Bottleneck (16)** | **Bottleneck (64)** | **Bottleneck (128)** |
> |-|-|-|-|-|-|-|
> | ETTh1 | 96 | **0.382**| **0.382** | **0.381** | **0.382** | 0.383|
> | | 192| **0.414**| 0.415| **0.412** | 0.415 | 0.416|
> | | 336| **0.438**| **0.438**| 0.440 | 0.443 | 0.442|
> | | 720| **0.462**| 0.467| 0.467 | 0.468 | **0.461**|
> | ECL | 96 | **0.221**| **0.236**| **0.236** | 0.241 | 0.240|
> | | 192| **0.238**| 0.255| **0.250** | 0.256 | 0.254|
> | | 336| **0.254**| **0.264**| 0.271 | 0.270 | 0.267|
> | | 720| **0.279**| 0.302| **0.291** | **0.291** | 0.295|
> | Traffic | 96 | **0.226**| **0.255**| **0.255** | 0.263 | 0.280|
> | | 192| **0.241**| **0.273**| 0.289 | 0.278 | 0.277|
> | | 336| **0.250**| **0.270**| 0.275 | 0.283 | 0.281|
> | | 720| **0.270**| 0.299| **0.296** | 0.298 | 0.301|
> | Weather | 96 | **0.190**| 0.192| **0.188** | **0.190** | **0.190**|
> | | 192| **0.235**| 0.240| **0.237** | 0.239 | 0.241|
> | | 336| **0.280**| 0.284| 0.287 | **0.281** | 0.282|
> | | 720| **0.333**| 0.338| **0.336** | 0.337 | 0.338|
> | PEMS03| 12 | **0.159**| 0.162| 0.161 | **0.160** | 0.162|
> | | 24 | **0.179**| 0.186| 0.183 | 0.182 | **0.181**|
> | | 48 | **0.210**| **0.223**| 0.224 | 0.224 | 0.224|
> | | 96 | **0.247**| 0.275| 0.266 | **0.264** | 0.271|
>
>
> | **Dataset** | **Metric** | **Ori.** | **Bottleneck (8)** | **Bottleneck (16)** | **Bottleneck (64)** | **Bottleneck (128)** |
> |-|-|-|-|-|-|-|
> | Traffic | Params(M)| 6.17 | **4.71** | **4.74**| 4.92| 5.16 |
> | | FLOPs(G) | 4.91 | **4.16** | **4.18**| 4.27| 4.39 |
> | | T.T. (s/iter)| 0.02 | **0.015**| **0.015** | 0.016 | 0.017|
> | | T.M.(GB) | 1.01 | **0.98** | **0.99**| 1.00| 1.01 |
> | | I.T.(ms/iter)| 5.71 | **5.60** | **5.63**| 5.71| 5.80 |
> | | I.M. (GB)| 0.43 | **0.39** | **0.39**| 0.40| 0.41 |
> | ECL | Params(M)| 4.79 | **4.59** | **4.60**| 4.67| 4.78 |
> | | FLOPs(G) | 1.65 | **1.55** | **1.56**| 1.59| 1.65 |
> | | T.T. (ms/iter) | 7.75 | **7.45** | **7.58**| 7.61| 7.80 |
> | | T.M.(GB) | **0.45** | **0.44** | **0.45**| **0.45**| 0.46 |
> | | I.T.(ms/iter)| 2.11 | **2.04** | **2.05**| 2.08| 2.09 |
> | | I.M. (GB)| 0.17 | **0.16** | **0.16**| **0.16**| 0.17 |
> | PEMS03| Params(M)| 4.84 | **4.60** | **4.61**| 4.69| 4.80 |
> | | FLOPs(G) | 1.85 | **1.73** | **1.74**| 1.77| 1.83 |
> | | T.T. (s/iter)| 8.14 | **8.05** | **8.06**| 8.12| 8.16 |
> | | T.M.(GB) | **0.48** | **0.48** | **0.48**| 0.49| 0.50 |
> | | I.T.(ms/iter)| 2.23 | **2.18** | **2.18**| 2.21| 2.24 |
> | | I.M. (GB)| 0.20 | **0.18** | **0.18**| **0.18**| 0.19 |
>
> - From above, we find that:
>   -  (1) the bottlenecked NormLin variants **incur approximately a 4% performance degradation**,
>   - but (2) **consume fewer resources**, especially on datasets with a large number of channels.
>
> - This provides readers with additional flexibility to decide whether the full quadratic version of NormLin is worthwhile for their specific use cases.
>
> ***
> We sincerely hope that our new comments can address your concerns. We’d deeply appreciate it if you could let us know whether your concerns have been addressed.
>
> Thanks for your time,
>
> Authors

---

> > ### Author Response · Authors · 2025-08-05
> >
> > Dear Reviewer CEMa,
> >
> > As the discussion deadline approaches, we are wondering whether our responses have properly addressed your concerns? Your feedback would be extremely helpful to us. If you have further comments or questions, we hope for the opportunity to respond to them.
> >
> > Many thanks,
> >
> > 10716 Authors

---

> > > ### Comment · Reviewer_CEMa · 2025-08-06
> > >
> > > Thank you for the comprehensive rebuttal that has clarified some of my concerns.
> > >
> > > Q1: While the justification for cov\_matrix / diag\_vec is now clear, my concern about the theoretical foundation remains. You've addressed the orthogonality of $Q$ (which is indeed guaranteed by construction), but my primary question was about the effectiveness of the decorrelation itself: can you show that $\text{CovMat}(Q_i^T x)$ is actually diagonal?
> > >
> > > More fundamentally, I question whether your sliding window approach actually estimates the autocorrelation of your input context. Your method computes correlations between overlapping subsequences $s_i$ and $s_j$, each with their own empirical means (in practice they could be almost equal indeed). However, true autocorrelation estimation requires:
> > > \begin{align}
> > > \gamma(k) = \mathbb{E}[(X_t - \mu)(X_{t+k} - \mu)]
> > > \end{align}
> > > where $\mu$ is a common reference point (typically the global mean). Your approach instead computes:
> > > \begin{align}
> > > \text{Corr}(s_i, s_j)
> > > \end{align}
> > > where $s_i$ has mean $\mu_i$ and $s_j$ has mean $\mu_j$.
> > >
> > > This is fundamentally different from autocorrelation estimation. For your input context of length $T$, the standard approach would estimate autocorrelation using the entire available time series with a consistent global mean, not through correlations between windowed subsequences with varying local means.
> > >
> > > Could you provide theoretical justification for why your windowing-based correlation matrix captures the same temporal dependencies as classical autocorrelation estimation? And more importantly, can you empirically show that this decorrelation actually works on unseen test data?
> > >
> > > Q2: Your justification for channel averaging (CA) over channel individual (CI) is reasonable given the computational and performance trade-offs. However, regarding the PCA comparison, that wasn't my point. Standard PCA would decompose the series into $N$ non-overlapping windows of length $T$ and center them. You would have a matrix $W$ of size $(N,T)$, and you would decompose $W^T W$ to obtain the $Q$ matrix and proceed with the next steps as you do. This is different from your approach that applies T overlapping sliding windows of length $M-T$ and computes the Pearson correlation between them.
> > >
> > > Q3: The dimension extension explanation is experimentally adequate.
> > >
> > > Q5: The clarification on how NormLin is applied in the temporal dimension is now clear.
> > >
> > > Q6: The comparison with recent lightweight baselines (SamFormer, SimpleTM, TQNet, TimePro, TimeBase) is valuable and shows competitive performance. The bottleneck analysis effectively addresses the computational complexity concerns while providing a practical trade-off between performance and efficiency.

---

> ### Author Response · Authors · 2025-08-07
>
> Dear Reviewer CEMa,
>
> We highly appreciate your constructive comments, and we are glad that you found our initial responses have addressed your concerns on recent lightweight baselines, quadratic computational complexity and implementation details. We address your questions point-to-point in the following.
>
>
>
> > Q1 (Part 1): Can you show that $\mathrm{CovMat} (\mathbf{Q}_i^T\mathbf{x})$ is actually diagonal?
>
> - Thank you for raising this point. Theoretically, suppose the temporal correlation matrix $\mathrm{CovMat}_t$ admits the eigendecomposition $\mathrm{CovMat}_t = \mathrm{Q}_i \mathbf{\Lambda} \mathrm{Q}_i^T$, where $\mathrm{Q}_i$ is an orthogonal matrix and $\mathbf{\Lambda}$ is a diagonal matrix. If the vector $\mathbf{x}$ is normalized (zero mean and unit variance), then the following holds:
>
> $$\mathrm{CovMat} (\mathbf{Q}_i^T\mathbf{x})=\mathbb{E}[\mathbf{Q}_i^T \mathbf{x} \mathbf{x}^T \mathbf{Q}_i]=\mathbf{Q}_i^T \mathbb{E}[\mathbf{x} \mathbf{x}^T]   \mathbf{Q}_i=\mathbf{Q}_i^T \mathrm{CorrMat}_t    \mathbf{Q}_i = \mathbf{Q}_i^T \mathbf{Q}_i \mathbf{\Lambda} \mathbf{Q}_i^T    \mathbf{Q}_i =\mathbf{\Lambda},  $$
>
> which is a diagonal matrix.
>
> - Empirically, we quantify the effectiveness of OrthoTrans by reporting the **Frobenius norm of the off-diagonal elements** in the correlation matrices for the original series, and for the series transformed by OrthoTrans, DFT, or Haar wavelets, using a window size of 96. For DFT, correlations are computed using only the real part of the transformed signal, while for Haar wavelets, only the high-frequency components are used. The results below show that OrthoTrans effectively mitigates temporal correlations and outperforms both DFT and Haar wavelets in decorrelation on both the training and test sets.
>
> | **Dataset** | **Training set** |  |  | | **Test set** |  |  | |
> |-|-|-|-|-|-|-|-|-|
> | | Original | OrthoTrans | DFT| Wavelet | Original| OrthoTrans | DFT| Wavelet |
> | ETTh1 | 61.36| **1.46** | 2.58 | 4.93| 47.86 | **1.89** | 2.81 | 5.35|
> | ETTm2 | 83.04| **1.26** | 5.22 | 2.13| 65.74 | **2.05** | 6.46 | 3.22|
> | ECL | 45.95| **1.78** | 2.26 | 11.20 | 45.69 | **2.03** | 2.57 | 11.26 |
> | Traffic | 34.80| **1.79** | 3.29 | 6.92| 34.77 | **2.16** | 3.46 | 7.01|
> | Weather | 61.70| **2.86** | 19.61| 4.33| 65.46 | **3.97** | 22.10| 4.31|
> | *Avg* | 57.37| **1.83** | 6.59 | 5.90| 51.90 | **2.42** | 7.48 | 6.23|
>
>
> > Q1 (Part 2): On global mean (std) and local mean (std)
>
> - We would like to show that the **local mean and standard deviation (std) of the sub-series are very close to the global mean and std**, since each sub-series contains only $T-1$ fewer elements than the global series (of length $M \gg T$). We report the following metrics: $\mu \_{err}=\max_{i} \left | \frac{\mathrm{mean} (\mathbf{s} \_i)}{\mu}-1  \right |$, and $\mathrm{std}  \_{err}=\max_{i} \left | \frac{\mathrm{std} (\mathbf{s} _i)}{\mathrm{STD} }-1  \right |  $, where $\mu$ and $\mathrm{STD}$ denote the global mean and std, respectively.
>
> | **Dataset** | **$\mu \_{err}$** | **$\mathrm{std}  \_{err}$** |
> |-|-|-|
> | ETTm1 | 1.4E-03| 8.0E-04 |
> | Solar | 1.5E-03| 9.0E-04 |
> | ECL | 1.4E-03| 2.5E-03 |
> | Traffic | 3.5E-03| 2.8E-03 |
> | Weather | 1.6E-03| 1.3E-03 |
>
> > Q1 (Part 3): On classic autocorrelation estimation (ACF)
>
> - We respectfully argue that for a univariate series, computing the ACF inherently involves comparing two subseries of the original signal, as it measures the similarity between the signal and its lagged versions.
>
> - Assume that the series $\mathbf{x} \in \mathbb{R}^M$ has been normalized (i.e., zero mean and unit variance). The ACF can be regarded as the covariance between two sub-series:
> $$ \gamma \_k = \mathrm{E} \left [ \mathbf{x} \_t \mathbf{x} \_{t+k} \right ]=\frac{1}{M-k} \sum_{i=0}^{M-k-1} \mathbf{x}\_i \mathbf{x}\_{i+k} = \mathrm{Cov}(\mathbf{s}\_{1,k}',\mathbf{s}\_{2,k}'  ), $$
>
> where $\mathbf{s}\_{1,k}'=\left [ \mathbf{x}\_0, \mathbf{x}\_1, \cdots,\mathbf{x}\_{M-k-1}  \right ] $, and $\mathbf{s}\_{2,k}'=\left [ \mathbf{x}\_k, \mathbf{x}\_{k+1}, \cdots,\mathbf{x}\_{M-1}  \right ] $. We assume that the means of $\mathbf{s}\_{1,k}'$ and $\mathbf{s}\_{2,k}'$ are 0.
> Both $\mathbf{s}\_{1,k}'$ and $\mathbf{s}\_{2,k}'$ have length $M-k$.
>
> - In OrthoTrans, $\gamma \_k = \mathrm{Cov} (\mathbf{s}\_i, \mathbf{s}\_{i+k} ),$
> where $\mathbf{s}\_{i}=\left [ \mathbf{x}\_i, \mathbf{x}\_{i+1}, \cdots,\mathbf{x}\_{M-T+i}  \right ] $ and $\mathbf{s}\_{i+k}=\left [ \mathbf{x}\_{i+k}, \mathbf{x}\_{i+k+1}, \cdots,\mathbf{x}\_{M-T+i+k}  \right ] $.
>
> - By comparing $\mathbf{s}\_{1,k}'$, $\mathbf{s}\_{2,k}'$ with $\mathbf{s}\_{i}$, $\mathbf{s}\_{i+k}$, we can observe that $\mathbf{s}\_{i}$ and $\mathbf{s}\_{i+k}$ are the **truncated versions** of $\mathbf{s}\_{1,k}'$ and $\mathbf{s}\_{2,k}'$ (with $T-k-1$ fewer entries) , respectively. In other words, **our sliding window approach is a (slightly) truncated version of the classical ACF**.

---

> ### Author Response · Authors · 2025-08-07
>
> > Q1 (Part 4): Performance comparison of OrthoTrans and ACF
>
> - We compute $\gamma \_k = \mathrm{E} \left [ \mathbf{x} \_t \mathbf{x} \_{t+k} \right ], k=1,2,\cdots, T-1$, and obtain  the vector $[1, \gamma \_1, \gamma \_2, \cdots, \gamma \_{T-1}]$. Based on this vector, we construct the symmetric Toeplitz correlation matrix and compute the corresponding Q matrix. We denote this method as ACF.
>
> - We directly compare the differences between the correlation matrices and $\mathbf{Q}$ matrices obtained from OrthoTrans and ACF using the metric $\mathrm{Diff}_{\mathrm{F\text{-}norm}}(\mathbf{A},\mathbf{B} ) = \left \|| \mathbf{A}-\mathbf{B}  \right \||_F / T^2 $, where $\left \|| \cdot  \right \||_F$ denotes the Frobenius norm. As shown below, the differences are negligible.
>
> | **Dataset** | **CorrMat_Diff_Fnorm** | **Q_Diff_Fnorm** |
> |-|-|-|
> | ETTh1 | 1.5E-04| 5.4E-04|
> | ETTm2 | 1.0E-06| 4.6E-04|
> | ECL | 8.0E-06| 6.2E-04|
> | Traffic | 1.1E-05| 4.4E-04|
> | Weather | 4.0E-06| 2.0E-03|
>
>
>
> - We report the off-diagonal Frobenius norm of the correlation matrices (size: $96 \times 96$) for the original series, and for the series transformed by OrthoTrans and ACF. The following results show that OrthoTrans and ACF achieve **comparable and strong decorrelation performance on both training and test sets**.
>
> | **Dataset** | **Training set** | | | **Test set** | | |
> |-|-|-|-|-|-|-|
> | | Original | Ours| ACF | Original| Ours| ACF  |
> | ETTh1 | 61.36| 1.46| **1.45**| 47.86 | 1.89| **1.87**|
> | ETTm2 | 83.04| **1.26**| 1.27| 65.74 | **2.05**| 2.06|
> | ECL | 45.95| **1.78**| 1.80| 45.69 | **2.03**| 2.05|
> | Traffic | 34.80| **1.79**| 1.82| 34.77 | **2.16**| 2.18|
> | Weather | 61.70| **2.86**| **2.86**| 65.46 | 3.97| **3.96**|
> | *Avg* | 57.37| **1.83**| 1.84| 51.90 | **2.42**| **2.42**|
>
> - We further compare the forecasting performance of OrthoTrans and ACF.  As shown below, **OrthoTrans performs comparably to ACF**.
>
> | **Dataset** | **Hor.** | **OrthoTrans** || **ACF** ||
> |-|-|-|-|-|-|
> | || MSE| MAE| MSE | MAE|
> | ETTh1 | 96 | 0.360| 0.382| 0.361 | 0.383|
> | | 192| 0.416| 0.414| 0.415 | 0.414|
> | | 336| 0.457| 0.438| 0.457 | 0.438|
> | | 720| 0.463| 0.462| 0.464 | 0.463|
> | | Avg| **0.424**| **0.424**| **0.424** | **0.424**|
> | ETTm2 | 96 | 0.169| 0.249| 0.169 | 0.249|
> | | 192| 0.232| 0.290| 0.232 | 0.290|
> | | 336| 0.291| 0.328| 0.291 | 0.329|
> | | 720| 0.389| 0.387| 0.389 | 0.387|
> | | Avg| **0.270**| **0.313**| **0.270** | 0.314|
> | ECL | 96 | 0.131| 0.221| 0.131 | 0.221|
> | | 192| 0.150| 0.238| 0.152 | 0.240|
> | | 336| 0.165| 0.254| 0.167 | 0.255|
> | | 720| 0.191| 0.279| 0.190 | 0.280|
> | | Avg| **0.159**| **0.248**| 0.160 | 0.249|
> | Traffic | 96 | 0.398| 0.226| 0.404 | 0.226|
> | | 192| 0.439| 0.241| 0.434 | 0.241|
> | | 336| 0.464| 0.250| 0.458 | 0.250|
> | | 720| 0.502| 0.270| 0.508 | 0.270|
> | | Avg| **0.451**| **0.247**| **0.451** | **0.247**|
> | Weather | 96 | 0.153| 0.190| 0.151 | 0.190|
> | | 192| 0.200| 0.235| 0.203 | 0.239|
> | | 336| 0.258| 0.280| 0.259 | 0.281|
> | | 720| 0.337| 0.333| 0.338 | 0.333|
> | | Avg| **0.237**| **0.260**| 0.238 | **0.260**|
>
>
> > Q2 (Part 1): On OrthoTrans and PCA
>
> - Thank you for your clear suggestion! We refer to the suggested variant as PCA. It first segments the series $\mathbf{x}\in \mathbb{R}^{M}$ into $N'$ non-overlapping windows of length $T$, forming the matrix $\mathbf{W}\in \mathbb{R}^{N' \times T}$. Then the correlations are computed among the columns of $\mathbf{W}$.
>
> - In essence, OrthoTrans and the PCA variant differ in their unfolding strategies: although both use a patch size of $T$, **OrthoTrans employs sliding windows with stride 1**, whereas **PCA uses a non-overlapping segmentation with stride $T$**.
>
> - We further introduce a variant, **OrthoTrans-C**, for comparison:
>   - In OrthoTrans-C, we do not use the CA strategy. Given the multivariate training series $\mathbf{X}^{train} \in \mathbb{R}^{N \times M} $, we apply PyTorch’s ```unfold``` function along the time dimension (second dimension) with patch size $T$ and stride 1, resulting in the matrix $\mathbf{X}^{train}\_{fold} \in \mathbb{R}^{N \times M' \times T} $. Then we flatten the first two dimensions of $\mathbf{X}^{train}\_{fold}$ to obtain a matrix of size $(N \cdot M') \times T$, and compute the correlation matrix across its columns.

---

> ### Author Response · Authors · 2025-08-07
>
> > Q2 (Part 2): Decorrelation performance of OrthoTrans, PCA and OrthoTrans-C
>
>
> - We compare the decorrelation effectiveness of OrthoTrans, PCA and OrthoTrans-C using the off-diagonal Frobenius norm with a window size of 96. In general, OrthoTrans and OrthoTrans-C achieve better decorrelation performance than PCA.
>
> | **Dataset** | **Training set** |||| **Test set** ||||
> |-|-|-|-|-|-|-|-|-|
> | | Original | OrthoTrans | PCA| OrthoTrans-C | Original| OrthoTrans | PCA| OrthoTrans-C |
> | ETTh1 | 61.36| 1.46 | 10.77| **1.43** | 47.86 | 1.89 | 11.66| **1.75** |
> | ETTm2 | 83.04| 1.26 | 7.98 | **1.01** | 65.74 | 2.05 | 9.37 | **1.55** |
> | ECL | 45.95| **1.78** | 11.87| 2.24 | 45.69 | **2.03** | 12.00| 2.71 |
> | Traffic | 34.80| **1.79** | 8.94 | 1.81 | 34.77 | **2.16** | 9.11 | 2.17 |
> | Weather | 61.70| **2.86** | 8.78 | 2.94 | 65.46 | 3.97 | 10.47| **3.78** |
> | Avg | 57.37| **1.83** | 9.67 | 1.89 | 51.90 | 2.42 | 10.52| **2.39** |
>
> - To further study the relationship between decorrelation performance and unfolding stride in PCA, we report the following results on the testsets of different datasets, which clearly show that **smaller strides lead to better decorrelation performance**.
>
> | **Dataset** | **Stride: 1** | **Stride: 16** | **Stride: 32** | **Stride: 64** | **Stride: 96** |
> |-|-|-|-|-|-|
> | ETTh1 | 1.89| 5.92 | 6.60 | 7.36 | 11.66|
> | ETTm2 | 2.05| 4.40 | 5.95 | 6.71 | 9.37 |
> | ECL | 2.03| 6.41 | 6.58 | 6.76 | 12.00|
> | Traffic | 2.16| 4.45 | 4.53 | 4.68 | 9.11 |
> | Weather | 3.97| 7.91 | 8.70 | 9.83 | 10.47|
> | Avg | 2.42| 5.82 | 6.47 | 7.07 | 10.52|
>
>
> > Q2 (Part 3): Forecasting performance of OrthoTrans, PCA and OrthoTrans-C
>
> - We apply these decorrelation methods to OLinear and observe that they achieve comparable performance. The results also verify the robustness of OLinear to different  $\mathbf{Q}$ matrices.
>
> | **Dataset** | **Hor.** | **OrthoTrans** || **PCA** || **OrthoTrans-C** ||
> |-|-|-|-|-|-|-|-|
> | || MSE| MAE| MSE | MAE| MSE| MAE|
> | ETTh1 | 96 | 0.360| 0.382| 0.361 | 0.382| 0.361| 0.383|
> | | 192| 0.416| 0.414| 0.415 | 0.414| 0.415| 0.414|
> | | 336| 0.457| 0.438| 0.457 | 0.439| 0.458| 0.439|
> | | 720| 0.463| 0.462| 0.463 | 0.462| 0.464| 0.463|
> | | Avg| **0.424**| **0.424**| **0.424** | **0.424**| **0.424**| **0.424**|
> | ETTm2 | 96 | 0.169| 0.249| 0.169 | 0.249| 0.169| 0.249|
> | | 192| 0.232| 0.290| 0.233 | 0.290| 0.232| 0.290|
> | | 336| 0.291| 0.328| 0.291 | 0.328| 0.291| 0.328|
> | | 720| 0.389| 0.387| 0.389 | 0.387| 0.389| 0.387|
> | | Avg| **0.270**| **0.313**| **0.270** | 0.314| **0.270**| 0.314|
> | ECL | 96 | 0.131| 0.221| 0.131 | 0.221| 0.131| 0.221|
> | | 192| 0.150| 0.238| 0.152 | 0.240| 0.153| 0.241|
> | | 336| 0.165| 0.254| 0.167 | 0.256| 0.167| 0.255|
> | | 720| 0.191| 0.279| 0.195 | 0.285| 0.191| 0.278|
> | | Avg| **0.159**| **0.248**| 0.161 | 0.250| 0.160| 0.249|
> | Traffic | 96 | 0.398| 0.226| 0.402 | 0.226| 0.402| 0.226|
> | | 192| 0.439| 0.241| 0.435 | 0.240| 0.437| 0.240|
> | | 336| 0.464| 0.250| 0.462 | 0.250| 0.463| 0.250|
> | | 720| 0.502| 0.270| 0.505 | 0.272| 0.509| 0.270|
> | | Avg| **0.451**| 0.247| **0.451** | 0.247| 0.453| **0.246**|
> | Weather | 96 | 0.153| 0.190| 0.152 | 0.189| 0.152| 0.189|
> | | 192| 0.200| 0.235| 0.201 | 0.236| 0.202| 0.238|
> | | 336| 0.258| 0.280| 0.260 | 0.282| 0.260| 0.283|
> | | 720| 0.337| 0.333| 0.339 | 0.333| 0.335| 0.332|
> | | Avg| **0.237**| **0.260**| 0.238 | **0.260**| **0.237**| **0.260**|
>
>
> - We also examine the robustness of OrthoTrans, PCA, and OrthoTrans-C **when the $\mathbf{Q}$ matrices are computed using only the first 10% of the training set**. We observe that OLinear performs robustly in this setting, consistent with the finding in Appendix I.8.
>
> | **Dataset** | **Hor.** | **OrthoTrans** || **PCA** || **OrthoTrans-C** ||
> |-|-|-|-|-|-|-|-|
> | || MSE| MAE| MSE | MAE| MSE| MAE|
> | ECL | 96 | 0.131| 0.221| 0.131 | 0.221| 0.131| 0.221|
> | | 192| 0.152| 0.240| 0.152 | 0.240| 0.154| 0.242|
> | | 336| 0.166| 0.255| 0.168 | 0.256| 0.165| 0.254|
> | | 720| 0.189| 0.278| 0.186 | 0.278| 0.186| 0.278|
> | | Avg| **0.159**| **0.248**| **0.159** | 0.249| **0.159**| **0.248**|
> | Traffic | 96 | 0.402| 0.226| 0.400 | 0.226| 0.404| 0.226|
> | | 192| 0.431| 0.240| 0.434 | 0.240| 0.438| 0.241|
> | | 336| 0.465| 0.250| 0.463 | 0.250| 0.464| 0.250|
> | | 720| 0.512| 0.271| 0.502 | 0.272| 0.500| 0.272|
> | | Avg| 0.453| **0.247**| **0.450** | **0.247**| 0.452| **0.247**|
> | Weather | 96 | 0.149| 0.187| 0.152 | 0.189| 0.152| 0.189|
> | | 192| 0.203| 0.239| 0.200 | 0.237| 0.202| 0.238|
> | | 336| 0.259| 0.281| 0.262 | 0.282| 0.264| 0.285|
> | | 720| 0.339| 0.332| 0.338 | 0.334| 0.347| 0.338|
> | | Avg| **0.237**| **0.260**| 0.238 | **0.260**| 0.241| 0.262|

---

> ### Author Response · Authors · 2025-08-07
>
> - The newly added experiments show that OrthoTrans, ACF, PCA, and OrthoTrans-C perform similarly when applied to OLinear, demonstrating its robustness to different decorrelation methods. We will include a discussion of these variants in a new appendix section and incorporate the results presented here to further strengthen our work. Thank you again for your constructive comments!
>
> > Q3-6: On the dimension extension, temporal NormLin, lightweight baselines, and computational complexity
>
> - We appreciate the reviewer’s constructive comments and are glad that our responses have addressed the concerns.
>
> ***
> We are eager to hear your feedback. We sincerely hope that our new comments can address your remaining concerns. We’d deeply appreciate it if you could let us know whether your concerns have been addressed. Thank you so much!
>
> Thanks for your time,
>
> Authors of #10716

---

> ### Comment · Reviewer_CEMa · 2025-08-08
> **Response to Authors**
>
> Thank you for your detailed response. You have treated most of my concerns, so I have decided to raise my score.
> However, I still would raise some important issues that I would like to ask the authors to solve.
> - Please be more mathematically rigorous. If I understand correctly, you assume that $CorrMat_t$  is close to $CovMat_t$ ( what is $CorrMat_t$ ? Is it the correlation matrix for a fixed channel?). In this case, you need to use $\approx$ in all transitions like "$CorrMat_t = Q_i \Lambda Q_i^\top$".
> - If ACF and your strategy give the same results, then I would suggest the authors to stick to ACF, because it is more intuitive and familiar to the broad reader. To me, it was very confusing to follow the paper, especially with $CovMat_t$ notation (by the way, it is confusing to write $CovMat_t$ and call it "the temporal correlation matrix").
> - When you compute the Frobenius norm of the off-diagonal elements, the values are not that meaningful as it depends on the size of the matrix. Maybe dividing by the number of off-diagonal elements would be more intuitive in order to have a value within [0,1].
>
> Please incorporate all the modifications we have discussed during the rebuttal. This will greatly improve the quality of the paper.
>
> Best regards,
> Reviewer CEMa

---

> ### Author Response · Authors · 2025-08-08
>
> Dear Reviewer CEMa,
>
> We highly appreciate your constructive responses and great efforts in our multi-round discussion. We are encouraged that you found our responses have addressed most of your concerns! We would like to respond to your remaining questions as follows.
>
> > Q1: If I understand correctly, you assume that $\mathrm{CorrMat}_t$ is close to $\mathrm{CovMat}_t$ ( what is $\mathrm{CorrMat}_t$? Is it the correlation matrix for a fixed channel?). In this case, you need to use $\approx$ in all transitions like "$\mathrm{CorrMat}_t=\mathbf{Q} _i \mathbf{\Lambda}  \mathbf{Q} _i^T$".
>
> - Thank you for raising this point. We sincerely apologize for any confusion caused by our mathematical notations, and would like to offer the clarifications as follows.
>
> - We use $\mathrm{CorrMat}$ and $\mathrm{CovMat}$ to denote the **correlation** and **covariance** matrices, respectively, throughout the paper. The subscript $t$ (or $v$, Line 262) indicates the temporal (or variate) dimension. Specifically, $\mathrm{CorrMat}_t$ denotes the correlation matrix among the time-lagged subseries, averaged across all variates (Line 146). For a specific variate $j$, we denote this as $\mathrm{CorrMat}_t^j$ (Line 140).
>
>
> - For a normalized series, it holds that $\mathrm{CorrMat}_t \approx \mathrm{CovMat}_t$ , based on the fact that the subseries share (almost) the same unit variance. However, since we explicitly define $\mathrm{CorrMat}_t=\mathbf{Q} _i \mathbf{\Lambda}  \mathbf{Q} _i^T$, rather than $\mathrm{CovMat}_t=\mathbf{Q} _i \mathbf{\Lambda}  \mathbf{Q} _i^T$,  on Line 149 in the paper, the use of the approximation symbol $\approx$ in the equation on Line 153 seems unnecessary.
>
> - We are sorry for making a mistake in our previous response for writing *"suppose the temporal correlation matrix $\mathrm{CovMat}_t$ admits the eigendecomposition $\mathrm{CovMat}_t = \mathrm{Q}_i \mathbf{\Lambda} \mathrm{Q}_i^T$"*.
>
> - The correct one should be $\mathrm{CorrMat}_t$ instead of $\mathrm{CovMat}_t$. We sincerely apologize for the confusion this has caused. We have thoroughly examined the paper, and are relieved to observe that these two notions are consistently and correctly used in the manuscript.
>
> > Q2: If ACF and your strategy give the same results, then I would suggest the authors to stick to ACF, because it is more intuitive and familiar to the broad reader. To me, it was very confusing to follow the paper, especially with $\mathrm{CovMat}_t$ notation (by the way, it is confusing to write $\mathrm{CovMat}_t$ and call it "the temporal correlation matrix").
>
> - Thank you for the suggestion! We will choose to stick **ACF** in Section 4.2 for our final version, which may be more familiar to a broad range of readers.
>
> - Please kindly note that the ACF here is only used for computing $\mathrm{CorrMat}_t$ in OrthoTrans, so that such revision does not affect the core contributions of our paper.
>
> - We have carefully reviewed the manuscript and confirmed that $\mathrm{CorrMat}_t$ is not mistakenly written as $\mathrm{CovMat}_t$ in our final version.
>
> > Q3: On the metric to quantify decorrelation
>
> - Thank you for the great suggestion! We divide the off-diagonal Frobenius norm by the number of off-diagonal elements, and report this new metric below.
>   - In our previous experiments of quantifying decorrelation, for DFT we used only the real part, and for Haar wavelets we used only the high-frequency components. We now adopt a fairer scheme: for DFT, we concatenate the real and imaginary parts; for Haar wavelets, we concatenate both the low- and high-frequency components.
>
> | **Dataset** | **Training Set** |  | | | **Test Set** |  | | |
> |-|-|-|-|-|-|-|-|-|
> | | Original | OrthoTrans | DFT | Wavelet | Original | OrthoTrans | DFT | Wavelet |
> | ETTh1 | 6.7E-03| **1.6E-04**| 5.8E-04 | 3.7E-03 | 5.2E-03| **2.1E-04**| 6.0E-04 | 3.1E-03 |
> | ETTm2 | 9.1E-03| **1.4E-04**| 1.1E-03 | 4.6E-03 | 7.2E-03| **2.2E-04**| 1.3E-03 | 3.7E-03 |
> | ECL | 5.0E-03| **1.9E-04**| 4.7E-04 | 3.6E-03 | 5.0E-03| **2.2E-04**| 4.9E-04 | 3.6E-03 |
> | Traffic | 3.8E-03| **1.9E-04**| 5.5E-04 | 2.6E-03 | 3.8E-03| **2.3E-04**| 5.7E-04 | 2.6E-03 |
> | Weather | 6.7E-03| **3.1E-04**| 4.1E-03 | 3.6E-03 | 7.2E-03| **4.0E-04**| 4.6E-03 | 3.7E-03 |
> | Avg | 6.3E-03| **2.0E-04**| 1.4E-03 | 3.6E-03 | 5.7E-03| **2.6E-04**| 1.5E-03 | 3.3E-03 |
>
>
> - Under this new metric, **OrthoTrans reduces the correlation in the original series by 96%, and outperforms DFT and Haar wavelet transforms by 84% and 93%, respectively**.
>
> - Following your suggestion, we will include the quantitation of temporal decorrelation of OrthoTrans in a new appendix section and incorporate the results presented here to further strengthen our work.
>
> ***
> We really appreciate your valuable suggestions which have undoubtedly contributed to improving the quality of our paper. Please let us know if we have properly addressed your questions and we are more than happy to discuss more!
>
> Thanks for your time,
>
> Authors of #10716

---

### Official Review · Reviewer_rQjb · 2025-07-02

**Clarity:** 3
**Significance:** 3
**Originality:** 3
**Rating:** 4
**Confidence:** 4

**Summary:**

This paper tackles entangled temporal dependencies in multivariate time series forecasting through two main contributions: a data-adaptive orthogonal transformation based on eigenvalue decomposition of Pearson correlation matrices, and a computationally efficient normalized linear layer (NormLin) that serves as an alternative to self-attention mechanisms. The proposed method leverages dataset-specific statistical properties to decorrelate temporal features before applying linear forecasting models. The comprehensive evaluation across multiple datasets and and tasks in this paper demonstrates consistent performance gains over strong baselines, with both components showing modularity as plug-in enhancements for existing architectures.

**Questions:**

1. How does the method perform on highly non-stationary time series where the correlation structure changes significantly over time? Have you considered adaptive or online updates to the orthogonal matrices?

2. You mention that OrthoTrans improves the rank of attention matrices in Transformer models. Can you provide more detailed analysis or proof of this phenomenon? What is the relationship between decorrelation and representation diversity?

3. Have you compared against other adaptive bases beyond the eigenvalue decomposition, such as those learned through neural networks or other unsupervised methods?

4. The orthogonal matrices Qi and Qo are computed from the training data. The paper only briefly mentions robustness to limited training data in the appendix but doesn't thoroughly investigate how performance degrades with smaller training sets or distribution shifts.

**Ethical Concerns:**

["NO or VERY MINOR ethics concerns only"]

**Final Justification:**

The authors have provided comprehensive response to my questions. I choose to maintain my current score.

**Limitations:**

Yes

**Quality:**

3

**Strengths And Weaknesses:**

**Strengths:**

This paper presents a well-executed combination of theoretical innovation and empirical rigor. The core contribution using eigenvalue decomposition of temporal Pearson correlation matrices for orthogonal transformation, provides an elegant theoretical foundation with clear justification. The work demonstrates state-of-the-art performance across multiple datasets and tasks, with both components serving as effective plug-in modules that consistently improve various baseline models. I am particularly impressed by the comprehensive experimental evaluation, which includes extensive ablation studies, and evaluation across carious tasks with different prediction horizons.


**Weaknesses:**

See questions below

---

> ### Author Rebuttal · Authors · 2025-07-26
>
> Thank you for your constructive response. We sincerely appreciate the reviewer’s recognition of OLinear as "**a well-executed combination of theoretical innovation and empirical rigor**". Below are our responses to the concerns:
>
> > Q1: On non-stationary time series
> - Financial data (e.g., stock prices and exchange rates) are widely recognized as non-stationary time series. As shown in Tables 2 and 3 of the paper, OLinear consistently achieves state-of-the-art performance on the real-world non-stationary time series datasets: Exchange, NASDAQ, SP500, and DowJones.
> - Regarding online updates of the Q matrices, we further conduct experiments where Q matrices are replaced by those computed on the validation set. After retraining with the updated Q matrices (using the same trainset), the model exhibits similar forecasting performance (as shown below). This phenomenon implies that the Q matrices obtained by OrthoTrans with the trainset can **generalize well to the full dataset**. Further discussion on online learning could be a **future direction** of this work.
>
> | **Dataset** | **Hor.** | **Data in paper** |   | **Val_Q and Retrain** |    |
> |-------------|----------|--------------------------------|--------|-----------------------|--------|
> |             |          | MSE                            | MAE    | MSE                   | MAE    |
> | Exchange    | 96       | 0.082                          | 0.200  | 0.082                 | 0.200  |
> |             | 192      | 0.171                          | 0.293  | 0.171                 | 0.293  |
> |             | 336      | 0.331                          | 0.414  | 0.331                 | 0.414  |
> |             | 720      | 0.837                          | 0.688  | 0.836                 | 0.688  |
> |             | Avg      | 0.355                          | 0.399  | 0.355                 | 0.399  |
> | NASDAQ      | 3        | 0.036                          | 0.092  | 0.036                 | 0.092  |
> |             | 6        | 0.049                          | 0.117  | 0.049                 | 0.117  |
> |             | 9        | 0.062                          | 0.137  | 0.062                 | 0.137  |
> |             | 12       | 0.073                          | 0.154  | 0.073                 | 0.154  |
> |             | Avg      | 0.055                          | 0.125  | 0.055                 | 0.125  |
> | SP500       | 3        | 0.035                          | 0.126  | 0.035                 | 0.127  |
> |             | 6        | 0.053                          | 0.158  | 0.054                 | 0.158  |
> |             | 9        | 0.070                          | 0.181  | 0.070                 | 0.181  |
> |             | 12       | 0.088                          | 0.204  | 0.088                 | 0.204  |
> |             | Avg      | 0.061                          | 0.167  | 0.061                 | 0.167  |
>
>
>
> > Q2: On the rank improvement of attention matrices
> - The observation that OrthoTrans improves the rank of attention matrices is interesting and partly explains why OrthoTrans, as a plug-in, consistently enhances Transformer-based forecasters (see Table 5 of the paper). A potential reason behind this phenomenon is that **OrthoTrans can mitigate the intrinsic low-rank property** of time series data. The empirical results are presented below.
>
> -  | **Dataset** | **Identity** | **OrthoTrans** | **DFT** | **Wavelet** |
> |-------------|--------------|----------------|---------|-------------|
> | ECL         | 0.162        | **0.446**          | 0.332   | 0.381       |
> | Traffic     | 0.002        | **0.568**          | 0.177   | 0.049       |
> | Weather     | 0.012        | **0.572**          | 0.352   | 0.292       |
> | Solar       | 0.007        | **0.481**          | 0.032   | 0.023       |
> | PEMS03      | 0.241        | **0.450**          | 0.356   | 0.395       |
>
>
> - Following TimeBase (ICML 2025), we compute the **median singular values** of the correlation matrix among non-overlapping time series patches (patch length = 24, number of patches = 30); see Figure 1 in TimeBase for more details. The results are presented above. As shown, OrthoTrans yields **larger singular values** than other transformation bases (with “Identity” denoting no transformation). In other words, OrthoTrans alleviates low-rank tendencies [1,2] and **enhances data diversity**.
>
>
> > Q3: On adaptive bases
>
> - We have considered learnable Q matrices for potential performance improvements. The most straightforward approach is adding a learnable matrix to Q. More advanced alternatives include using a learnable vector $\mathbf{v}$ with a Householder transformation to ensure orthogonality:
> $$\mathbf{Q}'=\mathbf{Q} \left ( \mathbf{I} -2\frac{\mathbf{v}\mathbf{v}^T}{\left \| \mathbf{v}  \right \|^2_2 }  \right ), $$
> or applying the Cayley transform:
> $$\mathbf{Q}'= \left ( \mathbf{I}+\mathbf{A} +\mathbf{V}   \right )^{-1} \left ( \mathbf{I}-\mathbf{A} -\mathbf{V}   \right ),$$
> where $\mathbf{V}$ is the learnable skew-symmetric matrix and $\mathbf{A}=\left ( \mathbf{I} -\mathbf{Q}  \right ) \left ( \mathbf{I} + \mathbf{Q}  \right )^{-1}$.
> We also explored an input-adaptive version: $$\mathbf{Q}'=\mathbf{Q}+ \alpha \cdot \mathrm{Linear_1}(\mathbf{x}) \mathrm{Linear_2}(\mathbf{x})^\mathrm{T} , $$
> where $\mathbf{x}$ is the input series and $\alpha$ is the learnable parameter. The results are reported below:
>
> | **Dataset** | **Horizon** | **OLinear** |   | **Plus Delta** |    | **Householder** |    | **Cayley** |    | **Adapt_Q** |   |
> |-------------|-------------|-------------|--------|----------------|--------|-----------------|--------|------------|--------|-------------|--------|
> |             |             | MSE         | MAE    | MSE            | MAE    | MSE             | MAE    | MSE        | MAE    | MSE         | MAE    |
> | ECL         | 96          | 0.131       | 0.221  | 0.131          | 0.221  | 0.131           | 0.221  | 0.130      | 0.221  | 0.131       | 0.221  |
> |             | 192         | 0.150       | 0.238  | 0.152          | 0.240  | 0.153           | 0.241  | 0.153      | 0.241  | 0.153       | 0.241  |
> |             | 336         | 0.165       | 0.254  | 0.166          | 0.256  | 0.166           | 0.255  | 0.165      | 0.255  | 0.164       | 0.254  |
> |             | 720         | 0.191       | 0.279  | 0.188          | 0.277  | 0.193           | 0.282  | 0.188      | 0.278  | 0.192       | 0.281  |
> |             | Avg         | 0.159       | 0.248  | 0.159          | 0.248  | 0.161           | 0.250  | 0.159      | 0.249  | 0.160       | 0.249  |
> | Traffic     | 96          | 0.398       | 0.226  | 0.397          | 0.227  | 0.400           | 0.226  | 0.401      | 0.226  | 0.399       | 0.226  |
> |             | 192         | 0.439       | 0.241  | 0.436          | 0.241  | 0.435           | 0.240  | 0.437      | 0.241  | 0.438       | 0.240  |
> |             | 336         | 0.464       | 0.250  | 0.460          | 0.250  | 0.461           | 0.250  | 0.466      | 0.250  | 0.460       | 0.250  |
> |             | 720         | 0.502       | 0.270  | 0.510          | 0.271  | 0.500           | 0.271  | 0.505      | 0.271  | 0.507       | 0.271  |
> |             | Avg         | 0.451       | 0.247  | 0.451          | 0.247  | 0.449           | 0.247  | 0.452      | 0.247  | 0.451       | 0.247  |
> | Weather     | 96          | 0.153       | 0.190  | 0.149          | 0.187  | 0.152           | 0.189  | 0.151      | 0.189  | 0.151       | 0.189  |
> |             | 192         | 0.200       | 0.235  | 0.200          | 0.236  | 0.204           | 0.239  | 0.205      | 0.240  | 0.203       | 0.238  |
> |             | 336         | 0.258       | 0.280  | 0.266          | 0.285  | 0.263           | 0.284  | 0.263      | 0.284  | 0.260       | 0.282  |
> |             | 720         | 0.337       | 0.333  | 0.334          | 0.331  | 0.339           | 0.334  | 0.343      | 0.337  | 0.337       | 0.332  |
> |             | Avg         | 0.237       | 0.260  | 0.237          | 0.260  | 0.239           | 0.261  | 0.240      | 0.262  | 0.238       | 0.260  |
>
> - However, these attempts do not yield performance gains. These results highlight the importance of temporal decorrelation: modifications to the Q matrices **reintroduce correlations** into the output series (since Q is theoretically the optimal orthogonal matrix for temporal decorrelation). This further demonstrates the superiority of the proposed OrthoTrans scheme.
>
> > Q4: On smaller training sets or distribution shifts
>
> - We report performance with smaller training sets in **Appendix I.4**. As a linear-based forecaster, OLinear does not heavily rely on large training data and adapts well when the training size is reduced from 100% to 5% (see Figure 16 on Page 32 of the paper).
>
> - Distribution shifts are challenging for any deep forecasting model. Tables 2 and 3 in the paper show that OLinear performs competitively on real-world non-stationary datasets. In Table 28, OLinear also demonstrates strong **zero-shot** and generalization capabilities.
>
>
> **We hope these responses can fully address your concerns. Thank you again for your detailed feedback!**
>
>
> [1] TimeBase: The Power of Minimalism in Efficient Long-term Time Series Forecasting. ICML 2025.
>
> [2] Robust recovery of subspace structures by low-rank representation. TPAMI 2012.

---

> > ### Author Response · Authors · 2025-08-04
> > **Thanks for your constructive comments!**
> >
> > Dear Reviewer rQjb,
> >
> > We would like to thank you once again for your efforts and constructive comments. We would be happy to address any further questions you may have.
> >
> > All the best,
> >
> > Authors

---

> > ### Comment · Reviewer_rQjb · 2025-08-05
> >
> > Thanks to the authors for the detailed response and experimental results, which are great additions to the paper. I have no further questions, and I will maintain my score.

---

> > > ### Author Response · Authors · 2025-08-05
> > >
> > > Dear Reviewer rQjb,
> > >
> > > We really appreciate your recognition of our work and your kind words, and we are happy to hear that you consider our responses are great additions to the paper and your concerns have been addressed. Again, thank you for your valuable suggestions which have undoubtedly contributed to improving the quality of our paper.
> > >
> > > Many thanks,
> > >
> > > The authors of #10716

---

### Author Response · Authors · 2025-08-01
**Summary of Rebuttal**

We sincerely thank all the reviewers for their insightful reviews and valuable comments, which are instructive for us to improve our paper.

In this work, we propose OLinear, a linear-based model for time series forecaseting in the orthogonally transformed domain. It mainly consists of the (plug-and-play) OrthoTrans and NormLin modules, and "demonstrates **state-of-the-art performance**" (Reviewer rQjb) across "**24 datasets and 140 configurations**" (Reviewer CEMa). Reviewer jUXb states that as a dataset-adaptive approach, OrthoTrans could "**inspire further work**" (Reviewer jUXb). When used as a replacement of self-attention, the NormLin module cuts FLOPs "**by roughly 50%**" (Reviewer 5fFn), "with validated improvements on iTransformer (+5.1%) and PatchTST (+10.1%), which is **really interesting**" (Reviewer CEMa). Reviewer rQjb further notes that "this paper presents a **well-executed** combination of **theoretical innovation** and **empirical rigor**".

In addition, we curate and publicly release **six new multivariate time series datasets**: SP500, DowJones, CarSales, Power, Website, and Unemp, to facilitate the development of the time series community.

The reviewers raised insightful and constructive comments. We are devoting all our efforts to addressing the concerns by providing sufficient evidence and the requested results. Below is a summary of the major responses in our rebuttal:

- **Further analysis**. Reviewer rQjb suggests further analysis of how OrthoTrans improves the rank of attention matrices in Transformer models. In response, we provide the perspective that OrthoTrans can **mitigate the intrinsic low-rank property of time series data**. Empirical results show that OrthoTrans yields larger singular values of the correlation matrix among non-overlapping time series patches than those obtained with other transformation bases. This phenomenon partly **explains the performance advantages of OrthoTrans over DFT and wavelet bases**.

- **More baselines**. As requested by Reviewers CEMa, 5fFn and jUXb, we compare OLinear with more baselines, including SimpleTM (ICLR 2025), TQNet (ICML 2025), TimePro (ICML 2025), and TimeBase (ICML 2025), SAMformer, RLinear, TSMixer, and Pyraformer. OLinear **consistently outperforms these baselines** on a wide range of datasets. Specifically, OLinear outperforms SAMformer by a substantial margin of 12.1%. Notably, replacing self-attention in SAMformer with the NormLin layer improves MAE performance by 10.6% and 5.8% on the Traffic and ECL datasets, respectively.

- **Elaboration on implementation details**. As requested by Reviewer CEMa, we provide detailed explanations of our implementation and use experiments to justify our design choices, including the computation of temporal correlation matrices, the design of the NormLin module, and the details of the “temporal NormLin” baseline. For the channel average (CA) scheme used to compute $\mathrm{CorrMat}_t$, we conduct **ablation studies demonstrating the performance advantages of CA over the channel-individual (CI) counterpart**.

- **Quantitation of temporal decorrelation**. Following Reviewer jUXb's suggestions, we report the Frobenius norm of the off-diagonal elements to **validate OrthoTrans's effectiveness in temporal decorrelation**. We also provide an intuitive solution to reduce the computational load of the NormLin layer. A statistical significance test using Student’s t-test between OLinear and iTransformer shows that **OLinear has a clear performance advantage over the classic iTransformer**. We also demonstrate the robustness of OLinear using **unified hyperparameter settings**.

- **Learnable Q matrices**. Reviewers rQjb and CEMa suggest using learnable Q matrices for potential improvements. To this end, we design several approaches (Plus Delta, Householder, Cayley, and Adapt_Q) to implement learnable Q matrices and compare them with the fixed ones. However, these variants do not outperform the original OrthoTrans. These results highlight the importance of temporal decorrelation and the challenges of learning matrices that perform better than the pre-computed Q matrices.


We are grateful for the reviewers’ valuable suggestions. We would be very happy to answer any further questions.

---

### Note · Authors · 2025-08-12

Dear AC and all reviewers,

We sincerely appreciate the considerable time and effort you have dedicated in evaluating our work. After reviewer-author discussion, we are encouraged that **all four reviewers are on the positive side of acceptance.** To facilitate the upcoming AC-Reviewer Discussions, we summarize the up-to-date status as follows.

- **Reviewer rQjb** comments ```our detailed response and experimental results are great additions to the paper``` and has ```no further questions```. We highlighted that OrthoTrans can mitigate the intrinsic low-rank property of time series, explaining why it improves the rank of attention matrices in Transformer-based forecasters -- we will include these in our final version.

- **Reviewer jUXb** comments that ```our comprehensive response have effectively addressed the initial concerns``` and they have ```no further questions```. In particular, he/she says ```the expanded discussion around Degrees-of-Freedom is very useful``` and ```the bottleneck experiment fully addresses my earlier concerns```. We will definitely follow the reviewer's suggestions to add those points in our camera ready version.

- **Reviewer 5fFn** praises Theorem 1 for ```giving the model a clear theoretical underpinning and enhancing interpretability```, and notes that ```both OrthoTrans and NormLin are modular, training-free components```, leading to both efficiency and effectiveness. He/she also recognized our ```experiments on 140 forecasting tasks drawn from 24 public datasets```, which ```compares against a wide range of baselines``` and is ```notably comprehensive```.

- **Reviewer CEMa** acknowledges that ```our detailed response have treated most of the concerns```, and raises some issues that would like us to solve. After careful double-checking, we realize some confusion may be caused by our typo (the correct one should be $\mathrm{CorrMat}_t$ instead of $\mathrm{CovMat}_t$). In addition, we choose to stick ACF in Section 4.2 for our final version, which may be more familiar to a broad range of readers. We also found the suggestion that "dividing by the number of off-diagonal elements when computing the Frobenius norm of the off-diagonal elements" very useful, and we have reported the updated results in light of this -- thank you so much!

We are confident that our responses can thoroughly address the reviewers' concerns. Thank you so much for the great efforts and thoughtfulness -- we truly appreciate it!

The authors of #10716

---

### Decision · Program_Chairs · 2025-09-17

**Decision:**

Accept (poster)

**Comment:**

This paper introduces OLinear, a new linear multivariate time series forecasting model, where the O indicates that the model operates in an orthogonally transformed domain. It includes two key components: OrthoTrans,  a data-adaptive orthogonal transformation that uses temporal Pearson correlation matrices; NormLin: a lightweight, normalized linear layer that effectively replaces more complex self-attention mechanisms, leading to gains in both accuracy and efficiency. The empirical study shows good performance.

All reviewers gave positive ratings, and indicated that the rebuttal comments are useful to clarify unclear things in the original submission. The authors promised to include the rebuttall comments into the final version to improve the paper. Overall, the paper presents an interesting linear model for time series forecasting.